# Machine-guided design of cell-type-targeting *cis*-regulatory elements

Sager J. Gosai[1,2,3,4,15]✉, Rodrigo I. Castro[5,15]✉, Natalia Fuentes[5,6], John C. Butts[5,7], Kousuke Mouri[5], Michael Alasoadura[5], Susan Kales[5], Thanh Thanh L. Nguyen[8], Ramil R. Noche[9,10], Arya S. Rao[1,11], Mary T. Joy[5], Pardis C. Sabeti[1,3,4,12,16], Steven K. Reilly[8,13,16]✉ & Ryan Tewhey[5,7,14,16]✉

*Cis*-regulatory elements (CREs) control gene expression, orchestrating tissue identity, developmental timing and stimulus responses, which collectively define the thousands of unique cell types in the body[1–3]. While there is great potential for strategically incorporating CREs in therapeutic or biotechnology applications that require tissue specificity, there is no guarantee that an optimal CRE for these intended purposes has arisen naturally. Here we present a platform to engineer and validate synthetic CREs capable of driving gene expression with programmed cell-type specificity. We take advantage of innovations in deep neural network modelling of CRE activity across three cell types, efficient in silico optimization and massively parallel reporter assays to design and empirically test thousands of CREs[4–8]. Through large-scale in vitro validation, we show that synthetic sequences are more effective at driving cell-type-specific expression in three cell lines compared with natural sequences from the human genome and achieve specificity in analogous tissues when tested in vivo. Synthetic sequences exhibit distinct motif vocabulary associated with activity in the on-target cell type and a simultaneous reduction in the activity of off-target cells. Together, we provide a generalizable framework to prospectively engineer CREs from massively parallel reporter assay models and demonstrate the required literacy to write fit-for-purpose regulatory code.

Our understanding of how CREs influence gene expression has been primarily derived from elements that exist naturally in the human genome[2,9–11]. Major efforts over the past decade have identified millions of putative CREs, yet sequences generated by evolution represent only a small subset of possible genetic sequences and may not meet expression objectives that are favourable for therapeutic applications[1,3,12]. Indeed, 200 bp of DNA can encompass over $2.58 \times 10^{120}$ possible sequences, more combinations than there are atoms in the observable universe. This unexplored DNA sequence space offers an untapped reservoir of potential CREs for clinical and biotechnological applications[13]. Bridging the gap in knowledge of regulatory grammar—the vocabulary of activating and repressing transcription factors (TFs), their combinatorial effects and higher-order syntax—has been a major goal of genomics for the past decade and would aid the development of application-specific CREs[1,3,14–17].

Recent advances are reshaping our ability to design CRE sequences with cell-type-specific activity by overcoming three gaps: (1) scalable methods to functionally characterize natural and synthetic CREs to produce generalizable insights; (2) accurate 'regulatory grammar' models of how genetic sequences lead to CRE activity across cell types; and

(3) the ability to repurpose predictive models for directed CRE generation. First, massively parallel reporter assays (MPRAs) can directly quantify the activity of hundreds of thousands of CREs across cell types[8,18–22], providing insights into regulatory syntax and cellular specificity[23–27]. Second, deep learning approaches have proven to be effective tools for predicting the relationships between a DNA sequence and proxies of regulatory activity, such as regions of open chromatin demarcated by DNase I hypersensitivity sites (DHSs), and have been more recently extended to reporter assays[28–37]. Last, although computational models are millions of times faster than experimentation, these models are still incapable of global searches over all possible sequence combinations within the size of a typical human CRE. Efficient frameworks to generate sequences from predictive models could help to address this gap and enable rational and interpretable design of candidate CREs[5,6,11,38–41], as highlighted by recent work designing synthetic CREs to drive cell-type specificity in *Drosophila*[42,43]. However, synthetic CREs designed using predictive models are untested in vertebrates, and their effectiveness compared with natural sequences remains unclear.

Programmed, highly precise, cell-type-specific CREs would contribute to the development of specialized reporters, CRISPR therapeutics,

[1]Broad Institute of MIT and Harvard, Cambridge, MA, USA. [2]Harvard Graduate Program in Biological and Biomedical Science, Boston, MA, USA. [3]Department Of Organismic and Evolutionary Biology, Harvard University, Cambridge, MA, USA. [4]Howard Hughes Medical Institute, Chevy Chase, MD, USA. [5]The Jackson Laboratory, Bar Harbor, ME, USA. [6]Harvard College, Harvard University, Cambridge, MA, USA. [7]Graduate School of Biomedical Sciences and Engineering, University of Maine, Orono, ME, USA. [8]Department of Genetics, Yale School of Medicine, New Haven, CT, USA. [9]Department of Comparative Medicine, Yale School of Medicine, New Haven, CT, USA. [10]Yale Zebrafish Research Core, Yale School of Medicine, New Haven, CT, USA. [11]Harvard Medical School, Boston, MA, USA. [12]Department of Immunology and Infectious Diseases, Harvard T H Chan School of Public Health, Harvard University, Boston, MA, USA. [13]Wu Tsai Institute, Yale University, New Haven, CT, USA. [14]Graduate School of Biomedical Sciences, Tufts University School of Medicine, Boston, MA, USA. [15]These authors contributed equally: S. J. Gosai, R. I. Castro. [16]These authors jointly supervised this work: P. C. Sabeti, S. K. Reilly, R. Tewhey. ✉e-mail: sgosai@broadinstitute.org; rodrigo.castro@jax.org; steven.k.reilly@yale.edu; ryan.tewhey@jax.org

gene-replacement approaches and more. In particular, the lack of robust cell-type-targeted delivery hinders gene therapies from ameliorating a rapidly growing list of human genetic diseases[44]. Being able to precisely fabricate synthetic CREs with highly tissue-specific functions could provide complementary tools to nanoparticle[45] and viral vector[46,47] technologies for gene delivery.

Here we present a method to engineer, ab initio, novel, synthetic CREs capable of driving cell-type-specific transgene expression across three transformed cell lines. We achieve this by integrating previous innovations in modelling regulatory grammar across cell types[4,37], efficient sequence space searching[5–7] and the MPRA experimental system that can validate thousands of CREs in parallel[8,25]. We used a recently generated database of uniformly processed MPRA experiments that characterized an unprecedented number of CREs[27] to train an accurate deep-learning model that can rapidly predict activity for any sequence in silico. Coupled with sequence-generation algorithms, we deploy our model to generate thousands of synthetic CREs with programmed specificity across three cell lines, which we functionally validate in vitro using MPRAs and in vivo by probing physiologically related tissues in mice and zebrafish.

## Models accurately predict CRE activity

We first built an accurate model of CRE activity from DNA sequence alone. Although previous models of CRE activity have primarily used epigenetic states correlated to CRE function[4,32,33,37,48], we trained our model on the regulatory output of 776,474 200-nucleotide sequences directly, as assayed by MPRA, a high-throughput reporter system that quantifies the effect of a given sequence on gene transcription (Methods and Supplementary Tables 1 and 2). These MPRAs were conducted by a single laboratory using a consistent experimental and analytical pipeline, yielding highly reproducible measurements[27] (Fig. 1a, Supplementary Fig. 1 and Supplementary Table 2). In total, we collected functional CRE measurements from 155.3 Mb of unique genomic sequence in each of three human cell types: K562 (erythroid precursors), HepG2 (hepatocytes) and SK-N-SH (neuroblastoma).

We created Malinois, a deep convolutional neural network (CNN) for the prediction of cell-type-informed CRE activity as measured by MPRA for any sequence. We adapted architectural components from Basset[4], a model of chromatin accessibility (Fig. 1b, Methods and Supplementary Fig. 2), and used Bayesian optimization[49,50] to iterate over hyperparameter settings to identify a high-performing model (Methods, Supplementary Note 1 and Supplementary Table 3). Malinois accurately models episomal CRE activity across cell types. For sequences held out from training (62,582 elements on chromosomes 7 and 13), Malinois predictions in K562, HepG2 and SK-N-SH cells correlate highly with empirical activity measurements (Pearson's $r = 0.88-0.89$; Spearman's $\rho = 0.81-0.83$; all $P < 10^{-300}$) (Fig. 1c) and estimate specificity on par with experimental results (Extended Data Fig. 1).

Given that Malinois can accurately and rapidly model CRE activity, we generated genome-wide predictions of sequence activity to compare with orthogonal approaches for characterizing CREs. We observed a strong correlation (Pearson's $r = 0.91$, $P < 10^{-300}$) between Malinois predictions and a comprehensive MPRA of sequences tiling a 2.1 Mb window encompassing $GATA1$ (Fig. 1d and Supplementary Fig. 3). We also found that Malinois K562 cell predictions have strong activity at known markers of CREs identified by DHSs[51] ($P < 10^{-300}$, two-sided paired $t$-test) and H3K27ac chromatin immunoprecipitation–sequencing (ChIP–seq) peaks[52,53] ($P < 10^{-114}$, two-sided paired $t$-test), and are correlated with STARR-seq peaks[52,54] ($P < 10^{-178}$, two-sided paired $t$-test), an orthogonal measure of CRE activity[12,55–57] (Fig. 1e, Extended Data Fig. 2 and Supplementary Table 1). This finding is consistent in HepG2 and SK-S-SH cells as well (Extended Data Fig. 2). Together, these data suggest that Malinois predictions provide accurate measurements of CREs.

## CODA designs CREs with desired functions

We next developed CODA (Computational Optimization of DNA Activity), a modular platform for designing novel CREs with programmed functionality. CODA follows an iterative loop of predicting the activity of sequences, quantifying how well sequences fit the design goals using an objective function, and then updating sequences to increase the objective value (Fig. 2a). Here our goal is to design CREs that drive cell-type-specific reporter transcription in one of the modelled cell lines. We quantify success by calculating the MinGap—the observed difference between the predicted MPRA activity in the targeted cell type and the maximum of the two off-target cell types (Methods). Sequence updates in CODA can be controlled using different sequence design algorithms. We implemented algorithms representative of three broad classes of optimization techniques (evolutionary, AdaLead[6]; probabilistic, Simulated Annealing[7]; and gradient based, Fast SeqProp[5]) for sequence generation (Supplementary Fig. 4). We selected these methodologies on the basis of their ease of implementation, previous documented successes or their ability to exploit the structure of deep-learning models. We found that the overall ability of these algorithms to design cell-type-specific elements is generally robust to hyperparameter choices (Supplementary Fig. 5). However, adjustments can be made to balance the trade-off between maximizing the objective and maintaining $k$-mer diversity in the set of designed elements.

To empirically test the effectiveness of CODA, we performed an MPRA to measure activity of the synthetic sequences. For each cell type, we generated 4,000 cell-type-specific sequences from each of the three sequence design algorithms in CODA, yielding a total of 36,000 synthetic candidates (Fig. 2b, Methods and Supplementary Table 4). We observed that Malinois induced strong preferences for certain sequence motifs when maximizing specificity (Supplementary Tables 5 and 6 and Supplementary Note 2). For this reason, we decided to encourage CODA to reduce the use of highly preferred motifs despite a potential decrease in predicted cell-type specificity by penalizing their inclusion in designs (Methods). Using Fast SeqProp, we designed a second group of 15,000 synthetic sequences with a motif penalty incorporated into the objective function, which diversified motif content (Fig. 2b, Methods, Supplementary Note 2 and Supplementary Table 5). Levenshtein distance and $k$-mer similarity analyses showed that all of the methods used to generate synthetic CREs resulted in sets of sufficiently diverse sequences (Methods and Supplementary Note 2).

We also selected naturally occurring CREs from the human genome to investigate how well these sequences drive cell-type-specific activity compared with our synthetic designs. H3K27ac histone marks and chromatin accessibility, as measured by DHSs, are common proxies for active CREs[1,51]. Thus, for each cell line, we identified 4,000 'DHS-natural' sequences with cell-type-specific chromatin accessibility and overlapping H3K27ac signals (12,000 total) (Methods and Supplementary Fig. 6). We then scanned the entire human genome for 200-mers predicted to be cell-type specific by Malinois and selected 4,000 'Malinois-natural' sequences with the greatest on-target expression and minimal off-target expression in each of the three cell lines (Methods and Extended Data Fig. 3a,b). Notably, there was low overlap between elements identified using DHSs and Malinois (0.10–4.1% intersection depending on the cell type of interest; Extended Data Fig. 3c). Although DHS-natural sequences displayed high levels of chromatin accessibility, Malinois-natural and synthetic sequences were predicted to have greater cell-type specificity, with non-penalized synthetic sequences surpassing all groups (Supplementary Fig. 7).

## Synthetic CREs are highly specific

We experimentally tested the library of 77,157 natural and synthetic sequences to determine whether machine-guided sequence design could reliably generate biologically functional elements with desired

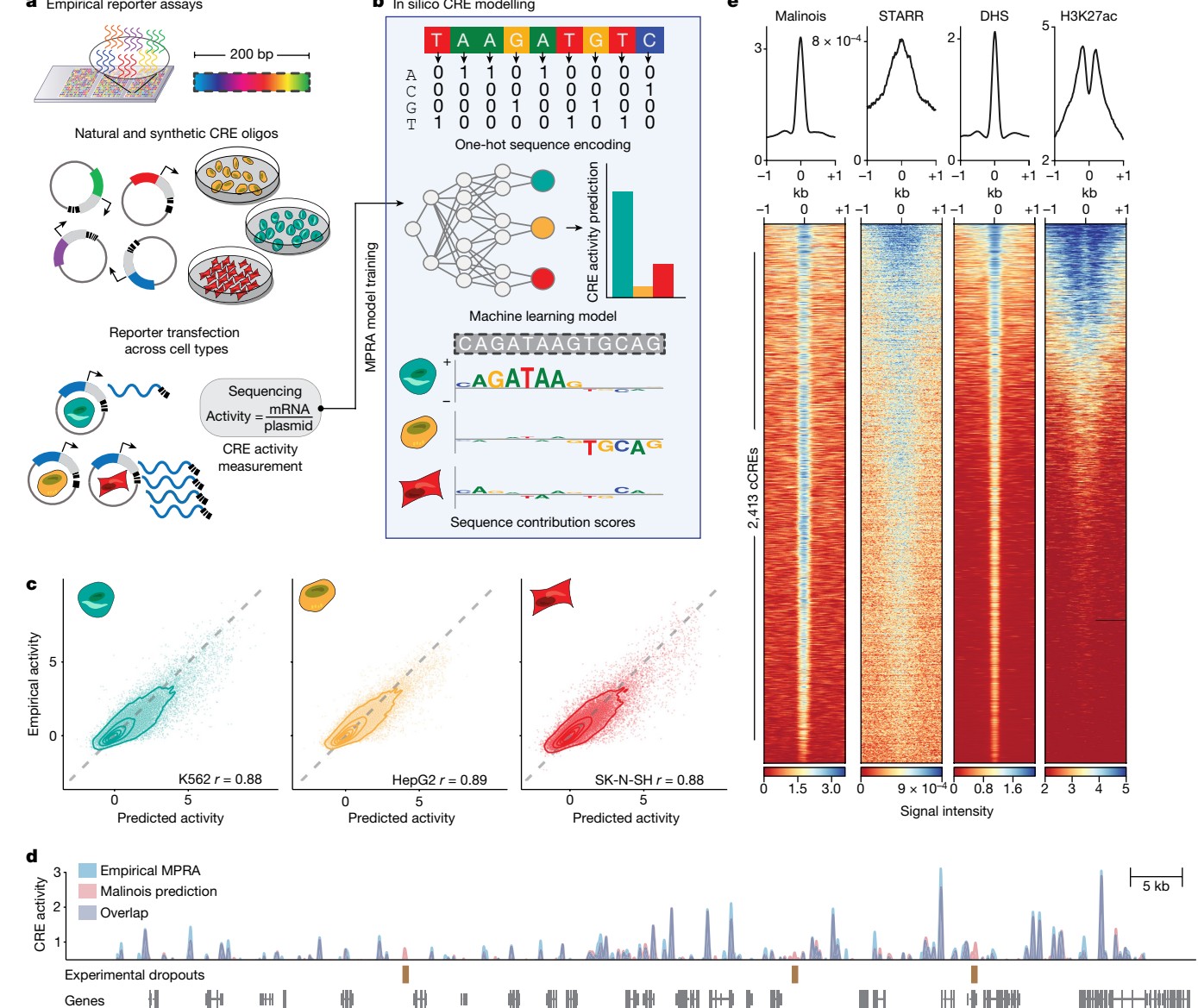

**Fig. 1 | Malinois accurately predicts transcriptional activation by CREs in episomal reporters. a**, Empirical MPRAs enable targeted functional characterization of the effects of hundreds of thousands of CREs on transcription in episomal reporters, and can quantify the impact of programmable 200 bp oligonucleotide sequences. MPRAs across multiple cell types enable the identification of cell-type-specific activity of CREs. **b**, Malinois is a deep CNN model that predicts cell-type-specific CRE effects directly from the nucleotide sequence in K562 (teal), HepG2 (yellow) and SK-N-SH (red) cells. Contribution scores extracted from the model determine how subsequences drive predicted function in each cell type. **c**, Malinois predictions are highly correlated with empirically measured MPRA activity across K562 (teal), HepG2 (yellow) and SK-N-SH (red) cells. The performance for each cell type was measured using Pearson correlation (*r*) analysis of a test set of sequences that were withheld from training (*n* = 62,562 oligos, *P* < 10⁻³⁰⁰).

Each point corresponds to the empirical and predicted activity of a single CRE in the corresponding cell type, and the topological lines indicate the point density (16.7%, 33.3%, 50%, 66.7%, 83.3%) in the scatter plots. Train–test splits were defined by chromosomes. **d**, Malinois predictions recapitulate an MPRA screen of overlapping fragments derived from a 2.1 Mb window centred on the *GATA1* gene (Pearson's *r* = 0.91, *n* = 51,242 oligos, *P* < 10⁻³⁰⁰; Supplementary Fig. 3). Purple signal indicates overlapping measurements, and the blue and red signals indicate either higher activity measurements or predictions by MPRA or Malinois, respectively, in the window chromosome X: 48000000–49000000. **e**, Malinois activity predictions for sequences centred on candidate CREs (cCRE) in chromosome 13 demarcated by DHS peaks in K562 cells (*n* = 2,413 peaks). This pattern of activation is concordant with quantitative signals measured using STARR-seq, DHS-seq and H3K27ac ChIP-seq.

activity. In total, the library included 51,000 synthetic sequences (36,000 standard and 15,000 motif-penalized), 24,000 natural sequences (12,000 DHS-natural and 12,000 Malinois-natural) and 2,157 experimental controls (Fig. 2b and Supplementary Note 3). We quantified the activity of an individual CRE as the log₂-transformed fold change (FC) in the expression of the reporter gene driven by the CRE compared with a set of negative controls (Fig. 2c). Malinois prospectively predicted empirical MPRA measurements of this library

with high accuracy (Pearson's *r* = 0.79–0.91; Spearman's *ρ* = 0.84–0.92; all *P* < 10⁻³⁰⁰; Extended Data Fig. 4 and Supplementary Fig. 8), suggesting that the predictive accuracy of Malinois is not limited to natural sequences.

Malinois also identified naturally occurring sequences with expression specificity in the modelled cell lines. Consistent with a priori Malinois activity predictions of genomic sequences, DHS-natural sequences in all three cell types performed poorly as cell-type-specific

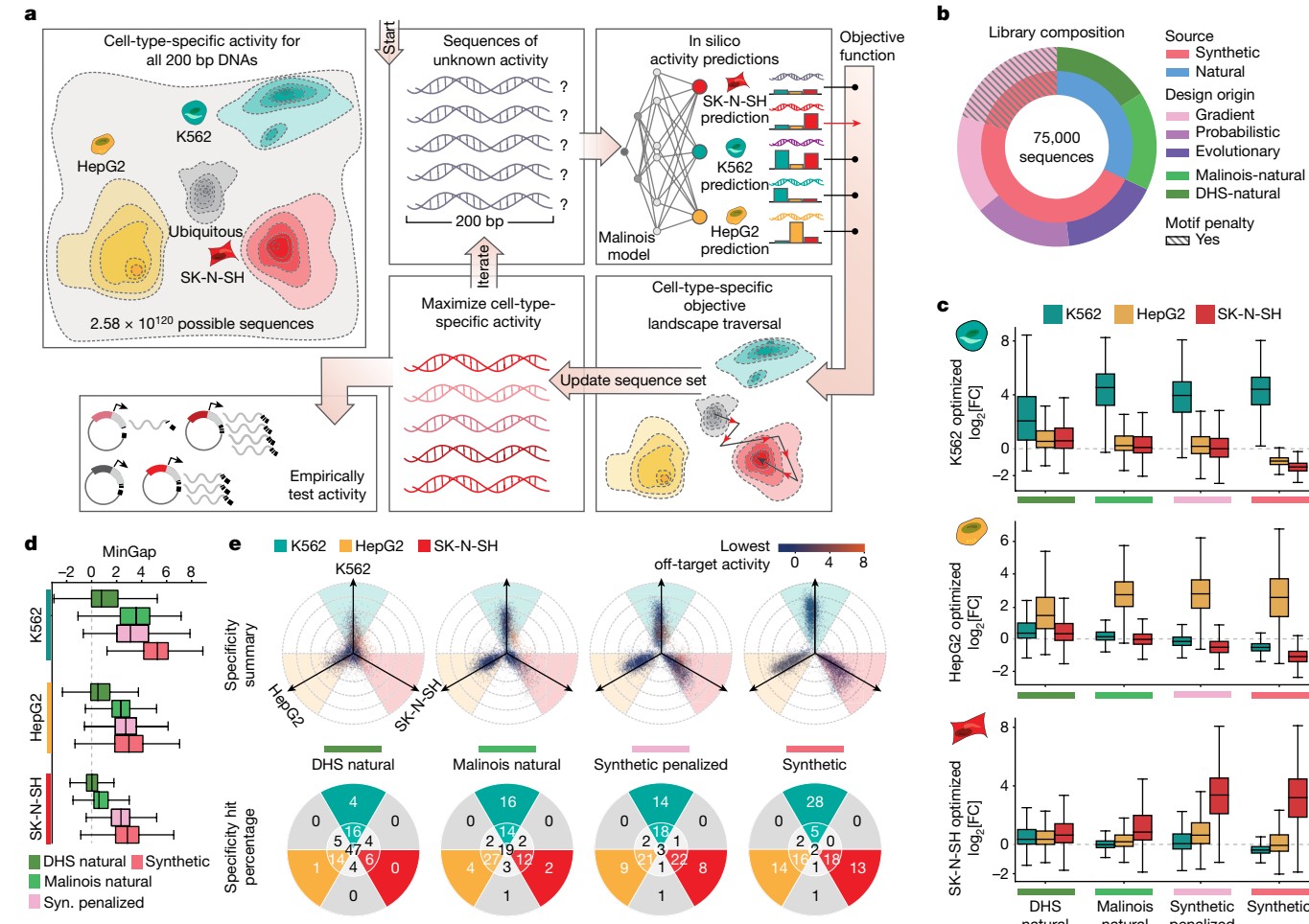

**Fig. 2 | CODA effectively designs cell-type-specific CREs. a**, CODA designs synthetic elements by iteratively updating sequences to improve predicted function. Malinois predicts CRE activity and an objective function directs sequence updates. After a stopping criteria is met, candidates are nominated for experimental validation. **b**, The MPRA library composition used to empirically evaluate candidate CREs. **c**, The distribution of MPRA log₂[FC] measurements, each row of the boxes corresponds to candidate CREs intended to drive specific expression in K562, HepG2 and SK-N-SH cells, respectively. Each box indicates measurements in either K562 (teal), HepG2 (yellow) or SK-N-SH (red) cells for the set of sequences nominated by the indicated design strategy on the *x* axis (left to right, top to bottom, *n* = 3,729; 3,410; 4,800; 10,867; 3,757; 3,727; 4,735; 10,917; 3,261; 3,804; 4,866; 11,677 elements). **d**, The distribution of MinGap scores (box plots), quantifying specificity; the colour indicates the

intended target cell type (K562, teal; HepG2, yellow; SK-N-SH, red; *n* values are as described in **c**, ordered from top to bottom). For the box plots in **c** and **d**, the centre line shows the 50th percentile, the box limits show the 25th and 75th percentiles, and the whiskers indicate the outermost point within 1.5× the interquartile range from the edges of the boxes. Syn. indicates synthetic. **e**, Propeller plots for each sequence group (top). The radial distance corresponds to the distance between the maximum and minimum cell type activity values, and the angle of deviation from an axis quantifies the relative activity of the highest off-target cell type (Methods). The dot colours indicate minimum activity across cell types. Bottom, the percentages of points in each delimited area rounded to the nearest integer. The groups synthetic and synthetic-penalized were randomly subsampled to resemble the size of the two natural groups. From left to right, *n* = 10,747; 10,941; 12,000 and 12,000 (Supplementary Fig. 8).

CREs compared with natural sequences identified by Malinois (median MinGap difference Malinois-natural versus DHS-natural: K562, 2.78; HepG2, 1.84; SK-N-SH, 0.57; *P* < 10⁻²⁵⁸ for all, one-sided Wilcoxon rank-sum tests; Fig. 2d, Extended Data Figs. 5 and 6 and Supplementary Fig. 7), suggesting that H3K27ac-positive DHS peaks are poor predictors of specificity measured by MPRA in the cell lines tested. These differences in MinGap were primarily driven by weaker on-target activity for DHS-natural sequences compared to Malinois-natural in K562 (median log₂[FC]: DHS-natural, 2.06; Malinois-natural, 4.54) and HepG2 (DHS-natural, 1.44; Malinois-natural, 2.72) cells, while low on-target activity in SK-N-SH cells in both groups (DHS-natural, 0.64; Malinois-natural, 0.84) resulted in a lower MinGap difference and reduced SK-N-SH cell specificity observed in natural sequences in general.

Synthetic sequences on aggregate outperformed both groups of natural sequences in MinGap measured across all three cell lines (median MinGap difference synthetics versus Malinois-natural: K562,

1.70; HepG2, 0.65; SK-N-SH, 2.28; *P* < 10⁻¹²¹ for all, one-sided Wilcoxon rank-sum tests; Fig. 2d and Extended Data Figs. 5 and 6). Between design methodologies, Fast SeqProp demonstrated greater consistency and slightly higher MinGap across all cell types (mean MinGap difference Fast SeqProp: 0.41 over Simulated Annealing, 0.62 over AdaLead; adjusted *P* < 10⁻³⁰⁰, Tukey's honest significant difference test). Performance gains for all synthetic groups were primarily driven by greater repression in off-target cell types (median off-target log₂[FC]: synthetic, −0.69; Malinois-natural, 0.09; DHS-natural, 0.41). Moreover, synthetic sequences had a higher on-target activity in SK-N-SH cells (median log₂[FC] 3.20) compared with both natural groups, and higher on-target activity for HepG2 and K562 cells compared with DHS-natural sequences (Fig. 2c). In summary, synthetic sequences consistently achieved the largest quantitative separation between target and off-target cell types when compared to both classes of naturally derived sequences.

In addition to evaluating specificity using MinGap, we quantified and visualized specificity using a radial coordinate system whereby the most specific sequences trend outwards along one of the three cell type axes, while sequences with uniform activity across cell types are drawn toward the origin. We categorize CREs as cell type-specific if two conditions are met: (1) the MaxGap is greater than 1 (the $\log_2$[FC] separation between the target cell type and minimum off-target); and (2) the maximum off-target cell type is closer to the minimum off-target than to the target (Fig. 2e and Methods).

Using our criteria to categorize cell-type-specific CREs, we observed that most (94.1%) synthetic sequences designed by CODA successfully drive cell-type specificity (Fig. 2e and Supplementary Figs. 9 and 10). Depletion of the most optimal motifs did not impact success substantially, with 92.4% of motif-penalized sequences still driving specificity. Comparatively, we observed that Malinois-natural (73.6%) and DHS-natural sequences (40.6%) were less successful (Fig. 2e). When increasing the stringency of the MaxGap by fourfold, synthetic sequences (54.7% specific) further outperformed Malinois-natural (21.5%) and DHS-natural (4.7%) sequences, as well as motif-penalized sequences (30.8%). Overall, synthetic CREs, which lack homology to the human genome, more consistently drive robust specificity in large part through repression of off-target activity, as well as through some increases in on-target activity (Methods).

## TF content drives cell-type specificity

Having found that synthetic CREs are more cell-type-specific than both classes of natural sequences, we sought to link sequence content to the responsible TF vocabulary. Transcription is controlled in part by individual TF binding to sequence motifs as well as interactions between TFs[15]. First, we used Malinois to predict nucleotide-resolution activity contribution scores for each sequence in the three cell types using a modified version of Integrated Gradients[58] (Methods and Supplementary Note 4). We next used TF-MoDISco[59,60] to identify 66 motif patterns informed by contribution scores, from which we extracted 36 non-redundant core motifs (7–18 bp) enriched in our MPRA-tested library, with 31 confidently aligning to a known human TF-binding motif[61,62] (Methods, Supplementary Figs. 11 and 12 and Supplementary Table 6).

The regulatory activity contribution scores identify the overall magnitude and direction of the effect of each motif in each of our three cell lines (Fig. 3a). Of the 36 core motifs, 28 had positive predicted contributions to sequence activity, while the remaining 8 were repressive (Fig. 3b). This included well-known activators such as GATA[63], an essential TF expressed in K562 cells that was predicted by Malinois to drive activity exclusively in K562 cells. Likewise, HNF1B and HNF4A, master regulators expressed in hepatocyte development[64–67], overlapped with high positive contribution scores exclusively in HepG2 cells. Motifs displaying negative contributions included the repressors GFI1B in K562 cells[68–70] and MEIS2 in HepG2 and SK-N-SH cells[71–73]. All motifs demonstrated predicted effects in accordance with their assigned contribution when embedded in a random background, as well as when replacing their instances in the library with random sequences (Methods and Supplementary Figs. 11 and 12).

We examined whether motif use differed between natural and synthetic sequences using a contribution-score-based motif scan (Methods and Supplementary Table 7). All of the 36 core motifs occur at least once in both synthetic and natural sequences, suggesting a shared vocabulary between the two classes (Fig. 3b and Extended Data Fig. 7). However, the use of motifs differed. For example, motifs for transcriptional activators GATA in K562 cells and HNF4A in HepG2 cells were deployed at higher rates in synthetic sequences (all synthetics: 92.3% and 77.1%, respectively; all naturals: 69.8% and 47.2%, respectively), as well as the repressors MEIS2 in K562 cells and GFI1B in HepG2 cells (all synthetics: 71.4% and 74.5%, respectively; all naturals:

24.6% and 40.8%, respectively) (Extended Data Fig. 7). Lexical analyses showed that synthetic sequences are typically composed of a greater number of unique motifs as well as more total motifs compared to naturals, while penalized synthetics showed a higher degree of non-redundant motif use than non-penalized synthetics (for all three comparisons, $P < 10^{-300}$, one-sided Wilcoxon rank-sum tests; Supplementary Note 2).

Notably, we also observed a higher use of particular motif combinations in synthetic sequences. For example, among non-penalized synthetic sequences, we see higher rates of GATA and MEIS2 in K562 cells (89.2%) and HNF4A and GFI1B in HepG2 cells (64.6%), compared with natural sequences (17.9% and 18.8%, respectively) (Fig. 3c, Methods and Supplementary Fig. 13). Pairs of distinct activating motifs were observed in most non-penalized synthetic and Malinois-natural sequences (95.7% and 93.4%, respectively), while activating–repressive and repressive–repressive motif pairs were observed at lower rates in naturals (activating–repressive: synthetic, 99.9%; Malinois-natural, 83.1%; repressive–repressive: synthetic, 98.9%; Malinois-natural, 57.6%).

## CRE groups display different semantics

In addition to single TF-motif usage and pair-wise co-occurrence, cell-type specificity is thought to arise through higher-order motif semantics, which can mediate the complex organization of many TFs to impart CRE activity[3,13,15,16]. To aggregate semantically related motifs into functional programs, we used non-negative matrix factorization (NMF)[74] to decompose sequences in our library into a mixture of 12 functional programs based on motif content calculated using contribution score-based motif mapping (Methods and Supplementary Fig. 14). These programs describe co-occurring TF vocabularies found in the elements that we tested. NMF identified five programs associated with clear cell-type-specific activity (one program in K562 cells, and two in each for HepG2 and SK-N-SH cells), with the seven remaining programs associated with pleiotropic activation and/or repression (Fig. 3d and Supplementary Fig. 15a).

Natural and synthetic sequences deploy distinct distributions of semantic programs (Fig. 3e and Supplementary Fig. 15b). While there are quantitative differences in program preference between the different synthetic sequence design methods, there are no programs unique to one method. Overall, synthetic elements have higher program content, consistent with higher motif usage, and greater program heterogeneity compared to natural CREs (Supplementary Fig. 16a,b). We also found that natural sequences primarily rely on activating programs, while synthetic sequences also frequently use programs with repressive effects in off-target cell types (median repressing program content: DHS-natural, 0.077; Malinois-natural, 0.064; synthetic, 0.123) (Supplementary Fig. 16c,d). The vast majority of synthetic sequences (91.9%) comprise both activating and repressing programs, each exceeding a threshold of 0.1, while relatively fewer DHSs (26.9%) and Malinois (25.3%) natural sequences show this combination (Methods and Supplementary Fig. 16e). These results support our motif-based observations that the improved performance of synthetic sequences is due to a combination of on-target activation and off-target repression.

## Selected CODA CREs are specific in vivo

We next sought to assess whether the specificity of synthetic CREs would generalize beyond the initial three cell lines used for design. To determine whether specificity is maintained when adding new cell lines, we trained additional models for A549 (lung epithelial cancer) and HCT116 (colon epithelial cancer) cells, observing that synthetic CREs retained maximum predicted activity in their target cell type over A549 and HCT116 cells, especially those generated using Fast SeqProp, albeit with a reduced MinGap (Supplementary Fig. 17).

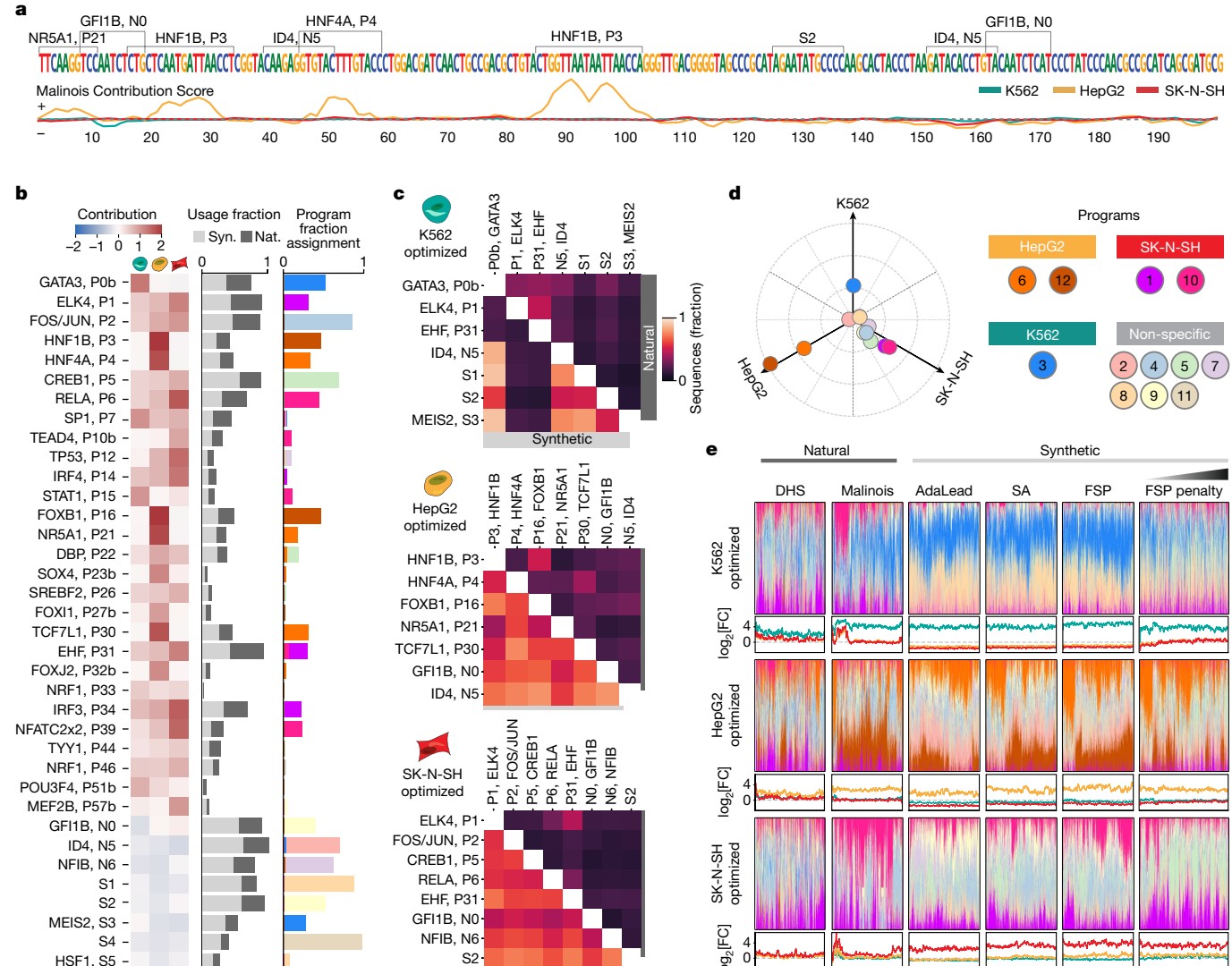

**Fig. 3 | Interpreting functional sequence content. a**, Malinois contribution scores of a representative synthetic CRE designed to drive HepG2-cell-specific expression. Enriched motifs are demarcated above the sequence and contribution scores are plotted below (K562, teal; HepG2, yellow; SK-N-SH, red) (Methods). **b**, The average contributions of core motifs in K562, HepG2 and SK-N-SH cells (left to right columns) (left). Middle, motif enrichment in synthetic (light grey) and natural (dark grey) sequences. The x axis represents fraction of sequences in each group containing the motif denoted on the y axis. Right, motif program association derived from the NMF feature matrix. The colours correspond to programs listed in **d**. **c**, Co-occurrences of enriched motifs. The colour indicates the percentage of sequences in each group containing a pair of motifs (Methods and Supplementary Fig. 13). The upper and lower triangular percentages correspond to natural and synthetic sequences, respectively. **d**, The empirical program function was calculated

using a weighted average of MPRA $\log_2$[FC] scores based on program mixture displayed in **e**. Ten specificity-driving programs were identified using the same criteria applied to sequences (bright coloured points). Seven programs are not associated with cell-type-specific transcription (pastel colours). Program 11 is overplotted by program 8, and program 4 partially obstructs program 9 on the plot. **e**, NMF decomposition of synthetic and natural sequences based on enriched motif content. For each sequence, programs are coloured based on the key in **d** and are plotted as a fraction of the total program content. Sequences not assigned to any program with any frequency yield a blank bar. Line plots display empirical activity in K562 (teal), HepG2 (yellow) and SK-N-SH (red) cells. SA, simulated annealing; FSP, Fast SeqProp. Sequences in each subpanel are sorted by hierarchical clustering based on program content (FSP penalty, n = 5,000; all others, n = 4,000).

To assess the specificity of synthetic CREs beyond an episomal reporter context in vitro, we evaluated selected sequences for their ability to drive cell-type-specific expression in vivo. Using Enformer, a deep learning model trained on gene regulatory signatures from primary tissues, we predicted the impact of synthetic CREs on epigenetic and transcriptional markers for gene activation[37] (Methods, Extended Data Fig. 8a and Supplementary Table 8). Specificity, as measured by MPRA in K562, HepG2 and SK-N-SH cells, was significantly correlated with tissue-specific Enformer scores in spleen, liver and neural structures, respectively (Extended Data Fig. 8b–d) and was higher in synthetic elements than in both groups of natural sequences (Extended Data Fig. 8e).

Encouraged by in vivo specificity of synthetic CREs as measured by in silico approaches, we established a pipeline to nominate and evaluate sequences directly in vertebrate models. Using empirical MPRA results, Malinois contribution scores, in silico predictions of tissue-specific epigenetic signals and manual inspection of motif organization, we nominated three HepG2- and three SK-N-SH-specific CREs, which we anticipated to be liver- and neuron-specific, respectively, for in vivo characterization in zebrafish embryos (Fig. 4a, Methods and Extended Data Fig. 9).

We inserted synthetic sequences upstream of a minimal promoter driving *GFP* in a zebrafish reporter to emulate the vector design used

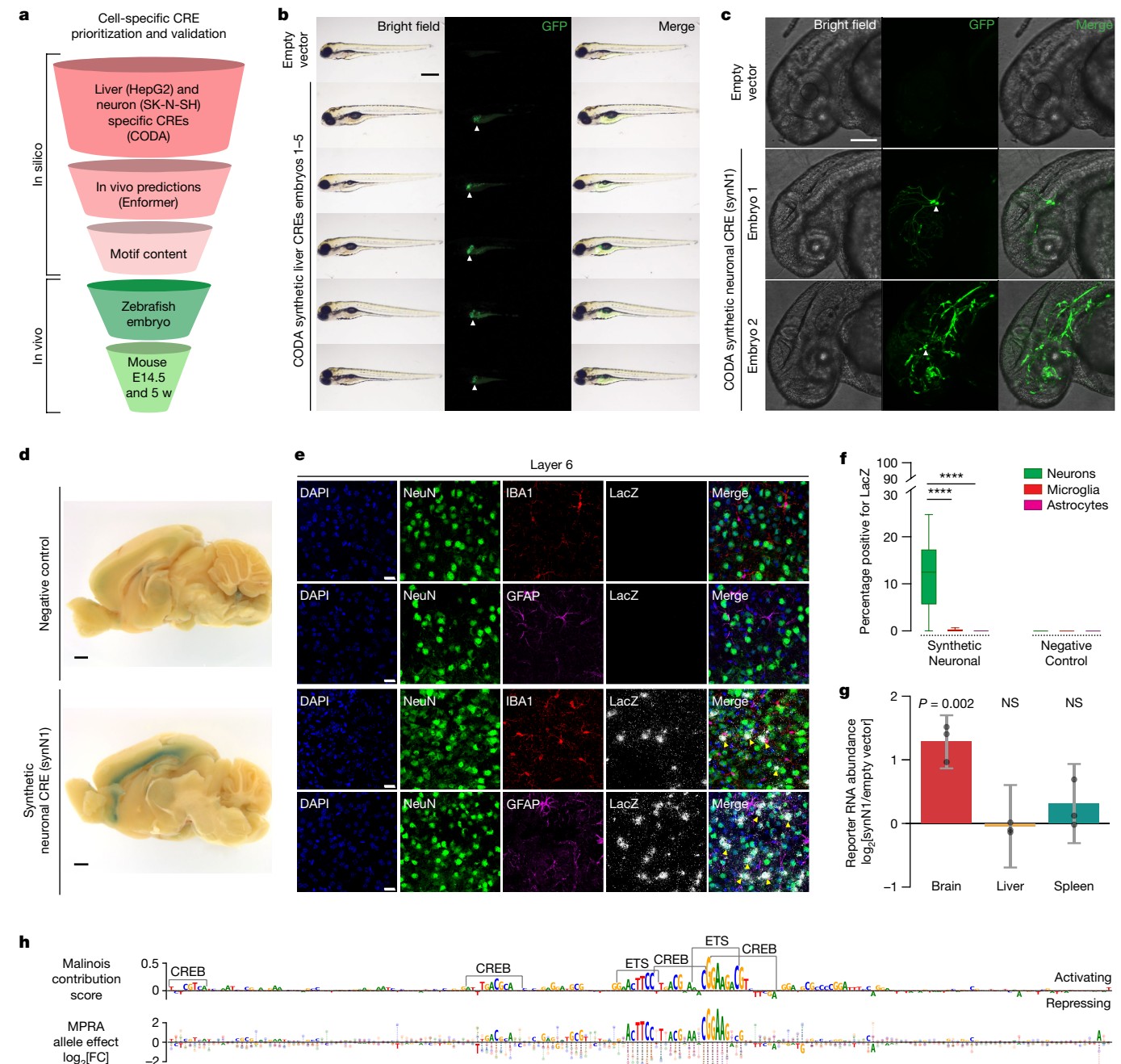

**Fig. 4 | In vivo validation of synthetic elements. a**, Synthetic CREs in vivo validation prioritization. **b**, CODA-designed HepG2-cell-specific CRE activity imaging in transgenic zebrafish at 96 h after fertilization. Lateral view, anterior on the left, dorsal up. The liver is indicated by white arrows. Liver expression was observed in 27 out of 36 animals (Supplementary Fig. 18). Scale bar, 500 μm. **c**, CODA-designed SK-N-SH-cell-specific CRE activity imaging in transgenic zebrafish at 48 h after fertilization. Lateral view of the brain and anterior spinal region, anterior left, dorsal up. The white arrows indicate GFP-positive neuronal cells. Embryo 2 shows incidental off-target vasculature expression. Neural staining was replicated in an additional animal. Scale bar, 125 μm. **d**, Synthetic SK-N-SH-cell-specific CRE (synN1) activity in 5-week-old postnatal mice measured by X-gal staining for LacZ in the medial brain section. Cortical X-gal signal was detected in *n* = 0 out 5 negative controls and *n* = 3 out of 6 synN1s. Scale bars, 1 mm. **e**, CRE activity (LacZ) in neocortex layer 6 with neuronal (NeuN), microglial (IBA1) and astrocyte (GFAP) co-staining. Top, control transgene activity. Bottom, synN1 activity. The arrows indicate colocalization between LacZ signals and neurons. Scale bars, 20 μm. Results

were replicated in *n* = 3 negative control and synN1 mice each, using *n* = 5 sagittal slices per mouse. **f**, The proportion of neurons, astrocytes and microglia positive for the transgene. *n* = 3 mice. Statistical analysis was performed using Kruskal–Wallis one-way analysis of variance; ****adjusted *P* < 10$^{-4}$. For the box plots, the centre line shows the 50th percentile, the box limits show the 25th and 75th percentiles, and the whiskers indicate the outermost point within 1.5× the interquartile range from the edges of the boxes. **g**, LacZ expression by synN1 was measured using RNA-sequencing (RNA-seq) normalized to the *lacZ* expression in transgenic mice for the minP empty vector. Data are mean ± s.e.m. Statistical analysis was performed using two-sided Wald tests. *n* = 3 mice per genotype. **h**, synN1 functional characterization. Top, SK-N-SH cell contribution scores. ETS- and CREB-like binding motifs are highlighted. Bottom, single-nucleotide MPRA saturation mutagenesis. The circles represent the expression change from each mutation (A, green; C, blue; G, yellow; T, red). The letter height represents the negative mean mutational expression change.

by CODA during in vitro testing[75]. To identify the unique expression patterns of each regulatory element, we integrated the reporter into the genome of zebrafish embryos and performed whole-animal imaging at 48 and 96 h after fertilization. For sequences designed to drive activity specifically in the liver, two out of three sequences demonstrated strong, consistent expression in the developing liver (Fig. 4b and Supplementary Figs. 18 and 19). Notably, we detected minimal off-target expression in non-targeted cell types outside the autofluorescent yolk sac. Sequences designed for neuronal specificity showed similar success (two out of three), driving expression in a subset of neuronal cell types (Fig. 4c and Supplementary Fig. 20). For both successful neuronal-nominated CREs, we observed GFP expression within cell bodies and axonal projections of the developing brain and spinal cord (Fig. 4c and Supplementary Fig. 20h).

We next evaluated whether the activity of the two sequences with neuronal specificity in zebrafish extended to a mammalian mouse model system. We placed each synthetic CRE sequence into a targeting vector upstream of a minimal promoter driving *lacZ* and *GFP*, and integrated the construct at the H11 safe-harbour locus of the mouse through zygote microinjection[76]. We collected embryos at embryonic day 14.5, a timepoint that is developmentally similar to that used in zebrafish, and examined the expression patterns of the reporter using lacZ staining of the transgenic embryos. Expression of the synthetic neuronal CRE 1 (synN1) was restricted to the developing cortex with no additional expression observed elsewhere (Supplementary Fig. 21a,b). To localize the expression patterns further within the cortex, we repeated the reporter assay with the synN1 CRE and performed in situ staining of the whole brain at 5 weeks postnatally (Fig. 4d and Supplementary Fig. 21c–h). We observed that cortex-specific expression is maintained in postnatal mice, with focal activity occurring in the neurons at neocortical layer 6 and at subplate neurons (Fig. 4e–g and Extended Data Fig. 10a,b).

Having designed and validated a novel CRE with strong neuronal specificity, we sought to further elucidate the factors responsible for transcriptional activity on the synN1 CRE in neuronal cells. Using Malinois' contribution scores in SK-N-SH cells, we observed two distinct motif classes as contributors to sequence activity: (1) two primary ETS GGA(A/T)-binding domains; and (2) four CREB-like TGACGCA-binding domains (Fig. 4h). ETS factors constitute one of the largest TF families, and its members exhibit highly similar binding motifs. Previous research has reported the potential of ETS factors to form heterodimers with CREB[77], and our contribution scores provided support for two heterodimer pairings in the sequence (Fig. 4h and Methods). To empirically validate Malinois contribution scores, we conducted a saturation mutagenesis MPRA in SK-N-SH cells, which confirmed high-contribution regions and supported ETS and CREB motif assignments (Fig. 4h and Methods). In the off-target cell types, contribution score profiles of ETS and CREB-like motifs were either reduced or absent, with the presence of two additional negatively contributing motifs, closely matching the repressor GFI1 (Extended Data Fig. 9d). This suggests that the specificity of neuronal synN1 could be partly attributed to the on-target transcriptional activity of cooperative heterodimers and off-target repression by GFI1.

## Discussion

Here we developed CODA, an effective strategy to design synthetic CREs that can direct cell-type-specific gene expression. CODA builds on previous sequenced-based methods to take advantage of the complex combinatorial rules of regulatory grammar learned by Malinois to identify cell-type-specific CREs from natural or rationally designed sequences[22,78–80], as well as more recent approaches for fully synthetic CREs[42,43]. We designed synthetic CREs in human cells and performed large-scale empirical validation, enabling well-powered comparisons between synthetic and natural sequences.

Synthetic sequences designed by CODA reliably outperform natural sequences in driving cell-type-specific gene expression in a reporter system assayed across transformed cell lines. We show that CODA can identify synthetic sequences that regularly outperform natural ones with far greater efficiency than random search[42,43] (Supplementary Fig. 22), but without assuredly identifying the global optima. CODA-designed synthetic CREs achieve higher specificity by deploying on-target activating and off-target repressing TFs in unique combinations that are not commonly found in the human genome. This suggests that our models have learned a component of the foundational rules governing CREs, and possess the ability to extrapolate this knowledge to rarely observed TF combinations.

Using Malinois, a direct model of a CRE's transcriptional output, we were able to identify genomic sequences with moderate proficiency for cell-type-specific activity, albeit to a lesser degree than synthetic sequences. Notably, Malinois was more proficient than traditional markers of CRE activity (such as DNase and H3K27ac) at identifying sequences in the genome capable of cell-type-specific reporter expression in the transformed cell lines studied here. This underscores the need to carefully consider sequences outside the typically studied candidate CREs when generating libraries with the intent to train high-performance models[11].

Our high success rate in modelling, generating and testing sequences specific to individual transformed cell lines in vitro prompted us to assess how that activity might extend to complex tissues in vivo. Despite potential challenges of incomplete conservation of tissue types, heterochrony and lineage-specific regulatory grammar, our CREs displayed conserved, tissue-level cross-species activity in zebrafish and mice. These findings show that it is feasible for CREs with novel functionalities developed in vitro to maintain specificity in analogous tissues in vivo. We were surprised that our neuronal synN1 CRE, designed from a single transformed SK-N-SH cell line, exhibited highly specific subcortical expression in mice. Further research is needed to develop optimal strategies to translate in vitro models to precise targeting in vivo. An integrated framework that combines human cell lines with whole-organism experimental models may be an effective approach to rapidly identify CREs capable of accomplishing new functions in humans.

Transgenic applications, such as gene therapies that require tissue, cell type or diseased cell state specificity, will probably benefit from design and validation of synthetic CREs with programmable functions. Training models on MPRA in additional cell types with greater clinical relevance could enable CODA to better design CREs with specificity tailored for therapeutic applications. As the technology underlying sequence-to-function models continues to evolve, we expect synthetic element designs to become even more reliable and reduce the experimental burden for in vitro and in vivo validation. Over the course of this study, several advances in DNA modelling have been reported by other groups that would probably yield such improvements[14,25,36,81,82], but were not tested here.

Although we successfully deployed CODA for cell-type specificity, the platform is designed to be flexible to any objective function. By combining alternative experimental platforms and models with CODA in the future, we could then explore the expansive landscape of synthetic CREs to achieve goals for which evolution may not have optimized, including drug responsiveness (for example, to glucocorticoids), to fine-tune expression outputs, or to respond to the complex syntax specific to cancer cells. This work, in conjunction with other recent studies, demonstrates that machine learning models are capable of writing transcriptional regulatory code tailored to diverse purposes, establishing a framework that can serve as a valuable catalyst for improving specificity of gene therapies.

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

# Methods

## Training Malinois, a model of MPRA activity of CREs

To enable systematic evaluation of parameters governing data pre-processing, model architecture and training, we developed tools for limited automatic machine learning in PyTorch (https://github.com/sjgosai/boda2). We implemented support for regression based on DNA sequences using CNNs. We deployed a containerized application based on this library in conjunction with the Vertex AI platform on Google Cloud to tune all hyperparameters using Bayesian optimization.

## Data preprocessing

**Malinois training.** To construct the train/validation/test dataset to train Malinois, we aggregated the $\log_2[FC]$ output of sequences tested in K562, HepG2 and SK-N-SH cells from multiple projects (OL indexed reference files are shown in Supplementary Table 1). The majority of projects focused on testing the allelic effects of human genetic variation with the remaining projects testing only the reference sequences of the human genome. In total, 798,064 unique oligos were aggregated, originating from ten independent experiments (from three different projects: UKBB (OL27, OL28, OL29, OL30, OL31, OL32, OL33), GTEx (OL41, OL42), OL15). The majority of the sequences used in our study (783,978) were designed to evaluate common human genetic variation associated with heritable complex traits. The majority of sequences (706,054) consisted of testing the reference and alternative allele, typically a single-nucleotide substitution, centred within 200 bp of flanking sequence. Additional sequences (77,924) evaluated the four pairwise combinations of two independent variants. Variants were selected on the basis of genetic fine-mapping, with most variants being linkage disequilibrium partners of causal alleles and therefore likely to not have a meaningful impact on cellular or organismal traits. The remaining sequences (14,086) originated from OL15, from which we selected the known DHSs and H3K27ac sequences. Oligos with a plasmid count of fewer than 20 or no RNA count in any cell type were discarded. If an oligo was present in more than one UKBB library, its $\log_2[FC]$ values were averaged across libraries. If an oligo in UKBB was also found in GTEx or OL15, only the UKBB readout was collected and the others were discarded. If an oligo in GTEx (but not in UKBB) was also found in OL15, only the GTEx readout was collected and the OL15 readout was discarded. Non-natural sequences from OL15 were discarded. Moreover, oligos with a $\log_2[FC]$ of 6 s.d. below the global mean were discarded (fewer than 10 oligos). Sequences were padded on both sides with constant sequences from the reporter vector backbone to form 600 bp sequences and converted into one-hot arrays (that is, A:= [1,0,0,0], C:= [0,1,0,0], G:= [0,0,1,0], T:= [0,0,0,1], N:= [0,0,0,0]). Oligos from chromosomes 19, 21 and X were held out from the parameter training loop as a validation set guide hyperparameter tuning. Oligos from chromosomes 7 and 13 were held out from both parameter training and hyperparameter tuning loops as a test set for reporting performance. Oligos from the remaining chromosomes were used in the training loop. Oligos containing alternative alleles were assigned to the same chromosomes as the reference allele oligos. Data augmentation was performed by including into the training set the reverse complement of the (600 bp) sequences, and duplicating oligos that had a $\log_2[FC]$ greater than 0.5 in any cell type. We also aggregated the $\log_2[FC]$ output of 318,247 and 442,482 sequences tested in A549 (OL27, OL28, OL29, OL30, OL31, OL32, OL33) and HCT116 (OL41, OL42) cells, respectively, according to the same count filtering steps as described above.

**Test set performance metrics and other analyses.** For analyses outside Malinois' training loop that leverage the train/validation/test sets, we aggregated the same 798,064 unique oligos mentioned above initially filtering out only oligos with an RNA count of zero before averaging the $\log_2[FC]$ across UKBB libraries (no plasmid count filter). Oligos with a $\log_2[FC]$ standard error greater than 1 in any cell type were then omitted

from performance metrics (in the case of oligos with multiple instances across UKBB libraries, oligos of which the highest $\log_2[FC]$ standard error across libraries was greater than 1 were omitted). For locus-specific benchmarking, we aggregated the $\log_2[FC]$ of oligos that tile the GATA1 locus (OL43) according to the same count filtering steps as described above. We generated per-genome-base activity measurements by averaging the MPRA activity of each oligo that overlaps that base pair. We removed oligo genomic coordinates that overlap with those in the UKBB and GTEx libraries in scatterplots and correlation calculations.

## Model architecture

The final Malinois model is composed of three functional segments: (1) three convolutional layers with batch normalization and maximum value pooling; (2) a fully connected linear layer to integrate positional and feature information from the previous hidden state after flattening; and (3) a stack of branched linear layers such that each output feature is a function of four independent transformations. As the first two segments are replicated from the Basset architecture[4], Malinois accepts batches of 4 × 600 arrays corresponding to one-hot encoded DNA sequences, so predictions for 200-nucleotide MPRA oligos are made by padding inputs on both sides with constant sequences from the reporter vector backbone. This strict input sizing requirement ensures that hidden states are appropriately shaped when transitioning between segments 1 and 2 of the model. Furthermore, this padding strategy enables us to use reverse complement data augmentation with awareness of the orientation of the 200-nucleotide MPRA inserts with respect to the transcription start site in the reporter backbone. Although it was not tested in this study, replacing the final strict max pooling layer with adaptive pooling or padding would allow flexibility in the input sizing requirements while maintaining all other components of the architecture. At training initiation, weights were initialized using pretrained weights from a PyTorch implementation of Basset when segments 1 and 2 were appropriately configured.

## Model fitting

We trained Malinois using the Vertex AI API on the Google Cloud Platform (GCP). This enabled optimization of all tuneable parameters controlling data preprocessing, model architecture and model training. To do this, first we generated a docker container (gcr.io/sabeti-encode/boda/production:0.0.11) with an installation of CODA using a GCP VM with the following specifications: Debian-based deep learning VM for Pytorch CPU/GPU operating system, a2-highgpu-1g machine type and 1 NVIDIA Tesla A100 40G GPU. The container entrypoint was set to a Python script for model training (boda2/src/main.py). Using this container, we deployed hyperparameter tuning jobs using the default algorithm to optimize the indicated hyperparameters (Supplementary Table 9). We include a notebook for deploying a hyperparameter tuning job using the Vertex AI SDK (boda2/tutorials/vertex_sdk_launch.ipynb). We finalized model selection for Malinois by benchmarking candidates on the validation set using predictions calculated as described in the next section. All test set benchmarking was retrospective and did not impact decision making in the study. Two additional models were fitted using a subset of sequences tested in either A549 or HCT116 cells using identical hyperparameter configurations to Malinois.

## Correlation of empirical and predicted MPRA activity

When comparing Malinois' predictions to empirical MPRA, we discard any oligo with a replicate $\log_2[FC]$ standard error greater than 1 in any cell type (see section Data preprocessing above for more details). Malinois' predictions for the (padded) forward and reverse complement sequences are averaged into a single prediction.

## Optimization of cell-type specificity

The objective function to guide the sequence design with simulated annealing (minimize energy) was the MinGap (Malinois $\log_2[FC]$

prediction in the target cell type minus the maximum off-target cell type log$_2$[FC] prediction). The objective function used with the algorithms Fast SeqProp and AdaLead (minimize or maximize, respectively) was the bent-MinGap, which is defined as follows. Let $y_+$ be the Malinois log$_2$[FC] prediction on the target cell type, and $y_-$ the maximum of the log$_2$[FC] predictions on the off-target cell types of a given sequence (so MinGap $= y_+ - y_-$). We constructed a bending function $g(x) = x - e^{-x} + 1$ to preprocess predictions such that the objective function becomes bent-MinGap $= g(y_+) - g(y_-)$. We applied $g(x)$ to the predictions to incentivize greater MinGaps with low expression in the off-target cell types. For three generative algorithms, to prevent pathologically extreme activity predictions that are common in deep learning methods when computing on sequences highly divergent from the training data, we constrained predictions to a limited interval (default: [−2, 6]) when generating sequences.

## Iterative maximization of sequence function using iterative, generative and evolutionary sequence generation algorithms

**Fast SeqProp.** Fast SeqProp[5] was selected as a representative gradient-based local optimization method that exploits the structure of deep learning models to conduct greedy search while retaining the ability to pass true one-hot encoded inputs to the model. We implemented this algorithm as described in previous work, but we removed the learnable affine transformation in the instance normalization layer and drew many one-hot encoded samples from the categorical nucleotide probability distribution in each optimization step to more confidently estimate the gradients of the learnable reparameterized input sequence. The input parameters were randomly initialized (drawn from a normal distribution) and optimized using the Pytorch implementation of the Adam optimization algorithm with a learning rate of 0.5, along with a cosine annealing scheduler with a minimum learning rate of $10^{-6}$ over 300 training steps. In each training step, the loss function value was the negative average bent-MinGap of 20 sequence samples drawn from the categorical nucleotide probability distribution at that step. Once optimization is finalized, instance normalization is applied to the learned input and 20 sequences were sampled from the obtained distribution and the sequence with the highest predicted bent-MinGap was collected unless the value was less than 3.6.

**AdaLead.** AdaLead[6], another greedy search algorithm, was selected as a representative evolutionary optimization algorithm for its ease of implementation and previously reported success in DNA sequence optimization. We implemented this algorithm as written in the GitHub repository associated with the original paper. In each run, 20 randomly initialized sequences are optimized over 30 generations with mu=1, recomb_rate=0.1, threshold=0.25, rho=2, using bent-MinGap as the fitness (objective) function. Once optimization is finalized, only the sequence with the highest predicted bent-MinGap is collected unless the MinGap was less than 2. We chose to collect only one sequence per run to maximize diversity in the global batch collected from all runs.

**Simulated annealing.** Simulated annealing[7] was selected as a representative probabilistic optimization algorithm based on a decades-long history of successful application to a wide range of domains for non-convex optimization. Simulated annealing starts by jumping between regions with different local optima by occasionally accepting proposals that deteriorate the objective when the sampling temperature is high early in the algorithm. In later stages, the algorithm shifts toward greedy hill climbing as low sampling temperatures only allow proposals that improve the objective to be accepted. We implemented simulated annealing based on the Metropolis–Hastings algorithm for Markov chain Monte Carlo simulations. Proposals were generated symmetrically at each step by mutating three random bases. We used negative MinGap (without bending) to simulate the energy landscape of the theoretical system. During optimization, the temperature term was reduced using a monotonically decreasing function with a diverging infinite sum:

$$\tau = \frac{1}{1 + s^{0.501}}.$$

To produce sequences with high target-specific activity we used negative MinGap (without bending) to simulate energy of the system.

## Motif penalization

To design a batch of sequences penalizing the enrichment of given motifs in the batch, we introduced to the loss function an additional term explained below. To penalize a single motif of length $l$, we construct the motif position–weight matrix (PWM; also known as position-specific scoring matrix, or log probabilities) and use it to score all possible subsequences $x_j$ of length $l$ in the batch. Let $s_j = \mathrm{PWM}(x_j)$ be the motif score of the subsequence $x_j$, $n$ the number of sequences in the batch, and $t$ a score threshold. Then, we define the motif penalty as

$$\frac{1}{n} \sum_{j:s_j \geq t} s_j,$$

where $j$ iterates over all the possible subsequences including their reverse complements. In other words, we sum all the motif scores above the score threshold and divide by the size of the batch. When penalizing $m$ motifs, the term we introduce is very close to simply averaging the $m$ motif penalties, except that we introduce a weighting factor for each motif penalty to emphasize the penalization of motifs with lower indices (or in our case below, to prioritize motifs based on their order of inclusion to the motif pool). If we let $s_j^{(i)} = \mathrm{PWM}^{(i)}(x_j)$ be the motif score of motif $i$ of the subsequence $x_j$, and $t^{(i)}$ the score threshold of motif $i$, then the total motif penalty given a motif pool $\{\mathrm{PWM}^{(1)}, ..., \mathrm{PWM}^{(m)}\}$ is defined as

$$\frac{1}{mn} \sum_{i \in [m]} (m - i + 1)^{\frac{1}{3}} \sum_{j:s_j^{(i)} \geq t^{(i)}} s_j^{(i)},$$

where the term $(m - i + 1)^{1/3}$ is the weighting factor increasing the value of the motif penalties with lower index $i$.

We used this motif penalty expression to iteratively design sequences subject to an increasing pool of motifs. We call these iterations penalization tracks. A single penalization track starts with the generation of a batch of 500 (non-penalized) sequences, which is then analysed for motif enrichment (top 10 motifs of length 8 to 15) using STREME through a Python wrapper function. We collect the top motif PWM$^{(1)}$ from the analysis and design a second batch of 250 sequences (which we call round-1 penalized sequences) penalizing the motif pool $\{\mathrm{PWM}^{(1)}\}$. We then extract the top motif PWM$^{(2)}$ enriched in the round-1 penalized sequences and design a third batch of 250 sequences (round-2 penalized sequences) penalizing the motif pool $\{\mathrm{PWM}^{(1)}, \mathrm{PWM}^{(2)}\}$. We continue this process till we generate 250 round-5 penalized sequences penalizing the motif pool $\{\mathrm{PWM}^{(1)}, \mathrm{PWM}^{(2)}, ..., \mathrm{PWM}^{(5)}\}$.

We generated four penalization tracks for each target cell type, for all three cell types. We defined the score threshold for each motif as a percentage of the motif score of its consensus sequence. The percentages used were 0 for K562-target sequences, and 0.25 for HepG2- and SK-N-SH-target sequences. The reason behind the different choice for K562 cells is that we found that the optimization process could more easily escape the penalization of GATA by still using suboptimal instances of the motif, so a more stringent penalty was of interest for us. The motivation for using a weighting factor was that we hypothesize that sequence design optimization gravitates more strongly to motifs captured in enrichment analyses of early penalization rounds, so we wanted to keep emphasizing the penalization of motifs extracted from earlier rounds.

In Supplementary Note 2, the motif-presence score (*y* axis) of a motif in each sequence was calculated by summing all the motif-match scores that pass the Patser score threshold (as defined in Biopython[83]), and then dividing by the maximum possible motif score (the match score of the motif consensus sequence).

## *k*-mer analysis

We calculated 4-mer and 7-mer content for sequences in the CODA MPRA library as well as various other sets of reference sequences including 200-mers upstream of RefGene annotated transcription start sites, shuffled CODA sequences and random 200-mers. We calculated the average Manhattan distance to the *k*-nearest neighbours distances for 200-mers (*k* = 4) by splitting sequences into groups based on design method, target cell line and penalty level and using the NearestNeighbors module from scikit-learn (v.1.2.2). We embedded sequences in two-dimensional space based on 4-mer content using the uniform manifold approximation and projection implemented by the umap-learn (v.0.5.2) Python package.

## Homology search using Nucleotide BLAST

We conducted a homology search using NCBI ElasticBLAST to determine whether synthetic sequences had measurable homology to any sequences in the Nucleotide Collection. We used the BLASTn algorithm, the dc-megablast task and a word size of 11 and maintained the defaults for all other settings.

## Selection of naturally occurring cell-type-specific sequences by DNase and Malinois-driven GenomeScan

**DHS-natural.** To identify CREs broadly replicating across experimental approaches, using a uniformly processed dataset from ENCODE, we first selected DNase peaks from each of the three cell lines (K562, HepG2 and SK-N-SH). To further select for active CREs, we subsetted DHS peaks that intersect with H3K27ac peaks from the same cell type. For each cell type, we then identified cell-type-specific peaks by requiring a that a DHS⁺H3K27ac⁺ peak had no overlap with a DHS peak in the other two cell types. For these DHS⁻H3K27ac peaks, in each cell type, we scored the K562, HepG2 and SK-N-SH DHS signal in the peak coordinates of the target cell type. We then selected the top 4,000 peaks with the highest ratio of on-target cell type's DHS signal to the maximal off-target cell type's DHS signal, mirroring our efforts to maximize MinGap of $\log_2$-space MPRA activity with other CREs.

**Malinois-natural.** To nominate cell-type-specific natural sequences with Malinois, we tiled the whole human genome into 200 bp windows using a 50 bp stride and generated predictions for each window sequence. The cell-type specificity of each sequence was obtained by evaluating the objective function mentioned above (bent-MinGap), and the top 4,000 best-performing sequences were selected for each cell type.

## Genome annotation of natural sequences

Malinois-natural sequences capture a unique component of the genome compared with DHS-natural, with 2.7% of Malinois-natural sequences overlapping sequences in our DHS-natural set, and 65.8% residing outside any previously annotated CREs. cCRE BED files for promoter-like sequences, proximal enhancer-like sequences, distal enhancer-like sequences and CTCF-only were downloaded from the ENCODE SCREEN Portal[12] and concatenated into a single BED file for intersection with DHS-natural and Malinois-natural BED files using a custom script. Intersections were performed using bedtools (v.2.30.0)[84] and pybedtools (v.0.9.0)[85] with the following command 'Malinois/DHS-natural_BED. intersect(ENCODE_cCRE_BED, wa=True, u=True)' and the number of intersections was reported. To determine the genomic features overlapping DHS-natural and Malinois-natural sequences, the same BED files were used as an input for annotatePeaks.pl from the homer suite (v.4.11)[86] with the following command 'annotatePeaks.pl inputBED

hg38 -annStats annStats.txt > annotatePeaksOut.txt'. Annotations for the whole genome (hg38) were generated by dividing the genome into 200 bp intervals using the bedtools makewindows command 'bedtools makewindows -g hg38.txt -w 200 > hg38_200bp.bed'. Annotations were generated for each cell type (K562, HepG2, SK-N-SH) and sequence selection method (DHS-natural, Malinois-natural).

## Sampled integrated gradients to compute contribution scores of Malinois predictions

We calculated nucleotide contribution scores for each sequence in the proposed library using an adaptation of the input attribution method Integrated Gradients[58]. Sampled Integrated Gradients (SIG) considers the expected gradients along the linear path in log-probability space from the background distribution to the distribution that samples the input sequence almost surely. In each point of the linear path, a sequence probability distribution (also known as a position probability matrix (PPM)) is obtained from the log-probability space parameters by applying the SoftMax function along the nucleotide axis, and a batch of sequences is sampled from that distribution to be fed into the model. We then calculate the gradients of the batch model predictions with respect to the parameters in the log-probability space, using the straight-through estimator to backpropagate through the sampling operation. The batch gradients are averaged for each point in the path and approximate the gradient integral as in the original formulation of the method. In our case, the subtraction of the baseline input from the input of interest involves the parameters in log-probability space. This adaptation of Integrated Gradients provides two useful features. First, the sequence inputs being fed to the model are always in one-hot form, avoiding evaluations of inputs off the vertices of the simplex on which the model was trained which could more easily lead to pathological predictions. Second, the original method relies on choosing an appropriate single baseline input against which to compare the input of interest, which might not always be straightforward, whereas our adaptation uses a background distribution of sequences as the baseline. Favourably, when choosing the uniform background (0.25, 0.25, 0.25, 0.25), the parameters in log-probability space where the line path is traversed become the zero matrix, which removes the need to subtract the baseline from the input of interest. We can then more easily extract integrated gradients for all tokens in all positions (by omitting masking the gradients with the one-hot input), which we found useful as hypothetical scores for TF-MoDISco.

## Contribution block ablation

To test the value of contribution scores obtained with SIG, we conducted an in silico ablation study of the library sequences using contribution blocks (defined below) to randomize segments of the sequences. The goal of the study was to investigate the predicted $\log_2$[FC] effects of randomizing positions within the sequences corresponding to blocks of either positive or negative contribution, or random positions outside blocks. The result of the study is summarized in Supplementary Note 4. Overall, randomizing segments of the sequences associated with negative contributions resulted in an increase in the predicted activity in either the target or off-target cell type, while randomizing those associated with positive contribution completely destroyed the activity in the target cell type, and marginally decreased the (already repressed) activity in off-target cell types. To make calls of contribution blocks in any given sequence, we took the 200 contribution scores and built a smoothed contribution signal using a one-dimensional Gaussian filter (scipy.ndimage.gaussian_filter1d) with a sigma of 1.15. We defined a positive contribution block whenever the smoothed signal was above a threshold of 0.015 for 4 contiguous positions or more, and negative whenever it was below 0.015 for 4 contiguous positions or more. Outside positions were those not assigned to a contribution block. For each target cell type group (25,000 sequences), contribution block calls and ablations were performed for all three prediction

tasks. For example, taking the K562-target sequences, three different ablations and call sets were carried out: (1) block calls using contribution scores in K562 cells assessing the K562 activity effect (target cell type); (2) block calls using contribution scores in HepG2 cells assessing the HepG2 activity effect (off-target cell type); (3) and block calls using contribution scores in SK-N-SH cells assessing the SK-N-SH activity effect (off-target cell type). This resulted in a total of nine sets of calls and ablations. When assessing the effect of disrupting positions outside contribution blocks, we subsampled the outside coverage (number of positions not in blocks) to match the upper half of the distribution of coverage sizes of positive and negative contribution blocks together, whenever possible. For the SK-N-SH-target group, for example, such a distribution match was not possible as the total number of available positions from which to sample was simply not large enough globally. The same was true for the target cell type outside ablation in K562 and HepG2 cells, which might be expected as positive contribution blocks alone have large coverages. We performed this outside subsampling to have comparable ablation sizes across categories, but also because disrupting all of the positions outside blocks that have low coverage (resulting in very high outside coverages) introduces too much noise into the sequence when most of the sequence is disrupted. We set a minimum of five positions to be disrupted by outside coverages.

## Propeller plots

A propeller dot plot (Fig. 2e (top row)) is a two-dimensional plot scheme of our own device that seeks to elucidate the cross-dimensional non-uniformity of three-dimensional points. In this coordinate system, a point's radial distance from the origin corresponds to the difference between the maximum and minimum values. Its deviant angle from the axis corresponding to the maximum value quantifies the position of the median value within the range of the minimum and maximum values. Namely, the angle is proportional to the ratio between two differences: (1) the difference of the median and minimum values; and (2) the difference of the maximum and minimum values. This ratio represents the 60°-angle fraction deviating from the axis corresponding to the maximum value towards the axis corresponding to the median value. A higher angle of deviation (maximum of 60°) indicates that the median value is closer to the maximum value, while a lower angle (minimum of 0°) of deviation indicates that the median value is closer to the minimum value.

This can also be formulated in terms of the MinGap (maximum − median) and MaxGap (maximum − minimum). In our coordinate system, the MaxGap corresponds to the radial distance. The difference $(1 - MinGap/MaxGap)$ corresponds to the 60°-angle fraction deviating from the axis corresponding to the maximum value towards the axis corresponding to the median value. The MinGap:MaxGap ratio controls how much a point gravitates toward a main axis and away from the in-between-axis areas. A ratio of 0 means that the MinGap is zero and therefore the median value is equal to the maximum, so the point will be exactly between two axes. If the ratio is 1, it means that the median and the minimum values are equal, therefore the point will fall exactly in the axis corresponding to the maximum value. Note that, for this point of view to work with target and off-target cell type activities, we assume that the maximum cell type activity is the intended target cell type. This implies that, when counting sequences that pass specificity thresholds in Fig. 2e, some sequences get their target cell type reassigned to the cell type with the maximum activity, with DHS-natural sequences being the group that most benefits from the reassignment. A total of 652 sequences pass the lenient specificity threshold of MaxGap > 1 and MinGap/MaxGap > 0.5 by getting their target cell type reassigned (DHS-natural, 565; Malinois-natural, 39; AdaLead, 12; Simulated Annealing, 5; Fast SeqProp, 0; Fast SeqProp penalized, 4). However, only 16 sequences pass the stringent specificity threshold of MaxGap > 4 and MinGap/MaxGap > 0.5 by getting their target cell type reassigned (DHS-natural, 15; Malinois-natural, 0; AdaLead, 1; Simulated Annealing, 0; Fast SeqProp, 0; Fast SeqProp penalized, 0).

As an example of coordinate calculation, take the point (5, 3, 1). This point would have a radial distance of $5 - 1 = 4$ and an angle of deviation from the axis of the first dimension of $(3 - 1)/(5 - 1) \times (60°) = 30°$ (in the direction of the axis of the second dimension). In terms of the MinGap:MaxGap ratio, the angle of deviation from the axis of the first dimension (the dimension of the maximum value) towards the axis of the second dimension would be $(1 - (5 - 3)/(5 - 1))(60°) = 30°$. Observe that all the points of the form $(x + 4, x + 2, x)$, for any real value of x, will have the same coordinates as the point (5, 3, 1).

A propeller count plot (Fig. 2e (bottom row)) shows the percentage of points that fall in each given area of a propeller dot plot. The teal, yellow and red regions capture sequences in which the median value is closer to the minimum value than to the maximum value. Teal, yellow and red areas represent sequences in which the MinGap:MaxGap ratio is greater than 0.5.

The two synthetic groups in Fig. 2e were randomly subsampled to have exactly 12,000 sequences each and avoid over-plotting compared to the plots of the two natural groups. Supplementary Fig. 9 shows the complete propeller plots broken down by design method.

Oligos with a replicate $\log_2[FC]$ standard error greater than 1 in any cell type were omitted from the plots.

## Motif discovery

We used TF-MoDISco Lite[59,60] to extract sequence motifs to be predicted as functional by Malinois through contribution scores obtained through SIG. As described above, SIG naturally provides hypothetical contribution scores (as defined by TF-MoDISco) when selecting the uniform random background by simply carrying out the equivalent of the full process minus masking out using the input sequence one-hot matrix. The final contribution scores can then be retrieved masking out the hypothetical contribution using the input sequence one-hot matrices, as required by TF-MoDISco. We computed hypothetical contribution scores for each of the three prediction tasks and ran TF-MoDISco Lite with 100,000 seqlets and a window size of 200 (equivalent results were obtained using 1,000,000 seqlets). We aggregated the discovered patterns across prediction tasks following their provided example using modiscolite.aggregator.SimilarPatternsCollapser. TF-MoDISco Lite results are provided as positive and negative patterns.

## TF-MoDISco patterns to PWMs

To convert a TF-MoDISco positive pattern living in the hypothetical-contribution-score space into a PWM, we divided the pattern scores by the maximum position score sum and multiplied by 10. To obtain the PPM, we applied the SoftMax function to each position vector. Some of our TF-MoDISco negative patterns are a combination of a negative pattern (negative contributions) and a positive one (positive contributions). Thus, to convert a TF-MoDISco negative pattern into a PWM, we first reversed the sign directionality of the negative portions (as informed by the pattern scores living in contribution-score space, not hypothetical) and compensated their magnitude by multiplying by 1.2 (because our negative contribution scores are in general smaller in magnitude than positive ones perhaps due to the nature of the training data target distribution that has a positive bias). We then proceed as for the positive patterns.

## Core motifs (TF-MoDISco)

As TF-MoDISco, in addition to capturing isolated ungapped motifs, is able to capture patterns that are combinations of motifs, we heuristically extracted core ungapped patterns that, to varying degrees, account for all the of the combinations observed in the TF-MoDISco merged results. To manually define the starts and stops of core motifs, we relied on scoring the full pattern PWMs against themselves using TOMTOM[87], information content contours and visual examination. The core motif IDs are derived from the IDs of the original patterns from which they were extracted. To convert the patterns into PWMs and

PPMs, we applied the same operations as described above. Matches to human known TF-binding motifs were assigned using TOMTOM with the default parameters against the databases JASPAR CORE (2022)[61] and HOCOMOCO Human (v11 FULL)[62].

### Core motifs (STREME)
In addition to extracting sequence motifs with TF-MoDISco, we also performed a motif enrichment analysis using STREME. First, to assess the agreement between a given STREME motif and its predicted functionality as measured by contribution scores, we performed weighted-averaging of the hypothetical contribution scores corresponding to all the sequence segments that were determined to be a match to the motif (as provided by FIMO with default parameters, using motif scores as weights), and compared the score averages (one set of averages per prediction task) to the motif's information-content matrix. We will refer to the weighted average hypothetical scores as the contribution-score projection. All motifs with overall positive contribution scores that had a strong agreement with their contribution-score projection had been already captured by TF-MoDISco, suggesting that the TF-MoDISco positive pattern results are very comprehensive. However, we found a small number of STREME motifs with negative contribution scores that had a strong agreement with their contribution-score projection, so we decided to include them to the list of core motifs. Note that these motifs had negative contribution scores with moderate-to-low magnitude. We speculate that the reason TF-MoDISco might not have been able to detect them is because the contribution allocated in the seqlets that would correspond to these motifs too often falls below the threshold of the distribution of negative scores, making it hard to discriminate them from noise or insignificant scores. Running TF-MoDISco with 1 million seqlets did not change the results. We retrieved 11 such STREME motifs with strong agreement with their contribution-score projection not captured by TF-MoDISco, 9 of which were clustered together into 3 groups with nearly identical contribution-score projection (up to 1 or 2 additional positions to the left or right). This gave us a total of five STREME negative patterns in contribution-score projection form that were included to the list of core motifs. Their conversion to PWM and PPM forms followed the same process as for the TF-MoDISco patterns. Matches to human known TF-binding motifs were assigned using TOMTOM with the default parameters against the databases JASPAR CORE (2022)[61] and HOCOMOCO Human (v11 FULL)[62].

### Contribution score-based motif scan
To find instances of the core motifs present in the CODA sequence library, we leveraged the hypothetical contribution scores of the sequences to match sequence segments to the core motifs in hypothetical-contribution-score form. First, we padded with zeros left and right all the sequence hypothetical contribution scores, yielding a matrix of dimensions $3 \times 75{,}000 \times 4 \times 210$. Second, for a core motif of length $l$, we computed all the Pearson correlation coefficients between every possible subsequence hypothetical contribution scores of length $l$ (matrices of size $75{,}000 \times 4 \times l$) and the core motif's hypothetical contribution scores in forward and reverse complement orientations. For each cell type dimension, we randomly sampled 500,000 Pearson correlation coefficients (arising from a single core motif) to obtain the value $\min(0.75, \mu + 4\sigma)$ to serve as a coefficient threshold, where $\mu$ and $\sigma$ represent the mean and the s.d., respectively, of the subsampled distribution. All subsequences for which the hypothetical contribution scores scored above their coefficient threshold were collected as motif hits for the given core motif. We repeated this process for all core motifs across all cell types.

### Motifs embedded in random background
We embedded single motifs in random sequences to measure their standalone predicted effect compared to fully random sequences. For each motif, we built a 200 × 4 PPM consisting of the motif's PPM in the middle and random background ([0.25, 0.25, 0.25, 0.25]) everywhere else. We sampled 5,000 sequences from it and fed them to Malinois to obtain predictions in each cell type. We also sampled 5,000 sequences from a 200 × 4 PPM of uniform background everywhere (no motif in the middle), and fed them to Malinois to serve as baseline.

### Motif ablation
We sought to assess the predicted effect of disrupting all instances of a single motif in our sequence library. For each motif, we collected the particular batch of sequences that had at least one instance of such motif, replaced all of the instances with random segments (sampled from uniform background), and fed them to Malinois to obtain predictions in each cell type. We performed this step five times, averaged the five predictions of each disrupted sequence and subtracted from the average the batch's original predicted activities to obtain the predicted disrupting effect. For example, say that a sequence has one instance of a given motif in positions 20–32. We inserted a random sequence segment in those positions and got the disrupted sequence's predictions. We did this five times, so five different random segments (with five different predictions) in positions 20–32, and averaged the five predictions (to mildly marginalize potential effects of replacing with random segments). The disrupting effect would be this average prediction minus the sequence's original predicted activity. We aggregated the disrupting effects by motif presence (as defined above in the last paragraph of motif penalization in this section). To find instances of core motifs, we used the contribution-score-based motif scan described above. To find instances of the original TF-MoDISco patterns, we used FIMO (with the default parameters), as our contribution score-based motif scan might not handle gapped patterns as well as FIMO. When submitting the pattern PPMs to FIMO, we trimmed the patterns at both ends such that the start/stop of the pattern is the first/last position to have an information content of at least 0.15 bits.

### Motif contributions
To get a motif's overall contribution, we performed a weighted average of the contribution score sums contained in all of the motif instances provided by our motif hit method across the three prediction tasks. The average was weighted using the motif scores corresponding to the Pearson correlation coefficients mentioned above. The overall regulatory directionality of a motif (activator or repressor) is given by the sign of the mean of the weighted averages across cell types. For all motifs, the overall regulatory directionality agrees with the original TF-MoDISco designation as a positive or negative pattern.

### Motif co-occurrence
We say that a pair of motifs co-occur whenever a sequence has at least one instance of each motif. By co-occurrence percentage of a motif pair, we mean the percentage of sequences in a given group in which the motif pair co-occurs.

### NMF analysis of motif programs
We used NMF, a parts-based representation of data[74], to model semantic relationships between motifs in our sequence library (scikit-learn v.1.2.2, initialized with NNDSVDAR, Frobenius loss). First we counted motif matches in each sequence with the contribution score-based motif scan described above[88] to generate $X \in \mathbb{N}^{n \times f}$, where rows represent sequences in the library and columns correspond to motifs. The sample matrix $X$ can then be decomposed into the coefficients and features matrices $W \in \mathbb{R}^{n \times k}$ and $H \in \mathbb{R}^{k \times f}$, respectively. These $k$-dimensional representations are referred to as 'topics' in natural language processing and 'programs' in gene expression analysis[89,90]. These programs capture the frequency of TF motifs appearing in semantically similar CREs, and the CREs themselves are modelled as compositions of programs. We tested decomposing sequences into $k \in [8,28]$ programs using bi-cross-validation[91] and identified an 'elbow'

in the reconstruction error at $k = 12$ (data not shown). When plotting the coefficient matrix comparative analysis, we normalize the coefficient matrix such that the rows sum to 1. We quantified the function of each decomposed program by calculating a weighted average of motif contributions (see the 'Motif contributions' section above) for each program using the motif weights in the features matrix. Motif contributions were clipped to an upper bound of 3 to mitigate the impact of extreme outliers.

## MPRA saturation mutagenesis plot

The saturation mutagenesis study (Supplementary Table 10) of the sequence in Fig. 4g consisted in empirically testing the activity of all the possible 600 variants of the sequence (3 variants per position, 200 positions). We followed an identical protocol to the previous MPRAs in SK-N-SH cells with this saturation mutagenesis library. We visualized the effect of each variant as the subtraction of the activity of the original sequence from each variant-sequence's activity, resulting in the lollipops in Fig. 4h. The mean variant effect is represented in the height of the logo sequence letters but in the opposite direction.

## CODA MPRA

**MPRA library construction.** The CODA MPRA library was constructed according to previously described protocols[8]. In brief, oligos were synthesized (Twist Bioscience) as 230 bp sequences containing 200 bp of genomic sequences and 15 bp of adaptor sequence on either end. The oligo library was PCR amplified with primers MPRA_v3_F and MPRA_v3_20I_R to add unique 20 bp barcodes along with arms for Gibson assembly into a backbone vector. The oligonucleotide library was assembled into pMPRAv3:Δluc:ΔxbaI (Addgene plasmid, 109035) and expanded by electroporation into *Escherichia coli*. Seven of the ten expanded cultures were purified using Qiagen Plasmid Plus Midi Kit to reach 200–300 colony-forming units (barcodes) per oligonucleotide. The expanded plasmid library was sequenced on the Illumina NovaSeq system using 2 × 150 bp chemistry to acquire oligo–barcode pairings. The library underwent AsiSI restriction digestion, and GFP with a minimal promoter amplified from pMPRAv3:minP-GFP (Addgene plasmid, 109036) using primers MPRA_v3_GFP_Fusion_F and MPRA_v3_GFP_Fusion_R was inserted by Gibson assembly resulting into the 200 bp oligo sequence positioned directly upstream of the promoter and the 20 bp barcode falling in the 3′ UTR of GFP. Finally, the library was expanded within *E. coli* and purified using the Qiagen Plasmid Plus Giga Kit.

**MPRA library transfection into cells.** All cell culture and transfection conditions followed previously established protocols[27]. For each of the three cell types, K562, SK-N-SH and HepG2, we collected two hundred million cells for transfections using the Neon Transfection System 100 μl Kit with 5 μg or 10 μg of the MPRA library per 10 million cells. Cells were collected 24 h after transfection, rinsed with PBS and collected by centrifugation. After adding RLT buffer (RNeasy Maxi kit), dithiothreitol and homogenization, cell pellets were frozen at −80 °C until further processing. For each cell type, three biological replicates were performed on different days. All cell lines were acquired from ATCC, authenticated using genotyping and gene expression signatures, and routinely tested for *Mycoplasma* and other common contaminants by The Jackson Laboratory's Molecular Diagnostic Laboratory.

**RNA isolation and MPRA RNA library generation.** RNA was extracted from frozen cell homogenates using the Qiagen RNeasy Maxi kit. After DNase treatment, a mixture of three GFP-specific biotinylated primers was used to capture *GFP* transcripts using Sera Mag Beads (Thermo Fisher Scientific). After a second round of DNase treatment, cDNA was synthesized using SuperScript III (Life Technologies) and the *GFP* mRNA abundance was quantified using quantitative PCR (qPCR) to determine the cycle at which linear amplification begins for each replicate.

Replicates were diluted to approximately the same concentration based on the qPCR results, and a first round of PCR (8 or 9 cycles) with primers MPRA_Illumina_GFP_F_v2 and Ilmn_P5_1stPCR_v2 was used to amplify barcodes associated with *GFP* mRNA sequences for each replicate. A second round of PCR (6 cycles) was used to add Illumina sequencing adaptors to the replicates. The resulting Illumina indexed MPRA barcode libraries were sequenced on the Illumina NovaSeq system using 1 × 20 bp chemistry.

## CRE prioritization for in vivo validation

**Enformer analysis of epigenetic signatures.** To simulate epigenetic and gene expression signatures in silico we collected the nucleotide sequence from chromosome 11: 3101137–3493091 of the mouse reference genome (mm10). The expected insertion sequence using an H11 targeting vector with a lacZ:P2A:GFP open reading frame was added. As a control, the expected CRE insertion site was simulated as a 200 nucleotide sequence of N. We simulated all possible CRE insertions corresponding to our cell-type-specific MPRA by replacing the oligo-N sequence with 200-mers from our library. We inferred epigenetic signatures for all of these sequences using Enformer by modifying the notebook available online (https://colab.research.google.com/github/deepmind/deepmind_research/blob/master/enformer/enformer-usage.ipynb). To estimate CRE-induced transcriptional activation in various tissues, we collected 128 nucleotide resolution DHS, H3K27ac, ATAC and CAGE datasets overlapping the expected insertion (35 bins). To calculate an aggregate effect for each tissue, we calculated the maximum signal for each feature over the insertion, followed by a feature-specific Yeo–Johnson power transformation. Normalized features were then selected based on tissue correspondence (Supplementary Table 8) and averaged to estimate CRE activity in ten different tissues. We calculated MinGap values for the spleen, liver and brain using these ten measurements for each CRE.

**Manual sequence prioritization.** Sequences were prioritized on the basis of review of empirical MPRA measurements, contribution scores, motif matches, sequence content and predicted epigenetic signatures. We looked for sequences that displayed a high separation between the MPRA measures of the target and the off-target cell types. We also looked to capture variations of combinations of motif matches, and we used the contribution scores to visually examine the motif matches and other potentially important sequence content and motif organization. Finally, we selected sequences with at least moderate tissue specificity in predicted epigenetic signatures.

## Transgenics

**Transient zebrafish synthetic enhancer assay.** To build the synthetic CRE eGFP reporter, double-stranded oligonucleotides corresponding to synthetic CREs (200 bp) were synthesized by IDT (GeneBlock). Synthetic CREs were amplified by PCR with primers that included homology to the plasmid vector E1b-GFP-Tol2 (Addgene plasmid, 37845)[75] and were cloned upstream of the minimal promoter (E1b) to generate the synthetic enhancer eGFP plasmid reporter (pTol2-synthetic CRE-E1b-eGFP-Tol2) using HiFi DNA Assembly according to the manufacturer's instructions (New England Biolabs). We also created 'empty vectors', which were identical to CODA CRE vectors except for the lack of a 200 bp insert. Reporter plasmid sequences were verified by Sanger sequencing. To transiently express the synthetic CRE reporter in zebrafish, plasmids were co-injected with tol2 transposase mRNA into one-cell-stage zebrafish embryos according to established methods[92]. A minimum of 15, one-cell zebrafish embryos of either sex were injected per construct. Injected embryos were imaged at the indicated days (2 or 4 days after fertilization) either using the dissecting (Olympus) or confocal fluorescence (Leica SP8) microscope. Injected embryos were not randomized, and researchers were not blinded. All zebrafish procedures were

approved by the Yale University Institutional Animal Care and Use Committee (2022-20274).

**Mouse transgenic reporter assay.** An H11 targeting vector with an lacZ:P2A:GFP open reading frame was linearized using PCR containing 2 ng of template, 1 µl of KOD Xtreme Hot Start DNA Polymerase (Sigma-Aldrich, 71975), 25 µl of Xtreme buffer and 0.5 µM forward and reverse primers (H11_bxb_lacZ:GFP_lin_F, pGL_minP_GFP_R; Supplementary Table 11) cycled with the following conditions: 94 °C for 2 min; 20 cycles of 98 °C for 10 s, 56 °C for 30 s and 68 °C for 13 min; and then 68 °C for 5 min. Amplified fragments were treated with 0.5 µl of DpnI (NEB, R0176S) for 30 min at 37 °C, purified using 1× volume of AMPure XP (Beckman Coulter, A63881) and eluted with water. Double-stranded oligonucleotides corresponding to synthetic enhancers with Gibson arms were synthesized by IDT (GeneBlock) and assembled into the targeting vector using 5 µl of NEBuilder HiFi DNA Assembly Master Mix (NEB, E2621S), 36 ng of linearized vector and 10 ng of the synthesized fragment in a total volume of 20 µl for 45 min at 50 °C. Transgenic mice were created according to the enSERT protocol[76]. A mixture of 20 ng µl$^{-1}$ Cas9 protein (IDT, 1074181), 50 ng µl$^{-1}$ single guide RNA (sgRNA_H11lacZ; Supplementary Table 11), 25 ng µl$^{-1}$ donor plasmid, 10 mM Tris, pH 7.5, and 0.1 mM EDTA was injected into pronuclear of FBV zygotes. Each group was tested with a predetermined sample size of 3 l and all of the samples were stained regardless of their genotype and sex. Embryos were collected and stained blindly with respect to their genotype. The whole embryo at embryonic day 14.5 or isolated brain at 5 weeks postnatal were fixed at 4 °C for 1 h in PBS supplemented with 2% paraformaldehyde, 0.2% glutaraldehyde and 0.2% IGEPAL CA-630. After washing with PBS, the embryos were stained at 37 °C overnight in a solution in PBS supplemented with 0.5 mg ml$^{-1}$ X-gal (Sigma-Aldrich, B4252), 5 mM potassium hexacyanoferrate(II) trihydrate, 5 mM potassium hexacyanoferrate(III), 2 mM MgCl$_2$ and 0.2% IGEPAL CA-630. The images were taken using the Leica M165 system for embryos or the Leica M125 system for brains. All mice were housed in duplexed pens containing five or less mice and under a 12 h–12 h light–dark cycle at 18–23 °C with 40–60% humidity. All mouse procedures were performed in accordance with the National Institutes of Health Guide for the Care and Use of Laboratory Animals, and were approved by the Institutional Animal Care and Use Committees of The Jackson Laboratory (18038).

**Histology and immunofluorescence staining.** After LacZ staining, mouse brains were sectioned with a vibratome (Leica VT100s) and free-floating 70-µm-thick sagittal sections were collected in ice-cold PBS. The sections were then rinsed in 1× PBS for 5 min and incubated for 30 min in a blocking solution consisting of 0.3% Triton X-100, 0.3% mouse on mouse blocking reagent (Vector laboratories, MKB-2213-1), 10% normal goat serum (Abcam, ab7481) and 5% BSA in 1× PBS with gentle agitation at room temperature. Immunostaining was then performed with a mixture of primary antibodies in the blocking solution at 4 °C on a shaker overnight. The sections were rinsed in 1× PBS three times for 5 min each and then incubated with corresponding fluorescence conjugated secondary antibodies for 2 h. After treatment with secondary antibodies, the slices were then further rinsed with PBS three times, followed by staining for nuclei with DAPI (Thermo Fisher Scientific, 62248). The sections were mounted onto slides with Prolong Gold antifade reagent (Cell Signalling Technology, 9071). The following primary antibodies were used during the staining procedure: mouse anti-NeuN (Abcam, ab104224), chicken anti-GFAP (OriGene Technologies, TA309150), rabbit anti-IBA1 (Abcam, ab178846). Secondary antibodies used were as follows: goat anti-mouse Alexa Fluor 488 (Thermo Fisher Scientific, AB_2534069), goat anti-chicken Alexa Fluor 568 (Thermo Fisher Scientific, AB_2534098), goat anti-rabbit Alexa Fluor 568 (Abcam, ab175471). All primary and secondary antibodies were used at 1:500 dilutions. Image acquisition for whole-brain

sagittal slice mosaic images was performed using the Thunder Imager (Leica Microsystems) system using a ×10/0.8 NA dry lens. Fluorescence imaging was combined with bright-field imaging to visualize LacZ staining. Computational tissue clearing was applied systematically to reduce background noise (Leica acquisition software). After obtaining mosaic scans, higher-magnification images of regions of interest (ROIs) were acquired on the Stellaris 8 (Leica Microsystems) equipped with a Diode, Ar gas and He/Ne adjustable wavelength lasers using ×40/1.2 NA and ×63/1.4 NA oil objectives for quantification and representative images, respectively. The pinhole size was set to 1 a.u. and the samples were illuminated with 405, 488, 561 and 633 nm lasers sequentially. Six-micrometre $z$-stack images with a 2 µm $z$-step size and with a 4,096 × 4,096 pixel resolution were acquired using HyD detectors with a line average of 3. Fluorescent LacZ staining was visualized using the confocal microscope using the 633 nm laser[93]. For the representative images shown, bright outliers were removed using the default 2-pixel radius and 20 threshold. A Gaussian blur was then applied with a sigma radius of 1.

**LacZ layer intensity analysis.** Acquired mosaic bright-field images underwent auto-thresholding using the default algorithm in the FIJI software (NIH). Quantification of LacZ signal intensity was achieved using the plot profile tool with ROIs drawn from superficial cortical layers down to the corpus callosum. Depth information for cortical layers was acquired from the Allen Brain atlas. Multiple ROIs were taken in different cortical areas to verify the distribution of the signal. Representative images are ROIs taken from the somatosensory and visual cortices. For cell quantification and overlap analysis, to quantify cell populations, using FIJI software, maximum-intensity projection of the $z$-stack of images acquired with a confocal microscope was performed, and background removal was applied with rolling ball radius of 50. The images were then processed for autothresholding using the Moments algorithm. The signal to noise ratio was uniform across ROIs and a single thresholding algorithm yielded reproducible results. Cells were then quantified using the Analyse particle function. By varying the particle size, accurate quantification of neurons, astrocytes and microglia was achieved. To calculate the overlap between LacZ expression and the cell-type-specific markers, each binarized LacZ image was multiplied with corresponding binarized neuronal, astrocytic and microglia ROIs and the residual signals were quantified using the Analyse particle function. In total, five sagittal slices were analysed per mouse and a total of $n$ = 3 mice was used for both controls and LacZ-positive brains.

**RNA-seq analysis.** Three replicates each from transgenic mice of CODA-designed SK-N-SH-specific CRE and empty vector were collected at 5 weeks postnatally. The liver, spleen and the right half of the brain were soaked into RNA later (Thermo Fisher Scientific) overnight at 4 °C and homogenized in QIAzol, followed by total RNA isolation using the RNeasy mini kit (Qiagen) with on-column DNase treatment. The RNA-seq library was generated from 1 µg of total RNA using the NEBNext Ultra II RNA Library Prep Kit for Illumina (NEB) and NEBNext Poly(A) mRNA Magnetic Isolation Module (NEB) according to the manufacturer's protocol. The libraries were indexed using i7 and i5 primers with the following conditions: 98 °C for 30 s; 10 cycles of 98 °C for 10 s, 65 °C for 75 s; then 65 °C for 5 min. Indexed samples were purified using 0.9× volume of AMpure XP, eluted in 20 µl of EB, pooled equimolarly and sequenced using 2 × 150 bp chemistry on the Illumina NovaSeq X+ instrument at the Jackson Laboratory. The sequencing reads were mapped onto a modified mouse genome (GRCm38/mm10) with the *lacZ-GFP* sequence as an additional chromosome using STAR[94] (v.2.5.2b). After removing duplicates using picard MarkDuplicates (MIT, v.3.1.1), the mapped reads were counted using featureCount (v.2.0.6, options: -p -B -Q 20 -T 16 -s 2 --countReadPairs). DESeq2 (v.1.32.0)[95] was used to normalize the read counts and calculate the log$_2$[FC], standard error and Wald-test $P$ values.

## Reporting summary

Further information on research design is available in the Nature Portfolio Reporting Summary linked to this article.

## Data availability

Reference datasets used in this study are linked and annotated in Supplementary Table 1. GRCh38/hg38 and GRCm38/mm10 were used as reference genomes in this study. Processed MPRA data used to train Malinois are available in Supplementary Table 2. Processed MPRA data and Malinois predictions for the cell-type-specific CRE library designed for this study are available in Supplementary Table 12. RNA-seq reads are available at the NCBI Gene Expression Omnibus (PRJNA1075667). Raw data, processing notebooks, model weights and immunofluorescence images are available at Zenodo[96] (https://zenodo.org/records/10698014).

## Code availability

CODA is available at GitHub (https://github.com/sjgosai/boda2).

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

**Acknowledgements** We thank the staff at The Jackson Laboratory Genome Technologies, Genetic Engineering Technologies and Microscopy Core, and the Yale Zebrafish Research Core for experimental support; K. Peterson for reagents and technical advice; T. Helenius, J. Xue, M. Stitzel, S. Rong, D. Kotliar, T. Sorrells, H. Dewey, N. Nerurkar and M. Noon for suggestions and conversations about the manuscript; and J. Schreiber and A. Kundaje for sharing an optimized version of TF-MoDISco called tf-modisco-lite (https://github.com/jmschrei/tfmodisco-lite). This work was supported by Howard Hughes Medical Institute and by US National Institutes of Health grants UM1HG009435, R00HG010669, R01HG012872 and R35HG011329.

**Author contributions** S.J.G. initiated the model-guided sequence design framework. S.J.G., R.I.C., S.K.R. and R.T. developed the full study and designed experiments. S.J.G. and R.I.C. developed the CODA software library, Malinois model and produced the synthetic sequences. N.F., S.K., T.T.L.N., and S.K.R. conducted in vitro experiments. M.A., M.T.J., R.R.N. and K.M. conducted in vivo experiments. S.J.G., R.I.C., N.F., J.C.B., M.A., A.S.R., M.T.J., S.K.R. and R.T. performed data analysis. S.J.G., R.I.C., J.C.B., P.C.S., S.K.R. and R.T. interpreted results and drafted the manuscript. P.C.S., S.K.R. and R.T. secured funding and supervised the study. All of the authors revised the manuscript and accepted its final version.

**Competing interests** P.C.S. is a co-founder of and consultant to Sherlock Biosciences and Board Member of Danaher Corporation. P.C.S. and R.T. have filed intellectual property related to MPRA. S.J.G., R.I.C., S.K.R., P.C.S. and R.T. have filed a provisional patent application related to work described here. The other authors declare no competing interests.

**Additional information**
**Correspondence and requests for materials** should be addressed to Sager J. Gosai, Rodrigo I. Castro, Steven K. Reilly or Ryan Tewhey.

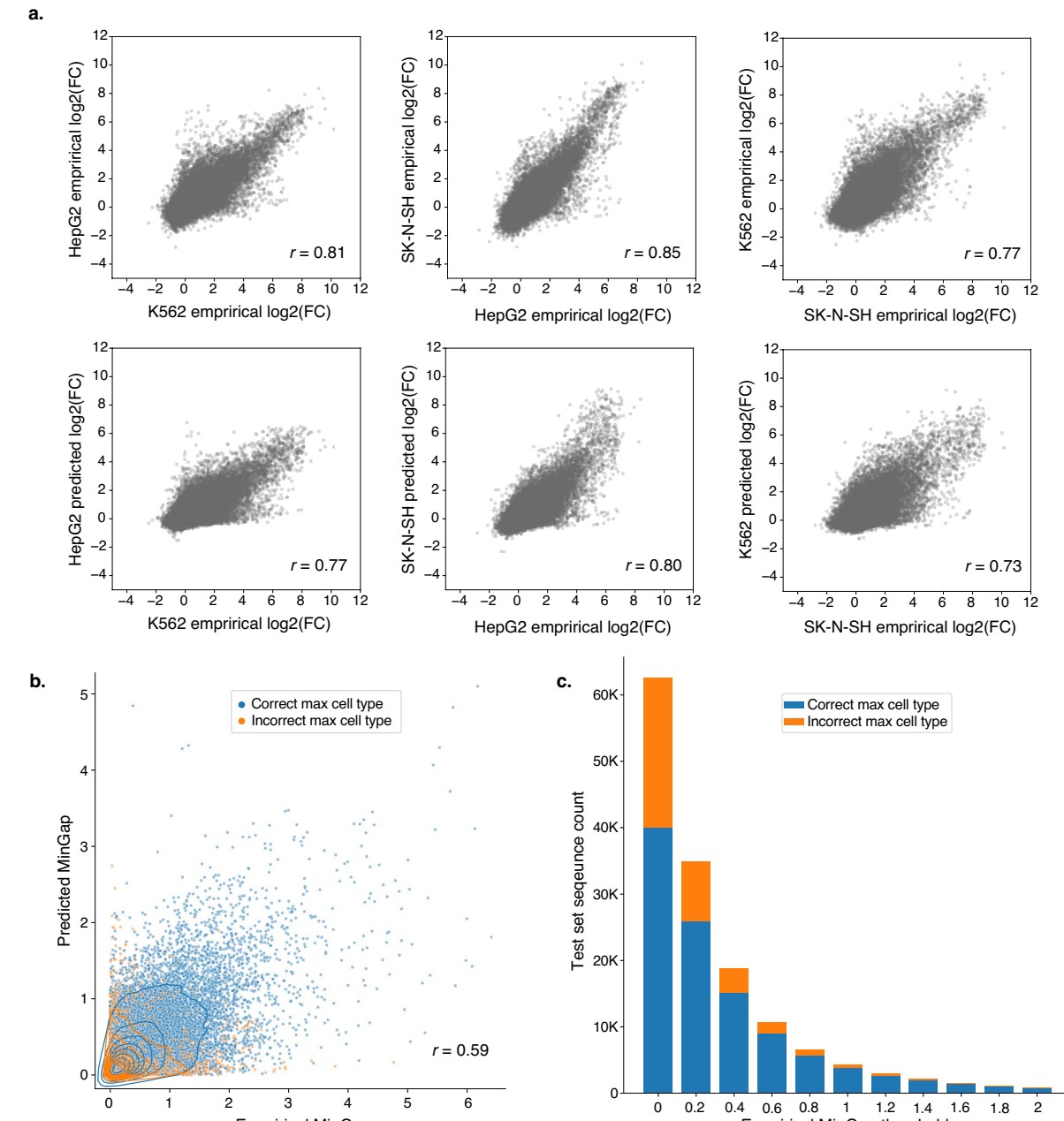

**Extended Data Fig. 1 | Cell type accuracy of model.** (**a**) Cross cell-type activity comparisons between empirical measurements and Malinois predictions organize and correlate similarly to empirical-to-empirical comparisons. Top scatter plots: empirical vs empirical cross-cell-type log2(FC). Bottom scatter plots: empirical vs predicted cross-cell-type log2(FC). Number of sequences n = 62,582. Pearson correlation coefficients are shown in the left-bottom corner of each scatter plot. All *p*-values < 1e-300. (**b**) Malinois can be used to identify highly active cell type-specific CREs. MinGap scores calculated using Malinois predictions correlate well with MPRA MinGap measurements for

sequences in the held-out test set. Points are coloured based on correct prediction of maximally active cell type by Malinois. (**c**) Malinois predictions of cell type associated with maximum CRE function are more accurate for sequences with high empirical specificity. Stacked bar plot displaying number of sequences in the test set falling into discrete bins based on an empirically measured MinGap threshold. Lower boundary of each bin is indicated on the *x*-axis and hue delineates sequences that are categorized correctly (blue) or incorrectly (orange). Number of sequences n = 62,582, *p*-value < 1e-300.

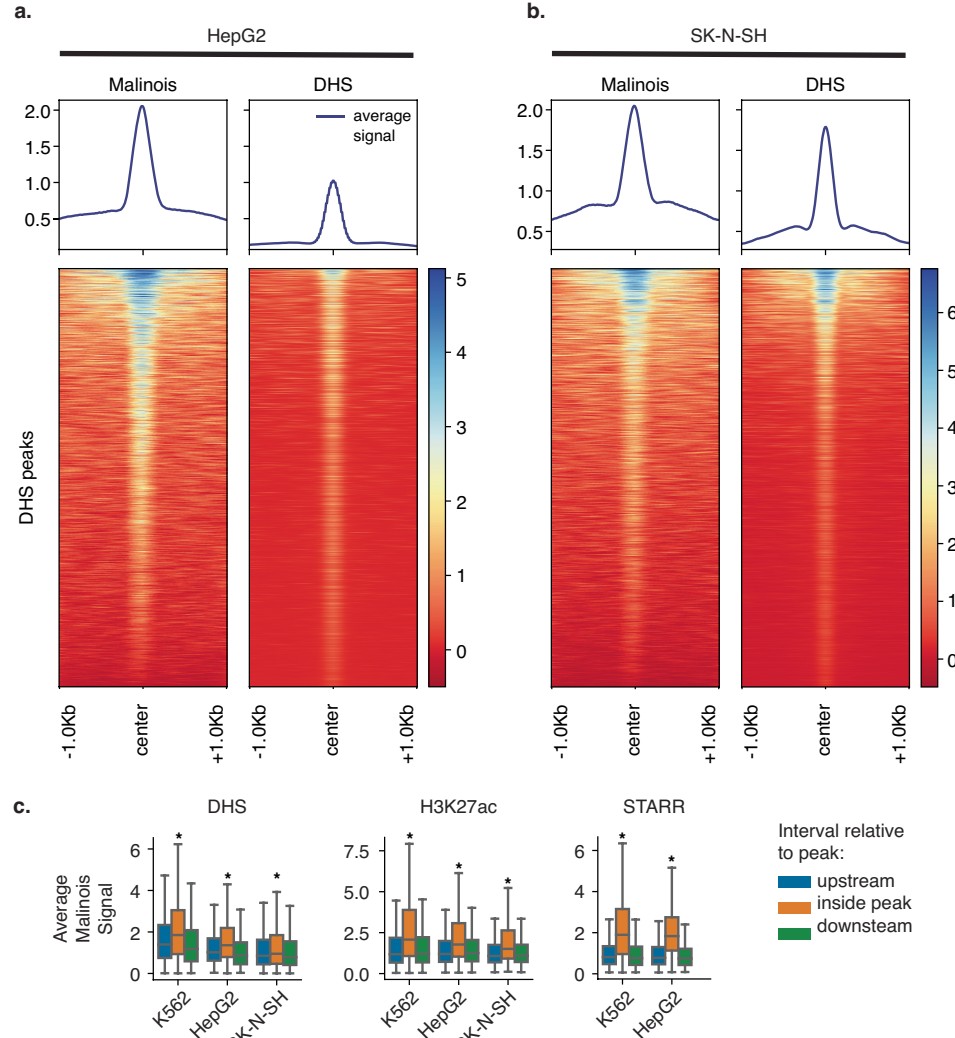

**Extended Data Fig. 2 | Malinois concordance with DHS/H3K27ac/STARR.**
(**a**) Malinois genome-wide predictions correspond well with DHS signal in
HepG2. Deeptools plots of Malinois genome-wide predictions and DHS signal
centred at DHS peaks in HepG2 cell lines on chromosome 13 (n = 1,188 peaks).
(**b**) DHS signal and Malinois genome-wide predictions are also similar in SK-N-SH.
Similar Deeptools plots to **a** except using SK-N-SH derived data (n = 3,512 peaks).
(**c**) Malinois genome-wide predictions are significantly associated with
candidate CRE mapping (DHS-seq, and H3K27ac ChIP-seq) and orthogonal
signals of CRE functional characterization (STARR-seq). Boxplots display

average signal generated by Malinois genome-wide predictions within peaks
on chromosome 13 annotated using DHS, H3K27ac, or STARR-seq (orange)
compared to paired upstream (blue) and downstream (green) flanking regions.
Boxes demarcate the 25th, 50th, and 75th percentile values, while whiskers
indicate the outermost point within 1.5 times the interquartile range from the
edges of the boxes. Stars indicate a significant (p-value < $10^{-100}$) for two $t$-tests
comparing signals within peaks and both upstream and downstream regions
outside of peaks (from left-to-right, comparisons made using n = 2,413; 1,188;
3,512; 836; 1,119; 1,993; 1,157; and 1,670 peaks).

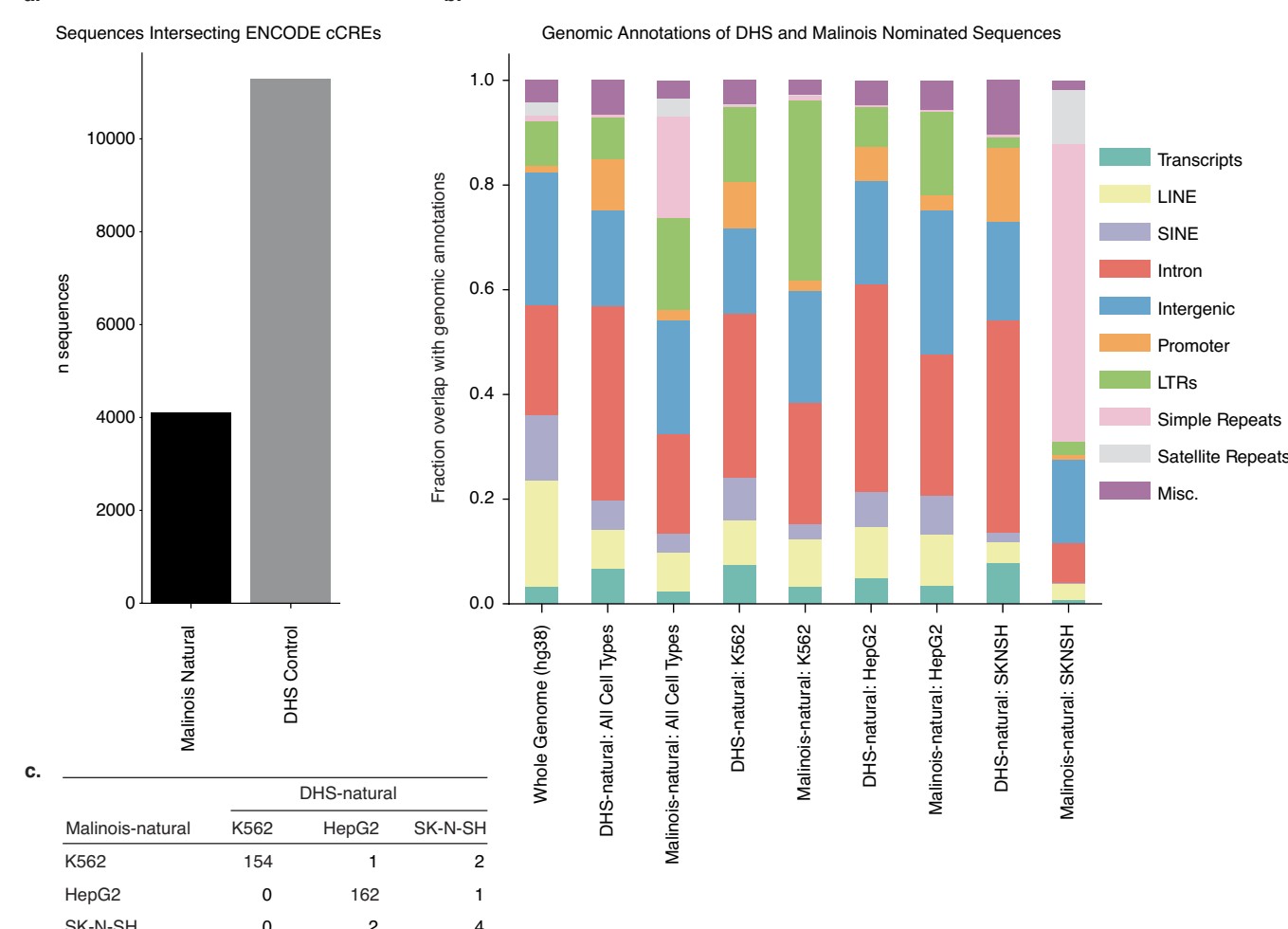

**a.** Sequences Intersecting ENCODE cCREs

**b.** Genomic Annotations of DHS and Malinois Nominated Sequences

**c.**

| Malinois-natural | DHS-natural | | |
|---|---|---|---|
| | K562 | HepG2 | SK-N-SH |
| K562 | 154 | 1 | 2 |
| HepG2 | 0 | 162 | 1 |
| SK-N-SH | 0 | 2 | 4 |

**Extended Data Fig. 3 | Annotation of naturally occurring sequences.**
(**a**) Sequences nominated by DHS accessibility (DHS-natural) and by Malinois (Malinois-natural) were intersected with ENCODE cCREs (promoter-like sequences, proximal enhancer-like sequences, distal enhancer-like sequences, and CTCF-only) to determine overlap with existing putative regulatory elements. 94% of DHS-natural sequences intersect a cCRE while only 34.2% of Malinois-natural sequences intersect a cCRE suggesting that Malinois may exploit sequences features not captured by typical cCRE measures to select a sequence that drives cell type-specific activity. (**b**) To explore additional genomic features that may overlap DHS-natural and Malinois-natural sequences were annotated using annotatePeaks.pl from the HOMER suite. Annotations were generated for the whole genome (hg38), the DHS-natural and Malinois-natural libraries as a whole, as well as DHS-natural and Malinois-natural by individual cell type. DHS-natural and Malinois-natural largely resemble the distribution of annotations genome-wide barring an overrepresentation of simple repeats in Malinois-natural sequences driven by SK-N-SH sequences. Despite this, selected sequences seem to be a representative sample of genomic features. (**c**) DHS-natural and Malinois-natural sequences were intersected to determine overlap between naturally occurring sequences. Notably overlap was minimal between selection methods (0.10%-4.1%) depending on cell type.

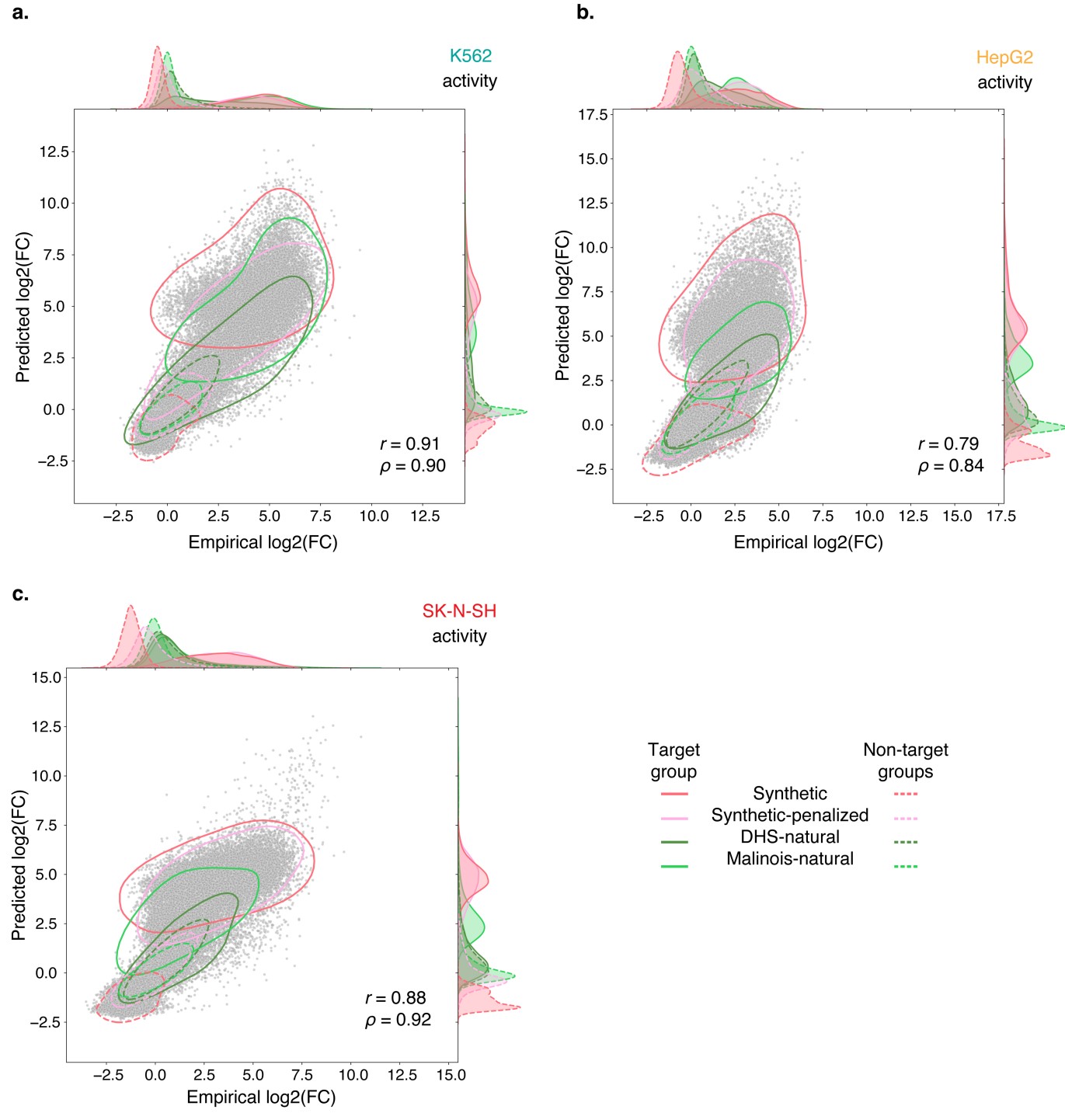

**Extended Data Fig. 4 | Library prediction validation plots. (a)** Prospective Malinois predictions of candidate cell type-specific CRE activity is correlated with experimental measurements across all three tested cell types. The scatter plot corresponds to predictions and measurements made in K562. Solid contour lines demarcate 95% density of points corresponding to candidate CRE expected to drive expression in K562. Dotted contour lines indicate 95% density of CREs expected to drive specific expression in one of the other two cell types. Colour indicates sequence selection or generation method. One-dimensional density estimates along axes share the same line style and colour associations. Sequences with a replicate $\log_2$FC standard error greater than 1 in any cell type were omitted from the plots. Number of sequences n = 69,550; $p$-values < 1e-300. **(b)** Same as **a**, but in HepG2. Number of sequences n = 69,550; $p$-values < 1e−300. **(c)** Same as **a**, but in SK-N-SH. Number of sequences n = 69,550; $p$-values < 1e-300.

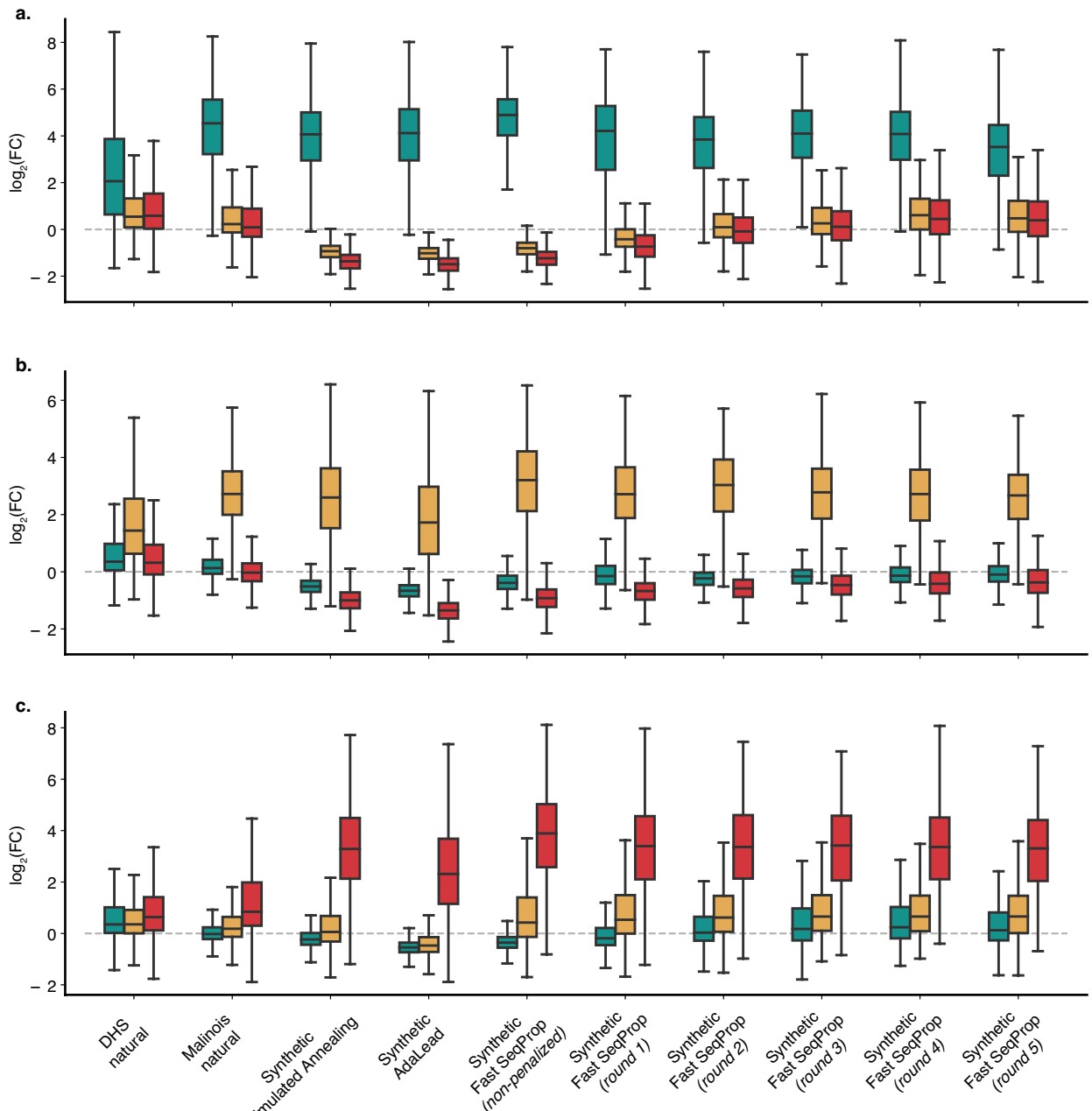

**Extended Data Fig. 5 | Empirical library activity.** (**a**) Empirical log2(Fold-Change) activity measured in K562 (teal), HepG2 (gold), and SK-N-SH (red) for sequences targeting K562 binned by design method group. Boxes demarcate the 25th, 50th, and 75th percentile values, while whiskers indicate the outermost point within 1.5 times the interquartile range from the edges of the boxes.

Number of sequences left-to-right n = 3,729; 3,410; 3,584; 3,545; 3,738; 955; 958; 967; 958; 962. (**b**) Same as (a) except sequences targeting HepG2. Number of sequences left to right n = 3,757; 3,727; 3,703; 3,531; 3,683; 917; 938; 961; 953; 966. (**c**) Same as (a) except sequences targeting SK-N-SH. Number of sequences left to right n = 3,261; 3,804; 3,894; 3,868; 3,915; 978; 968; 976; 972; 972.

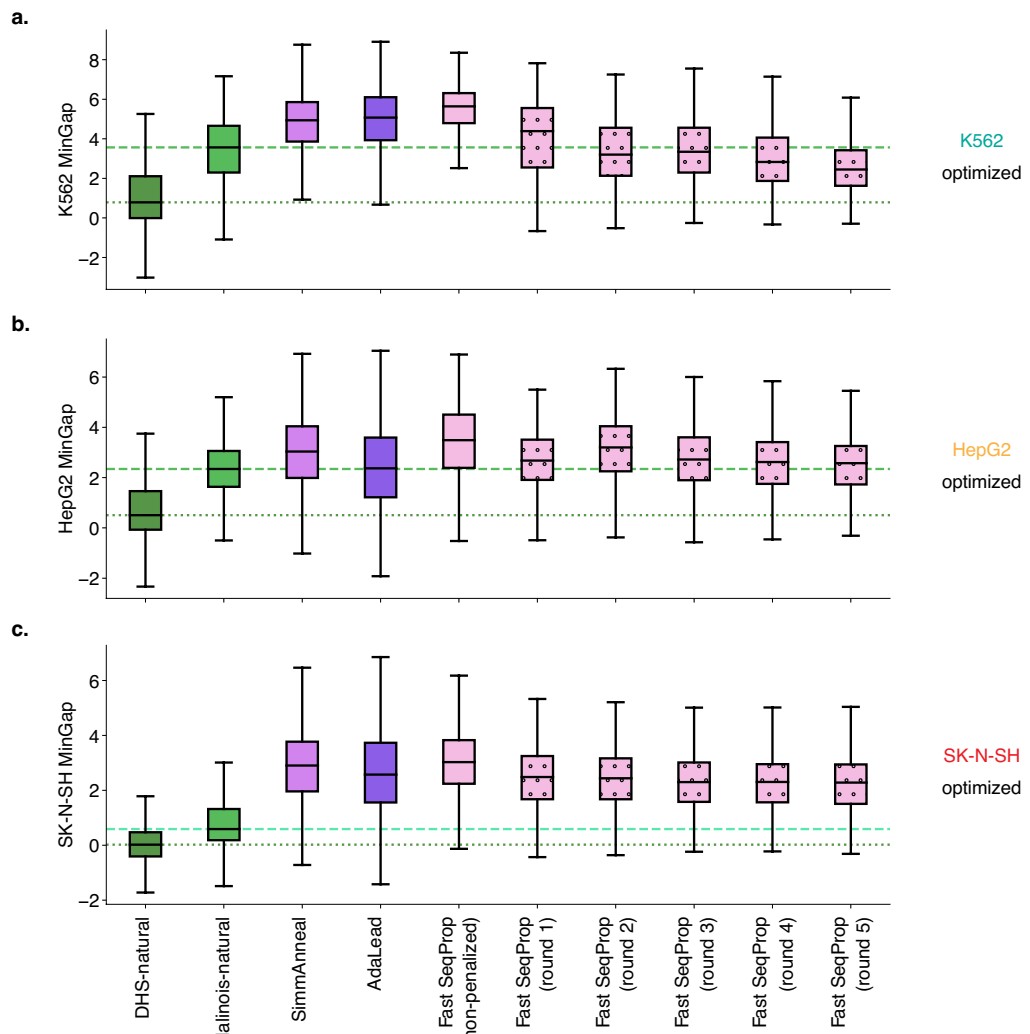

**Extended Data Fig. 6 | Library MinGap. (a)** Malinois improves identification of CREs with K562-specific activity and synthetic sequence generation enables creation of CREs with enhanced functions. Distribution of MPRA-measured K562-specific activity in various candidate CRE groups. Green and aquamarine lines indicate median MinGap of DHS-natural and Malinois-natural candidates respectively. Sequences with a replicate $\log_2$FC standard error greater than 1 in any cell type were omitted from the plots. Boxes demarcate the 25th, 50th, and 75th percentile values, while whiskers indicate the outermost point within 1.5 times the interquartile range from the edges of the boxes. Number of sequences left to right n = 3,729; 3,410; 3,584; 3,545; 3,738; 955; 958; 967; 958; 962. **(b)** Same as (a) except quantification of candidate sequences targeting HepG2. Number of sequences left to right n = 3,757; 3,727; 3,703; 3,531; 3,683; 917; 938; 961; 953; 966. **(c)** Same as (a) except quantification of candidate sequences targeting SK-N-SH. Number of sequences left to right n = 3,261; 3,804; 3,894; 3,868; 3,915; 978; 968; 976; 972; 972.

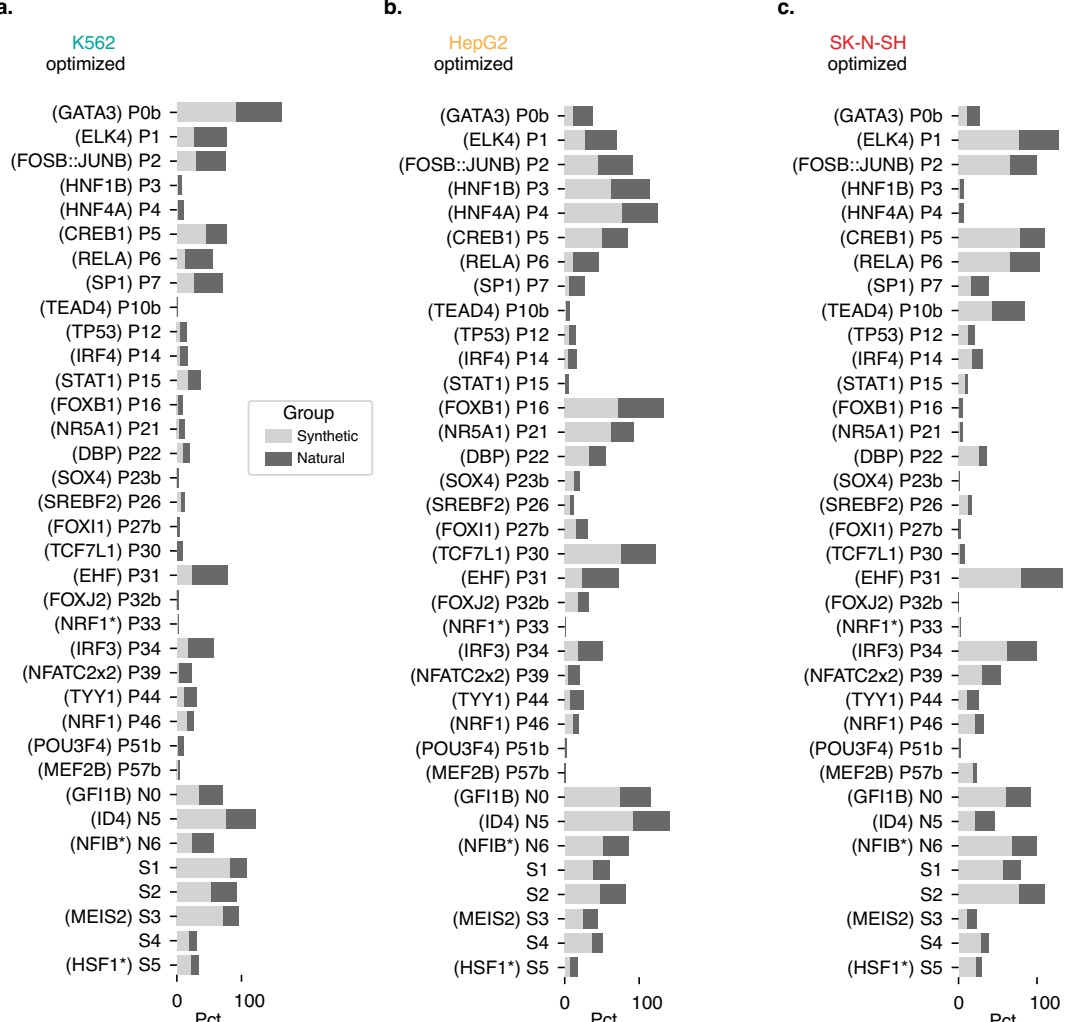

**Extended Data Fig. 7 | Motif enrichment by cell type target. (a)** Motif representation in K562-optimized sequences only. Bar width indicates the fraction of natural (dark grey) or synthetic (light grey) K562-optimized sequences containing the motif. (**b**) Same as (a) but in HepG2-optimized. (**c**) Same as (a) but in SK-N-SH-optimized.

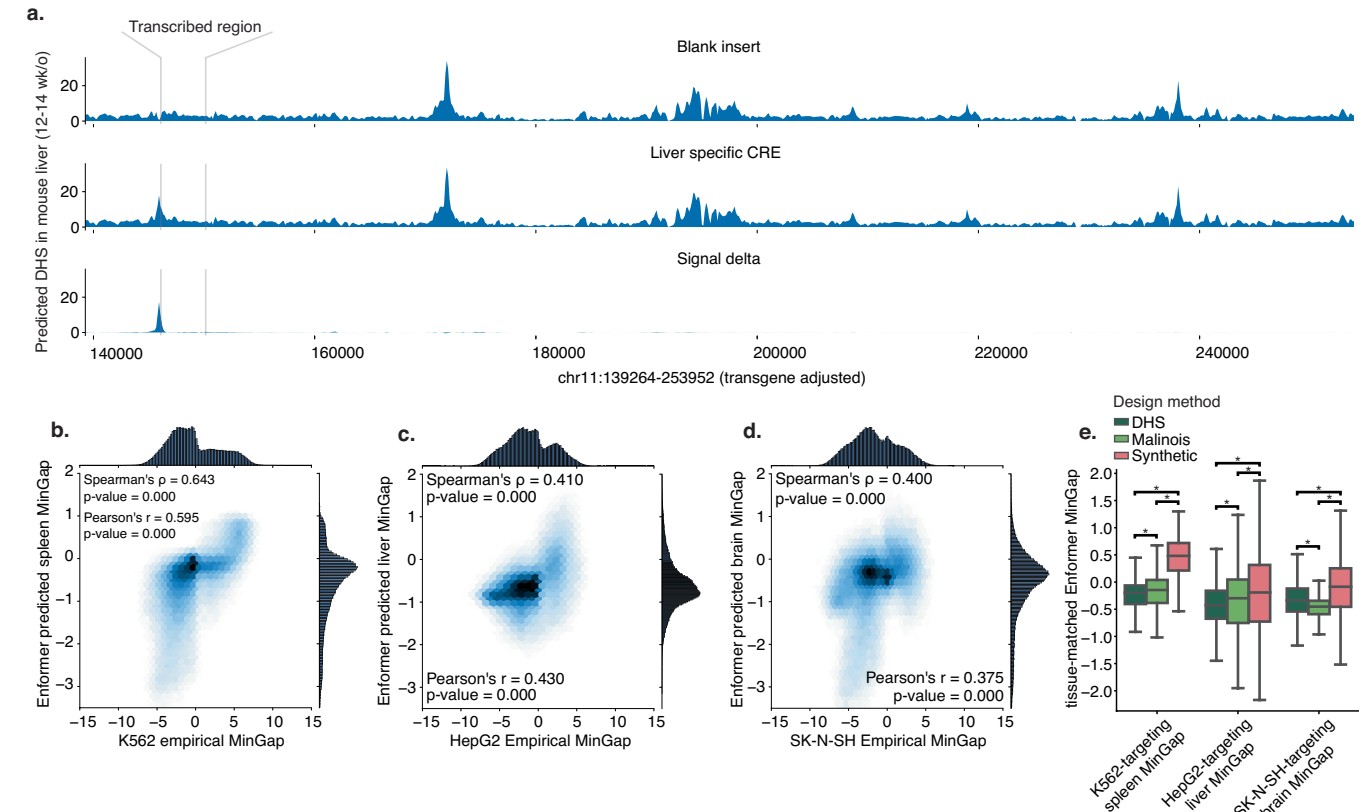

**Extended Data Fig. 8 | Enformer based prioritization of oligos for in vivo tests.** (**a**) Enformer can predict CRE-driven changes in epigenetic and transcription dynamics of transgenes inserted into the H11 safe harbour locus in mice. Three example sequence tracks display predicted DHS signals observed in the livers of 15.5 day old mice. Transgene transcription start site and poly-adenylation signal are indicated by the grey bars. The first track is the predicted signal when the input sequence at the CRE insertion site is all Ns. The second track is an example predicting using a validated HepG2-specific synthetic CRE. The third displays the differential DHS effect. (**b**) Empirical K562 MinGap measurements are well correlated with Enformer-predicted features of spleen-specific transcriptional activation (**Methods**). (**c**) Empirical HepG2 MinGap measurements are also well correlated with Enformer-predicted features of liver-specific transcriptional activation. (**d**) Empirical SK-N-SH MinGap measurements are also well correlated with Enfomer-predicted features of neural-specific transcriptional activation. (**e**) Enformer-based cell type matched tissue-specific transcriptional activation predictions (K562 matched to spleen, HepG2 matched to liver, SK-N-SH matched to adult brain). Stars indicate family-wise error rate corrected p-values < 1e-4 (In each trio of boxes, n = 4,000; 4,000; 12,000 elements for the DHS, Malinois, and synthetic groups, respectively).

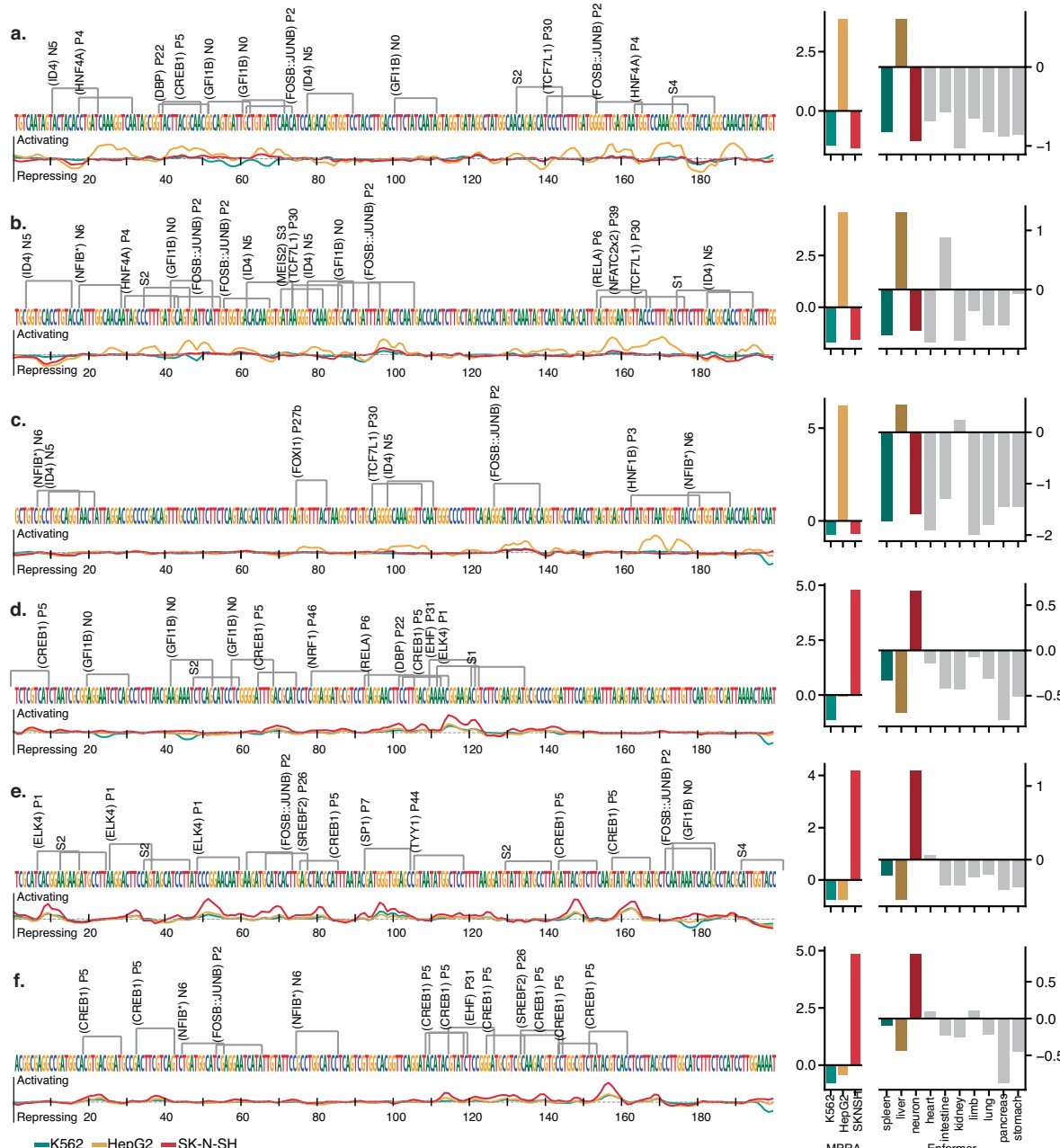

**Extended Data Fig. 9 | Malinois contribution scores/Enformer/MPRA results for in vivo sequences.** Collection of synthetic sequences prioritized for in vivo validation. Sequences in panels (**a-c**) and (**d-f**) are expected to drive expression in liver and neurons, respectively. Left column: Nucleotide sequence, motif matches, and contribution score tracks for each candidate.

Right column: Bar plots of empirical MPRA signal (left *y*-axis) in K562 (teal), HepG2 (gold), and SK-N-SH (red) as well as aggregated Enformer predictions (right *y*-axis) of epigenetic signals reflecting transcriptional activation in mouse spleen (dim teal), liver (dim gold), neural tissue (dim red), heart, intestine, kidney, limb buds, lung, pancreas, and stomach.

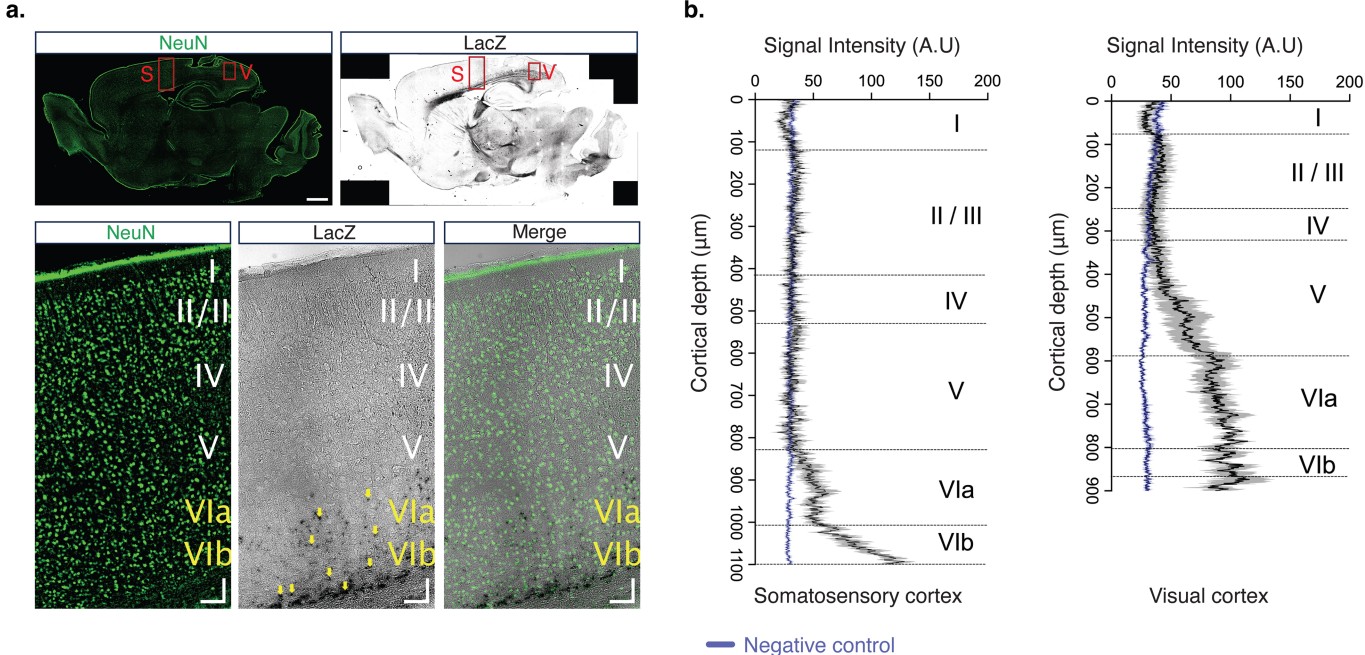

**a.**

**b.**

**Extended Data Fig. 10 | Immunohistochemistry of N1 CRE activity in the mouse cortex.** (**a**) Representative fluorescence and brightfield images showing expression patterns of neuronal marker, NeuN (top left) and LacZ (top right) across the whole brain. Boxed regions represent the somatosensory cortex (S) and visual cortex (V), digitally zoomed in bottom image; scale bars: 1 mm (top images) and 100 µm (bottom images). Yellow arrows indicate LacZ expression in layer 6. (**b**) Fluorescence intensity profile plots from quantification of LacZ signal intensity across layers in the somatosensory cortex and visual cortex for non-transgenic control (blue) and N1 CRE transgenic mouse (black).

Ryan Tewhey  
Steven K Reilly  
Rodrigo I Castro  
Sager J Gosai

# Reporting Summary

## Statistics

For all statistical analyses, confirm that the following items are present in the figure legend, table legend, main text, or Methods section.

| n/a | Confirmed | |
|---|---|---|
| ☐ | ☒ | The exact sample size (*n*) for each experimental group/condition, given as a discrete number and unit of measurement |
| ☐ | ☒ | A statement on whether measurements were taken from distinct samples or whether the same sample was measured repeatedly |
| ☐ | ☒ | The statistical test(s) used AND whether they are one- or two-sided *Only common tests should be described solely by name; describe more complex techniques in the Methods section.* |
| ☒ | ☐ | A description of all covariates tested |
| ☐ | ☒ | A description of any assumptions or corrections, such as tests of normality and adjustment for multiple comparisons |
| ☐ | ☒ | A full description of the statistical parameters including central tendency (e.g. means) or other basic estimates (e.g. regression coefficient) AND variation (e.g. standard deviation) or associated estimates of uncertainty (e.g. confidence intervals) |
| ☐ | ☒ | For null hypothesis testing, the test statistic (e.g. *F, t, r*) with confidence intervals, effect sizes, degrees of freedom and *P* value noted *Give P values as exact values whenever suitable.* |
| ☒ | ☐ | For Bayesian analysis, information on the choice of priors and Markov chain Monte Carlo settings |
| ☒ | ☐ | For hierarchical and complex designs, identification of the appropriate level for tests and full reporting of outcomes |
| ☐ | ☒ | Estimates of effect sizes (e.g. Cohen's *d*, Pearson's *r*), indicating how they were calculated |

*Our web collection on statistics for biologists contains articles on many of the points above.*

## Software and code

Policy information about availability of computer code

| | |
|---|---|
| Data collection | Reference data collection: downloaded data from respective databases with default versions of Internet browsers, wget 1.21.2, and curl 7.81.0. Zebrafish imaging: Leica Application Suite LAS X 3.5.5.19976, Olympus cellSens Standard 2.3 (Build 18987). Mouse whole organism and tissue imaging: Leica Application Suite X 3.7.5.24914, FIJI 2.11.0. Mouse IHC imaging: Leica Thunder Imager, Leica Stellaris 8, FIJI 2.11.0. |
| Data analysis | MPRA results were analyzed with: MPRAmodel (v1.0.1) and MPRAcount (v1.0.1). Statistical tests and topic modeling: MPRAmodel (v1.0.2), DESeq2 1.32.0, Scipy 1.10.1, Sklearn 1.2.2, Pandas 1.5.3, Biopython 1.81. DNA sequence function modeling was performed using the CODA python library (github.com/sjgosai/boda2). Docker images for interactive development and running CODA-based applications can be found at gcr.io/sabeti-encode/boda. RNA-seq analysis conducted with: STAR (v2.5.2b), picard MarkDuplicates (MIT, v3.1.1), featureCount (v2.0.6), DESeq2 (v1.32.0). Mouse IHC co-staining quantification analyzed with Prism (v10.1). |

For manuscripts utilizing custom algorithms or software that are central to the research but not yet described in published literature, software must be made available to editors and reviewers. We strongly encourage code deposition in a community repository (e.g. GitHub). See the Nature Portfolio guidelines for submitting code & software for further information.

## Data

Policy information about availability of data

All manuscripts must include a data availability statement. This statement should provide the following information, where applicable:

- Accession codes, unique identifiers, or web links for publicly available datasets
- A description of any restrictions on data availability
- For clinical datasets or third party data, please ensure that the statement adheres to our policy

Reference data sets collected from the ENCODE and Zoonomia databases that were used in this study are linked and annotated in Supplementary Table 1. GRCh38/hg38 and GRCm38/mm10 were used as reference genomes in this study. Processed MPRA data used to train Malinois is available in Supplementary Table 2. Processed MPRA data and Malinois predictions for the cell type-specific CRE library designed for this study are available in Supplementary Table 12. Sequencing reads for RNA-seq are available in NCBI GEO (PRJNA1075667). Raw data, processing notebooks, model weights, and immunofluorescence images are available at https://zenodo.org/records/10698014.

## Research involving human participants, their data, or biological material

Policy information about studies with human participants or human data. See also policy information about sex, gender (identity/presentation), and sexual orientation and race, ethnicity and racism.

| | |
|---|---|
| Reporting on sex and gender | N/A |
| Reporting on race, ethnicity, or other socially relevant groupings | N/A |
| Population characteristics | N/A |
| Recruitment | N/A |
| Ethics oversight | N/A |

Note that full information on the approval of the study protocol must also be provided in the manuscript.

# Field-specific reporting

Please select the one below that is the best fit for your research. If you are not sure, read the appropriate sections before making your selection.

☒ Life sciences  ☐ Behavioural & social sciences  ☐ Ecological, evolutionary & environmental sciences

For a reference copy of the document with all sections, see nature.com/documents/nr-reporting-summary-flat.pdf

# Life sciences study design

All studies must disclose on these points even when the disclosure is negative.

| | |
|---|---|
| Sample size | Our sample size for each group of similarly derived candidate DNA elements was set to a minimum of 1000 and a maximum of 4000 for in vitro . This was based on intuition that downstream group-to-group comparisons would be well powered at these sample sizes. We were limited by the maximum tractable size of a single MPRA library at the time the experiments were done.<br><br>Sample size for sequences used for in vivo validation are limited by available experimental capacity. Results are reported for all synthetic CREs undergoing in vivo validation, including examples which failed to validate as expected. All in vivo data shown should be taken as representative. |
| Data exclusions | Data were not excluded. |
| Replication | MPRA assays were conducted in triplicate. All replicates of MPRA were successful. Three HepG2 specific synthetic CREs were used to produce transgenic zebrafish and replicated liver expression in 27/36; 0/17; and 7/18 animals with an additional 0/32 control animal demonstrating no expression. Three synthetic SK-N-SH specific CREs were used to produce transgenic zebrafish and and replicated neural transgene expression in 3/3; 3/3; and 0/3 animals imaged, with an additional 0/3 control animals demonstrating no expression. Two synthetic SK-N-SH specific CREs were used to produce transgenic mice. Only 1/2 synthetic SK-N-SH generated neural expression in mice. The synthetic SK-N-SH specific CRE that generated neural expression in mice was found to drive transgene expression in 3/6 5-week-old mouse brains. Negative control mice demonstrated transgene expression in 0/5 5-week-old mouse brains. |
| Randomization | Individual cells and animals were chosen randomly for this study from their existing cultures or colonies. MPRA randomizes CREs transfected into individual cells. Control and experimental animals were randomly assigned before transformation. Each group was tested with a predetermined sample size of 3 liters and all samples were stained regardless of their genotype and sex. |

Candidate cell type-specific CREs were selected or generated algorithmically without human intervention once algorithms were deployed. Additionally synthetic sequence generation begins with sequence randomization.

| Blinding | Embryos were harvested and stained blindly with respect to their genotype. |

# Reporting for specific materials, systems and methods

We require information from authors about some types of materials, experimental systems and methods used in many studies. Here, indicate whether each material, system or method listed is relevant to your study. If you are not sure if a list item applies to your research, read the appropriate section before selecting a response.

## Materials & experimental systems

| n/a | Involved in the study |
|---|---|
| ☐ | ☒ Antibodies |
| ☐ | ☒ Eukaryotic cell lines |
| ☒ | ☐ Palaeontology and archaeology |
| ☐ | ☒ Animals and other organisms |
| ☒ | ☐ Clinical data |
| ☒ | ☐ Dual use research of concern |
| ☒ | ☐ Plants |

## Methods

| n/a | Involved in the study |
|---|---|
| ☒ | ☐ ChIP-seq |
| ☒ | ☐ Flow cytometry |
| ☒ | ☐ MRI-based neuroimaging |

## Antibodies

| Antibodies used | mouse anti-NeuN (abcam ab104224), chicken anti-GFAP (OriGene Technologies TA309150), rabbit anti-Iba1 (abcam ab178846). Secondary antibodies used were Goat anti-mouse Alexa Flour 488 (ThermoFisher Scientific, AB_ 2534069), Goat anti-chicken Alexa Flour 568 (ThermoFisher Scientific, AB_ 2534098), Goat anti-rabbit Alexa fluor 568 (abcam, ab175471). |

| Validation | All antibodies used are available from commercial vendors. Below lists vendor validated applications, total number of publications and relevant citations listed on the suppliers' websites.<br><br>• Mouse anti-NeuN (abcam ab104224): Monoclonal. Suitable for ICC/IF, WB, IHC-P. 582 publications. (https://pubmed.ncbi.nlm.nih.gov/27325769/)<br>• chicken anti-GFAP (OriGene Technologies TA309150): Polyclonal. Suitable for IF, WB. 3 publications. (https://pubmed.ncbi.nlm.nih.gov/34157194/)<br>• Mouse anti-Iba1 (abcam ab178846) Recombinant monoclonal. Suitable for WB, ICC/IF, Flow Cyt (Intra), IHC-P. 390 publications. (https://pubmed.ncbi.nlm.nih.gov/29769726/)<br>• Goat anti-mouse Alexa Flour 488 (ThermoFisher Scientific, AB_ 2534069): Polyclonal secondary Suitable IHC, ICC/IF, Flow Cytometry, 86 publications (https://pubmed.ncbi.nlm.nih.gov/36450710/).<br>• Goat anti-chicken Alexa Flour 568 (ThermoFisher Scientific, AB_ 2534098): Polyclonal secondary. Suitable for WB, IHC, ICC/IF, flow cytometry, 3 publications (https://pubmed.ncbi.nlm.nih.gov/32290848/).<br>• Goat anti-rabbit Alexa fluor 568 (abcam, ab175471): Polyclonal. Suitable for ELISA, IHC-Fr, IHC-P, Flow Cyt, ICC/IF, 187 publications. (https://pubmed.ncbi.nlm.nih.gov/33503434/) |

## Eukaryotic cell lines

Policy information about cell lines and Sex and Gender in Research

| Cell line source(s) | All cell lines were purchased from ATCC (atcc.org). This study used K562, HepG2, and SK-N-SH cell lines. |

| Authentication | All cell lines were acquired from ATCC, authenticated using genotyping and gene expression signatures, routinely. |

| Mycoplasma contamination | All cell lines are tested monthly for mycoplasma and other common contaminants by The Jackson Laboratory's Molecular Diagnostic Laboratory. |

| Commonly misidentified lines<br>(See ICLAC register) | *Name any commonly misidentified cell lines used in the study and provide a rationale for their use.* |

## Animals and other research organisms

Policy information about studies involving animals; ARRIVE guidelines recommended for reporting animal research, and Sex and Gender in Research

| Laboratory animals | Mouse: Species - mus musculus, Strain - FVB (JAX ID #001800), Age - embryo to 5 weeks post natal. Zebrafish: Species - Danio rerio, [AB wild-type strain], Age - embryos up to 5 days post fertilization. All mice were housed in duplexed pens containing five or less mice and a 12-hour light/dark cycle at 18-23°C with 40-60% humidity. |

| Wild animals | This study did not involve wild animals. |
| Reporting on sex | Sex was not considered in this study. |
| Field-collected samples | This study did not involve field-collected samples. |
| Ethics oversight | All zebrafish procedures were approved by the Yale University Institutional Animal Care and Use Committee (IACUC) (Protocol Number 2022-20274). All mouse procedures were performed in accordance with the National Institutes of Health Guide for the Care and Use of Laboratory Animals, and were approved by the Institutional Animal Care and Use Committees of The Jackson Laboratory (protocol number 18038). |

Note that full information on the approval of the study protocol must also be provided in the manuscript.

