## [Peer Review File · Nature]

Manuscript Title: Machine-guided design of cell type-specific cis-regulatory elements

Reviewer Comments & Author Rebuttals

Reviewer Reports on the Initial Version:

Referee #1 (Remarks to the Author):

The manuscript titled "Machine-guided design of synthetic cell type-specific cis-regulatory elements" by Gosai et al. presents a framework, called CODA, which uses a deep learning model in conjunction with sequence space exploration algorithms to synthetically design sequences to drive cell type-specific expression in human cells. Specifically, the authors first trained a convolutional neural network (CNN), called Malinois, using data from a large-scale MPRA experiment and demonstrate that its performance is good. Then, they use Malinois as an "oracle" to guide local search algorithms in sequence space, either as a scoring function or through input optimization via backprop using established methods. The authors validate designed sequences in a follow up MPRA and find that synthetic sequences outperform natural sequences at driving cell type-specific expression. They also perform validation experiments across species in zebrafish and mouse.

Overall, CODA seems to work well and so this could serve as a useful tool that could advance the agenda of designing cell-type specific regulatory sequences in medicine and basic research. The power of this approach really seems to come from the training data being quite large, beyond the standard approach of just using a reference genome. Notwithstanding, CODA seems to work despite many unknowns that were not explored in this study. Performing a more thorough, expanded analysis is important to help bridge this gap. Moreover, the motif analysis makes strong claims but provides weak support. In addition, it would be beneficial to clean up the overall presentation with more precise claims that are sufficiently supported with the evidence provided, otherwise the strength of the claims should be toned down. Below, we detail our concerns.

Major concerns:

Malinois: In summary, it is not clear whether Malinois is a worthwhile model as there was no comparison with other models.

- The description of the model and training procedure is not sufficiently detailed to allow someone to reproduce the work. This makes it difficult to fully evaluate Malinois. One part of Malinois that was clear is that it has at least one very odd design choice -- 200nt padding on each side of the central 200nt MPRA sequence, making the total input sequence 600nt. This seems excessive and unnecessary, unless if they performed transfer learning with pretrained Basset model weights, which was not discussed. In any case, the authors need to do more standard benchmarking of their model choices and compare against other models and perform proper control experiments if the field is to appreciate one of the main contributions of this paper, Malinois.

- The authors state that Bayesian inference was used to find the best hyperparameters for their CNN (out of 1000 other contenders). It is not clear whether Malinois provides significantly improved performance compared to the other 1000 contenders. It is likely that there are hundreds of settings that would likely provide comparable results. Showing a distribution of performance of each model in the hyperparameter search could help understand this better. It is also not clear whether the other models would score the proposed synthetic sequences in the same way. If so, this would actually be beneficial as it shows that the pipeline is robust to CNN, instead of trying to claim that a single model was the reason why the pipeline was possible.
- An ablation study should be performed on Malinois to identify key components that led to high predictive performance and good sequence design. Further model exploration should be done to improve efficiency of the model and remove any unnecessary components that don't specifically improve model performance.
- In a comparison with new models, the focus should not only be on performance but also the design aspect. Rather than running new experiments, it could be sufficient to perform a retroactive study that shows how the other models would score the synthetic sequences. If other models perform just as well as Malinois, the takeaway should de-emphasize a specific model and the takeaway should be more about any CNN trained on large amounts of MPRA data can be quite powerful for sequence design. If Malinois is the only model that does well, then it is critical to perform the aforementioned ablations to understand what components enabled it to do better.

Sequence space searching algorithms: Overall, readers will want to not just know if the pipeline works but also why the pipeline works. A more careful evaluation of the sensitivities to various aspects of the CODA pipeline should be presented.

- The sequence design algorithms have significant limitations – they are all local search algorithms. The descriptions of the guarantees should be written more precisely so as not to be misleading.
- The evaluation of the sequence design methods is lacking; they are grouped together under the label, synthetic (Fig. 2c). Does one method work better than another? How similar are the sequences generated from each generation method? Do they fall into similar functional programs for instance?
- Evaluating the diversity of sequences is tricky. The authors propose a k-nearest neighbor approach. Other approaches should be performed to improve robustness of the study, including a comparison of sequence context, such as k-mer frequencies of a set of sequences generated by each design method.
- The description of motif penalization is confusing. The description should be made clearer.
- How sensitive are the search algorithms to their hyperparameter choices?

Motif analysis: Overall, the analysis is observational without any validation or sensitivity analysis.

- Many claims are based on observational motif analysis. It is not clear that motifs and motif combinations identified from basic motif enrichment analysis is directly tied to functional use. To validate the motif claims, ideally, motifs and motif combination should be embedded in random backgrounds and measured experimentally in follow up MPRA. In silico validation with Malinois could suffice if the in silico experiments are performed systematically and are comprehensively. Such interventional experiments are becoming the norm in model interpretability in genomics. Any hypotheses of what motifs/motif combinations are important can be directly tested in a controlled environment to uncover the quantitative effect sizes. [PMID: 33603233; PMID: 33983921; PMID: 35551305]
- A robust modified version of Integrated gradients was used to "identify magnitude and direction of effect of each motif in each of our three cell lines. The description needs to be clearer in the methods and evidence should be provided why it is more robust. Generally, it's not clear what value this robust modified version of IG is? Does it provide insights that couldn't be achieved with standard attribution analysis?
- The motif statistics presented in the results reads as an overly confident analysis. It is not clear how motifs were defined as repressive for instance. How sensitive are these numbers given the heuristic choices to make these definitions? It could be more conservative to state what heuristic was used and give the motif statistics with that disclaimer.
- With regards to the functional programs analysis that are based on standard motif analysis (using MEME suite), the main concern is that these motifs may or may not be functional. An opportunity to use Malinois to annotate functional motifs using their learned sequence-function relationship seems squandered. As their oracle, this should be more informative for "functional motifs". A similar protocol using tf-Modisco followed by contribution-weight matrix scans could identify functional motifs and annotate each sequence.
- Further comments for the NMF analysis on motif enrichment. The strength of motif binding sites is not considered -- weak motifs and stronger motifs are binarized as a functional motif according to FIMO, which is known to lead to many false positives. One could in principle use motif scores instead of counts for this analysis – attribution scores or the PWM scan. Also, the sensitivity of the NMF analysis should be explored based on different hyperparameters of FIMO.
- The validation of the regulatory programs is lacking. It is just an observational analysis which may or may not be right. One way to prove regulatory programs are known is to perform in silico experiments - controlled motif embeddings in random background sequences to quantify that predictions behave as one expects.
- "to improve interpretability of the topic modeling, we generated an additional 4000 sequences for each cell type which prioritized off-target expression." It is not clear whether the presence/absence of these sequences improves interpretability.
- With regards to: "The regulatory activity contribution scores identify the overall magnitude and direction of the effect of each motif in each of our three cell lines" Attribution methods quantify nucleotide importance, not motif importance.

General concerns

- Reproducibility is a major concern. The code repository is not organized well, there is no roadmap for how to execute code, and the code is not commented well. Code should be provided to: 1. take supp table 2 and processes it into training, validation and test sets; 2. build and train Malinois on this dataset; 3. execute each sequence design method; and 4. run inference on a trained model to replicate the figures in the main text. Also, the processed dataset and synthetically generated sequences should be saved as npz or h5 or even pickle.
- It is not always clear what is novel by the authors versus what was already known or previously established. Being clearer on this in the text could be beneficial.
- A more thoughtful description of limitations should be included. For instance, modeling based on MPRA, sequence design algos, pathological behaviors outside the data distribution (i.e. out-of-distribution). These of course were not fully explored but could at least be mentioned as a warning to others that blindly think that DNNs that make good predictions on test data can generalize everywhere in sequence space.
- There is no exploration for which components of CODA/Malinois and so it is not clear what enabled these results. For instance, is MPRA design better than training a model on chromatin accessibility sites from ATAC-seq for these 3 cell types? This is a very different question than what was probed in the study, which compares natural accessible sites.
- The authors claim that CODA is generalizable and extendable. I can see how CODA is generalizable and extendable in principle. But the framework is not novel. It is standard practice to use oracles and have either search algos or ML models to help navigate the oracle's functional landscape. This is done quite commonly in ML-guided protein function landscapes. The code itself is far from a usable toolkit or an extensible framework that others would be able to build upon.

Minor concerns:

- On line 64, meaning is ambiguous: "computational models are still only capable of characterizing a fraction of all possible sequence combinations". The CNN has learned a function that defines a value to each input sequence in the domain. So, does this refer to the poor out-of-distribution generalization or the intractability of training and querying a model on a limited subspace.
- "optimization guarantees" (line 154) should be clarified. The simple sequence design methods are all greedy search algorithms infamous for their lack of ability to navigate complex surfaces.
- Colors in Fig 1e could benefit from something that contrasts better, perhaps red. Also, the authors could play with the transparency.
- Equations should be formatted so that they are clearer (see Methods; e.g., lines 676, 652, 835, 836, etc.)
- Grammar: Line 349; Line 652 (equation typo); Line 688.

Referee #2 (Remarks to the Author):

In this manuscript, the authors use MPRA, epigenomic marks, and machine learning to design tissue-specific enhancers. This impressive body of work tests many enhancers in different cell types to create tissue-specific enhancers. They develop computational tools to predict functional tissue-specific enhancers using MPRA and epigenomic data. They compare synthetic and natural sequences' ability to drive tissue-specific expression as defined by activity in 3 cell lines - erythroid, hepatocytes, and neuroblastoma. They claim synthetic enhancers show greater cell type specificity than the predicted functional natural sequences. The authors test 6 synthetic enhancers via reporter assay in zebrafish and two in mouse, although the nature of the specificity of these enhancers within the embryos is poorly described. The number of enhancers tested in this manuscript is impressive, and the computational tools will be helpful to the research community. The design of tissue-specific enhancers is a novel and exciting area.

The premise of the approach and comparison between natural and synthetic enhancers and their ability to drive cell type-specific expression is flawed. The cell types the authors are studying are derived from cancer cell lines; thus, these are not the environment in which the genome evolved. Therefore, it makes sense that their synthetic enhancers would drive better cell type specificity than natural sequences, but this does not mean that, in general, synthetic enhancers are better at driving cell type specificity. The authors make claims about synthetic enhancers vs. natural ones, but they cannot translate their results in transformed stable cell lines to normal cell types in an organism.

The summary statement states that their synthetic sequences can outperform natural sequences in driving tissue-specific gene expression. It is also not clear what out-perform natural sequences means. To demonstrate that synthetic sequences outperform natural sequences, they would need to remove all enhancers contributing to the expression of a gene and replace them with synthetic enhancers, and show the organism survives. Or if there is a therapeutic goal, show that the synthetic sequence drives expression of the target gene more effectively than the natural sequence. It is likely within the genome using endogenous promoters, the natural sequences would outperform the synthetic sequences in non-transformed cells. I feel that the comparison of natural and synthetic sequences detracts from their ability to design tissue-specific enhancers.

In terms of the synthetic and natural enhancers that function, they test these in 3 cell lines they trained their data on. This is impressive, and there is a clear preference for the predicted tissue-specific enhancers to be active in the respective cell types. I would like to know how these libraries containing their predicted functional tissue-specific enhancers work on other cell lines? Such as fibroblasts or other epithelial cells and neural cell types for example. Do we know that these enhancers are only active specifically within one cell type?

They claim that synthetic CREs are fit for purpose in vivo, and yet we do not know this for several reasons. Firstly the information provided about the specificity is very limited, are the three enhancers seen in the liver specifically expressed in hepatocytes only? There are four major cell types in the liver, which cell types are their two enhancers active within? I would like to see staining with co-markers to see this. They also need more controls that test a similar number of sequences they anticipate would not be liver-specific or would be inert and yet contain similar motifs and show these do not drive liver expression. I would like to see testing of three natural sequences that they predict are hepatocyte-specific as well.

Regarding the neural enhancers, the ability to design tissue-specific neural enhancers is also unclear to me and not validated in the current manuscript. The enhancers are active in neural cell types, but how do the neural cell types in the zebrafish relate to neuroblastoma cells or mouse cortex? Would their neuroblastoma active enhancers be active in all neural cell types? If so, is this really cell specificity?

More enhancers need to be tested and more details provided on the precise location of expression. They would also need to show that synthetic enhancers that are not predicted to work and yet show the same motifs are non-functional. If the authors want to show natural sequences are less specific than synthetic they would also need to test natural sequences and see the location of expression.

The authors state, "There is no guarantee that an optimal CRE for an intended purpose has arisen naturally through evolution." I agree, but I would caution against trying to make statements about evolution when the analysis is within transformed cell types. There are several statements about evolution, and I would be careful with these types of statements as they can be misleading. Evolution has not been working on transformed cell lines, and thus one would not expect the natural sequences to be optimized to drive expression in these cell lines.

Regarding the discovery of syntax, the authors state "Synthetic sequences leverage unique sequence syntax to promote activity in the on-target cell type and simultaneously reduce activity in off-target cells."

The syntax is not clearly defined, and I don't think this manuscript contributes to our understanding of syntax within enhancers. I understand that TF motifs are found, but are unique syntaxes that drive on-target and reduce off-target demonstrated within their data? Can they make a library where they delete these syntax elements and show that they lose on-target and gain off-target expression? This experiment would be required to validate their claims.

The authors state that "Malinois contribution scores enable nucleotide resolution interpretation of sequence activity." In Figure 3a, the authors show a synthetic enhancer that drives expression in HepG2 Cells, they highlight motifs found by Malinois that are important for the activity. To demonstrate that Malinois finds the functional features within the enhancer with base pair resolution, the authors need to mutate these features and show that this renders the enhancer inactive or no longer tissue-specific. Can the authors do this validation on a library of enhancers?

Referee #3 (Remarks to the Author):

Summary

Massively parallel reporter assays (MPRAs) are increasingly common technique to interpret human genetic variation, screen for enhancer activity, and understand the principle of gene regulation. One application of MPRA data is the prioritization and design of sequences that can label specific cell types. Gosai et. al. develop computational methods to learn a “regulatory grammar” for a specific cell types based on MPRA data. Then, they apply those models to generate synthetic sequences that have greater specificity for the cell type of interest. The research presented here is cutting edge and of broad interest to multiple scientific communities. The strength of the manuscript is the careful benchmarking of different evolutionary algorithms, the interpretations of transcription factor binding site motifs, and the in vitro validation their results. In spite of these positives, there are major issues to address. First, the manuscript depends on unpublished MPRA data for which there is no described quality control or experimental design included. Second, the Malinois model is not adequately benchmarked for its cell type specificity or compared to alternative computational models. This is crucial, given that Malinois is used as a benchmark for CODA. Third, the application of the approach to label cell types in vivo requires a strong similarity between the regulatory grammar of that cell type and the in vitro model of that cell type. This relationship isn’t adequately explored. Fourth, Malinois and CODA as a method for finding sequences to label specific populations aren’t benchmarked against published methods. In summary, there is strong potential for this manuscript to have an impact, but additional rigor in some components of the analysis are required to demonstrate the author’s claims.

Major Points

The increasing amount of available MPRA data makes this manuscript especially timely. There is a need of innovative computational methods to better leverage and understand this type of data. This work fills an important gap and represents a departure from the use of natural sequences towards potential engineering with synthetic sequences.

The CODA portion of the manuscript is well motivated and well written. The choice of evolutionary algorithms makes sense in the context of the study. The in vitro validation provides a good demonstration of CODA’s utility.

The research depends on large amounts of unpublished MPRA data, which seems to have been collected in the laboratory of the senior author. The details of how the MPRA experiments were designed (especially which sequences were included) and the quality control metrics of that dataset have a bearing on what conclusions can be drawn from the study.

For Figure 1, it’s not clear why the GATA1 locus is the focus. It would be more helpful to stratify the accuracy across the different input datasets. Hyper parameters seemed to be tuned on chromosome X, where GATA1 is locus, which could introduce circularity and inflate the model performance.

The cell type-specificity of Malinois is key to the study, but there are several issues with how that component of the project is validated. First, Malinois is presented as a new computational method, but it is not benchmarked relative to others that could be trained on similar data, including an ENFORMER-like model. Second, the cell type-specificity of Malinois isn't compared to the cell type-specificity of open chromatin or other measurements, just the raw values. Differences in signal and depth could play a role in accuracy. Third, based on Supplemental figure 3a, I get the impression there is a low proportion of cell type specific CREs. To account for this, there should be a greater exploration and visualization of the number of CREs enriched for each cell type. A statistical test that better handle class imbalance would be helpful to quantify the cell type-specificity. The spearman correlation is currently applied. These concerns are especially important given that CODA is compared to Malinois alone. If Malinois isn't efficient at determining cell type specificity due to these concerns, then CODA's improvement over Malinois could be overestimated.

The ultimate goal of the research is to label specific types of cells. There are other methods for predicting the how sequence specific certain cell type are. Currently, there is no comparison of this method to those previous methods. These include:

<https://doi.org/10.7554/eLife.48089>
<https://doi.org/10.7554/eLife.69571>
<https://doi.org/10.1038/s41434-021-00227-z>
<https://doi.org/10.1093/nar/gkad375>
<https://doi.org/10.1101/2022.07.26.501466>

The final claim of the abstract is that the method is able to “write regulatory code that is fit-for-purpose in vivo across vertebrates.” The experiments in brain and liver are promising, but are not sufficient to support that claim. It isn't clear if it is possible to collect a in vivo dataset large enough to apply CODA to. Alternatively, the method requires that the regulatory grammar of the in vivo cell type is similar enough to the in vitro system for CODA to be useful. That demonstration is also not shown at a large scale.

Minor Points

- In figure 1, a continuous representation should be used to describe the relationship between Malinois predictions and the open chromatin and other genomic measurements. P values are difficult to interpret given the large number of data points.
- Supplemental figure 3 does not have scatter plot comparisons for the SK-N-SH cells.
- Starr-seq is a continuous signal and an assay similar to MPRA. It would be helpful to see both treated as a continuous signal to look for correspondence.
- In figure 2c, the performance of the DHS natural and Malinois natural are surprisingly poor. Why is open chromatin and H3K27ac a poor indicator of specificity? Does the relatively poor Malinois performance suggest reproducibility issues?

- For each cell type (and across cell types) how well do the new MPRA results correlate with the previous MPRA results for the same sequences? Are there experimental differences, either procedure or source of the cell line, that could be driving performance?

- The proportion of oligos with error that are too high to serve a reliable indicator should be more thoroughly described and addressed. A computational model that produces sequences with high variability should be considered inferior.

- The propeller plots are a useful visualization and do confirm that the sequences with the largest fold difference are also the ones high specificity. However, it would also be useful to have visualization for each pair of cell types that compare the specificity in 2D space.

Reviewer #1

The manuscript titled "Machine-guided design of synthetic cell type-specific cis-regulatory elements" by Gosai et al. presents a framework, called CODA, which uses a deep learning model in conjunction with sequence space exploration algorithms to synthetically design sequences to drive cell type-specific expression in human cells. Specifically, the authors first trained a convolutional neural network (CNN), called Malinois, using data from a large-scale MPRA experiment and demonstrate that its performance is good. Then, they use Malinois as an "oracle" to guide local search algorithms in sequence space, either as a scoring function or through input optimization via backprop using established methods. The authors validate designed sequences in a follow up MPRA and find that synthetic sequences outperform natural sequences at driving cell type-specific expression. They also perform validation experiments across species in zebrafish and mouse.

Overall, CODA seems to work well and so this could serve as a useful tool that could advance the agenda of designing cell-type specific regulatory sequences in medicine and basic research. The power of this approach really seems to come from the training data being quite large, beyond the standard approach of just using a reference genome. Notwithstanding, CODA seems to work despite many unknowns that were not explored in this study. Performing a more thorough, expanded analysis is important to help bridge this gap. Moreover, the motif analysis makes strong claims but provides weak support. In addition, it would be beneficial to clean up the overall presentation with more precise claims that are sufficiently supported with the evidence provided, otherwise the strength of the claims should be toned down. Below, we detail our concerns.

We thank the reviewer for their careful review and thoughtful summary of the work. We appreciate their recognition of the work's utility for advancing design of cell type-specific regulatory sequences. We also understand their request for expanded analysis to better support the claims. We have aimed to do so in responding to the specific comments below, and more broadly through additional work and manuscript revision.

Malinois: In summary, it is not clear whether Malinois is a worthwhile model as there was no comparison with other models.

Author note: Our responses to reviewer 1's summary point and comments 1 and 4 are related, so we are combining them here.

R1C1. The description of the model and training procedure is not sufficiently detailed to allow someone to reproduce the work. This makes it difficult to fully evaluate Malinois. One part of Malinois that was clear is that it has at least one very odd design choice -- 200nt padding on each side of the central 200nt MPRA sequence, making the total input sequence 600nt. This seems excessive and unnecessary, unless if they performed transfer learning with pretrained Basset model weights, which was not discussed. In any case, the authors need to do more standard benchmarking of their model choices and compare against other models and perform proper control experiments if the field is to appreciate one of the main contributions of this paper, Malinois.

R1C4. In a comparison with new models, the focus should not only be on performance but also the design aspect. Rather than running new experiments, it could be sufficient to perform a retroactive study that shows how the other models would score the synthetic sequences. If other models perform just as well as Malinois, the takeaway should de-emphasize a specific model and the takeaway should be more about any CNN trained on large amounts of MPRA data can be quite powerful for sequence design. If Malinois is the only model that does well, then it is critical to perform the aforementioned ablations to understand what components enabled it to do better.

Model description and Basset architecture

We appreciate that our methods and design motivations were not clear, as noted in this and other related points raised by reviewers. We agree with the reviewer that a more detailed description of our design decisions will provide important context to the reader for why Malinois works well. We've added text to the **Main** and **Methods** sections to make the model and training of Malinois much clearer. Specifically:

- We clarify in the **Main** section that Malinois is adapted from the Basset architecture. We provide additional details in the Methods section explicitly describing our adaptation.
- We explain in the **Main** section our decision to use transfer learning to better train an MPRA model from a data set that, by modern deep-learning standards, is considered small.
- We have edited **Supplementary Figure 2** and part of the **Results** text to highlight portions of the architecture with weights inherited from Basset at the start of training.
- We now explain in the **Methods** that padding input sequences to 600 nucleotides is required to maintain compatibility with the architecture components of Malinois that are inherited from Basset, which we did not clearly do before.

We provide the key modified sections below with key new text in blue:

Updated Results Line 109:

*"We created Malinois, a deep convolutional neural network (CNN) for **prediction of cell type-informed CRE activity of any arbitrary sequence as measured by MPRA. We adapted architectural components from Basset⁴⁷, a model of chromatin accessibility (Figure 1c, Supplementary Figure 2, Methods), and leveraged Bayesian optimization^{57,58} to iterate over hyperparameter settings to identify a high performing model (Supplementary Figure 3a). We observed several design choices that impacted the model including the use of transfer learning from Basset (Supplementary Figure 3b-d, Supplementary Table 3, Methods).**"*

Updated Methods Line 658:

*"Malinois accepts batches of 4 x 600 arrays corresponding to one-hot encoded DNA sequences, so predictions **for 200-nt MPRA oligos** are made by padding inputs on both sides with constant sequences from the reporter vector backbone. **This strict input sizing requirement ensures hidden states are appropriately shaped when transitioning between segments (1) and (2) of the model. At training initiation weights were initialized using pre-trained weights from a PyTorch implementation of Basset when (1) and (2) were appropriately configured.**"*

Exchangeability of predictive models for sequence design

As the reviewer suggests, we do not expect the architecture of Malinois to be the sole unique factor behind its effectiveness in this study, but rather the large amount of high-quality MPRA data it is trained on, generated in the same labs as those building the models. We only aimed to ensure that Malinois was as good, if not better, than currently used methods such as Enformer and MPRA-DracoNN, comparisons to which we have now included in response to revisions. We, however, agree that newer architectures such as ResNets,

Transformers, or Structured State Space Sequence models could likely outperform Malinois after careful development. By extension, these other sequence-to-function models are also likely able to be used for the development of synthetic elements, so we have clarified this in the **Discussion**.

Line 597:

“Applying MPRA in additional cell types with greater clinical relevance and training new models on these data could enable CODA to better design CREs with specificity tailored for therapeutic applications. As the technology underlying sequence-to-function models continues to evolve, are mechanistically interrogated through ablation studies, and are trained on high-quality MPRA data sets, we expect synthetic element designs to become even more reliable and reduce the experimental burden for in vitro and in vivo validation.”

Comparison to alternative contemporary models to Malinois for sequence design

We appreciate the reviewer's concern that we did not sufficiently show the value of the Malinois model in comparison to other models. It had not been our intention to focus the paper on the advances of our Malinois model, but rather as a means to an end of designing best-in-class synthetic sequences. We are however very pleased to provide comparisons on the method.

When this study was initiated, there were no models perfectly suited to predict the impact of arbitrary 200nt DNA sequences in MPRA in K562, HepG2, and SK-N-SH, and still none published to date. While there were two well-established models for analyzing MPRA data, they had limitations that motivated us to train a new MPRA-based model on data being generated from our lab.

- MPRA-DracoNN (Movva et al. 2019) (<https://kipoi.org/models/MPRA-DracoNN/ConvModel/>) had been developed to predict MPRA activity in K562 and HepG2, but not SK-N-SH. Furthermore, MPRA-DracoNN was trained on very early MPRA experiments which have undergone dramatic technical improvement in recent years.
- DNA sequence models of chromatin architecture, such as Enformer (Avsec et al. 2021) (<https://github.com/google-deepmind/deepmind-research/tree/master/enformer>), could serve as another imperfect substitute, however there is ample evidence of discordance between the impact of DNA sequences on chromatin architecture and in reporter assays. We quote two examples from the literature:
 - “We found that 43.3% of the active sequences were marked as DHS” (Tewhey et al. 2016)
 - “We observed low correlation between MPRA expression and observed ATAC-seq signal (Pearson $\rho = 0.097$), predicted ATAC-seq signal (Pearson $\rho = 0.088$) and observed H3K27ac signal (Pearson $\rho = 0.061$)” (Kim et al. 2021)

We have now compared Malinois to Enformer and MPRA-DracoNN. As expected, we show that Malinois better predicts the function of CREs designed for cell type-specific transcriptional activation than either method as shown in Reviewer Figure R1C1 below:

Reviewer Figure R1C1. Benchmarking MPRA-DracoNN and Enformer predictions on CREs designed for cell type-specific reporter activation. MPRA-DracoNN predicts 6 features corresponding to reporter assay activity in each K562 and HepG2. Colored horizontal bars above each boxplot indicate Pearson's correlation of Malinois with MPRA. (a) Pearson's correlation coefficients for different MPRA-DracoNN features with measured MPRA activity of natural and synthetic CREs. (b) Correlations re-calculated using synthetic CREs only. (c) Pearson's correlation coefficients calculated using Enformer features for the same reference sequences in (a). (d) Enformer correlations re-calculated using synthetic CREs only.

While these results validate our decision to train a new model on newly generated data, including this analysis to the manuscript could lead a reader to incorrectly conclude that we are implying MPRA-DracoNN or Enformer are poor models rather than imperfect surrogates that were designed for other contexts. Our approach to benchmarking is to exclude direct comparisons when there are considerable limitations to the analysis *and* results that could unfavorably critique someone else's work. In this situation, we believe it is unjust to compare our model against all pre-existing models due to the reasons described above. However, if the reviewers strongly believe we should include this analysis in the manuscript we will consider inclusion.

Comparison of Malinois to state-of-the-art alternatives in development

While its architecture is relatively simple by today's standards, Malinois archives high predictive performance and scaled effectively for the rest of the applications in this study. Through correspondence with other experts in the field, we have also tested two upcoming alternate models on our data (**Reviewer Figure R1C4**).

- MPRA-LegNet was trained on reporter assay data generated in parallel to the training set used in Malinois by another lab as part of ENCODE Phase 4. An ensemble of weights from 90-fold cross-validation was used to make predictions in K562 and HepG2 (correspondence with Ivan Kulakovskiy, Dmitry Penzar, and Vikram Agarwal).
- ReporterNet was trained on the *identical* data set as Malinois and benchmarked using an ensemble of weights from 5-fold cross validation (correspondence with Ziwei Chen and Anshul Kundaje). It is worth noting that, during the development of ReporterNet, collaborative discussions on Malinois' training

methodology and architecture influenced ReporterNet to converge to some of our strategies, and contributed to enhancing its performance. For instance, the group reported that ReporterNet improved performance by [REDACTED]

Reviewer Figure R1C4. Malinois compares favorably to (unpublished) state-of-the-art alternatives.

Scatterplots capture correlation between MPRA measurements on the x-axis and model predictions on the y-axis. On each plot, outer and inner black lines demarcate the 90th and 50th percentile kernel density estimates, respectively. (a) Malinois predictions. (b) MPRA-legNet predictions (through correspondence with: Ivan Kulakovskiy, Dmitry Penzar, and Vikram Agarwal). (c) ReporterNet predictions (through correspondence with: Ziwei Chen and Anshul Kundaje).

Our benchmarking on cell type-specific sequence proposals shows Malinois (Pearson’s $r = 0.77-0.90$; Spearman’s $\rho = 0.83-0.89$) performs comparably to MPRA-LegNet (Pearson’s $r = [REDACTED]$; Spearman’s $\rho = [REDACTED]$) and ReporterNet (Pearson’s $r = [REDACTED]$; Spearman’s $\rho = [REDACTED]$). Interestingly, this is despite the extensive ensembling used by other techniques and separate weights trained to model each cell type using

LegNet and ReporterNet. In contrast, Malinois is deployed with a single set of weights for multi-task prediction across cell types.

Both MPRA-LegNet and ReporterNet papers are still unpublished, and both are focused largely on the development and extensive comparison of CRE modeling. We respect their request to allow these comparisons to be carried out in their manuscripts. As suggested by this reviewer, we instead de-emphasize Malinois, while expanding our description of its implementation for future development, and instead clarify the focus this manuscript is on analyzing synthetic sequences.

R1C2. The authors state that Bayesian inference was used to find the best hyperparameters for their CNN (out of 1000 other contenders). It is not clear whether Malinois provides significantly improved performance compared to the other 1000 contenders. It is likely that there are hundreds of settings that would likely provide comparable results. Showing a distribution of performance of each model in the hyperparameter search could help understand this better. It is also not clear whether the other models would score the proposed synthetic sequences in the same way. If so, this would actually be beneficial as it shows that the pipeline is robust to CNN, instead of trying to claim that a single model was the reason why the pipeline was possible.

Clarification of hyperparameter tuning

We now include a new Bayesian Optimization experiment that jointly optimized over all relevant hyperparameters we varied during the development of Malinois. This experiment is more comprehensive than previous hyperparameter optimization experiments. These prior experiments were done very early in the study and piecemeal over subsets of hyperparameter combinations during the initial development of Malinois. Importantly, this new experiment assessed the importance of branching cell-specific heads, transfer learning weights from Basset, and various data set augmentations.

The results of the Bayesian Optimization experiment are included in a new supplementary figure (**Supplementary Figure 3**) to better enable others to use and expand on our models. The key takeaways from this experiment are:

- Bayesian optimization finds reasonable hyperparameter settings within 100 proposals and additional adjustments produce models with only incremental improvements in performance (**Supplementary Figure 3a**).
- Initializing weights using Basset, a model of chromatin accessibility, is more effective than random initialization (**Supplementary Figure 3b**).
- Duplicating and augmenting the training data by taking the reverse complements of the input sequences improves predictions (**Supplementary Figure 3c**).
- Using branched linear layers in place of fully connected layers in the final layers of the model can produce some of the highest performing individual models, but does not result in a dramatic overall improvement and introduces substantial variance likely due to the lack of transfer learning at these new layers (**Supplementary Figure 3d**).
- Malinois, the model and weights of which we finalized in 2021 and used to generate synthetic elements, is only slightly outperformed by the top configurations from this latest experiment (Malinois: Pearson's $r = 0.877-0.886$; Top BayesOpt model: Pearson's $r = 0.880-0.890$). We agree with the reviewer that these slight additional improvements are unlikely to have a meaningful impact on other aspects of this study.

We highlight results in the **Main** text and a **Supplementary Figure** and add details regarding experimental setup to the **Methods**. For ease of review, we include the text and figures below, with relevant new text in blue italics:

Line 109:

“We created Malinois, a deep convolutional neural network (CNN) for prediction of cell type-informed CRE activity of any arbitrary sequence as measured by MPRA. We adapted architectural components from Basset⁴⁸, a model of chromatin accessibility (Figure 1c, Supplementary Figure 2, Methods), and leveraged Bayesian optimization^{58,59} to iterate over hyperparameter settings to identify a high performing model (Supplementary Figure 3a). We observed several design choices that impacted the model including the use of transfer learning from Basset (Supplementary Figure 3b-d, Supplementary Table 3, Methods).”

“Supplementary Figure 3. Bayesian optimization effectively finds reasonable hyperparameter settings. (a) Validation and test set performance of models from hyperparameter proposals picked by Bayesian Optimization, in order. Dotted lines indicate test set performance of Malinois. (b) Transfer learning by initializing weights from Basset results in less variation and overall improvement in training outcomes. (c) Duplicating and augmenting the training data by taking the reverse compliments of the input sequences improves modeling accuracy. (d) Replacing fully-connected layers in the decoder segment of CNNs increases variance in fitted model performance, although the top performing branched decoder models show improvement comparatively.”

We are also including a notebook to allow others to deploy this experiment on the Google Cloud Platform (boda2/tutorials/vertex_sdk_launch.ipynb) and the complete results as a supplementary table (**Supplementary Table 3**) that readers can use to interrogate any detailed questions about hyper parameter combinations. We note this in the **Methods**:

Line 675:

“We include a notebook for deploying a Hyperparameter Tuning Job using the Vertex AI SDK (boda2/tutorials/vertex_sdk_launch.ipynb). We finalized model selection for Malinois by benchmarking candidates on the validation set using predictions calculated as described in the next section. All test set benchmarking was retrospective and did not impact decision making in the study.”

R1C3. An ablation study should be performed on Malinois to identify key components that led to high predictive performance and good sequence design. Further model exploration should be done to improve efficiency of the model and remove any unnecessary components that don't specifically improve model performance.

While isolating the impact of individual modeling decisions on performance is of interest to the field, our primary goal in designing Malinois was to develop a model specific to our MPRA data that was sufficient for sequence design, not necessarily a comprehensive search for the optimal predictor. Our subsequent reliance on high-throughput assays to test our predictions also underlied our decision to rapidly develop our models. Currently, we have generated a sizable amount of experimental validation on top of this current model. Making post-hoc adjustments to Malinois would thus unlink the model from all of the downstream data generation and analysis. Similarly, it would not change the major finding from this study, that synthetic CREs can drive more cell type-specific activity than natural CREs.

In the prior response above, we clarified in the results and discussion that Malinois is not uniquely behind our ability to design synthetic sequences. However, the algorithmic selection of modeling parameters by Bayesian Optimization does provide key insights into which modeling choices are most effective (**Supplementary Figure 3**). We also included a point in the discussion that focused improvement on both modeling and data, and ablation studies to interrogate the mechanism underlying model performance, are useful future goals.

Line 597:

“Applying MPRA in additional cell types with greater clinical relevance and training new models on these data could enable CODA to better design CREs with specificity tailored for therapeutic applications. As the technology underlying sequence-to-function models continues to evolve, are mechanistically interrogated through ablation studies, and are trained on high-quality MPRA data sets, we expect synthetic element designs to become even more reliable and reduce the experimental burden for in vitro and in vivo validation.”

Sequence space searching algorithms: Overall, readers will want to not just know if the pipeline works but also why the pipeline works. A more careful evaluation of the sensitivities to various aspects of the CODA pipeline should be presented.

R1C5. The sequence design algorithms have significant limitations – they are all local search algorithms. The descriptions of the guarantees should be written more precisely so as not to be misleading.

Improved descriptions of sequence design algorithm guarantees

We agree with this review that each sequence design algorithm has specific benefits and drawbacks. We have included a detailed description for the reviewer of our rationale in picking these methods:

- We selected AdaLead primarily for its ease of implementation.

- We selected Simulated Annealing because it can approximate the global optimum of any function given enough iterations (though it is intractable to conduct enough iterations to confidently identify the global optimum), and has a decades-long history of successful use for non-convex optimization in a wide range of domains. We now specifically highlight this limitation.
- We selected Fast SeqProp for its ability to exploit the structure of deep-learning models while retaining the ability to pass true one-hots to the model via the straight-through estimator.

We summarize these details in the **Methods** section:

Line 705-745:

“Fast SeqProp³⁶ was selected as a representative gradient-based local optimization method that exploits the structure of deep learning models to conduct greedy search while retaining the ability to pass true one-hot encoded inputs to the model.”

“AdaLead³⁷, another greedy search algorithm, was selected as a representative evolutionary optimization algorithm for its ease of implementation and previously reported success in DNA sequence optimization.”

“Simulated Annealing⁶⁵ was selected as a representative probabilistic optimization algorithm based on a decades-long history of successful application to a wide range of domains for non-convex optimization. Simulated Annealing starts by jumping between regions with different local optima by occasionally accepting proposals that deteriorate the objective when the sampling temperature is high early in the algorithm. In later stages, the algorithm shifts toward greedy hill climbing as low sampling temperatures only allow proposals that improve the objective to be accepted.”

We have also updated **Main** and **Method** text to better communicate our design choices:

Line 160:

“We implemented three algorithms representative of three broad classes of optimization techniques (evolutionary: AdaLead³⁷, probabilistic: Simulated Annealing⁶⁵, and gradient-based: Fast SeqProp³⁶) for sequence generation. We selected these methodologies based on their ease of implementation, prior documented successes, or their ability to exploit the structure of deep-learning models.”

We also realized in response to the reviewer's other comment (R1C24) we were unclear with our assertion that effectively being restricted to local search is a major limitation of sequence design. To our knowledge, no search algorithms within the scope of this project are capable of searching the 4^{200} possibilities. We have stated this more plainly now:

Line 65:

“Lastly, although computational models are millions of times faster than experimentation, they are incapable of global searches over all possible sequence combinations within the size of a typical human CRE.”

Finally, we have explicitly included text in the discussion to make it clear there are no guarantees we will identify the optimal sequence and that conclusions we make are based solely on the specific sequences tested:

Line 553:

“Due to the intractability of fully searching sequence space, CODA cannot assuredly identify global specificity maxima, but our exhaustive evaluation of natural sequences demonstrates the design methods we used can identify synthetic sequences that regularly outperform natural ones with 1000-fold greater efficiency compared to previous methods using a zero-order Markov approach (Supplementary Figure 38)^{41,42}.”

R1C6. The evaluation of the sequence design methods is lacking; they are grouped together under the label, synthetic (Fig. 2c). Does one method work better than another? How similar are the sequences generated from each generation method? Do they fall into similar functional programs for instance?

Our initial manuscript collapsed all the synthetic sequences together due to their overall similar performance, especially compared to natural sequences. While we did include some comparisons in the supplement, we agree that a deeper comparison is warranted. A comprehensive analysis of these and other design methods would be beyond the scope or goals of this manuscript, but we have replotted our data and performed additional analyses to address this question using the tools in the original paper. We summarize our comparison of sequence design methods in three major areas: (1) Sequence design method specificity performance comparison; (2) Sequence design method impact on functional programs; (3) Differences between design methods in off-target activity

Sequence design method specificity performance comparison

We added **Supplementary Figure 17** which displays the same results as the combined synthetic sequences in figure 2c but with synthetic sequences displayed separately. We also note **Supplementary Figures 18** and **19** summarized cell type specificity of sequences separated by design method. We now highlight these observed differences across design methods in the **Main** text describing slight improvements in MinGap from sequences designed by Fast SeqProp.

Line 303:

“Between design methodologies, Fast SeqProp demonstrated greater consistency and slightly higher MinGap across all cell types (Mean MinGap difference Fast SeqProp: 0.41 over Simulated Annealing, 0.62 over AdaLead; $p\text{-adj} < 10^{-300}$, Tukey's HSD test).”

Sequence design method impact on sequence diversity

We now provide a detailed analysis of how design methods impact sequence diversity, which has been substantially updated in response to the next reviewer comment, R1C7. We describe the findings in depth there as well as the updates to the text. Overall, we find sequence content of synthetic elements is primarily determined by the target cell line, not design method, and generally distinct from natural CREs (**Supplementary Figure 13**).

Sequence design method impact on functional programs

While it is possible to show significant differences in program usage between the synthetic design methodologies, the more striking observation is that synthetic sequences share similar trends of program

usage compared to natural sequences (**Supplementary Figure 28b**). We make the following update to the **Main** text to reflect this view:

Line 437:

“While there are quantitative differences in program preference between the different synthetic sequence design methods, there are no programs unique to one method.”

Differences between design methods in off-target activity

As part of our response to R2C2, we studied the robustness of sequences to remain cell type-specific when introducing 2 additional cell types into the analysis that were not originally modeled by Malinois or otherwise used during sequence design. Intriguingly, Fast SeqProp generated sequences with notably higher retention of overall cell specific function when considering two new cell types. It is unclear if this pattern is coincidental or due to the specific qualities of the design algorithms, but would be a useful feature if shown by future studies to generalize. We note this finding in the **Results** section and discuss comprehensive design method comparison as important future work:

Line 450:

“Selected synthetic CREs drive desired tissue-specific activity in vivo
*We next sought to assess if the specificity of synthetic CREs would generalize beyond the initial three cell lines used for design. To determine if low off-target activity is maintained in additional cell lines we trained two new CNN models for A549 (lung epithelial cancer; prediction Pearson’s $r = 0.78$) and HCT116 (colon epithelial cancer; prediction Pearson’s $r = 0.84$) cells, which were not included in the original model used for CODA (**Supplementary Figure 30a-d, Methods**). Synthetic CREs maintained maximum activity for their target cell type after inclusion of A549 and HCT116, especially those generated using Fast SeqProp (**Supplementary Figure 30e-h**).”*

“Supplementary Figure 30. MPRA models for A549 and HCT116 predict synthetic CREs. Additional MPRA measurements were made in A549 and HCT116 for 318,247 and 442,482 elements and used to model CRE activity in these cell lines, respectively. (a-b) Pairplot showing distribution of activity for sequences measured in (a) A549 and (b) HCT116 and other cell types. (c-d) A model trained on sequences with (c) A549 and (d) HCT116 measurements with the same settings as Malinois accurately predicts MPRA measurements of CRE function. Scatterplots show model performance on held out test data. (e) Predicted activity of K562-targeting CREs across 5 cell lines. CREs are separated into frames based on design methodology. Text inset indicates percentage of CREs where the intended target had the highest prediction before and after A549 and HCT116 predictions were considered. (f) Same as (e) except for HepG2-targeting CREs. (g) same as (e) and (f) except for SK-N-SH-targeting CREs. (h) On-target predicted activity of CREs summarized by minGap before and after A549 and HCT116 predictions were included in the calculation. Each frame collects CREs from the five frames to the left. Each box represents CREs from a different design method.”

R1C7. Evaluating the diversity of sequences is tricky. The authors propose a k-nearest neighbor approach. Other approaches should be performed to improve robustness of the study, including a comparison of sequence context, such as k-mer frequencies of a set of sequences generated by each design method.

Expanding sequence diversity analysis

We agree that assessing sequence diversity is challenging and appreciate the reviewer's helpful recommendation for expanding this analysis. In the previous version of the manuscript, we used k-nearest neighbor distances between sequences to avoid intractably large pairwise distance comparisons when summarizing diversity of groups of sequences. We used two metrics to measure sequence-to-sequence distance for this analysis shown in the original **Supplementary Figure 12**: A) Levenshtein distance, which counts the number of single nucleotide edits between elements, and B) L1 distance between 7-mer content of sequences.

Per the reviewer's suggestion, we have also included two new analyses comparing k-mer content between natural and synthetic (non-penalized) CREs:

- First, we summarize the frequency distribution of all possible 4-mers in each group of sequences and measure the pairwise L1 distances between these group-level distributions (**Supplementary Figure 13a**). We find groups of synthetic sequences are closest to one another if they target the same cell type while groups of natural sequences are closest regardless of targeted cell type.
- Second, we visualize the complex structure of the 4-mer content of sequences using a 2-D embedding by UMAP (4-mers chosen over 7-mers due to tractability of computation on 256 versus 16384 dimensions). We confirm synthetic elements display divergent and specialized 4-mer usage compared to random 200-mers (**Supplementary Figure 13b-c**). Natural (**Supplementary Figure 13d-e**) and synthetic sequences (**Supplementary Figure 13f-i**) show distinct embeddings compared to random oligos. Notably, all synthetic element design methods generate sequences with similar 4-mer content with AdaLead generating sequences with the qualitatively tightest 4-mer content distribution (**Supplementary Figure 13i**). These embeddings show 4-mer content of synthetic elements is largely determined by target cell type.

We have included these updated and new results in the manuscript with the following **Results** text and in **Supplementary Figure 13**:

Line 261:

*“Finally, embedding the 4-mer content of the sequences into two-dimensions using UMAP we observed synthetic elements separated by target cell type and from natural elements (**Supplementary Figure 13a-i**) supporting the observation that the synthetic sequences are distinct to sequences found in the human genome ⁶⁸.”*

“Supplementary Figure 13. Variation in 4-mer content between natural and synthetic cell type specific elements. (a) L1 distance between groups of designed CREs based on marginalized 4-mer frequencies in each group. (b) UMAP embedding of all non-penalized CREs in the designed cell type specific sequence element library colored by synthetic (pink) or natural (blue) provenance. (c) 12,000 random 200-mers embedded in the same UMAP as (a). (d) The subset of points in (a) that are natural CREs selected to be cell type specific based on DHS or Malinois predictions, colored by target cell type. (e) A kernel density estimate from the natural CREs in (d) but recolored by if the element was selected using DHS (orange) or Malinois (green). (f) The subset of points in (a) that are synthetic CREs, colored by target cell type. (g) A kernel density estimate from synthetic CREs designed by Fast SeqProp, colored by target cell type. (h) Same as (g) except from CREs designed by Simulated annealing. (i) Same as (g) except CREs designed by AdaLead. The UMAP region containing 90% of random sequences is indicated by a gray line in (d)-(i).”

R1C8. The description of motif penalization is confusing. The description should be made clearer.

Additional methodological details regarding motif penalization

We extensively modified the explanation of the motif penalization in the **Methods** section with a focus on clarity. We have also included an explicit form of the penalization term that is added to the objective function, and explained the process more concisely.

Line 747:

“Motif Penalization

In order to design a batch of sequences penalizing the enrichment of given motifs in the batch, we introduced to the loss function an additional term explained below. To penalize a single motif of length l , we construct the motif PWM (position-weight matrix, a.k.a. Position-Specific Scoring Matrix, or log probabilities) and use it to score all possible subsequences x_j of length l in the batch. Let $s_j = PWM(x_j)$ be the motif score of the subsequence x_j , n the number of sequences in the batch, and t a score threshold. Then, we define the motif penalty as

$$\frac{1}{n} \sum_{j:s_j \geq t} s_j$$

where j iterates over all the possible subsequences including their reverse complements. In other words, we sum all the motif scores above the score threshold and divide by the size of the batch. When penalizing m motifs, the term we introduce is very close to simply averaging the m motif penalties, except that we introduce a weighting factor for each motif penalty to emphasize the penalization of motifs with lower indices (or in our case below, to prioritize motifs based on their order of inclusion to the motif pool). If we let $s_j^{(i)} = PWM^{(i)}(x_j)$ be the motif score of motif i of the subsequence x_j , and $t^{(i)}$ the score threshold of motif i , then the total motif penalty given a motif pool $\{PWM^{(1)}, \dots, PWM^{(m)}\}$ is defined as

$$\frac{1}{mn} \sum_{i \in [m]} (m - i + 1)^{\frac{1}{3}} \sum_{j:s_j^{(i)} \geq t^{(i)}} s_j^{(i)}$$

where the term $(m - i + 1)^{1/3}$ is the weighting factor increasing the value of the motif penalties with lower index i .

We used this motif penalty expression to iteratively design sequences subject to an increasing pool of motifs. We call these iterations penalization tracks. A single penalization track starts with the generation of a batch of 500 (non-penalized) sequences, which is then analyzed for motif enrichment (top 10 motifs of length 8 to 15) using STREME via a python wrapper function. We collect the top motif $PWM^{(1)}$ from the analysis and design a second batch of 250 sequences (which we call round-1 penalized sequences) penalizing the motif pool $\{PWM^{(1)}\}$. Then we extract the top motif $PWM^{(2)}$ enriched in the round-1 penalized sequences and design a third batch of 250 sequences (round-2 penalized sequences) penalizing the motif pool $\{PWM^{(1)}, PWM^{(2)}\}$. We continue this process till we generate 250 round-5 penalized sequences penalizing the motif pool $\{PWM^{(1)}, PWM^{(2)}, \dots, PWM^{(5)}\}$.

We generated 4 penalization tracks for each target cell type, for all three cell types. We defined the score threshold for each motif as a percentage of the motif score of its consensus sequence. The percentages used were 0 for K562-target sequences, and 0.25 for HepG2- and SK-N-SH-target sequences. The reason behind the different choice for K562 is that we found that the optimization process could more easily escape the penalization of GATA by still using suboptimal instances of the motif, so a more stringent penalty was of interest for us. The motivation for using a weighting factor was that we hypothesize that sequence design

optimization gravitates more strongly to motifs captured in enrichment analyses of early penalization rounds, so we seeked to keep emphasizing the penalization of motifs extracted from earlier rounds.”

R1C9. How sensitive are the search algorithms to their hyperparameter choices?

Sequence design hyperparameter testing

As an extension of our new *in silico* analysis of hyperparameter choices in **Supplementary Figure 3** assessing Malinois performance, we now also include analysis of how parameter choices impact the generation of K562-specific CREs using two search algorithms: Fast SeqProp and Simulated Annealing. Overall, our hyperparameter selections for the design of *in vitro* tested sequences reflect our desire to balance high predicted cell-specific activity with diversity in the generated sequence proposals. Our implementation of AdaLead utilized all the algorithm’s default parameters except for the parameter governing sequence batch size. As an evolutionary algorithm, AdaLead generates a sequence generation consisting primarily of offspring sequences derived through mutation or crossover from a pool of parent sequences. Upon completion of a run, the resulting batch typically contains sequences that can be organized into families of closely related sequences. Therefore, in the interest of maximizing sequence diversity, we decided to retrieve only one sequence per run, so setting the sequence batch size to a low value (20) was appropriate for our design goals. Due to the extended time it would take to deploy to generate comparable sequence diversity as the other two methods, we did not explore hyperparameter settings with AdaLead.

We visualize the impact of these choices on estimated activity in each cell type, predicted minGap specificity, 4-mer variety, and GC content in a new **Supplementary Figure 7**. In summary, we find:

- Sequence generation by Simulated Annealing to be very robust to all hyperparameters tested.
- Fast SeqProp is robust to most of its hyperparameters, but there can be a tradeoff between high predicted MinGap and high k-mer diversity.
- Using learnable affine parameters in the Instance Normalization layer during Fast SeqProp is detrimental to producing sequences with high predicted activity and high diversity.

We have added the following to the **Main** text:

Line 166:

“We find the overall ability of these algorithms to design cell-specific elements are generally robust to hyperparameter choices. However, adjustments can be made to balance the tradeoff between maximizing the objective and maintaining k-mer diversity in the set of designed elements (Supplementary Figure 7).”

“Supplementary Figure 7. Screening sequence design hyperparameters for generating synthetic CREs. Different hyperparameter combinations for Fast SeqProp (a)-(f) and Simulated Annealing (g)-(k) were tested to generate predicted K562-specific synthetic CREs. Predicted \log_2 -fold-change, predicted minGap activity, 4-mer heterogeneity, and GC content was measured for each sequence and plotted as a function of hyperparameter choices.”

Author note: Our responses to reviewer 1's comments 13 and 10 are related, so we are combining them here.

R1C13. With regards to the functional programs analysis that are based on standard motif analysis (using MEME suite), the main concern is that these motifs may or may not be functional. An opportunity to use Malinois to annotate functional motifs using their learned sequence-function relationship seems squandered. As their oracle, this should be more informative for "functional motifs". A similar protocol using tf-Modisco followed by contribution-weight matrix scans could identify functional motifs and annotate each sequence.

R1C10. Many claims are based on observational motif analysis. It is not clear that motifs and motif combinations identified from basic motif enrichment analysis is directly tied to functional use. To validate the motif claims, ideally, motifs and motif combination should be embedded in random backgrounds and measured experimentally in follow up MPRA. In silico validation with Malinois could suffice if the in silico experiments are performed systematically and are comprehensively. Such interventional experiments are becoming the norm in model interpretability in genomics. Any hypotheses of what motifs/motif combinations are important can be directly tested in a controlled environment to uncover the quantitative effect sizes. [PMID: 33603233; PMID: 33983921; PMID: 35551305]

As R1C13 suggested an improved method for nominating motifs, which in turn impacts all future analysis with those motifs, we have combined and reordered the two reviewer comments.

(R1C13) Functional motif discovery and mapping using TF-MoDISco and nucleotide contributions

In response to the reviewer's recommendation, we substituted our initial approach which relied on the MEME suite and instead leveraged our contributions scores with TF-MoDISco to uncover motif patterns perceived as functional by Malinois. We identified a set of non-redundant, ungapped, core motifs that account for all of the TF-MoDISco motif combinations and patterns, as well as for repressive motifs present in the original STREME motif list that align well with contribution scores but that were not found by TF-MoDISco. We used hypothetical contribution-score matrix scans to nominate sequence matches to these core motifs, aggregating Pearson correlation coefficients of the scans as motif scores. This updated analysis is described the **Main** text:

Line 342:

*“First, we used Malinois to predict nucleotide-resolution activity contribution scores for each sequence in the three cell types using a modified version of Integrated Gradients (**Methods**)⁶⁹. We consistently observed that disrupting blocks of positive contribution led to a decrease in predicted activity, while disrupting blocks of negative contribution resulted in an increase (**Supplementary Figure 21, Methods**). This alignment with expected prediction effects supports the functional relevance of the contribution scores as perceived by the model. Next, we employed TF-MoDISco Lite^{70,71} to identify 66 motif patterns informed by contribution scores, from which we extracted 36 non-redundant core motifs (7-18 bp) enriched in our MPRA-tested library, with 31 confidently aligning to a known human TF binding motif (**Methods, Supplementary Tables 5 and 6**)^{72,73}.”*

We now describe our new motif discovery and *in silico* interventional experiments in the **Methods**:

Line 959:

*“**Motif discovery**
We used TF-MoDISco Lite^{70,71} to extract sequence motifs to be predicted as functional by Malinois through contribution scores obtained through Sampled Integrated Gradients (SIG). As described above, SIG naturally provides hypothetical contribution scores (as defined by*

TF-MoDISco) when selecting the uniform random background by simply carrying out the equivalent of the full process minus masking out using the input sequence one-hot matrix. The final contribution scores can then be retrieved masking out the hypothetical contribution using the input sequence one-hot matrices, as required by TF-MoDISco. We computed hypothetical contribution scores for each of the three prediction tasks and ran TF-MoDISco Lite with 100,000 seqlets and a window size of 200 (equivalent results were obtained using 1,000,000 seqlets). We aggregated the discovered patterns across prediction tasks following their provided example using `modiscolite.aggregator.SimilarPatternsCollapser`. TF-MoDISco Lite results are provided as positive and negative patterns.

TF-MoDISco patterns to PWMs

To convert a TF-MoDISco positive pattern living in the hypothetical-contribution-score space into a Position-Weight Matrix (PWM), we divided the pattern scores by the maximum position score sum and multiplied by 10. To obtain the Position-Probability Matrix (PPM) we applied the Softmax function to each position vector. Some of our TF-MoDISco negative patterns are a combination of a negative pattern (negative contributions) and a positive one (positive contributions). Thus, in order to convert a TF-MoDISco negative pattern into a PWM, we first reversed the sign directionality of the negative portions (as informed by the pattern scores living in contribution-score space, not hypothetical) and compensated their magnitude by multiplying by 1.2 (because our negative contribution scores are in general smaller in magnitude than positive ones perhaps due to the nature of the training data target distribution that has a positive bias). Then, we proceed as with the positive patterns.

Core motifs (TF-MoDISco)

Since TF-MoDISco, in addition to capturing isolated ungapped motifs, is able to capture patterns that are combinations of motifs, we heuristically extracted core ungapped patterns that, to varying degrees, account for all the combinations observed in the TF-MoDISco merged results. To manually define the starts and stops of core motifs, we relied on scoring the full pattern PWMs against themselves using TOMTOM⁹⁷, information content contours, and visual examination. The core motif IDs are derived from the IDs of the original patterns from which they were extracted. To convert the patterns into PWMs and PPMs, we applied the same operations as described above. Matches to human known TF binding motifs were assigned using TOMTOM with default parameters against the databases JASPAR CORE (2022)⁷¹ and HOCOMOCO Human (v11 FULL)⁷².

Core motifs (STREME)

In addition to extracting sequence motifs with TF-MoDISco, we also performed a motif enrichment analysis using STREME. First, to assess the agreement between a given STREME motif and its predicted functionality as measured by contribution scores, we weighted-averaged the hypothetical contribution scores corresponding to all the sequence segments determined to be a match to the motif (as provided by FIMO with default parameters, using motif scores as weights), and compared the score averages (one set of averages per each prediction task) to the motif's Information-Content Matrix (ICM). We will refer to the weighted average hypothetical scores as the "contribution-score" projection. All motifs with overall positive contribution scores that had a strong agreement with their contribution-score projection had been already captured by TF-MoDISco, suggesting that the TF-MoDISco positive pattern results are very comprehensive. However, we found a small number of STREME motifs with negative

contribution scores that had a strong agreement with their contribution-score projection, so we decided to include them to the list of core motifs. It is worth noting that these motifs had negative contribution scores with moderate-to-low magnitude. We speculate that the reason TF-MoDISco might not have been able to detect them is because the contribution allocated in the seqlets that would correspond to these motifs too often falls below the threshold of the distribution of negative scores, making it hard to discriminate them from noise or insignificant scores. Running TF-MoDISco with 1M seqlets did not change the results. We retrieved 11 such STREME motifs with strong agreement with their contribution-score projection not captured by TF-MoDISco, 9 of which were clustered together into 3 groups with nearly identical contribution-score projection (up to 1 or 2 additional positions to the left or right). This gave us a total of 5 STREME negative patterns in contribution-score projection form that were included to the list of core motifs. Their conversion to PWM and PPM forms followed the same process as with the TF-MoDISco patterns. Matches to human known TF binding motifs were assigned using TOMTOM with default parameters against the databases JASPAR CORE (2022)⁷¹ and HOCOMOCO Human (v11 FULL)⁷².”

(R1C10) *In silico* interventional motif analyses

We agree with the reviewer that our previous observations on motifs were based on enrichment analysis, and that interventional experiments could provide insight into the causal role of sequence content on function. We start this analysis using the non-redundant core motif set extracted from the motif patterns nominated by TF-MoDISco as described above, to focus on functional motifs.

Per the reviewer’s recommendation, we conduct a comprehensive *in silico* analysis testing the predicted impact of *embedding nominated functional sequence motifs* into random sequences, similar to the approaches used in the references cited by this reviewer. We do this for all core motifs as well as the for the original TF-MoDISco motif patterns (**Supplementary Figures 22c and 23c**). We find strong concordance with predicted motif function based on contribution scores (**Figure 3b**). Additionally, we perform an orthogonal analysis measuring the predicted activity impact of disrupting motif instances present in our sequence library for both sets of motifs. Again, we observe that all motif contributions were in close concordance to their effects when ablated (**Supplementary Figures 22d and 23d**).

We describe the results of the motif embeddings and ablations in the main text as follows:

Line 361:

*“All motifs demonstrated predicted effects in accordance with their assigned contribution when embedded in a random background, as well as when replacing their instances in the library with random sequences (**Supplementary Figures 22 and 23, Methods**).”*

As this is a large change from our initial manuscript, we have made extensive additions describing our new methodology.

Line 1041:

*“Motifs embedded in random background
We embedded single motifs in random sequences to measure their standalone predicted effect compared to fully random sequences. For each motif, we built a 200x4 Position-Probability Matrix (PPM) consisting of the motif’s PPM in the middle and random background ([0.25, 0.25, 0.25, 0.25]) everywhere else. We sampled 5000 sequences from it and fed them to Malinois to*

obtain predictions in each cell type. We also sampled 5000 sequences from a 200x4 PPM of uniform background everywhere (no motif in the middle), and fed them to Malinois to serve as baseline.

Motif ablation

We sought to assess the predicted effect of disrupting all instances of a single motif in our sequence library. For each motif, we collected the particular batch of sequences that had at least one instance of such motif, replaced all the instances with random segments (sampled from uniform background), and fed them to Malinois to obtain predictions in each cell type. We performed this step 5 times, averaged the 5 predictions of each disrupted sequence, and subtracted from the average the batch's original predicted activities to obtain the predicted disrupting effect. For example, say that a sequence has one instance of a given motif in positions 20-32. We inserted a random sequence segment in those positions and got the disrupted sequence's predictions. We did this 5 times, so 5 different random segments (with 5 different predictions) in positions 20-32, and averaged the 5 predictions (to mildly marginalize potential effects of replacing with random segments). The disrupting effect would be this average prediction minus the sequence's original predicted activity. We aggregated the disrupting effects by motif presence (as defined above in the last paragraph of motif penalization in this section). To find instances of core motifs, we used the contribution score-based motif hit method described above. To find instances of the original TF-MoDISco patterns, we used FIMO (with the default parameters), since our contribution score-based motif hit method might not handle gapped patterns as well as FIMO. When submitting the pattern PPMs to FIMO, we trimmed the patterns at both ends such that the start/stop of the pattern is the first/last position to have an information content of at least 0.15 bits."

(R1C13) Updating functional program analysis with additional, new TF-MoDISco nominated motifs

Subsequently, we employed these matches in a completely updated NMF functional program analysis, now based on this reviewer's apt suggestion to have Malinois directly annotate functional motifs. This is described in the main text as:

Line 427:

*"To aggregate semantically-related motifs into functional programs, we used Non-negative Matrix Factorization (NMF)⁸⁵ to decompose sequences in our library into a mixture of 12 functional programs based on motif content **calculated using contribution score-based motif mapping (Supplementary Figure 27, Methods).**"*

We now describe our new motif mapping strategy for NMF extensively in the **Methods**:

Line 1026:

"Contribution score-based motif hit mapping

To find instances of the core motifs present in the CODA sequence library, we leveraged the hypothetical contribution scores of the sequences to match sequence segments to the core motifs in hypothetical-contribution-score form. First, we padded with zeros left and right all the sequence hypothetical contribution scores, yielding a matrix of dimensions 3x75000x4x210. Second, for a core motif of length l, we computed all the Pearson correlation coefficients between every possible subsequence hypothetical contribution scores of length l (matrices of size 75000x4xl) and the core motif's hypothetical contribution scores in forward and reverse

complement orientations. For each cell type dimension, we randomly sampled 500,000 Pearson correlation coefficients (arising from a single core motif) to obtain the value $\min(0.75, \mu + 4\sigma)$ to serve as a coefficient threshold, where μ , σ represent the mean and the standard deviation, respectively, of the subsampled distribution. All subsequences for which their hypothetical contribution scores scored above their coefficient threshold were collected as motif hits for the given core motif. We repeated this process for all core motifs across all cell types.”

R1C11. A robust modified version of Integrated gradients was used to "identify magnitude and direction of effect of each motif in each of our three cell lines. The description needs to be clearer in the methods and evidence should be provided why it is more robust. Generally, it's not clear what value this robust modified version of IG is? Does it provide insights that couldn't be achieved with standard attribution analysis?"

We regret that a clear methodological description motivating the value of Sampled Integrated Gradients was not clearly described in the methods. We have added the text below in the **Methods** highlighting the practical advantages and value of our adaptation. We have also omitted the qualifier “robust” in the main text since that claim would require a thorough study that would be out of the scope of the paper.

Line 844:

*“Sampled Integrated Gradients to compute contribution scores of Malinois predictions
We calculated nucleotide contribution scores for each sequence in the proposed library using an adaptation of the input attribution method Integrated Gradients⁶⁸. Sampled Integrated Gradients considers the expected gradients along the linear path in log-probability space from the background distribution to the distribution that samples the input sequence almost surely. In each point of the linear path, a sequence probability distribution (a.k.a. Position Probability Matrix) is obtained from the log-probability space parameters by applying the Softmax function along the nucleotide axis, and a batch of sequences is sampled from that distribution to be fed into the model. We then calculate the gradients of the batch model predictions with respect to the parameters in the log-probability space, using the straight-through estimator to backpropagate through the sampling operation. The batch gradients are averaged for each point in the path and approximate the gradient integral as in the original formulation of the method. In our case, the subtraction of the baseline input from the input of interest involves the parameters in log-probability space. This adaptation of Integrated Gradients provides two useful features. First, the sequence inputs being fed to the model are always in one-hot form, avoiding evaluations of inputs that exist off the vertices of the simplex on which the model was trained which could more easily lead to pathological predictions. Second, the original method relies on choosing an appropriate single baseline input against which to compare the input of interest which might not always be straight forward, whereas our adaptation uses a background distribution of sequences as the baseline. Favorably, when choosing the uniform background (0.25, 0.25, 0.25, 0.25), the parameters in log-probability space where the line path is traversed become the zero matrix, which removes the need to subtract the baseline from the input of interest. We can then more easily extract integrated gradients for all tokens in all positions (by omitting masking the gradients with the one-hot input), which we found useful as hypothetical scores for TF-ModISco.”*

R1C12. The motif statistics presented in the results reads as an overly confident analysis. It is not clear how motifs were defined as repressive for instance. How sensitive are these numbers given the heuristic choices to make these definitions? It could be more conservative to state what heuristic was used and give the motif statistics with that disclaimer.

Improved clarity and support of predicted motif function.

First, we designate motifs as activators or repressors based on their positive or negative average motif contributions as calculated via our implementation of integrated gradients, averaged across cell lines. We also now note this directionality agrees with the TF-MoDISco analysis. We make the following addition to the **Methods** to describe how motif contributions are calculated and label motifs as activators or repressors:

Line 1071:

Motif contributions

To get a motif's overall contribution, we performed a weighted average of the contribution score sums contained in all the motif instances provided by our motif hit method across the three prediction tasks. The average was weighted using the motif scores corresponding to the Pearson correlation coefficients mentioned above. The overall regulatory directionality of a motif (activator or repressor) is given by the sign of the mean of the weighted averages across cell types. For all motifs, the overall regulatory directionality agrees with the original TF-MoDISco designation as a positive or negative pattern."

Sensitivity of motif analysis to analysis choices

We chose to address the question of sensitivity of our analyses to heuristic choices by expanding these analyses with additional, orthogonal experiments. How motifs function in the tested elements and the mechanistic insight into their function is an important advance in our paper. On this reviewer's previous recommendations, we have extended our analysis of motif function to embedding and ablation *in silico* analyses that should provide complementary evidence to support reported motif function statistics in **Figure 3b**. Each of these analyses are described in detail in their respective comments:

- (R1C10) Motif embedding in random sequence backgrounds (**Supplementary Figure 22 and 23**).
- (R1C10) Motif ablation from CODA library sequences (**Supplementary Figure 22 and 23**).

We highlight that the three complementary strategies for determining motif function are all in agreement with the motif statistics presented in **Figure 3b** in the **Main text** :

Line 361:

*"All motifs demonstrated predicted effects in accordance with their assigned contribution when embedded in a random background, as well as when replacing their instances in the library with random sequences (**Supplementary Figures 22 and 23, Methods**)."*

As we see strong concordance across these methodologies, the findings appear quite robust. However, while orthogonal, all three methods are *in silico*. We clarify in the main text that the supporting *in silico* motif embedding and ablation experiments are based on model predictions. When introduced, we also explicitly note that Malinois is used to derive nucleotide contributions. Additionally, we also include a caveat in the discussion and point towards the utility of future empirical analysis. :

Line 342:

“First, we used Malinois to predict nucleotide-resolution activity contribution scores for each sequence in the three cell types using a modified version of Integrated Gradients (Methods)⁶⁸.”

Line 573:

“Future empirical analysis of motif ablation or embedding could be used to further validate how the model interprets regulatory sequences and improve training.”

R1C14. Further comments for the NMF analysis on motif enrichment. The strength of motif binding sites is not considered -- weak motifs and stronger motifs are binarized as a functional motif according to FIMO, which is known to lead to many false positives. One could in principle use motif scores instead of counts for this analysis – attribution scores or the PWM scan. Also, the sensitivity of the NMF analysis should be explored based on different hyperparameters of FIMO.

Updating motif mapping in sequences prior to NMF:

We agree with the reviewer that our initial analysis could have been more sophisticated. We have also observed visually that our original FIMO approach struggles to generate expected motif matches at high stringency and quickly generates spurious motif matches when these thresholds are reduced. However, in our experience, we find NMF, and topic modeling generally, is computed from count data and worry that using alternative featurization could confound our ability to interpret the analysis. Upon further exploration based on this comment, we were especially concerned about using FIMO motif scores because PWMs with different information content will have different ranges of possible motif scores. Additionally, continuing to use FIMO motifs would require a detailed analysis of FIMO hyperparameter impacts on our NMF analysis.

In our revision, we decided to build upon our new TF-MoDISco analysis to improve the quality of our motif counting rather than use FIMO match scores. We use TF-MoDISco and contribution score-based motif scanning for motif discovery and counting (**Methods**), respectively, for NMF based on the reviewer's suggestion. TF-MoDISco constructs motif PWMs based on model derived nucleotide contribution scores, thus only identifying motifs that are predicted to have activating or repressing function. By mapping these PWMs back to sequences based on nucleotide contribution scores, we ensure hits are mapped only when the corresponding subsequence is predicted to be functional. While we cannot avoid binarization of motif hits and use NMF, we believe focusing on functional motifs improves the analysis. By mapping motif hits using PWM concordance with contribution scores, we believe we are mitigating spurious motif matches with non-functional sequences.

R1C15. The validation of the regulatory programs is lacking. It is just an observational analysis which may or may not be right. One way to prove regulatory programs are known is to perform in silico experiments - controlled motif embeddings in random background sequences to quantify that predictions behave as one expects.

Constructing sequences from NMF programs and testing with Malinois

Our original intention was for NMF to serve as observational analysis to show broad patterns of differences between groups of sequences. However, the reviewer raises an interesting question that we agree should be evaluated if we were using NMF programs to define cell specificity. We agree we lack the data to establish a causal relationship between program content and sequence function (e.g. cell type specificity). While it is clear how to embed individual motifs into sequences, as we have done in response to R1C10, we find embedding

programs into sequences non-trivial. Our method of featurizing sequences and computing NMF programs is not end-to-end, making it difficult to generate sequences with well controlled program embeddings. One way to overcome this is similar to the approach applied by Taskiran et al. which used a model trained on program assignments as the oracle for designing synthetic CREs which can then be evaluated for specificity (Taskiran et al. 2024). There is no straightforward way to do that without training a new model to approximate our NMF results and applying sequence design techniques. We look forward to future studies attempting to develop methods that could directly assess how well program assignments reflect regulatory programs at-scale.

We have now updated the manuscript to clarify our NMF is intended only to provide a broad overview of relationships between sequences tested in our study. To limit overinterpretation, we have removed our detailed analysis of differences in the content of specific programs in different groups of sequences. We instead focus on higher level patterns of synthetic elements deploying activating and repressing programs to a higher degree than natural sequences, which is in agreement with the analysis of individual activators and repressors. We reproduce the altered section below (deleted text not shown).

Line 423:

"Complex semantic architectures are syntactically differentially deployed in natural and synthetic sequences

In addition to single TF-motif usage and pair-wise co-occurrence, cell type specificity is thought to arise through higher-order motif semantics, which can mediate the complex organization of many TFs to impart CRE activity^{7,8,11,12}. To aggregate semantically-related motifs into functional programs, we used Non-negative Matrix Factorization (NMF)⁸⁵ to decompose sequences in our library into a mixture of 12 functional programs based on motif content calculated using contribution score-based motif mapping (Supplementary Figure 27, Methods). These programs broadly describe related sequences found in the elements we tested. NMF identified 5 programs associated with clear cell type-specific activity (1 program in K562, and 2 in each HepG2 and SK-N-SH), with the 7 remaining programs associated with pleiotropic activation and/or repression (Figure 3d, Supplementary Figure 28a).

Natural and synthetic sequences deploy distinct distributions of semantic programs (Figure 3e, Supplementary Figure 28b). While there are quantitative differences in program preference between the different synthetic sequence design methods, there are no programs unique to one method. Overall, synthetic elements have higher program content and program heterogeneity compared to natural CREs (Supplementary Figure 29a-b). We also found that natural sequences primarily rely on activating programs while synthetic sequences also frequently utilize programs with repressive effects in off-target cell types (median repressing program content: DHS-natural 0.077; Malinois-natural 0.064; synthetic 0.123) (Supplementary Figure 29c,d). The vast majority of synthetic sequences (91.9%) are composed of both activating and repressing programs each exceeding a threshold of 0.1, while relatively fewer DHS (26.9%) and Malinois (25.3%) natural sequences show this combination (Methods, Supplementary Figure 29e). These results support our motif-based observations that the improved performance of synthetic sequences is due to a combination of on-target activations and off-target repression."

R1C16. "to improve interpretability of the topic modeling, we generated an additional 4000 sequences for each cell type which prioritized off-target expression." It is not clear whether the presence/absence of these sequences improves interpretability.

We originally found the sequences helped NMF more clearly organize motifs associated with activators and repressors into their own programs. However, this has been removed after the addition of TF-MoDISco based motif annotation which more clearly identifies functional motifs.

R1C17. With regards to: "The regulatory activity contribution scores identify the overall magnitude and direction of the effect of each motif in each of our three cell lines" Attribution methods quantify nucleotide importance, not motif importance.

Aggregating nucleotide-resolution contribution scores into motif contributions

We regret not having clarified how we quantify motif importance. We agree with the reviewer that attribution methods quantify nucleotide-resolution importance. To address this, we have added to the methods section the text below explaining how we aggregate nucleotide contributions into motif contributions.

Line 1071:

"Motif contributions

To get a motif's overall contribution, we performed a weighted average of the contribution score sums contained in all the motif instances provided by our motif hit method across the three prediction tasks. The average was weighted using the motif scores corresponding to the Pearson correlation coefficients mentioned above. The overall regulatory directionality of a motif (activator or repressor) is given by the sign of the mean of the weighted averages across cell types. For all motifs, the overall regulatory directionality agrees with the original TF-MoDISco designation as a positive or negative pattern."

General concerns

R1C18. Reproducibility is a major concern. The code repository is not organized well, there is no roadmap for how to execute code, and the code is not commented well. Code should be provided to: 1. take supp table 2 and processes it into training, validation and test sets; 2. build and train Malinois on this dataset; 3. execute each sequence design method; and 4. run inference on a trained model to replicate the figures in the main text. Also, the processed dataset and synthetically generated sequences should be saved as npz or h5 or even pickle.

We appreciate the reviewers' concerns about reproducibility and organization. We have made major updates to the codebase since the original submission. These include adding:

- Code comments for most functions and classes in the submodules of our library.
- A tutorial notebook to interactively load/pre-process data and train a new model using modules from the library.
- A terminal command example to also execute the same as point 2.
- A notebook to deploy hyperparameter optimization on GCP's Vertex AI.
- A notebook to load trained models for interactive inference.
- A terminal example to run sequence generation with each design algorithm.
- A tutorial notebook to interactively combine models with design algorithms.

Additionally:

- The processed dataset and synthetically generated sequences are provided as text-based supplementary tables and have already been analyzed by at least one independent group (Gupta, Lal, Gunsalus & Biancalani et al 2023 biorXiv). We are happy to provide these data in additional formats as supplementary information at the reviewer and Journal’s discretion.
- The containerized deployments on a public Google Container Registry for training and inference also offer options for readers to ensure reproducibility.
- We are committed to continue working to improve usability of the codebase.

If the reviewers or the journal suggest other ways to disseminate our results and methods, we would be happy to utilize those options before publication.

R1C19. It is not always clear what is novel by the authors versus what was already known or previously established. Being clearer on this in the text could be beneficial.

We appreciate that we need to be more clear in delineating what is novel in our work versus established approaches. A major comment from another reviewer (R3C6) requested a detailed comparison to existing, related methods. In that response we comprehensively highlight what is novel in our work and highlight changes to the text to clarify.

R1C20. A more thoughtful description of limitations should be included. For instance, modeling based on MPRA, sequence design algos, pathological behaviors outside the data distribution (i.e. out-of-distribution). These of course were not fully explored but could at least be mentioned as a warning to others that blindly think that DNNs that make good predictions on test data can generalize everywhere in sequence space.

We agree with the reviewer that readers should be aware of the potential limited generalizability of deep learning models. We now introduce the concept of unreliable predictions for examples with extreme divergence from the training data in the **Methods** text and explain our strategy for mitigation:

Line 698:

“To prevent pathologically extreme activity predictions that are common to deep learning methods when computing on sequences highly divergent from the training data, we constrained predictions to a limited interval (default: [-2, 6]) when generating sequences.”

We also highlight out-of-distribution detection as a future aim that will improve the reliability of model guided sequence design in the **Discussion**:

Line 603:

“With increasingly complex models, it will be essential to determine the bounds of reliable predictions across sequence space to ensure synthetic sequence designs are not based on pathological model predictions.”

R1C21. There is no exploration for which components of CODA/Malinois and so it is not clear what enabled these results. For instance, is MPRA design better than training a model on chromatin accessibility sites from ATAC-seq for these 3 cell types? This is a very different question than what was probed in the study, which compares natural accessible sites.

Overall, our goal is to identify elements that can drive transgene expression, making MPRA, with its similar episomal nature, an ideal proxy. We have better clarified in the main text to state our specific goal.

Line 157:

“Here, our goal is to design CREs that drive cell-specific transcription in one of the modeled cell lines, as measured by MPRA.”

We also specify in the discussion that our results are limited to our specific system:

Line 551:

“Synthetic sequences designed by CODA easily outperform natural sequences in driving cell type-specific gene expression in a reporter system”

However, how well open chromatin models could be used for synthetic sequence design is an interesting question we expect many readers may have. Chromatin accessibility has been shown to be only moderately correlated with gene expression. DNA-seq’s first papers showed that correlation with a relatively few handpicked genes was at best Spearman’s $\rho = 0.744$ (Boyle et al. 2008). The most current indexes of DHS sites do not attempt to correlate proximal DHS signal to RNA levels (Meuleman et al. 2020), and instead are most useful for categorizing active vs inactive genes. ATAC-seq is highly correlated $R=0.79-0.83$ with DNase-seq (Buenrostro et al. 2013) and again has been shown to have low correlation ($r = 0.2-0.35$) between signal at promoters and gene expression (Nair et al. 2021). As briefly mentioned in R1C1, several studies have shown MPRA has only modest correlation with signals of chromatin accessibility (Kim et al. 2021). This could be due to predictions being directly on a specific 200 bp of sequence, measuring CREs outside of the endogenous context of the genome, nearby sequences repressing on-target activity or inducing off-topic expression, cooperative effects of multiple CREs acting on a specific promoter, and other endogenous impacts may explain the difference between open chromatin measures and both empirical and predicted reporter results. Because our goal is to drive transgene expression, we focused experimental resources towards an MPRA-based model that could be used for reliable synthetic sequence design.

While we do not test if a chromatin accessibility model can design sequences with the desired activity in MPRA, we show that Malinois provides more accurate predictions for synthetic sequences in MPRA than Enformer predictions of DHS, CAGE, H3K4me3, and H3K4me1 predictions in K562, HepG2, or SK-N-SH (Reviewer Figure R1C1). This suggests even a state-of-the-art chromatin architecture model might be poorly suited for designing CREs for transgenic applications in human cells. We believe this is also consistent with our observation that natural CREs nominated by DHS as cell type-specific underperform compared to Malinois-nominated elements. However, synthetic sequences built using these models have been demonstrated to be successful at designing CREs for different applications (de Almeida et al. 2024; Taskiran et al. 2024) and future work will be required to identify the best approach for specific applications.

R1C22. The authors claim that CODA is generalizable and extendable. I can see how CODA is generalizable and extendable in principle. But the framework is not novel. It is standard practice to use oracles and have either search algos or ML models to help navigate the oracle's functional landscape. This is done quite commonly in ML-guided protein function landscapes. The code itself is far from a usable toolkit or an extensible framework that others would be able to build upon.

We agree with the reviewer that the paradigm of using ML models with search algorithms to navigate a functional landscape is not novel. Indeed, we have implemented three such algorithms that were already established in the existing literature. We present our study not as a novel ML paradigm but rather as a complete and high-throughput pipeline for the nomination of synthetic CREs in human cell lines. Rather than focusing primarily on a novel model architecture, optimization algorithm, or a specific synthetic enhancer, our intention has been to meticulously hone efforts across all steps to ensure their effective performance. The platform's novelty lies in its thoroughness and robustness as a whole with respect to the application of CRE design. We anticipate future research to build upon and refine these results.

With regards to the usability and extensibility of CODA, as mentioned in R1C18, we have incorporated notebooks featuring code examples to assist fellow researchers in navigating the CODA pipeline. Our Python library follows a modular structure, offering a collection of base classes intended as foundational building blocks for the creation of custom training and sequence design pipelines. These base classes enable the development of diverse models, facilitate the utilization of other (pre-processed) datasets, support the implementation of various optimization algorithms, and allow for other optimization objectives through the customization of the energy and parameters classes. Hence, our assertion in the text aimed to express that the CODA platform can be extended through: (i) *integrating advances in deep learning*, such as implementing other training schemes or model architectures; (ii) *conditioning models on orthogonal data modalities*, such as training models on other types of data; (iii) *modeling CRE function in more tissue types*, which he have implemented in R2C2; and (iv) *tasking different biological objectives*, such as customizing our energy functions for a different optimization objective. We acknowledge that while our platform is extendable, the process may involve a learning curve, so successful extension will require proper effort and engagement.

Minor concerns

R1C24. On line 64, meaning is ambiguous: "computational models are still only capable of characterizing a fraction of all possible sequence combinations". The CNN has learned a function that defines a value to each input sequence in the domain. So, does this refer to the poor out-of-distribution generalization or the intractability of training and querying a model on a limited subspace.

We had included this sentence as a reference to the vastly increased throughput of computational methods compared to MPRA, but that the entirety of the search space (4^{200} combination) is still inaccessible with the current algorithms. We have updated the sentence below to improve clarity:

Line 65:

"Lastly, although computational models are millions of times faster than experimentation, they are incapable of global searches over all possible sequence combinations with the size of a typical human CRE."

R1C25. "optimization guarantees" (line 154) should be clarified. The simple sequence design methods are all greedy search algorithms infamous for their lack of ability to navigate complex surfaces.

Clarifying algorithms ability to navigate complex surfaces

In our response to R1C5, we describe in detail the rationale behind the choice of the sequence generation algorithms and have removed language that used confusing terminology around optimization guarantees. We agree with the reviewer that AdaLead and Fast SeqProp are greedy search algorithms and have clarified that in the description of their use in the methods. Comparatively, Simulated Annealing can accept worse proposals

with respect to the objective function with some probability early in the annealing schedule allowing escape from local optima.

We have made sure to better describe why we chose these algorithms, and highlight these issues in the **Methods**:

Line 705:

“Fast SeqProp³⁶ was selected as a representative gradient-based local optimization method that exploits the structure of deep learning models to conduct greedy search while retaining the ability to pass true one-hot encoded inputs to the model.”

Line 721:

“AdaLead³⁷, another greedy search algorithm, was selected as a representative evolutionary optimization algorithm for its ease of implementation and previously reported success in DNA sequence optimization.”

Line 731:

“Simulated Annealing⁶⁵ was selected as a representative probabilistic optimization algorithm based on a decades-long history of successful application to a wide range of domains for non-convex optimization. Simulated Annealing starts by jumping between regions with different local optima by occasionally accepting proposals that deteriorate the objective when the sampling temperature is high early in the algorithm. In later stages, the algorithm shifts toward greedy hill climbing as low sampling temperatures only allow proposals that improve the objective to be accepted.”

Overall, the performance of these three methods is largely similar except for execution time and k-mer diversity (described briefly in R1C9). We expect future, more focused work using the design-test loop we describe in this paper will more fully explore a more broad range of sequence generator methodologies.

R1C26. Colors in Fig 1e could benefit from something that contrasts better, perhaps red. Also, the authors could play with the transparency.

We have updated this figure to take into account these suggestions, including a better contrasting red vs blue with transparency. We have included the updated **Fig 1e** panel below.

Figure 1e

R1C27. Equations should be formatted so that they are clearer (see Methods; e.g., lines 676, 652, 835, 836, etc.)

We have reformatted the equations in the **Methods** to be more clear.

R1C29. Grammar: Line 349; Line 652 (equation typo); Line 688.

Thank you for identifying these grammar issues and typos, we have corrected them.

Reviewer #2

In this manuscript, the authors use MPRA, epigenomic marks, and machine learning to design tissue-specific enhancers. This impressive body of work tests many enhancers in different cell types to create tissue-specific enhancers. They develop computational tools to predict functional tissue-specific enhancers using MPRA and epigenomic data. They compare synthetic and natural sequences' ability to drive tissue-specific expression as defined by activity in 3 cell lines - erythroid, hepatocytes, and neuroblastoma. They claim synthetic enhancers show greater cell type specificity than the predicted functional natural sequences. The authors test 6 synthetic enhancers via reporter assay in zebrafish and two in mouse, although the nature of the specificity of these enhancers within the embryos is poorly described. The number of enhancers tested in this manuscript is impressive, and the computational tools will be helpful to the research community. The design of tissue-specific enhancers is a novel and exciting area.

The premise of the approach and comparison between natural and synthetic enhancers and their ability to drive cell type-specific expression is flawed. The cell types the authors are studying are derived from cancer cell lines; thus, these are not the environment in which the genome evolved. Therefore, it makes sense that their synthetic enhancers would drive better cell type specificity than natural sequences, but this does not mean that, in general, synthetic enhancers are better at driving cell type specificity. The authors make claims about synthetic enhancers vs. natural ones, but they cannot translate their results in transformed stable cell lines to normal cell types in an organism.

We are grateful for the reviewer's feedback and pleased that they appreciate the extensive scale and importance of our study. The comments were instrumental in highlighting sections of the manuscript that required clearer articulation of our study's goals and conclusions, especially for terms used with different meanings in different fields. We are confident that the revisions made in response to the reviewers' questions have significantly improved the manuscript's clarity.

R2C1. The summary statement states that their synthetic sequences can outperform natural sequences in driving tissue-specific gene expression. It is also not clear what out-perform natural sequences means. To demonstrate that synthetic sequences outperform natural sequences, they would need to remove all enhancers contributing to the expression of a gene and replace them with synthetic enhancers, and show the organism survives. Or if there is a therapeutic goal, show that the synthetic sequence drives expression of the target gene more effectively than the natural sequence. It is likely within the genome using endogenous promoters, the natural sequences would outperform the synthetic sequences in nontransformed cells. I feel that the comparison of natural and synthetic sequences detracts from their ability to design tissue-specific enhancers.

As the reviewer notes, these studies are conducted in transformed cell lines which do not accurately reflect selective pressures on the genome. Furthermore, our system uses an episomal reporter system for both model training and experimental validation. Our use of the term 'outperform' refers not to our ability to design synthetic sequences that would increase organismal fitness or that can completely substitute all CREs associated with a single locus, but rather to our ability to design sequences that best drive cell type-specific reporter expression in the three cell lines in our study, a task that is outside the scope of pressures on the genome. We anticipate that future applications of synthetic enhancers will have similar objectives of ectopically driving transgene expression patterns, which may not have been under the necessary selective pressure. Importantly, our result demonstrates that it is feasible to generate *de novo* such sequences and that sequences which best achieve the design criteria we sought cannot (maybe not surprisingly) be discovered by mining the human genome. We realize we failed to properly define what 'outperform' refers to in our study. Outperforming

is the engineering goal of designing active (high on-target activity) and specific (low off-target activity) elements in three transformed cell lines. We have updated text in multiple areas, to make this more clear”

More broadly, we appreciate the reviewer’s comments and the opportunity to revise the text to more accurately reflect the goals and results of the study. In light of the reviewer’s comments, and upon careful review of the language used in our manuscript, we realize the certain terminology used can have distinct interpretations across the various fields we believe this work will be of interest. For example, our use of “fitness” refers to a fitness function as used in the machine learning field and more specifically in relation to evolutionary algorithms. We realize this term is well-defined with different meanings in other fields and may lead to unnecessary confusion for many readers. As a result we have now replaced “fitness” with “objective” and other alternative language throughout the manuscript.

We provide the modified sections below with new text in blue:

Updated Abstract Line 31:

*“Through **large-scale in vitro validation**, we show that synthetic sequences **are more effective at driving cell type-specific expression compared to** natural sequences from the human genome, **and maintain specificity when tested in vivo.**”*

Updated Results Line 155:

*“CODA follows an iterative loop of predicting the activity of sequences, **quantifying** how well sequences fit the design goals **using an objective function**, and then updating sequences to **increase the objective value**. Here, our goal is to design CREs that drive cell-specific transcription in one of the modeled cell lines, as measured by MPRA.”*

Updated Discussion Line 551:

*“Synthetic sequences **designed by CODA** easily outperform natural sequences in driving cell type-specific gene expression **in a reporter system**, which suggests that novel functions can be programmed into CREs and interpreted by human cells.”*

Finally, we would like to further clarify that our method is not attempting to modulate expression levels of endogenous genes. Our goal is to drive cell-specific expression of an episomal gene cassette. The reviewer is right that we expect these sequences to have relevance for a therapeutic goal, however, the target system is expression from a transgene delivered by a system such as AAV, lipid particles or other delivery methods. Because we are optimizing for transgene expression and using an episomal reporter system, we do not anticipate that our synthetic enhancers could substitute for natural CREs. Additionally, it is uncertain how synthetic enhancers might affect the overall fitness of the organism, apart from the potential benefits or drawbacks resulting from the expression of the transgene in specific cell types.

R2C2. In terms of the synthetic and natural enhancers that function, they test these in 3 cell lines they trained their data on. This is impressive, and there is a clear preference for the predicted tissue-specific enhancers to be active in the respective cell types. I would like to know how these libraries containing their predicted functional tissue-specific enhancers work on other cell lines? Such as fibroblasts or other epithelial cells and neural cell types for example. Do we know that these enhancers are only active specifically within one cell type?

Measuring cell type specificity in new cell types with MPRA.

We agree with the reviewer that cell type specificity beyond the three cell types that Malinois was trained is an important and interesting question. We are pleased to address it in our revision by significant expansion of both wet lab experiments and extensive modeling. We aim to pursue this question more comprehensively in future work with additional technologies we are developing.

We have built two new models incorporating large amounts of MPRA data from two additional cell types, A549s (lung-cancer epithelium) or HCT116 (colon-cancer epithelial) cells. We included either 318,247 or 442,482 sequences that were tested in the original three cell lines in additional MPRA in either A549 or HCT116 cells, respectively (**Supplementary Figure 30a-b**). We then built two new models, either incorporating A549 or HCT116 data but otherwise using identical hyperparameter settings as the initial 3 cell models to generate comparable predictors (now included in the **Methods**). Both the new A549- and HCT116-inclusive models retained similar predictive performance (A549 model: Pearson's $r = 0.78-0.80$, Spearman's $\rho = 0.70-0.77$; HCT116 model: Pearson's $r = 0.85-0.87$, Spearman's $\rho = 0.75-0.80$; **Supplementary Figure 30c-d**) compared to the original, 3 cell-type Malinois models (A549 test set: Pearson's $r = 0.83-0.85$, Spearman's $\rho = 0.80-0.81$; HCT116 test set: Pearson's $r = 0.88-0.89$, Spearman's $\rho = 0.78-0.83$). We have included these new models and underlying data in the methods and supplementary information.

Having demonstrated the accuracy of the additional cell type models, we next predicted the activity of the synthetic and natural CREs tested in the original manuscript in these cell types. Given the high correlation of activity for the specific sequences measured by MPRA between both A549 and HCT116 with the originally modeled cell lines (A549 vs old cell types: Spearman's $\rho = 0.75-0.81$; HCT116 vs old cell types: Spearman's $\rho = 0.85-0.89$, **Supplementary Figure 30a-b**), it would be reasonable to expect CODA might design synthetic elements with high off-target expression in these cell types. However, we are pleased to report that the synthetic sequences remain clearly strongest in the tissue they were engineered to be most active in (**Supplementary Figure 30e-g**).

A summary of the results include:

- All K562 synthetic CREs are predicted to maintain maximum activity in K562 (**Supplementary Figure 30e**), while most HepG2 and SK-N-SH synthetic CREs are predicted to display the strongest activity in the originally intended target, depending on the design algorithm (HepG2: 97.0-100%; SK-N-SH: 89.2-97.4%; **Supplementary Figure 30g-h**). This may be due to a higher overlap in transcriptional programs across epithelial morphologies (HepG2, SK-N-SH, A549, HCT-116), compared to K562 cells.
- Synthetic CREs designed for all three original cell types do not display repressed transcriptional activity in A549 or HCT-116 as they do in the original off-target cell types.
- Together these new results demonstrate that CODA can identify activating functions directed to specific cells, without knowledge of other cell types. However, it cannot include repressive functions for cell types the model was not trained on. This emphasizes the need to train models on empirical data from any cell types that are targeted for repression or the need for an orthogonal method to mitigate function in non-modeled off-target cell types.

We include these results and discussion in the updated manuscript

Line 450:

"Selected synthetic CREs drive desired tissue-specific activity in vivo
We next sought to assess if the specificity of synthetic CREs would generalize beyond the initial three cell lines used for design. To determine if low off-target activity is maintained in additional

cell lines we trained two new CNN models for A549 (lung epithelial cancer; prediction Pearson's $r = 0.78$) and HCT116 (colon epithelial cancer; prediction Pearson's $r = 0.84$) cells, which were not included in the original model used for CODA (**Supplementary Figure 30a-d, Methods**). Synthetic CREs maintained maximum activity for their target cell type after inclusion of A549 and HCT116, especially those generated using Fast SeqProp (**Supplementary Figure 30e-h**).”

“**Supplementary Figure 30. MPRA models for A549 and HCT116 predict synthetic CREs.** Additional MPRA measurements were made in A549 and HCT116 for 318,247 and 442,482 elements and used to model CRE activity in these cell lines, respectively. (a-b) Pairplot showing distribution of activity for sequences measured in (a) A549 and (b) HCT116 and other cell types. (c-d) A model trained on sequences with (c) A549 and (d) HCT116 measurements with the same settings as Malinois accurately predicts MPRA measurements of CRE function. Scatterplots show model performance on held out test data. (e) Predicted activity of K562-targeting CREs across 5 cell lines. CREs are separated into frames based on design methodology. Text inset indicates percentage of CREs where the intended target had the highest prediction before and after A549 and HCT116 predictions were considered. (f) Same as (e) except for HepG2-targeting CREs. (g) same as (e) and (f) except for SK-N-SH-targeting CREs. (h) On-target predicted activity of CREs summarized by minGap before and after A549 and HCT116 predictions were included in the calculation. Each frame collects CREs from the five frames to the left. Each box represents CREs from a different design method.”

Estimating cell type specificity in whole organisms orthogonal machine-learning models.

We also provide a more explicit and detailed review of our *in silico* analysis of tissue-specific chromatin signature measurements using Enformer. Enformer predicts chromatin feature profiles for enhancers and promoters across many mouse cells and tissues, for a total of 1643 mouse-specific feature tracks. To capture

predicted effects in K562, HEPG2, and SK-N-SH, we created a composite score for 10 tissues including spleen, liver, and neuronal structures, respectively. This score is derived from manually curated predicted chromatin features, such as DHS, ATAC, H3K27ac, and CAGE signals, associated with transcriptional activation in the tissue. The features from Enformer associated with each tissue are collated in **Supplementary Table 8**. In this way, we use Enformer to predict the effect of synthetic CREs in many different tissues in mice (**Methods** subsection: **Enformer analysis of epigenetic signatures, Supplementary Figures 31 and 32**). We added the following text to the main text and highlight **Supplementary Table 8**:

Line 458:

*“To assess specificity of synthetic CREs beyond an episomal reporter context *in vitro*, we evaluated selected sequences for their ability to drive cell type-specific expression *in vivo*. Using Enformer, a deep learning model trained on gene regulatory signatures from primary tissues, we predicted the impact of synthetic CREs on epigenetic and transcriptional markers for gene activation (**Methods, Supplementary Table 8, Supplementary Figure 31a**)³⁴. Specificity as measured by MPRA in K562, HepG2, and SK-N-SH was significantly correlated with tissue specific Enformer scores in spleen, liver, and neural structures, respectively (**Supplementary Figure 31b-d**) and was higher in synthetic elements than both groups of natural sequences (**Supplementary Figure 31e**).”*

Finally, we adjust the frames on the right hand side of **Supplementary Figure 32** to explicitly show the Enformer composite predictions in 10 different mouse tissues for each synthetic element that underwent *in vivo* mouse and/or zebrafish experiments.

“Supplementary Figure 32. Malinois contribution scores/Enformer/MPRA results for in vivo sequences.

Collection of synthetic sequences prioritized for in vivo validation. Sequences in panels (a-c) and (d-f) are expected to drive expression in liver and neurons, respectively. Left column: Nucleotide sequence, motif matches, and contribution score tracks for each candidate. Right column: Bar plots of empirical MPRA signal (left y-axis) in K562 (teal), HepG2 (gold), and SK-N-SH (red) as well as aggregated Enformer predictions (right y-axis) of epigenetic signals reflecting transcriptional activation in mouse spleen (dim teal), liver (dim gold), neural tissue (dim red), heart, intestine, kidney, limb buds, lung, pancreas, and stomach.”

R2C3. They claim that synthetic CREs are fit for purpose in vivo, and yet we do not know this for several reasons. Firstly the information provided about the specificity is very limited, are the three enhancers seen in the liver specifically expressed in hepatocytes only? There are four major cell types in the liver, which cell types are their two enhancers active within? I would like to see staining with co-markers to see this. They also need more controls that test a similar number of sequences they anticipate would not be liverspecific or would be inert and yet contain similar motifs and show these do not drive liver expression. I would like to see testing of three natural sequences that they predict are hepatocyte-specific as well.

We apologize for the lack of clarity regarding the goal of our *in vivo* experiments. They serve as a proof-of-concept that fully synthetic elements would function in animals. While we agree that experiments would be required to critically evaluate the effectiveness of synthetic versus natural sequences *in vivo*, these are not tractable at scale. Instead we provide an *in silico* comparison of natural and synthetic CRE function in mice, in which we are less limited by the number of CREs we can interrogate or number of tissue types we can assess expression in. We provide a detailed discussion of these points below.

Clarifying language on scale of *in-vivo* proof of concepts

The testing of synthetic sequences *in vivo* was not intended to be a comprehensive breakdown of CRE activity but rather a proof-of-concept to confirm that our *in vitro* derived sequences did indeed drive expression in the animals tested (mouse and zebrafish). We've updated this section title (Line 450) to be "Selected Synthetic CREs drive desired tissue-specific activity *in vivo*" to make it clear we did not test all CREs *in vivo*. The success rate of reporter expression in both mouse and zebrafish suggest that our strategy of coupling *in vitro* design of cell-specific enhancers with additional *in silico* prioritization can identify synthetic sequences likely to drive reporter expression. While it is true that we are unable to differentiate between the four liver cell types with our imaging, at a tissue level expression is restricted to the liver. We replace 'hepatocyte' with 'developing liver' at line 479 to reflect this correction. Lastly, for each CRE presented here, we also tested a negative control 'empty vector' nearly identical to CODA vectors except for the lack of the 200-bp synthetic CRE. These were injected at similar rates as the experimental CREs, and showed very little to no expression signal. We now better highlight this in the methods by including the sentence:

Line 1179:

"We also created 'empty vectors' which were identical to CODA CRE vectors except for the lack of a 200-bp insert."

Feasibility of quantitative *in-vivo* validation

If we extrapolate that liver-specific synthetic CREs will continue to validate at a 66% rate and estimate that liver-nonspecific elements would fail (i.e., drive expression in the liver) at a 5% rate, we estimate we would need to test 14 liver-specific and 14 liver-nonspecific elements to achieve >90% power to show a significant difference in validation rate by χ^2 contingency test. This projection increases to over 60 elements if liver-nonspecific elements fail at a higher but still modest rate of 20%. This would translate to testing hundreds to thousands of animals. The costs required to test these many synthetic sequences *in vivo* with sufficient power are unfortunately prohibitive, outside the scope of this project and beyond that of publications with similar goals (de Almeida et al. 2024; Taskiran et al. 2024).

In silico simulation of *in vivo* validation using Enformer

Despite the intractability of *in vivo* extended validation, we agree with the reviewer that scaled validation of elements in whole organisms would be highly informative. In lieu of experimental validation, we used Enformer to predict the impact of natural and synthetic CREs based on expression (CAGE) and chromatin state (DHS, ATAC, & H3K27ac) in mouse tissues, matching K562 to spleen, HepG2 to liver, and SK-N-SH to adult brain tissues. In addition, we made similar predictions in 7 additional off target tissues (heart, intestine, kidney, limb bud, lung, pancreas, and stomach). We reused the predicted transcription changes from *in silico* insertion of our CREs that we had done previously for CRE prioritization (**Methods:** Enformer analysis of epigenetic signatures). We observe that in all tissues similar to our targeted cell types, synthetic CREs are predicted to be more specific (as measured by MinGap of Enformer transcription activation predictions using all 10 tissue predictions) than natural sequences. We include these results as part of **Supplementary Figure 31e**, which is included below for the reviewer. We have also updated the **Main** text:

Line 462:

“Specificity as measured by MPRA in K562, HepG2, and SK-N-SH was significantly correlated with tissue specific Enformer scores in spleen, liver, and neural structures, respectively (Supplementary Figure 31b-d) and was higher in synthetic elements than both groups of natural sequences (Supplementary Figure 31e).”

“Supplementary Figure 31. Enformer based prioritization of oligos for in vivo tests. (a) Enformer can predict CRE-driven changes in epigenetic and transcription dynamics of transgenes inserted into the H11 safe harbor locus in mice. Three example sequence tracks display predicted DHS signals observed in the livers of 15.5 day old mice. Transgene transcription start site and poly-adenylation signal are indicated by the gray bars. The first track is the predicted signal when the input sequence at the CRE insertion site is all Ns. The second track is an example predicting using a validated HepG2-specific synthetic CRE. The third displays the differential DHS effect. **(b)** Empirical K562 MinGap measurements are well correlated with Enformer-predicted features of spleen-specific transcriptional activation (Methods). **(c)** Empirical HepG2 MinGap measurements are also well correlated with Enformer-predicted features of liver-specific transcriptional activation. **(d)** Empirical SK-N-SH MinGap measurements are also well correlated with Enformer-predicted features of neural-specific transcriptional activation. **(e)** Enformer-based cell type matched tissue-specific transcriptional activation predictions (K562 matched to spleen,

HepG2 matched to liver, SK-N-SH matched to adult brain). Stars indicate family-wise error rate corrected p-values < 1e-4.”

R2C4. Regarding the neural enhancers, the ability to design tissue-specific neural enhancers is also unclear to me and not validated in the current manuscript. The enhancers are active in neural cell types, but how do the neural cell types in the zebrafish relate to neuroblastoma cells or mouse cortex? Would their neuroblastoma active enhancers be active in all neural cell types? If so, is this really cell specificity? More enhancers need to be tested and more details provided on the precise location of expression. They would also need to show that synthetic enhancers that are not predicted to work and yet show the same motifs are non-functional. If the authors want to show natural sequences are less specific than synthetic they would also need to test natural sequences and see the location of expression.

The SK-N-SH cell line is derived from a neuroblastoma, and given this origin, we anticipated neuronal expression would be most likely if indeed these sequences were capable of driving expression *in vivo*. We agree with the reviewer that substantial future work is needed to explore the design and testing of tissue-specific neural enhancers *in vivo* at-scale. This is an exciting area of exploration for which we are currently trying to develop new technologies to feasibly address. They however currently require significant time and resources beyond the scale of recently published papers, and outside the scope of what we set out to assess in our current manuscript. Our intention is that this current body of work will serve as a starting point to explore exactly the excellent questions raised by the reviewer. In response to the reviewer’s feedback, however, we have carried out further characterization of our *in vivo* findings by performing (i) immunohistochemistry on the mouse brain, (ii) RNA-seq experiments in the brain, liver, and spleen, and (iii) new, saturation mutagenesis MPRA. We describe each of these new experiments and their findings below.

Validating a synthetic neuronal CRE

Our only initial objective with CODA was to achieve cell type specificity *in vitro*. Given the resounding success, we elected to nominate several sequences for *in vivo* testing. We did this to establish a baseline on the transferability of an *in vitro* designed sequence to an *in vivo* system for future studies. To be clear, we are not intending to make statements regarding if CODA can design sequences that are superior for *in vivo* expression. However, we believe the inclusion of the *in vivo* testing is important and demonstrates that *in vitro* screening procedures of ML-designed elements coupled with additional prioritization can generate sequences with exceptional *in vivo* specificity. To strengthen this claim, in line with this reviewer’s suggestion, we have performed immunohistochemistry on the mouse brain samples that show transgene expression is isolated to neurons within layer VI of the cortex and the subplate (**Figure 4e,f**, and detailed analysis in **Supplementary Figure 37**). We reference this in the main text as well as included updated methods for these experiments. We reproduce the new panel and legend for the reviewer below:

“Figure 4: (e) LacZ expression in deep cortical layers is neuron-specific. Top panel: representative confocal images of layer 6 neurons, microglia, astrocytes, and merged image demonstrating the absence of transgene in control mice. Lower panel: confocal images show that transgene expression is exclusive to cortical neurons with arrows indicating colocalization between LacZ signal and neurons. Scale bars: 20um. (f) Box plot showing proportion of neurons, astrocytes, and microglia positive for the transgene. Neurons exclusively express LacZ. **: adj $p < 0.0001$ for Kruskal-Wallis One-Way ANOVA.”**

Molecular validation of CRE targeting specificity

To address questions of specificity with more sensitivity than microscopy, we performed new RNA-seq experiments in brain, liver, and spleen. Again we see that transgene LacZ expression is highly specific to the brain (**Figure 4g**). We have updated the main text and methods to reflect these analyses. Results from this analysis have been added to Figure 4 (above) with the legend for the reviewer below:

“Figure 4: (g) Synthetic N1 CRE drives specific transgene expression in the brain. LacZ expression by synthetic N1 CRE is measured using RNA-seq and normalized by the expression of LacZ in mice transgenic for the minP empty vector.”

We would like to test more sequences in order to refine how we design and nominate synthetic sequences and to compare them to natural sequences; however, the additional animals required to comprehensively test this is outside the scope of our current goal of determining whether or not these synthetic sequences, designed to drive reporter expression in transformed cell lines, are capable of driving *in vivo* expression.

Interrogating functional motifs in our neuronal CRE

Following this reviewer’s suggestion to better understand the mechanisms of our synthetic neuronal CRE, we completed a new, saturation mutagenesis MPRA. Originally, we used Malinois contributions scores in SK-N-SH to identify two primary ETS binding domains and four CREB-like binding domains as driving activity. In our MPRA saturation mutagenesis, we mutate each nucleotide in the CRE to every other nucleotide, allowing us to generate empirical functional maps of nucleotide importance to CRE activity, complementing our predictions. Both ETS and three of CREB motifs are shown to drive the CRE’s activation. We’ve now included this as an update to main **Figure 4h**, updated methods, and included this update to the main text:

Line 509:

“To assess contribution scores from Malinois we conducted an empirical saturation mutagenesis MPRA in SK-N-SH, which confirmed high-contribution regions and supported motif assignments identified from the contribution scores (Figure 4h, Methods).”

Figure 4: (h) Nucleotide level effects of synthetic neuronal CRE N1. Top track: Malinois contribution scores reveal the role of ETS and CREB-like binding domains in mediating synthetic CRE activity in neurons. Subsequences of high predicted contribution to SK-N-SH activity overlap with ETS- and CREB-like binding motifs based on visual inspection. Bottom track: Single nucleotide effects measured experimentally using MPRA saturation mutagenesis. Circular points represent the expression change measure by MPRA when only that position is mutated in N1. Letters represent the reference nucleotide of the N1 sequence at that position with the height corresponding to the mean expression change at that position with opposite sign.”

R2C5. The authors state, “There is no guarantee that an optimal CRE for an intended purpose has arisen naturally through evolution.” I agree, but I would caution against trying to make statements about evolution when the analysis is within transformed cell types. There are several statements about evolution, and I would be careful with these types of statements as they can be misleading. Evolution has not been working on transformed cell lines, and thus one would not expect the natural sequences to be optimized to drive expression in these cell lines.

We agree with this reviewer that the evolutionary pressures shaping the genome have not acted on transformed cell lines, nor should we expect explicit selection for gene expression objectives like those presented here. After considering the reviewers comment, we now appreciate that our statement could be understood in multiple ways. We were trying to convey that, given our experimental goal to design sequences capable of driving reporter assay expression in a single transformed cell line, it is unlikely that sequences optimal for that task would exist naturally due to evolution. Given the redundancy and sharing of CREs at individual loci (Osterwalder et al. 2018), and emerging evidence that suboptimal CREs are preferred by evolution (Farley et al. 2015; Jindal and Farley 2021), it is unclear if we should expect sequences in the genome to be highly specific for transformed or even natural tissues. Furthermore, the human genome has a severely constrained search space further hindering our ability to identify sequences necessary for bespoke objectives, such as those with therapeutic purposes (de Boer and Taipale 2024). Our work suggests that some objectives are likely outside the scope of typical evolutionary pressures but not outside what may be possible with synthetically designed sequences.

To better articulate these ideas to the reader, we have made the following modifications 1) we have removed ‘through evolution’ to make it clear we are not suggesting the functions we are trying to design are the same functions evolution acted on, and 2) added “these intended purposes” which links to the clearly non-natural “therapeutic or biotechnology applications” described earlier in the sentence.

Line 25:

*“While there is great potential for strategically incorporating CREs in therapeutic or biotechnology applications that require tissue specificity, there is no guarantee that an optimal CRE for **these intended purposes** has arisen naturally ~~through evolution~~. ”*

This point was reiterated in the Discussion, so we have also made a similar change to emphasize evolution is not acting on transformed cell lines:

Line 562:

*“The dearth of natural sequences capable of achieving exquisite specificity **in a desired cell type** in our study highlights the difficulty of using human genomic sequences to achieve non-natural objectives for which evolution **may not have acted on**. ”*

R2C6. Regarding the discovery of syntax, the authors state “Synthetic sequences leverage unique sequence syntax to promote activity in the on-target cell type and simultaneously reduce activity in off-target cells.” The syntax is not clearly defined, and I don’t think this manuscript contributes to our understanding of syntax within enhancers. I understand that TF motifs are found, but are unique syntaxes that drive on target and reduce off-target demonstrated within their data? Can they make a library where they delete these syntax elements and show that they lose on-target and gain off-target expression? This experiment would be required to validate their claims.

Sequence content and motif differences in synthetic sequences

Our descriptions of regulatory syntax aim primarily to describe the features, namely TFs and their combinations, utilized by different classes of sequences. We have not outlined prescriptive rules for how regulatory syntax is encoded in our sequences, but instead have classified sequences based on their usage of specific syntactic elements by multiple observational metrics including motif co-occurrence, NMF, and k-mer analysis. Additionally, analyses by an external group showed that our designed sequences differed significantly from natural sequences and random sequences (Gupta, Lal, Gunsalus & Biancalani et al 2023 biorXiv). Similarly, our new analysis of k-mer content in Supplementary Figure 13 shows clear differences in 4-mer content between natural and synthetic cell type-specific elements. Together, these data suggest that designed synthetic sequences are distinct from natural sequences chosen for accomplishing the same objective.

As described in depth in R1C10, we completely redid our motif analysis and subsequently our program analysis, using predicted functional motif patterns nominated by TF-MoDISco. To address this and reviewer 1’s concerns about the importance of individual motifs, we used *in-silico* motif ablation and embedding studies to show that disruption and creation of a motif has the expected impact. Across **Supplementary Figures 22 and 23**, we observe very strong correspondence between expected CRE activity impacts when a motif is ablated or embedded, suggesting that our identified motifs are a functional part of that cell type’s regulatory syntax learned by Malinois. While further study is necessary to define the specific rules governing the syntax of our designed sequences, the evidence presented supports the claim that they are employing functional regulatory syntax to drive cell type specificity.

Updating program analysis

Using the new NMF program analysis on TF-MoDISco-nominated motifs, we restricted our exploration of motif syntax to only motifs with predicted function (as deeply described in R1C10). With this analysis, we again see

distinct, quantitative differences in semantic program usage such as higher rates of deploying both activating and repressive programs. To clarify, in this analysis we observe unique patterns of program usage in synthetic elements, but not see programs that are unique to synthetic elements. We agree that the original statement in the **Abstract** about *unique* programs was unclear and could be construed as an overstatement. We have updated this to use more clear and conservative language:

Line 34:

“Synthetic sequences exhibit distinct sequence syntax associated with activity in the on-target cell type and simultaneously reduce activity in off-target cells.”

We also now describe the results of our program analysis in more detail in the main text, specify that we do not find unique programs, and include the updated statistics using the new NMF analysis, none of which change our initial conclusions:

Line 436:

“Natural and synthetic sequences deploy distinct distributions of semantic programs (Figure 3e, Supplementary Figure 28b). While there are quantitative differences in program preference between the different synthetic sequence design methods, there are no programs unique to one method. Overall, synthetic elements have higher program content and program heterogeneity compared to natural CREs (Supplementary Figure 29a-b). We also found that natural sequences primarily rely on activating programs while synthetic sequences also frequently utilize programs with repressive effects in off-target cell types (median repressing program content: DHS-natural 0.077; Malinois-natural 0.064; synthetic 0.123) (Supplementary Figure 29c,d). The vast majority of synthetic sequences (91.9%) are composed of both activating and repressing programs each exceeding a threshold of 0.1, while relatively fewer DHS (26.9%) and Malinois (25.3%) natural sequences show this combination (Methods, Supplementary Figure 29e).”

R2C7. The authors state that “Malinois contribution scores enable nucleotide resolution interpretation of sequence activity.” In Figure 3a, the authors show a synthetic enhancer that drives expression in HepG2 Cells, they highlight motifs found by Malinois that are important for the activity. To demonstrate that Malinois finds the functional features within the enhancer with base pair resolution, the authors need to mutate these features and show that this renders the enhancer inactive or no longer tissue-specific. Can the authors do this validation on a library of enhancers?

Addressing the reviewer’s comment, we conducted the aforementioned analysis *in silico* using Malinois as a proxy of MPRA as suggested by reviewer 1. To test whether contribution scores capture functional features learned by Malinois, we disrupted blocks of clustered positive or negative contributions by randomly mutating the corresponding sequence segments corresponding to block calls, and obtained Malinois’ predictions of the mutated sequences. Similarly, we tested sequences randomly mutating positions outside positive and negative block calls. We observed that mutating blocks of positive or negative contributions globally decreased or increased the predicted activity, respectively, when compared to the undisrupted sequences. The effect was most dramatically observed for the target cell type, where disruption of positive contributions completely annihilated the predicted activity. In addition, the disruption of negative contributions further increased the predicted activity in the target and off-target cell types. In contrast, disrupting positions not present in block calls had a more neutral effect in activity. We summarized our results in **Supplementary Figure 21**.

Supplementary Figure 21. Contribution block ablation. (a) Predicted activity (labeled as initial) in K562 (teal), HepG2 (gold), and SK-N-SH (red) of the library sequences targeting K562. Activity predictions of disrupted sequences when ablating segments corresponding to negative (gray), positive (dark gray) contribution blocks, or outside blocks (light gray) determined by contribution scores in each cell type. The number above each box denotes the number of sequences for which a contribution block type was found. All initial activity boxes correspond to 25,000 sequences. Boxes demarcate the 25th, 50th, and 75th percentile values, while whiskers indicate the outermost point with 1.5 times the interquartile range from the edges of the boxes. (b) Same as (a) but library sequences targeting HepG2. (c) Same as (a) but library sequences targeting SK-N-SH. (d) Distributions denoting the number of positions disrupted in (a) by negative (gray), positive (dark gray) contribution blocks, or outside blocks (light gray). Boxes demarcate the 25th, 50th, and 75th percentile values, while whiskers indicate the outermost point with 1.5 times the interquartile range from the edges of the boxes. (e) Same as (d) but disrupted in (b). (f) Same as (d) but disrupted in (c).

We have included the complete details of the *in silico* experiment in the methods section, and added the lines below to the main text referencing the analysis.

Line 342:

“First, we used Malinois to predict nucleotide-resolution activity contribution scores for each sequence in the three cell types using a modified version of Integrated Gradients (**Methods**)⁶⁹. We consistently observed that disrupting blocks of positive contribution led to a decrease in predicted activity, while disrupting blocks of negative contribution resulted in an increase (**Supplementary Figure 21, Methods**). This alignment with expected prediction effects supports the functional relevance of the contribution scores as perceived by the model.”

In addition, as a single empirical example, we performed saturation mutagenesis MPRA (MPRA-satmut) on the synthetic CRE (N1) that was originally evaluated *in vivo*. This experiment used MPRA to directly measure the effect of every possible mutation along the 200-bp sequence. The resulting activity profile from MPRA-satmut is highly concordant with the contribution scores we initially provided in **Figure 4h**. We have now updated **Figure 4** to include both the contribution scores from Malinois and the experimental results from MPRA-satmut. This single example demonstrates how contribution scores compare to an experimental approach.

Figure 4h: Nucleotide level effects of synthetic neuronal CRE N1. Top track: Malinois contribution scores reveal the role of ETS and CREB-like binding domains in mediating synthetic CRE activity in neurons. Subsequences of high predicted contribution to SK-N-SH activity overlap with ETS- and CREB-like binding motifs based on visual inspection. Bottom track: Single nucleotide effects measured experimentally using MPRA saturation mutagenesis. Circular points represent the expression change measure by MPRA when only that position is mutated in N1. Letters represent the reference nucleotide of the N1 sequence at that position with the height corresponding to the mean expression change at that position with opposite sign.”

Reviewer #3

Massively parallel reporter assays (MPRAs) are increasingly common technique to interpret human genetic variation, screen for enhancer activity, and understand the principle of gene regulation. One application of MPRA data is the prioritization and design of sequences that can label specific cell types. Gosai et. al. develop computational methods to learn a “regulatory grammar” for a specific cell types based on MPRA data. Then, they apply those models to generate synthetic sequences that have greater specificity for the cell type of interest. The research presented here is cutting edge and of broad interest to multiple scientific communities. The strength of the manuscript is the careful benchmarking of different evolutionary algorithms, the interpretations of transcription factor binding site motifs, and the in vitro validation their results. In spite of these positives, there are major issues to address. First, the manuscript depends on unpublished MPRA data for which there is no described quality control or experimental design included. Second, the Malinois model is not adequately benchmarked for its cell type specificity or compared to alternative computational models. This is crucial, given that Malinois is used as a benchmark for CODA. Third, the application of the approach to label cell types in vivo requires a strong similarity between the regulatory grammar of that cell type and the in vitro model of that cell type. This relationship isn’t adequately explored. Fourth, Malinois and CODA as a method for finding sequences to label specific populations aren’t benchmarked against published methods. In summary, there is strong potential for this manuscript to have an impact, but additional rigor in some components of the analysis are required to demonstrate the author’s claims.

We appreciate the enthusiasm for this work and agree with the reviewer one of the key strengths of this paper is the careful benchmarking and validation of results. We hope we address the reviewers' concerns below.

R3C1. The increasing amount of available MPRA data makes this manuscript especially timely. There is a need of innovative computational methods to better leverage and understand this type of data. This work fills an important gap and represents a departure from the use of natural sequences towards potential engineering with synthetic sequences.

R3C2. The CODA portion of the manuscript is well motivated and well written. The choice of evolutionary algorithms makes sense in the context of the study. The in vitro validation provides a good demonstration of CODA’s utility.

We agree with the reviewer there is a gap in the field studying comparing synthetic and natural approaches to sequence engineering. In addition to significantly advancing synthetic engineering, our comprehensive evaluation of synthetic and natural sequences together provides a true assessment of the benefits synthetic designs have. We also appreciate the reviewers assessment that our study is well motivated and well written. We believe the effective communication of a study to its scientific audience is paramount and appreciate all three reviewers' contributions towards improving towards that goal.

R3C3. The research depends on large amounts of unpublished MPRA data, which seems to have been collected in the laboratory of the senior author. The details of how the MPRA experiments were designed (especially which sequences were included) and the quality control metrics of that dataset have a bearing on what conclusions can be drawn from the study.

We regret that the details of the MPRA design were lacking and underlying data unpublished at the time of initial submission. We are near completion of a preprint describing the experiments used in the training set and expect the preprint to be released early-March in parallel with other ENCODE 4 manuscripts. To facilitate review of this manuscript, we would be happy to provide an early draft of this work upon request. We will note,

all data has been publically available since February 2023 on the ENCODE portal (ENC IDs are included in **Supplementary Table 1** and indeed many groups both internal and external to ENCODE have already downloaded, assessed, and used the datasets in their own research programs. Methods, metadata, and experimental details about the generation of this data are also available on the ENCODE portal.

R3C4. For Figure 1, it's not clear why the GATA1 locus is the focus. It would be more helpful to stratify the accuracy across the different input datasets. Hyper parameters seemed to be tuned on chromosome X, where GATA1 is locus, which could introduce circularity and inflate the model performance.

We chose GATA1 as it is one of the few comprehensively tiled loci that previously existed using our MPRA design strategy of synthesized 200-bp elements, and the only one not in a training set chromosome. This screen consists of 53,662 elements tiling 2.09Mb of sequence with a 50-bp step size. As we later use Malinois to assess the entire genome for CRE specificity, we reasoned that a display of Malinois ability to match MPRA signal across a genomic locus would be useful for the reader.

The reviewer is right to be concerned about the circularity of testing generalizability on the validation set used for hyperparameter tuning. However, while chromosome of origin is a useful way to organize oligos into the train/validation/test splits, the oligos do not comprehensively cover chromosomes. Therefore, there are vast segments of the training and validation chromosomes that were never used to fit or select a model, and elements from these regions are suitable for benchmarking. Furthermore, only 407 out of 52,906 oligos from the validation set overlap the tiled region around GATA1, so it is unlikely for model selection to be appreciably biased towards high performance at the GATA1 locus specifically.

To improve our benchmarking and address the reviewer's valid concerns, we now exclude confounding sequences by removing oligos from the GATA1 tiling screen containing any overlap with oligos in the validation set and analyze the remaining oligos. Note that each individual validation set oligo dispersed throughout the GATA1 locus will overlap multiple GATA1 tiling oligos due to the 50-bp tiling step size. We updated **Supplementary Figure 5a** and note that the performance metrics are nearly unchanged when using oligos with no validation set overlap (before filtering: Pearson's $r = 0.91$, Spearman's $\rho = 0.85$; after filtering: Pearson's $r = 0.91$, Spearman's $\rho = 0.84$). This filtering is now also mentioned in **Methods** and **Supplementary Figure 5** legend:

a.

“Supplementary Figure 5. Correlation of Malinois predictions and empirical MPRA tiling data. (a) Malinois predictions are highly correlated with empirical MPRA measurements of tiled sequences in the GATA locus (chrX:47,785,602:49,880,397)^{5,48-50} in K562 (Pearson's $r = 0.91$, Spearman's $\rho = 0.84$). X-axis and y-axis correspond to empirical measurements and Malinois predictions, respectively for oligos in the library ($n = 51242$ oligos). Sequences which overlap with oligos from the validation data split used for model selection were removed from this plot and correlation calculations ($n = 2420$ oligos omitted). Additionally, oligos with a replicate \log_2FC standard error greater than 1 in any cell type were omitted from the plots.”

We provide the modified sections below with new text in blue:

Updated Methods Line 644:

“For locus-specific benchmarking we aggregated the \log_2FC of oligos that tile the GATA1 locus (OL43) following the same counts filtering steps as described above. We generated per-genome-base activity measurements by averaging the MPRA activity of each oligo that overlaps that base pair. We remove oligos genomic coordinates which overlap those in the UKBB and GTEx libraries in scatterplots and correlation calculations. We also aggregated the \log_2FC output of 318,247 and 442,482 sequences tested in A549 (OL27, OL28, OL29, OL30, OL31, OL32, OL33) and HCT116 (OL41, OL42), respectively following the same counts filtering steps as described above.”

We also have access to MPRA tiling data from 6 additional loci that overlap with chromosomes included in the training split. Here again, we removed oligos containing any overlap with oligos in the training data split. Overall, Malinois maintains consistent, high-quality predictions across all 7 of these loci individually (Pearson's $r = 0.85-0.91$; Spearman's $\rho = 0.79-0.88$) and the two libraries on aggregate (Pearson's $r = 0.88-0.90$; Spearman's $\rho = 0.82-0.83$).

Reviewer Figure R3C4. Malinois predicts the results of additional tiling screens with high accuracy. (a) Aggregate and per-locus performance of Malinois predictions on an MPRA using OL3 which contains oligos tiling the GATA1 and MYC loci. **(b)** Aggregate and per-locus performance of Malinois predictions on an MPRA using OL45 which tiles the BCL11A, HBA2, HBE1, LMO2, and RBM38 loci. For both **(a)** and **(b)** sequences which overlap with oligos from the training and validation data splits are omitted from the plots and correlation calculations. Additionally, oligos with a replicate log₂FC standard error greater than 1 in any cell type were omitted from the analysis.

We feel these comparisons are trustworthy due to careful removal of sequences overlapping the training data. However, we are concerned including them may distract readers because these loci (excluding GATA1) are on chromosomes from where oligos in the training split were derived.

R3C5. The cell type-specificity of Malinois is key to the study, but there are several issues with how that component of the project is validated. First, Malinois is presented as a new computational method, but it is not benchmarked relative to others that could be trained on similar data, including an ENFORMER-like model.

Second, the cell type-specificity of Malinois isn't compared to the cell type-specificity of open chromatin or other measurements, just the raw values. Differences in signal and depth could play a role in accuracy.

Third, based on Supplemental figure 3a, I get the impression there is a low proportion of cell type specific CREs. To account for this, there should be a greater exploration and visualization of the number of CREs enriched for each cell type. A statistical test that better handle class imbalance would be helpful to quantify the cell type-specificity. The spearman correlation is currently applied. These concerns are especially important given that CODA is compared to Malinois alone. If Malinois isn't efficient at determining cell type specificity due to these concerns, then CODA's improvement over Malinois could be overestimated.

Benchmarking of Malinois activity predictions

We agree with the reviewer that benchmarking additional models on our data set would provide useful context to evaluate Malinois. To this end, and also addressed extensively in R1C1/4, we benchmarked MPRA-DracoNN and Enformer, published models for MPRA and chromatin state, respectively. We now include a comparison of these models' predictions and Malinois' to empirical MPRA results of 77k elements in Reviewer Figure R1C1. We show Malinois outperforms these previously trained models on these data. We caution overinterpretation of Enformer' lower performance, as it was not initially trained on any MPRA data, and is instead tasked with predicting features like open chromatin. Retraining the model with our data would take extensive resources and expertise beyond the scope of this paper. Furthermore, the computational efficiency of Malinois compared to a Transformer model such as Enformer vastly improves the tractability of iterative sequence design, as done in this study.

Since we finalized Malinois in November 2021, additional advances in deep learning have been proposed that could likely improve on Malinois predictions. Our benchmarking on unpublished, privately communicated models we have tested through correspondence with multiple labs in the field indicates Malinois remains relatively strong. As described in R1C1/4, ReporterNet, which was trained on the identical dataset as Malinois, achieves similar overall performance. We included these comparisons to unpublished studies for the reviewer, and are confident similar analyses will be made public in those paper's respective preprints.

We agree with the reviewer that chromatin accessibility measurements may not be of uniform quality genome-wide. To mitigate this we conducted comparisons between Malinois predictions and raw DHS signal at and around DHS peaks where experimental confidence is high (**Figure 1f, Supplementary Figure 6**).

Improved visualization/quantification of specific CRE numbers

We agree with the reviewer that only a small proportion of sequences in the training, validation, and test dataset plotted in **Supplementary Figure 4** (previously Supplementary Figure 3) were truly cell type-specific. One of the striking results of this study is that synthetic elements with strong cell type-specific activity can be reliably designed by a model trained on data with a dramatic imbalance between specific and non-specific elements. As seen in the top three plots of **Supplementary Figure 4a** comparing empirical values from two different cell types, it is evident how few points are captured along the axis (cell type-specific). More important is how the shape of the plots containing these sequences compare to those containing cell type-specific synthetic sequences (**Supplemental Figure 20**). For context, the sequences in the training, validation, and test datasets were selected for overlapping a variant associated with either a complex trait in the human population

or an eQTL in GTEx. The majority of the variants tested were either controls (negative control variants matched to trait associated variants) or non-causal variants in linkage disequilibrium with the causal variant. As a result, for practical purposes we can consider this dataset to be mostly random with a possible slight preference toward distal CREs (complex traits and eQTLs) and promoters (eQTLs). The significant proportion of shared signal seen in **Supplementary Figure 4a** is reflective of what we typically observe by MPRA, that most sequences from the human genome have shared activity across cell types, likely a reflection of a set of shared activating factors (e.g. ETS, SP, etc). Furthermore, we have a working internal hypothesis that cell type specificity is harder to achieve for episomal assays than for a chromatinized. The fact that shared elements are common highlights one of the significant capabilities of CODA: that Malinois can learn cell type-specific signals despite there being only a few examples in the training data, and that its accurate interpretation of these features is correctly exploited by CODA when designing synthetic cell type-specific sequences.

R3C6. The ultimate goal of the research is to label specific types of cells. There are other methods for predicting the how sequence specific certain cell type are. Currently, there is no comparison of this method to those previous methods. These include:

<https://doi.org/10.7554/eLife.48089>

<https://doi.org/10.7554/eLife.69571>

<https://doi.org/10.1038/s41434-021-00227-z>

<https://doi.org/10.1093/nar/gkad375>

<https://doi.org/10.1101/2022.07.26.501466>

Firstly, we appreciate the reviewer highlighting these important additional citations, and apologize for our oversight in not including them in our submission. We had already begun to note the oversight, and incorporated some of these citations in our *bioRxiv* preprint released in the days following our initial submission to *Nature*. We agree each paper represents a valuable advance in the field, and we have now included all five citations throughout our manuscript, as well as added additional citations published after submission. One notable addition is de Almeida et al. which was published alongside Taskiran et al. (cited above by the reviewer) at *Nature* on Feb 1st (de Almeida et al. 2024; Taskiran et al. 2024).

Importantly, we also provide a detailed analysis of each of these papers below for the reviewer, and we note how our work both compliments and extends each. Broadly, the largest difference between our study and all prior work to our knowledge is that ours performs large-scale experimental validation of cell type-specific sequences. Previous work has only validated a small number of sequences (10-287 CREs and only several controls). Our study uses MPRA to test 51,000 synthetic CREs, more than all previous studies combined. This well-powered analysis allows us to directly compare the success rate between synthetic and natural creating a benchmark not previously established.

We have separated our discussion of previous work into two categories: approaches that use/identify natural sequences and those that generate synthetic sequences from model predictions.

(i) Natural sequence prioritization or rational design using natural sequences: Four of the citations provided above employ various approaches to identify, prioritize and test naturally existing sequences to accomplish cell type-specific expression.

PESCA and SNAIL

- PESCA (Hrvatin et al, eLife 2019), uses ATAC-seq and sequence conservation to nominate CREs specific for neuronal subpopulations (Hrvatin et al. 2019).
- Similarly, an effort by Lawler et al. called SNAIL, uses ATAC and DHS datasets to identify sequences specific to parvalbumin-expressing brain cell types (Lawler et al. 2022). Unlike PESCA, SNAIL uses machine learning to prioritize candidates with an SVM-based classifier to assign a binary on- or off-target classification.
- We believe both methods have significant similarities to the selection approach of DHS sequences used in our manuscript and to a lesser degree that of Malinois nominated sequence.
- A major strength of both studies is the *use of in vivo* validation. Hrvatin et al. tests 287 human CREs while Lawler et al. evaluates 2 sequences, both in mice.
- However, unlike our approach, they only evaluate sequences they nominate, failing to provide any baseline expectations to judge the success of their methods.
- We tested 24,000 natural sequences (DHS and Malinois nominated) by MPRA and conclusively demonstrated synthetic sequences outperform natural sequences for all three cell types. Our extensive comparison of natural and synthetic sequences establishes for the first time that for certain objectives natural sequences in the genome are not the optimal solution.

Minipromoters: Simpson and Wasserman labs

- Equally impressive efforts in two studies by the Simpson lab and Wasserman labs (Korecki et al. Gene Therapy 2021 & Fornes et al. NAR 2023) use a related approach of nominating natural sequences termed “Minipromoters” (Fornes et al. 2023; Korecki et al. 2021).
- Their approach combines promoters with distal CREs and further manipulates these sequences using rational design principles. We especially appreciate this approach because it uses combinations of CREs to create cell-specific elements which likely better reflects how the genome actually accomplishes cell specificity.
- Again, these works only evaluated a small number of sequences, testing 24 sequences across both papers. It will be interesting in the future to evaluate how well this combinatorial approach works compared to our synthetic sequences.
- We note that Minipromoters are substantially larger (800-2500 bp) than the 200-bp sequences designed by CODA which limits their direct evaluation at-scale and also limits their therapeutic applications due to the design constraints of some viral vectors

(ii) Design of fully-synthetic sequences: The work of de Almeida et al. and Taskiran et al. both use deep learning models of CRE activity to inform the design of synthetic sequences (de Almeida et al. 2024; Taskiran et al. 2024). Both approaches rely on ATAC-seq data for model training, with de Almeida et al. fine-tuning their model on *in situ* reporter assays as a binary classifier. Each paper provides significant contributions to the fields of gene regulation and synthetic CRE design, and we enjoyed reading both papers. However, we believe there are several aspects of our study that provide significant advancements that were not the focus of de Almeida et al. or Taskiran et al.

de Almeida et al.

- In de Almeida et al. CREs were generated by scoring up to 3 billion random sequences and selecting the top performing sequences. They then functionally validate 40 synthetic CREs in *Drosophila* embryos across 8 tissues, which while not enough to be statistically powered, gives us a reasonable comparator to our large-scale validation in humans.
- In their study, 68% were active in the target tissue, but 25% also had off-target tissue specificity. This is lower than the 94.1% success rate of CREs we identified in our work, but not

incompatible with our findings due to difference in methodologies, and de Almeida et al. employing more extensive testing of off-target effects.

- Another important distinction between our work and that described in de Almeida et al. is the efficiency of our design process. In de Almeida et al. they score random sequences selecting those that score best with their design criteria. In our work, we utilize search algorithms to identify local optima greatly increasing the efficiency of our process and also likely leading to CREs with greater specificity.
- To measure these efficiency differences we used the same zero-order Markov process deployed by de Almeida within CODA for sequence design. We ran this approach for roughly 37 hours to generate 15000 elements, screening 6.144 billion elements in the process and compared the resulting sequences to 15000 sequences designed using FastSeqProp (52.1 minutes) and Simulated Annealing (31.5 minutes). Both Fast SeqProp and Simulated Annealing identified CREs that significantly outperformed a random search approach (**Supplementary Figure 38**). In fact, if we linearly extrapolate the 2.5%-tile bound (e.g., the 2.5% best elements) we estimate the zero-order Markov process will take 1106 hours to intersect with the same 2.5%-tile bound of Fast SeqProp. Thus Fast SeqProp and Simulated Annealing improve efficiency by over 1000-fold, which is a conservative estimate given our linear projection ignores the decay in rate of improvement over time of zero-order Markov processes.
- We have added this important observation of our increased efficiency to the **Discussion**.

Line 553:

“Due to the intractability of fully searching sequence space, CODA cannot assuredly identify global specificity maxima, but our exhaustive evaluation of natural sequences demonstrates the design methods we used can identify synthetic sequences that regularly outperform natural ones with 1000-fold greater efficiency compared to previous methods using a zero-order Markov approach (Supplementary Figure 38)^{40,41}.”

“Supplementary Figure 38. Projection of efficiency of zero-order Markov chains for model directed sequence design. 200-mers were uniformly randomly sampled (i.e., sampled from a

zero-order Markov chain) and tested using Malinois to calculate MinGap for K562 targeting sequences. We plotted the negative MinGap of the cumulatively best 15000 elements collected over 3000000 steps with 2048 samples taken at each step (total of 6.144 billion elements screened). We plot the median (blue line) and 95%-tile interval (blue shaded region) of the negative MinGap trajectory of the best element collection. As a comparison, we designed 15000 elements using Fast SeqProp (52.1 minutes) and Simulated Annealing (31.5 minutes) with the same objective and plotted the median and 95%-tile intervals of predicted MinGap for these groups.”

Taskiran et al.

- In Taskiran et al. the authors used three design approaches, (i) random sequences with single base changes added in a stepwise progression, (ii) motif embedding, and (iii) generative adversarial networks.
- Out of all the papers described here, Taskiran et al most closely matches our approach.
- The work however does not include large-scale validation of the comparisons to baseline controls that allow meaningful conclusions to be drawn regarding the success of synthetic CREs.
- The work is also primarily focused on design within drosophila with some additional work done in two human cell lines.
- The majority of validation performed by Taskiran et al. is performed *in silico*, with only ~30 unique sequences tested *in vivo* (Drosophila) and ~30 unique sequences confirmed in a human cell line model (we note the approximate count is due to a lack of clarity what was tested and how *unique* sequence is defined).
- Our strategy to model CRE activity across cell types directly is fundamentally different from the approach by Taskiran et al. which deploys a neural approximation of topic modeling from sequences. Our model enables us to construct highly flexible objective functions to rationally design sequences with any possible activity profile across the modeled cell types. In contrast, the approach taken by Taskiran et al engineers sequences by optimizing for topic predictions with predetermined cellular activity profiles, some of which are cell-specific. While it will be interesting to see in future work which process is optimal, we note that therapeutic applications could demand CREs of novel activity profiles across cell types which are unlikely to be captured by topic modeling.
- Our large-scale MPRA of synthetic elements allows us, to our knowledge for the first time, to empirically investigate how multiple sequence generation methods, cell type targets, sequence programs, and synthetic vs natural sequences impact specificity with statistical power.
- We have added qualitative statements to the paper noting that highly specific sequences in the fly brain have been developed with conceptually similar approaches:

Line 67:

“Efficient frameworks to generate sequences from predictive models could enable rational and interpretable design of candidate CREs^{4,34-39}, as demonstrated by recent work designing synthetic CREs to drive cell type specificity in drosophila^{40,41}. However, synthetic CREs designed using predictive models are untested in vertebrates, and their effectiveness compared to natural sequences remains unknown.”

In summary, we have included text to the paper describing important commonalities shared across all these works (Lines 53-71 & 542-560), that 1) synthetic CREs have been shown to drive specificity in different tissues and organisms and 2) sequence-based models can learn fundamental logics of regulatory grammar to drive

this specificity. We also have clarified through the text the specific advances and approaches unique to our paper, briefly summarized here:

- Our models are trained on large scale uniformly processed reporter data that are direct readouts of transcription rather than epigenetically correlated markers like ATAC/DHS. This distinction provides a high-performance model that is more directly related to the objectives of many therapeutically relevant design tasks (e.g. AAV and LNP based gene therapy)
- Our objective function designs sequences for cell type specificity as predicted directly by the oracle. This is fundamentally different from the approach by Taskiran et al. which optimizes for cell-specific topics that are predetermined to be cell-specific. We believe our approach is more robust and has greater flexibility than a function which reflects a linear combination of individual topic functions. It will be interesting to see in future work which process is shown as the optimal approach.
- Our large-scale testing of 51K synthetic elements and 24K genomic elements allowed us to directly compare specificity between different synthetic approaches and natural sequences.
- We have included in-vivo, whole-organism validation in vertebrates, including 6 elements in zebrafish and 2 elements in mice. This is the first instance that we know of that a truly synthetic CRE generated by a deep learning model has been demonstrated to work in mammals.

R3C7. The final claim of the abstract is that the method is able to “write regulatory code that is fit-for-purpose in vivo across vertebrates.” The experiments in brain and liver are promising, but are not sufficient to support that claim. It isn’t clear if it is possible to collect a in vivo dataset large enough to apply CODA to. Alternatively, the method requires that the regulatory grammar of the in vivo cell type is similar enough to the in vitro system for CODA to be useful. That demonstration is also not shown at a large scale.

We agree that this claim is an overstatement as written, and does not convey that it is based on a smaller number of in-vivo experiments than the large-scale MPRA datasets. Accordingly, we have updated the abstract to now state our ability to engineer sequences without the suggestion we have demonstrated reliable success in vivo:

Line 36:

“Together, we provide a generalizable framework to prospectively engineer CREs from MPRA models and demonstrate the required literacy to write fit-for-purpose regulatory code”

Minor Points

R3C8. In figure 1, a continuous representation should be used to describe the relationship between Malinois predictions and the open chromatin and other genomic measurements. P values are difficult to interpret given the large number of data points.

We have ensured that all of the values shown in Fig 1e are continuous. We realized that a **Methods** description for this analysis was lacking and have since updated it. Briefly, the MPRA over the 2.1MB GATA1 locus includes nearly all (dropouts indicated on figure) 200-bp oligos spanning the region, using a step size of 50bp, as generated for the ENCODE project (ENC IDs are included in **Supplementary Table 1**). The signal for each base pair is generated by averaging the MPRA activity of each oligo that overlaps that base pair. The Malinois activity is generated following the same strategy, using predictions for each oligo rather than empirical data. Signal is directly overlaid and colored to show non-overlapping sections. Pearson’s r and Spearman’s rho between the two tracks, as well as DHS signal from K562 are also included in **Fig S4B**.

We have now included an updated **Methods** section for this analysis:

Line 644:

“For locus-specific benchmarking we aggregated the \log_2FC of oligos that tile the GATA1 locus (OL43) following the same counts filtering steps as described above. We generated per-genome-base activity measurements by averaging the MPRA activity of each oligo that overlaps that base pair.”

R3C9. Supplemental figure 3 does not have scatter plot comparisons for the SK-N-SH cells.

The axis labels for this plot initially were listed only once and denoted all plots within a column. This was unclear, and could make it appear like the scatter plots for SK-N-SH were missing. We've updated the labeling to make it clear in the manuscript and provide the updated panels below:

Updated Supplementary Figure 4a

R3C10. Starr-seq is a continuous signal and an assay similar to MPRA. It would be helpful to see both treated as a continuous signal to look for correspondence.

We have included an additional analysis outside of the deeptools pileup plots shown in Figure 2f, comparing Malinos signal to STARR-seq signal in peaks of activity identified by STARR-seq as well as upstream and downstream flanking regions. Shown in **Supplementary Fig 6C** for the two cell lines (K562 and HepG2) that have genome-wide STARR-seq, we see the Malinois signal is significantly different in peaks vs flanking regions, highlight correspondence between a Malinois and an MPRA-orthogonal reporter assay. The panel is included for the reviewer below as well:

Supplementary Figure 6. Malinois concordance with DHS/H3K27ac/STARR. (c) Malinois genome-wide predictions are significantly associated with candidate CRE mapping (DHS-seq, and H3K27ac ChIP-seq) and orthogonal signals of CRE functional characterization (STARR-seq). Boxplots display average signal generated by Malinois genome-wide predictions within peaks annotated using DHS, H3K27ac, or STARR-seq (orange) compared to paired upstream (blue) and downstream (green) flanking regions. Boxes demarcate the 25th, 50th, and 75th percentile values, while whiskers indicate the outermost point with 1.5 times the interquartile range from the edges of the boxes. Stars indicate a significant ($-\log_{10} p\text{-value} > 100$) for two t-tests comparing signals within peaks and both upstream and downstream regions outside of peaks.

R3C11. In figure 2c, the performance of the DHS natural and Malinois natural are surprisingly poor. Why is open chromatin and H3K27ac a poor indicator of specificity? Does the relatively poor Malinois performance suggest reproducibility issues?

We agree with the reviewer that the observation that open chromatin combined with H3K27ac filtering is a poor indicator of specificity could be surprising to many readers. Figure 2c displays that natural sequences, either nominated from DHS signal or Malinois MPRA predictions, do generally have worse performance at identifying highly specific sequences. Empirically, we observe that natural DHS sequences are generally less active than synthetic sequences, and have higher amounts of pleiotropy as defined by activity in the MPRA. This could be an endogenous feature of natural sequences, perhaps driven by evolution's reuse and cooption of regulatory elements in different tissues, or due to discrepancies between quantitative DHS signal and transcriptional output. As we discuss in detail in R1C21, open chromatin signals display only a weak, but significant, correlation with MPRA (Kim et al. 2021; Puig-Alcaraz et al. 2016) or gene expression (Nair et al. 2021). For Malinois-nominated natural sequences, we see that they actually perform better in terms of specificity. This could be due to predictions being directly on a specific 200 bp outside of the endogenous context of the genome, nearby sequences repressing on-target activity or inducing off-target expression, cooperative effects of multiple CREs acting on a specific promoter, and other endogenous impacts may explain the difference between open chromatin measures and both empirical and predicted reporter results. We have included these ideas in the discussion:

Line 565:

"This is possibly a reflection of selective pressure that has shaped DHS elements across mammalian evolution to be optimized for redundancy, versatility, and modular function^{91,92} or alternatively, a weak correlation between quantitative DHS signal and CRE activity."

While the performance of natural sequences vis a vis specificity may be relatively poor, as an activity predictor, Malinos actually performs well on both natural and synthetic sequences. If we separate correlations between Malinois predictions vs DHS natural using K562 as an example, Malinois-natural sequences have a Pearson correlation to empirical results of 0.91 while DHS-natural have a correlation of 0.86. This suggests that Malinois does not have a reproducibility issue related to the source of the sequence, and instead points to

biological differences between the sequence sources. We summarize these results in more depth in **Supplementary Figure 16** and reproduce the figure in our response to R3C13.

R3C12. For each cell type (and across cell types) how well do the new MPRA results correlate with the previous MPRA results for the same sequences? Are there experimental differences, either procedure or source of the cell line, that could be driving performance?

Cross library reproducibility

To remove the possibility of experimental noise impacting performance assessments for different sequence design methods, all natural and synthetic cell-specific designs are tested in one library. Therefore, technical variability between experiments will not induce differences in function between natural and synthetic elements. Moreover, we retained all experimental procedures, cell lines, and analyzes consistent between MPRA data used in training and final CODA MPRA that compared natural and synthetic CREs simultaneously. To assess the possibility of systematic experimental differences between old and new experiments impacting the generalizability of our models and observations, we compared MPRA measurements for 594 control elements that were tested in both the initial training set and the final CODA library. We show MPRA measurements for individual sequences are highly correlated between experiments (**Supplementary Figure 14**). We do observe a small subset (<5%) of elements that display differences between each library in the HepG2 cell line. We have not been able to discern any sequence-specific or technical reason for these differences, but it does not appear to impact sequence generation abilities for HepG2.

Line 268-277:

*“We experimentally tested the library of 77,157 natural and synthetic sequences (**Figure 2b**) to determine if machine-guided sequence design could reliably generate biologically functional elements with desired activity. In total, the library included 51,000 synthetic sequences (36,000 standard and 15,000 motif-penalized), 24,000 natural sequences (12,000 DHS-natural and 12,000 Malinois-natural), and 2,157 experimental controls. We quantified activity of an individual CRE as the \log_2 fold change (\log_2FC) of expression of the reporter gene driven by the CRE compared to a set of negative controls (**Figure 2b,c**). A set of 594 control elements shared with the training data libraries confirms the high reproducibility of MPRA measurements across experiments (Pearson’s r 0.97, 0.81, and 0.98 for K562, HepG2, and SK-N-SH, respectively; **Supplementary Figure 14**).”*

“Supplementary Figure 14. MPRA measurements for individual elements are reproducible between different experiments and libraries. MPRA activity measurements made in the training data plotted on the x-axis are highly

correlated with later measurements made in the CODA library on the y-axis. Measurements were made in K562 (teal), HepG2 (gold), and SK-N-SH (red).”

R3C13. The proportion of oligos with error that are too high to serve a reliable indicator should be more thoroughly described and addressed. A computational model that produces sequences with high variability should be considered inferior.

We agree with the reviewer that a model, whose predictions of designed sequences correlate with MPRA measurements better than Malinois', would be considered a superior model. However, to our knowledge, such a model has not yet been published. In fact, we have been impressed with the ability of Malinois to accurately predict synthetic sequences. Prior to receiving the experimental validation results, we had concerns that synthetic sequences were out-of-distribution compared to the training dataset resulting in spurious predictions. Instead, prediction performance for synthetic elements appears to be similar, if not slightly better than natural sequences selected for the same objective (**Supplementary Figure 16**). This suggests the model is generally performing well, and more importantly, it accurately predicts the synthetic sequences it assists in designing. Furthermore, Malinois is outperforming all published models and performs similarly to the best unpublished models of which we have knowledge. In **R1C4**, we describe how we retrospectively benchmark Malinois' predictions of synthetic sequences to two published models, MPRA-DracoNN and Enformer, and to two models still in development, MPRA-LegNet and ReporterNet. In panels (b) and (d) in **Reviewer Figure R1C1**, we show that both published models exhibit poor performance compared to Malinois when predicting the activity of synthetic sequences. On the other hand, in **Reviewer Figure R1C4**, we show that Malinois performs comparably to MPRA-LegNet and ReporterNet in that task. We recognize the possibility that the models above could achieve better performance when predicting sequence designs of their own creation. Unfortunately, none of those models have been used to design sequences that are also empirically validated for specificity.

Supplementary Figure 16. Granular Malinois prediction performance of CODA library. Pearson correlation coefficient values between Malinois activity predictions and MPRA empirical measurements in K562 (teal), HepG2 (gold), and SK-N-SH (red) of the CODA library broken down by method group.

In response to multiple comments, we added text in the discussion that future, more accurate models will benefit sequence generation:

Line 599:

“As the technology underlying sequence-to-function models continues to evolve, are mechanistically interrogated through ablation studies, and are trained on high-quality MPRA data sets, we expect synthetic element designs to become even more reliable and reduce the experimental burden for in vitro and in vivo validation.”

R3C14. The propeller plots are a useful visualization and do confirm that the sequences with the largest fold difference are also the ones high specificity. However, it would also be useful to have visualization for each pair of cell types that compare the specificity in 2D space.

Following the reviewer's suggestion, we have included 2D scatter plots (**Supplementary Figure 20**) comparing each pair of cell types for each sequence group.

“Supplementary Figure 20. Cell-type activity comparisons. Scatter plots comparing empirical $\log_2(\text{Fold-Change})$ activity in each pair of cell types for each design group. Color indicates the target cell type for which sequences were designed (synthetic) or selected (natural).”

References

- de Almeida BP, Schaub C, Pagani M, Secchia S, Furlong EEM & Stark A (2024) Targeted design of synthetic enhancers for selected tissues in the *Drosophila* embryo. *Nature* 626, 207–211.
- Avsec Ž, Agarwal V, Visentin D, Ledsam JR, Grabska-Barwinska A, Taylor KR, Assael Y, Jumper J, Kohli P & Kelley DR (2021) Effective gene expression prediction from sequence by integrating long-range interactions. *Nat. Methods* 18, 1196–1203.
- de Boer CG & Taipale J (2024) Hold out the genome: a roadmap to solving the cis-regulatory code. *Nature* 625, 41–50.
- Boyle AP, Davis S, Shulha HP, Meltzer P, Margulies EH, Weng Z, Furey TS & Crawford GE (2008) High-resolution mapping and characterization of open chromatin across the genome. *Cell* 132, 311–322.
- Buenrostro JD, Giresi PG, Zaba LC, Chang HY & Greenleaf WJ (2013) Transposition of native chromatin for fast and sensitive epigenomic profiling of open chromatin, DNA-binding proteins and nucleosome position. *Nat. Methods* 10, 1213–1218.
- Farley EK, Olson KM, Zhang W, Brandt AJ, Rokhsar DS & Levine MS (2015) Suboptimization of developmental enhancers. *Science* 350, 325–328.
- Fornes O, Av-Shalom TV, Korecki AJ, Farkas RA, Arenillas DJ, Mathelier A, Simpson EM & Wasserman WW (2023) OnTarget: in silico design of MiniPromoters for targeted delivery of expression. *Nucleic Acids Res.* 51, W379–W386.
- Hrvatin S, Tzeng CP, Nagy MA, Stroud H, Koutsoumpa C, Wilcox OF, Assad EG, Green J, Harvey CD, Griffith EC & Greenberg ME (2019) A scalable platform for the development of cell-type-specific viral drivers. *Elife* 8. Available at: <http://dx.doi.org/10.7554/eLife.48089>.
- Jindal GA & Farley EK (2021) Enhancer grammar in development, evolution, and disease: dependencies and interplay. *Dev. Cell* 56, 575–587.
- Kim DS, Risca VI, Reynolds DL, Chappell J, Rubin AJ, Jung N, Donohue LKH, Lopez-Pajares V, Kathiria A, Shi M, Zhao Z, Deep H, Sharmin M, Rao D, Lin S, Chang HY, Snyder MP, Greenleaf WJ, Kundaje A & Khavari PA (2021) The dynamic, combinatorial cis-regulatory lexicon of epidermal differentiation. *Nat. Genet.* 53, 1564–1576.
- Korecki AJ, Cueva-Vargas JL, Fornes O, Agostinone J, Farkas RA, Hickmott JW, Lam SL, Mathelier A, Zhou M, Wasserman WW, Di Polo A & Simpson EM (2021) Human MiniPromoters for ocular-rAAV expression in ON bipolar, cone, corneal, endothelial, Müller glial, and PAX6 cells. *Gene Ther.* 28, 351–372.
- van Laarhoven PJM & Aarts EHL (1987) Simulated annealing. In P. J. M. van Laarhoven & E. H. L. Aarts, eds. *Simulated Annealing: Theory and Applications*. Dordrecht: Springer Netherlands, pp.7–15.
- Lawler AJ, Ramamurthy E, Brown AR, Shin N, Kim Y, Toong N, Kaplow IM, Wirthlin M, Zhang X, Phan BN, Fox GA, Wade K, He J, Ozturk BE, Byrne LC, Stauffer WR, Fish KN & Pfenning AR (2022) Machine learning sequence prioritization for cell type-specific enhancer design. *Elife* 11. Available at: <http://dx.doi.org/10.7554/eLife.69571>.
- Linder J & Seelig G (2021) Fast activation maximization for molecular sequence design. *BMC Bioinformatics* 22, 510.
- Meuleman W, Muratov A, Rynes E, Halow J, Lee K, Bates D, Diegel M, Dunn D, Neri F, Teodosiadis A, Reynolds A, Haugen E, Nelson J, Johnson A, Frerker M, Buckley M, Sandstrom R, Vierstra J, Kaul R & Stamatoyannopoulos J (2020) Index and biological spectrum of human DNase I hypersensitive sites. *Nature* 584, 244–251.
- Movva R, Greenside P, Marinov GK, Nair S, Shrikumar A & Kundaje A (2019) Deciphering regulatory DNA sequences and noncoding genetic variants using neural network models of massively parallel reporter assays. *PLoS One* 14, e0218073.
- Nair VD, Vasoya M, Nair V, Smith GR, Pincas H, Ge Y, Douglas CM, Esser KA & Sealfon SC (2021) Differential analysis of chromatin accessibility and gene expression profiles identifies cis-regulatory elements in rat adipose and muscle. *Genomics* 113, 3827–3841.
- Osterwalder M, Barozzi I, Tissières V, Fukuda-Yuzawa Y, Mannion BJ, Afzal SY, Lee EA, Zhu Y, Plajzer-Frick I, Pickle CS, Kato M, Garvin TH, Pham QT, Harrington AN, Akiyama JA, Afzal V, Lopez-Rios J, Dickel DE, Visel A & Pennacchio LA (2018) Enhancer redundancy provides phenotypic robustness in mammalian development. *Nature* 554, 239–243.
- Puig-Alcaraz C, Fuentes-Albero M & Cauli O (2016) Relationship between adipic acid concentration and the core symptoms of autism spectrum disorders. *Psychiatry Res.* 242, 39–45.
- Sinai S, Wang R, Whatley A, Slocum S, Locane E & Kelsic ED (2020) AdaLead: A simple and robust adaptive greedy search algorithm for sequence design. *arXiv [cs.LG]*. Available at: <http://arxiv.org/abs/2010.02141>.
- Taskiran II, Spanier KI, Dickmanken H, Kempynck N, Pančíková A, Ekşi EC, Hulselmans G, Ismail JN, Theunis K, Vandepoel R, Christiaens V, Mauduit D & Aerts S (2024) Cell-type-directed design of synthetic enhancers. *Nature* 626, 212–220.
- Tewhey R, Kotliar D, Park DS, Liu B, Winnicki S, Reilly SK, Andersen KG, Mikkelsen TS, Lander ES, Schaffner SF & Sabeti PC (2016) Direct Identification of Hundreds of Expression-Modulating Variants using a Multiplexed Reporter Assay. *Cell* 165, 1519–1529.

Reviewer Reports on the First Revision:

Referee #1 (Remarks to the Author):

The authors have adequately addressed previous concerns, significantly improving clarity and coherence. I have some minor concerns regarding the problem statements, which appear as "straw man" arguments.

For example, in the intro (page 2, line 62-67), the authors state that genomic DNNs have been largely applied to chromatin demarcated by DHS rather than direct CRE activity. This is misleading as there are plenty of studies that fit quantitative CRE activity data from MPRAs, lentiMPRA, and STARR-seq. Proper citations should be made and the statement should be made more precise.

Also, the authors state that experimentation are incapable of global searches over all possible sequence combinations within the size of human CREs. This is also misleading as computational approaches are also limited in this search. For a sequence of length 200, there are $2.58225e+120$ possible combinations! The motivation for the method should be more precise and to the point. There is no need to create a gap in the field where there is no gap.

The data splits are unclear. Information about the MPRAs testing allelic effects is not provided in a form that is comprehensible to a reader without having to go through each dataset. How the loci probed were split so as to ensure no train-test leakage is not clear.

Malinois is clearly a very odd architecture choice due to its inefficiency in processing 400 nts of padding as opposed to the 200 variable sequences that are important. The authors should highlight this oddity and explain the inefficiencies of this choice. The authors could have just initialized the filters of a CNN that considers 200 nts; even with all filters from 3 conv layers of basset with the same pooling is just outside the 200 nts, so the necessary padding would be much less and handled within the backend using 'zero padding'. The lack of alternative model comparisons, beyond a 2016-based model, such as LegNet and beyond, should also be highlighted as Malinois was used just for the purposes of this study but is not likely to be close to state-of-the-art for CRE activity prediction tasks (see random promoters DREAM challenge). I understand a comprehensive comparison is beyond the scope of the manuscript -- any expectation and confusion can be remedied by carefully wording that malinois is not a central part of the main thesis, which (I believe) is that trained sequence-function DNNs can be used to design cell-type specific regulatory sequences as they can act as a scoring function. The challenge is that sequence space is large and so navigating it can be computationally prohibitive. CODA framework can help navigate this search space relatively efficiently, compared to 0-order Markov model. More interestingly, the local search methods all work relatively well (including single-nucleotide evolution (Ibrahim et al 2023, Vaishnav et al 2023), which suggests that there are many modes within the local sequence space that generate valid functional sequences.

Referee #2 (Remarks to the Author):

I appreciate that the authors have done extensive work to revise their manuscript. These have addressed many of my concerns, however, the way in which this data is presented still does not accurately reflect the data and this main concern remains. The manuscript needs extensive rewriting to accurately reflect the data and to tone down some of the conclusions. This would make the paper far stronger. The main concerns relate to referring to cell type specificity or cell specificity when referring to specific expression in a particular transformed cell line and how these transformed cell lines relate to in vivo expression patterns. Another key issue is the use of syntax when looking at combinations of TFs rather than syntax.

The data and findings in the manuscript are compelling and I'm disappointed to see how it is presented. I do not feel my comments have been addressed sufficiently and to a reader the conclusions are still misleading. The main one being that synthetic enhancers can be designed to give more tissue specificity than genomic elements. This is true for the specific transformed cell lines they have tested and may be true more globally but is not validated in this manuscript. This manuscript shows that synthetic elements outperform genomic elements to drive expression specifically in transformed cell lines which may be logical given that evolution has not selected for elements that drive expression in a particular cancer cell line. It is impressive that these elements do drive expression in related cell types in vivo, but they are related, and it is not the same cell-specificity and this needs to be clarified. The data is good but there is no need to overstate and inaccurately report the data. The focus on genomic vs synthetic to me is still a distraction for the main point of this paper, which nicely demonstrates that you can design enhancers with desired specificity using synthetic sequences and MPRA data and that using synthetic enhancers increases search space and can find novel mechanisms of driving the desired specificity.

Specific examples of problematic statements:

Through large-scale in vitro validation, we show that synthetic sequences are more effective at driving cell type-specific expression compared to natural sequences from the human genome and maintain specificity when tested in vivo.

The synthetic sequences outperform natural sequences in the human genome in driving in expression in transformed cell lines, there is not enough data to make this claim so sweepingly about cell-type specific expression globally as the assay was done in cell lines in which the genome has not evolved. The in vivo expression is within the same organ systems as the cell lines of the original study, which is impressive and interesting, but it is not accurate to say maintain specificity when tested in vivo here. This is still very misleading.

Synthetic sequences exhibit distinct sequence syntax associated with activity in the on-target cell type and simultaneously reduce activity in off-target cells.

There is no syntax shown in the manuscript, it would be more accurate to use the term different combinations of TFs or TF vocabulary.

Main text:

Here we present a method to engineer novel synthetic CREs capable of driving gene expression with cell type specificity.

It is cell specific within transformed cell lines. This should be clarified.

We leverage innovations in modeling regulatory grammar across cell types

What are the innovations in modeling regulatory grammar? Isn't the innovation the use of MPRA and models that look at TFBS combinations and vocab?

*Coupled to sequence generation algorithms, we deploy our model to generate thousands of **cell type-specific, synthetic CREs, which we functionally validate using MPRA and in vivo using mouse and zebrafish.***

It would be more accurate to say And we see the expression in related cell types within the same organs in vivo using mouse and zebrafish.

Results:

We were able to identify naturally occurring sequences with cell type specificity.

Would be better and more accurate to state ...With expression specifically in particular cell lines.

Synthetic sequences from all three algorithms outperformed both groups of natural sequences as cell type-specific CREs in all three cell types.

This needs to be rewritten to reduce the misleading nature of this statement to something like synthetic sequences outperformed genomic sequences as cell line specific CREs in all three transformed cell lines. Although this may be expected given the nature of these transformed cell lines and the fact that the genome has not evolved to drive expression specifically in transformed cell lines.

We sought to link sequence content to the responsible regulatory syntax.

The manuscript as it stands only finds combinations of TFs and TF vocab, I cannot see any evidence of regulatory syntax. Please do not refer to syntax when the syntax is not studied.

suggesting that natural sequences are less likely to use repressive grammar in constructing cell type-specific CREs.

I find this misleading as it is at over-generalization about cell type specificity and the fact that genomic sequences are less likely to use repressors, while this may be the case in the transformed cell lines it does not mean that this is the case in general. It would be more accurate to say:

This suggests that genomic sequences that show specific expression within the transformed cell lines are less likely to use repressive TFs in constructing transformed cell line specific CREs.

The section starting: *Complex semantic architectures are syntactically differentially deployed in natural and synthetic sequences*

This section is hard to understand and ultimately boils down to different combinations of TFs are seen in different types of CREs. The use of semantic architectures seems overly complex to me, and it would be better to simply state what the data finds. The section title suggests finding different syntax in different enhancers, but they are talking about the use of different TFs namely activators and repressors.

It is unclear to me what these “programs” identified by NMF are supposed to be. It is unclear how these relate to more than just the combo of binding sites.

Using empirical MPRA results, Malinois contribution scores, in silico predictions of tissue-specific epigenetic signals, and element syntax, we nominated three liver and three neuronal-specific CREs for in vivo characterization in zebrafish embryos (

These are 3 liver transformed cell line and neuroblastoma cell lines CREs, that they decided to see if they had liver, and neural activity in vivo. This should be stated. It is impressive that these do drive expression in the organs from which the transformed cell line is derived, and this should be discussed, rather than oversimplifying to suggest that the cell line and the location of expression are demonstrating conserved cell-type specificity.

It is unclear how they used element syntax to select enhancers, it appears to me that they used combination of TFBS. In the methods section they don't mention anything about looking at binding site arrangement. Thus is it really syntax or just combinations of TFs?

Remarkably, we detected minimal off-target expression in non-targeted cell types

Please expand in the main text. It looks like there is expression in non-liver and neural cells in the zebrafish assays. This is fine but needs to be explained to the reader.

We confirmed cortex specific expression with focal activity occurring in the neurons at neocortical layer 6 and at subplate neurons (Figure 4e-g, Supplementary Figure 37a,b). Please explain the link between the cell line used for the MPRA assays (the neuroblastoma cell line) and the cell types in which the

enhancer is found beyond the fact that they are neural. Why would one expect these CREs to be only expressed within neocortical layer 6?

Discussion:

Due to the intractability of fully searching sequence space, CODA cannot assuredly identify global specificity maxima, but our exhaustive evaluation of natural sequences demonstrates the design methods we used can identify synthetic sequences that regularly outperform natural ones with 1000-fold greater efficiency compared to previous methods using a zero-order Markov approach (Supplementary Figure 38) 40,41 .

Again the sequences outperform in designing enhancers that drive expression specifically in transformed cell lines. I understand the use of outperform in different fields but this needs to be clear, so it is not misleading.

Synthetic sequences designed by CODA easily outperform natural sequences in driving cell type-specific gene expression in a reporter system, which suggests that novel functions can be programmed into CREs and interpreted by human cells.

.... In driving cell line specific gene expression, which suggests that synthetic enhancer can be programmed into CREs and interpreted by transformed human cell lines. Again this is likely to be the case when you use a cancer cell line to do the assays and needs to be stated for the reader.

The dearth of natural sequences capable of achieving exquisite specificity in a desired cell type in our study highlights the difficulty of using human genomic sequences to achieve non-natural objectives for which evolution may not have acted on.

My comments relating to this have not been addressed sufficiently and this needs to be spelled out to the reader. The synthetic elements will of course do better as the focus is on elements that are driving expression in transformed cell lines which the genome has not evolved to do.

This suggests that our models have learned a component of the foundational rules governing CREs and possess the ability to extrapolate this knowledge to unobserved or rarely observed syntax combinations. Replace syntax with TFs combinations.

we were able to identify natural sequences in the genome with moderate proficiency for cell-specific activity, albeit to a lesser degree than synthetics.

we were able to identify sequences in the genome with moderate proficiency for cell-specific activity, albeit to a lesser degree than synthetics in the transformed cell lines.

(Remarks on code availability): The code is easier to understand than it was in the previous version. I'll defer to the other two reviewers on this who are more computational experts than I.

Referee #3 (Remarks to the Author):

Summary

The authors do a very thorough job addressing my concerns and those of the other reviewers. In particular, the comparisons to other methods (Enformer, MPRA-DracoNN) provide better benchmarking. The new TF-ModISco results improve the confidence that the models are learning the relevant regulatory code. However, there are still two major concerns. First, there are still key details missing in the transfer learning procedure, including whether potential circularity. Second, the manuscript still relies heavily on MPRA experiments that have still not been adequately described.

Major points:

The motivation behind Malinois and its role in the manuscript are made much more clear by the additions to the text. However, it also raises new questions:

- Given the improvement that transfer learning provides, how does Bassett on its own do in predicting MPRA activity? This is critical because it gets to whether the neural network model requires MPRA data or whether open chromatin is sufficient.
- How much of Malinois' performance boost relative to MPRA-DracoNN and Enformer are due to transfer learning?
- The authors have addressed my original concerns of circularity, but new ones have been introduced. It seems like the parameters from the Bassett architecture, which is trained on a global library of open chromatin, have been used to initialize the models. This could be then be introducing circularity across many of the loci.

The lack of detail on the MPRA experiments are still a major concern. There is still no preprint.

Other Items:

The addition of TF-ModISco substantially improves the manuscript. The interpretation of individual motifs, especially those recognized by cell type-specific transcription factors.

The comparisons to Enformer and MPRA-DracoNN improve the manuscript.

The thorough comparison to previous methods, including the newly published ones, is helpful.

The new supplemental figure 20 provides a very compelling visualization of cell type-specificity.

Reviewer #1

The authors have adequately addressed previous concerns, significantly improving clarity and coherence. I have some minor concerns regarding the problem statements, which appear as "straw man" arguments.

We thank the reviewer again for the constructive comments during the review process and are pleased to know we satisfactorily addressed previous concerns. We hope the following responses help to address and clarify the reviewer's remaining minor concerns. New in-line references are provided in long form for ease of review.

R1C1. For example, in the intro (page 2, line 62-67), the authors state that genomic DNNs have been largely applied to chromatin demarcated by DHS rather than direct CRE activity. This is misleading as there are plenty of studies that fit quantitative CRE activity data from MPRA, lentiMPRA, and STARR-seq. Proper citations should be made and the statement should be made more precise.

We agree with the reviewer that our original text did not properly highlight recently published approaches that model CRE activity measured by reporter assays. We have updated the text and added citations to address this issue, clearly stating that while the majority of DNNs have been trained on chromatin features, recent work has shown advances when trained on functional CRE activity data.

Line 63:

"While these sequence models are promising tools for the interpretation of genetic sequences^{27,28,31,33}, they have largely been trained on, and predict, proxies of regulatory activity such as regions of open chromatin demarcated by DNase Hypersensitivity sites (DHS), rather than direct CRE activity measured by reporter assays. Recent works, such as DeepSTARR³¹ and EnformerMPRA²¹ have demonstrated that training such models directly on reporter assays can provide substantial performance gains."

R1C2. Also, the authors state that experimentation are incapable of global searches over all possible sequence combinations within the size of human CREs. This is also misleading as computational approaches are also limited in this search. For a sequence of length 200, there are 2.58225e+120 possible combinations! The motivation for the method should be more precise and to the point. There is no need to create a gap in the field where there is no gap.

We wholeheartedly agree with the reviewer that it is unfeasible for **both** experimental and computational analysis to perform global searches, and had meant this as the gap that exists. We had previously noted at lines 65-67 (numbering corresponds to previous submission) in the manuscript that computational approaches are limited in global searches, even with their million times increased speed over experimental methods. However, we see that the original sentence might not have been clearly structured as originally written. We have updated the text to help make clear that global searches are impossible for computational approaches.

Line 68:

“Lastly, although computational models are millions of times faster than experimentation, these models are still incapable of global searches over all possible sequence combinations within the size of a typical human CRE. Efficient frameworks to generate sequences from predictive models could help address this gap and enable rational and interpretable design of candidate CREs^{4,34–39}, as highlighted by recent work designing synthetic CREs to drive cell type specificity in drosophila^{40,41}.”

We expect the scale of the problem is underappreciated by many in the field. As a result, we reference the vast size of the search space at line 45 to underscore the problem and the opportunities for discovery. We are in full agreement with the reviewer that *both* computational and experimental approaches are limited in their search capabilities, and we welcome further suggestions from the reviewer if they believe there are additional modifications we can make to emphasize this.

Line 45:

“Indeed, 200 base pairs of DNA can encompass over 2.58×10^{120} possible sequences, more combinations than atoms in the observable universe. This unexplored CRE sequence space, combined with our current poor understanding of the underlying principles driving CRE function, leave available a vast untapped reservoir of potential CREs for clinical and biotechnological applications⁸.”

R1C3. The data splits are unclear. Information about the MPRA testing allelic effects is not provided in a form that is comprehensible to a reader without having to go through each dataset. How the loci probed were split so as to ensure no train-test leakage is not clear.

We appreciate the reviewers attention to how we handle and describe our data splits, and the underlying data, to ensure there is no train-test leakage.

Clarifying data splits.

We have updated the **Methods** section to help clarify the data was split in a manner that ensures no train-test data leakage.

Line 657:

“Oligos from chromosomes 19, 21, and X were held out from the parameter training loop as a validation set guide hyperparameter tuning. Oligos from chromosomes 7, 13 were held out from both parameter training and hyperparameter tuning loops as a test set for reporting performance. Oligos from the remaining chromosomes were used in the training loop. Oligos that contain alternative alleles are assigned to the same chromosomes as the reference allele oligos.”

Detailed information on allelic effects.

We appreciate the reviewer highlighting a unique feature of our MPRA dataset. To provide additional details that may be of interest to the reader, we have expanded the **Methods** section that describes the dataset. We now provide the proportion of sequences in our dataset that were designed to test allelic effects and precisely how the sequences are designed.

Additional details can be found in our newly released preprint that describes the design and analysis of the entire dataset used for training in our manuscript (Siraj et al. 2024) (DOI:

<https://doi.org/10.1101/2024.05.05.592437>). All data related to the preprint and this manuscript is also publically available on the ENCODE portal (**Supplemental Table 1**, Link to Encode Portal).

Line 635:

“The majority of projects focused on testing the allelic effects of human genetic variation with the remaining projects testing only the reference sequences of the human genome. In total, 798,064 unique oligos were aggregated, originating from 10 independent experiments (from three different projects: UKBB [OL27, OL28, OL29, OL30, OL31, OL32, OL33], GTEx [OL41, OL42], OL15). The majority of the sequences used in our study (783,978) were designed to evaluate common human genetic variation associated with heritable complex traits. The majority of sequences (706,054) consisted of testing the reference and alternative allele, typically a single nucleotide substitution, centered within 200 bp of flanking sequence. Additional sequences (77,924) evaluated the 4 pairwise combinations of two independent variants. Variants were selected based on genetic fine-mapping with most variants being LD-partners of causal alleles and thus likely to not have a meaningful impact on cellular or organismal traits. The remaining sequences (14,086) originated from OL15 from which we selected the known DHS and H3K27ac sequences.”

R1C4. Malinois is clearly a very odd architecture choice due to its inefficiency in processing 400 nts of padding as opposed to the 200 variable sequences that are important. The authors should highlight this oddity and explain the inefficiencies of this choice. The authors could have just initialized the filters of a CNN that considers 200 nts; even with all filters from 3 conv layers of basset with the same pooling is just outside the 200 nts, so the necessary padding would be much less and handled within the backend using 'zero padding'. The lack of alternative model comparisons, beyond a 2016-based model, such as LegNet and beyond, should also be highlighted as Malinois was used just for the purposes of this study but is not likely to be close to state-of-the-art for CRE activity prediction tasks (see random promoters DREAM challenge). I understand a comprehensive comparison is beyond the scope of the manuscript -- any expectation and confusion can be remedied by carefully wording that malinois is not a central part of the main thesis, which (I believe) is that trained sequence-function DNNs can be used to design cell-type specific regulatory sequences as they can act as a scoring function. The challenge is that sequence space is large and so navigating it can be computationally prohibitive. CODA framework can help navigate this search space relatively efficiently, compared to 0-order Markov model. More interestingly, the local search methods all work relatively well (including single-nucleotide evolution (Ibrahim et al 2023, Vaishnav et al 2023), which suggests that there are many modes within the local sequence space that generate valid functional sequences.

We appreciate the reviewer’s question and take the opportunity to further describe here our choice of architecture and input size. We recognize that it would have been possible to design Malinois in a way that maintains the settings of Basset’s convolutional layers and reduces the input size to 216 nt. However, during the development phase of Malinois, we found it more straightforward to explore transfer learning from Basset by directly importing its weights from the convolutional layers and some of the weights from the *fully-connected layers*, thus requiring 600 nt inputs to ensure the preservation of hidden state shape after flattening the sequence length dimension. Although we acknowledge there are other ways of pursuing this, designing Malinois to be the *most* lightweight model possible was not a primary focus during our development process.

We have not yet encountered any practical disadvantages of having Malinois' input size requirements mirror those of the model from which it inherits its weights, and envisioned potential advantages that might be leveraged at latter stages. In addition, Malinois' input size of 600 nt may facilitate straightforward fine tuning on datasets with longer sequence lengths, a desirable feature for future engineering goals. Also, padding with portions of the plasmid vector flanking sequences opens up the possibility of capturing the potential difference in activity between an oligo and its reverse complement without sacrificing the ability to reverse complement both the input and flanking sequence as a data augmentation for training. Augmenting the training dataset by including the reverse complement of the 600 nt consisting of an oligo and the flanking sequences preserves the ability to predict the activity of the reverse complement oligo as in MPRA with respect to its orientation to the transcription start site. Otherwise, performing such an augmentation with no flanking sequences or with zero padding would be to assume that an oligo and its reverse complement will always have the same MPRA activity when inserted in the plasmid vector.

In order to ascertain inefficiencies in our choice, we would need to demonstrate that models with a trimmed-down input size are noticeably faster and will perform just as well when doing transfer learning. The reviewer's expectation of inefficiency may be true, however, we hope the reviewers appreciate such work would require careful and methodological analyses that are outside of the scope of this manuscript, or required for our main findings. We make the following updates to the methods to better explain the retention of the 600 nt input size requirement from Basset, acknowledge the impact on data augmentation by reverse complementation, and highlight reasonable options to mitigate issues surrounding variable input sizes.

Line 681:

"The final Malinois model is composed of three functional segments: (1) three convolutional layers with batch normalization and maximum value pooling, (2) a fully-connected linear layer to integrate positional and feature information from the previous hidden state after flattening, and (3) a stack of branched linear layers such that each output feature is a function of 4 independent transformations. As the first two segments are replicated from the Basset architecture⁴⁷, Malinois accepts batches of 4 x 600 arrays corresponding to one-hot encoded DNA sequences, so predictions for 200-nt MPRA oligos are made by padding inputs on both sides with constant sequences from the reporter vector backbone. This strict input sizing requirement ensures hidden states are appropriately shaped when transitioning between segments (1) and (2) of the model. Furthermore, this padding strategy enables us to use reverse complement data augmentation with awareness of the orientation of the 200-nt MPRA inserts with respect to the transcription start site in the reporter backbone. While not tested in this study, replacing the final strict max pooling layer with adaptive pooling or padding would allow flexibility in the input sizing requirements while maintaining all other components of the architecture."

We have gone through the manuscript and ensured that nowhere is Malinois referred to as 'state-of-the-art'. It is not our intention to position Malinois as state-of-the-art, but rather as sufficient for reliable CRE engineering, and we are open to changing any text in the manuscript that may suggest otherwise. During the revision process we have altered language, thanks to concerns raised by the reviewers, to inform the reader that other CNNs exist that could yield similar performance when used within CODA. In addition, we have now added to the discussion that several advances in DNA

modeling have been reported by other groups over the course of our study. We note in the manuscript we did not test the efficacy of BP-net, LegNet or other novel architectures highlighting immediate opportunities for improving efficacy of MPRA modeling (Linder et al. 2023; Penzar et al. 2023; Avsec et al. 2021; Agarwal et al. 2023; Rafi et al. 2024). Overall, we hope these changes convey that Malinois is not the central finding in the paper but instead an important component of a larger framework. If there are additional locations the reviewer thinks we could further improve clarity, we would be open to making changes.

Line 607:

“As the technology underlying sequence-to-function models continues to evolve, are mechanistically interrogated through ablation studies, and are trained on high-quality MPRA data sets, we expect synthetic element designs to become even more reliable and reduce the experimental burden for in vitro and in vivo validation. Over the course of this study, several advances in DNA modeling have been reported by other groups that would likely yield such improvements^{9,21,32,92,93}, but are not tested here.”

Reviewer #2

I appreciate that the authors have done extensive work to revise their manuscript. These have addressed many of my concerns, however, the way in which this data is presented still does not accurately reflect the data and this main concern remains. The manuscript needs extensive rewriting to accurately reflect the data and to tone down some of the conclusions. This would make the paper far stronger. The main concerns relate to referring to cell type specificity or cell specificity when referring to specific expression in a particular transformed cell line and how these transformed cell lines relate to *in vivo* expression patterns. Another key issue is the use of syntax when looking at combinations of TFs rather than syntax.

The data and findings in the manuscript are compelling and I'm disappointed to see how it is presented. I do not feel my comments have been addressed sufficiently and to a reader the conclusions are still misleading. The main one being that synthetic enhancers can be designed to give more tissue specificity than genomic elements. This is true for the specific transformed cell lines they have tested and may be true more globally but is not validated in this manuscript. This manuscript shows that synthetic elements outperform genomic elements to drive expression specifically in transformed cell lines which may be logical given that evolution has not selected for elements that drive expression in a particular cancer cell line. It is impressive that these elements do drive expression in related cell types *in vivo*, but they are related, and it is not the same cell-specificity and this needs to be clarified. The data is good but there is no need to overstate and inaccurately report the data. The focus on genomic vs synthetic to me is still a distraction for the main point of this paper, which nicely demonstrates that you can design enhancers with desired specificity using synthetic sequences and MPRA data and that using synthetic enhancers increases search space and can find novel mechanisms of driving the desired specificity.

We thank the reviewer for their comments and appreciate their continued important and detailed feedback as we refine the language in the manuscript. We are grateful that they can see the 'extensive work' we have performed in revisions and that they think the results are 'compelling'. We also wholeheartedly agree with their perspective that being very measured in what we say we can do will make the manuscript 'far stronger' and we hope we have achieved this in our latest revision.

Clarifying scope of *in vitro* described cell type-specificity

In drafting the manuscript, we believed the extensive focus early in the text highlighting data generation in transformed cell lines K562, HepG2, and SK-N-SH would provide the appropriate context for readers to later interpret comments about cell type-specificity. Based on this reviewer's suggestions, we appreciate this may not universally be the case.

We have now clarified the text early in the **Main** section and throughout the **Results** in accordance with the reviewer's specific comments below to delineate the scope of cell types used to design CREs and the quantification of cell type-specificity. We also clarify the distinction between the cell lines and analogous but distinct cells and tissues *in vivo*. Changes related to this concern can be found at lines 31, 87, 89, 94, 288, 304, 557, 562, 571, and 583 and are presented in detail in the point-by-point responses below.

Refining terminology pertaining to motif analysis

Based on the reviewer's comments, we realize that CRE "syntax" can be justifiably interpreted as the quantitative measurement of spatial TF organization. We agree that our analysis of motif usage does not account for ordering and have **adjusted our language to make clear the focus of our model interpretation is on TF content and combinations**, and highlight the relevant changes in the

responses to the specific comments below. In summary, we make changes at lines 34, 201, 345, and 578 as directed by the reviewer.

R2C1. *“Through large-scale in vitro validation, we show that synthetic sequences are more effective at driving cell type-specific expression compared to natural sequences from the human genome and maintain specificity when tested in vivo.”*

The synthetic sequences outperform natural sequences in the human genome in driving in expression in transformed cell lines, there is not enough data to make this claim so sweepingly about cell-type specific expression globally as the assay was done in cell lines in which the genome has not evolved. The in vivo expression is within the same organ systems as the cell lines of the original study, which is impressive and interesting, but it is not accurate to say maintain specificity when tested in vivo here. This is still very misleading.

We have added details to the **Abstract** shown below to address the reviewer’s clarity concerns.

Line 31:

“Through large-scale in vitro validation, we show that synthetic sequences are more effective at driving cell type-specific expression in three cell lines compared to natural sequences from the human genome, and achieve specificity in analogous tissues when tested in vivo.”

R2C2. *“Synthetic sequences exhibit distinct sequence syntax associated with activity in the on-target cell type and simultaneously reduce activity in off-target cells.”*

There is no syntax shown in the manuscript, it would be more accurate to use the term different combinations of TFs or TF vocabulary.

After careful consideration and literature searches, we speculate this comment arises due to a broad, and sometimes loose definition of sequence syntax that currently exists in the field. We suspect the reviewer interprets syntax as the spacing and ordering of TF motifs. We have used alternative terms in statements previously containing the word syntax. Addressing this comment, we have made the following edits to sentences describing syntax.

Line 34:

“Synthetic sequences exhibit distinct motif vocabulary associated with activity in the on-target cell type and the simultaneous reduction of activity in off-target cells.”

Line 201:

“We observed successful reduction in initially enriched motifs and a simultaneous increase in motifs underutilized in earlier rounds (Supplementary Figure 9b), diversifying the motif content of CODA-proposed sequences for experimental evaluation.”

Line 345:

“Having found that synthetic CREs are more cell type-specific than both classes of natural sequences, we sought to link sequence content to the responsible TF vocabulary.”

Figure 3 has a new title removing the word syntax:

“Interpreting functional sequence content in engineered elements.”

Line 578:

“This suggests that our models have learned a component of the foundational rules governing CREs, and possess the ability to extrapolate this knowledge to rarely observed TF combinations.”

R2C3. *“Here we present a method to engineer novel synthetic CREs capable of driving gene expression with cell type specificity.”*

It is cell specific within transformed cell lines. This should be clarified.

While this study is a proof-of-concept in cell lines, we note that the methodology to engineer CREs based on MPRA data is applicable to any system where the assay can be applied. Several studies have clearly demonstrated MPRA is not limited to transformed cell lines and can be applied to *ex vivo* primary cell isolations (Bourges et al. 2020; Deng et al. 2023; Kim et al. 2021) and *in vivo* screens using AAV delivery (Chan et al. 2023; Hrvatin et al. 2019; Lagunas et al. 2023; Brown et al. 2022). We introduce a new sentence to clearly delineate the scope of data and experiments used to engineer CREs and provide context for the remainder of the manuscript.

Line 87:

“Here we present a method to engineer novel synthetic CREs capable of driving gene expression with cell type specificity which we deploy to design elements ab initio that regulate transgene expression across three transformed cell lines.”

R2C4. *“We leverage innovations in modeling regulatory grammar across cell types.”*

What are the innovations in modeling regulatory grammar? Isn't the innovation the use of MPRA and models that look at TFBS combinations and vocab?

We presume that the sentence as originally written might have given the impression that we are referring to innovations proposed by our study and not innovations in the field. To more clearly communicate the latter idea, we have explicitly noted prior work and added citations related to the innovations which we combined in this study.

Line 89:

“We achieve this by integrating prior innovations in modeling regulatory grammar across cell types^{33,47}, efficient sequence space searching^{35,36,48}, and the MPRA experimental system that can validate thousands of CREs in parallel^{13,21}.”

R2C5. *“Coupled to sequence generation algorithms, we deploy our model to generate thousands of cell type-specific, synthetic CREs, which we functionally validate using MPRA and in vivo using mouse and zebrafish.”*

It would be more accurate to say And we see the expression in related cell types within the same organs in vivo using mouse and zebrafish.

We agree with the reviewer's suggestion to more accurately summarize the *in vivo* studies. We have modified the suggested sentence incorporating the reviewer's suggestion.

Line 94:

"Coupled to sequence generation algorithms, we deploy our model to generate thousands of synthetic CREs with programmed specificity across three cell lines, which we functionally validate in vitro using MPRA and in vivo by probing physiologically related tissues in mouse and zebrafish."

R2C6. *"We were able to identify naturally occurring sequences with cell type specificity."*

Would be better and more accurate to state ...With expression specifically in particular cell lines.

We agree with the reviewer's suggestion to reinforce that the engineering effort was initially directed towards expression in particular cell lines.

Line 288:

"We were able to identify naturally occurring sequences with expression specificity in particular cell lines"

R2C7. *"Synthetic sequences from all three algorithms outperformed both groups of natural sequences as cell type-specific CREs in all three cell types."*

This needs to be rewritten to reduce the misleading nature of this statement to something like synthetic sequences outperformed genomic sequences as cell line specific CREs in all three transformed cell lines. Although this may be expected given the nature of these transformed cell lines and the fact that the genome has not evolved to drive expression specifically in transformed cell lines.

We have added the following text to the sentence highlighted by the reviewer to further emphasize the nature of the cell types of the study. Together with the edits presented in previous comments, we hope it will now be clear to the reader we are operating in the context of three transformed cell lines.

Additionally, in our response to R2C16 we address the impact of the lack of evolutionary pressures on the genome to create CREs that have specificity in transformed cell lines and the updates made to the **Discussion**.

Line 304:

"Synthetic sequences from all three algorithms outperformed both groups of natural sequences as cell type-specific CREs across all three cell lines."

R2C8. *"We sought to link sequence content to the responsible regulatory syntax."*

The manuscript as it stands only finds combinations of TFs and TF vocab, I cannot see any evidence of regulatory syntax. Please do not refer to syntax when the syntax is not studied.

Following the reviewer's comment, we have modified the suggested text. Please refer to our above responses to the summary of comments and R2C2.

Line 345:

*“Having found that synthetic CREs are more cell type-specific than both classes of natural sequences, we sought to link sequence content to the responsible **TF vocabulary**.”*

R2C9. *“suggesting that natural sequences are less likely to use repressive grammar in constructing cell type-specific CREs.”*

I find this misleading as it is an over-generalization about cell type specificity and the fact that genomic sequences are less likely to use repressors, while this may be the case in the transformed cell lines it does not mean that this is the case in general. It would be more accurate to say:

This suggests that genomic sequences that show specific expression within the transformed cell lines are less likely to use repressive TFBs in constructing transformed cell line specific CREs.

We agree with the reviewer that the sentence as originally written could inaccurately suggest a global lack of repressive grammar in natural sequences to achieve broad cell type specificity. Upon closer inspection of the text, we realized the statement was unnecessary and elected to remove it in its entirety. The deleted text was previously located at line 380.

R2C10. It is unclear to me what these “programs” identified by NMF are supposed to be. It is unclear how these relate to more than just the combo of binding sites.

We add additional text to the Methods section to clarify terminology corresponding to NMF for the general reader.

Line 1119:

*“We used non-negative matrix factorization (NMF), a **parts-based representation of data**⁸⁴, to model semantic relationships between motifs in our sequence library (scikit-learn version 1.2.2, initialized with NNDSVDAR, Frobenius loss). First we counted motif matches in each sequence with the contribution score-based motif hit mapping described above (Grant, Bailey, and Noble 2011) to generate $\mathbf{X} \in \mathbb{N}^{n \times f}$ where rows represent sequences in the library and columns correspond to motifs. The sample matrix \mathbf{X} can then be decomposed into the coefficients and features matrices $\mathbf{W} \in \mathbb{R}^{n \times k}$ and $\mathbf{H} \in \mathbb{R}^{k \times f}$, respectively. **These k -dimensional representations are referred to as “topics” in natural language processing and “programs” in gene expression analysis^{100,101}. These programs capture the frequency of TF motifs appearing in semantically similar CREs, and the CREs themselves are modeled as compositions of programs.**”*

R2C11. “Using empirical MPRA results, Malinois contribution scores, in silico predictions of tissue-specific epigenetic signals, and element syntax, we nominated three liver and three neuronal-specific CREs for in vivo characterization in zebrafish embryos”

These are 3 liver transformed cell line and neuroblastoma cell lines CREs, that they decided to see if they had liver, and neural activity in vivo. This should be stated. It is impressive that these do drive expression in the organs from which the transformed cell line is derived, and this should be discussed, rather than oversimplifying to suggest that the cell line and the location of expression are demonstrating conserved cell-type specificity.

It is unclear how they used element syntax to select enhancers, it appears to me that they used combination of TFBS. In the methods section they don't mention anything about looking at binding site arrangement. Thus is it really syntax or just combinations of TFs?

We agree with the reviewer that the sentence as originally written could have suggested a straightforward connection between a cell line and the organ from where it is derived. We have edited the sentence to better clarify our reasoning and expectations. We also note for the reviewer that further details are provided in the **Methods** subsection “CRE prioritization for *in vivo* validation”.

Line 473:

“Using empirical MPRA results, Malinois contribution scores, in silico predictions of tissue-specific epigenetic signals, and *manual inspection of motif organization*, we nominated three *HepG2-* and three *SK-N-SH-specific* CREs *which we anticipated to be liver- and neuronal-specific, respectively*, for in vivo characterization in zebrafish embryos (**Figure 4a, Methods, Supplementary Figure 32**).”

R2C12. “Remarkably, we detected minimal off-target expression in non-targeted cell types”

Please expand in the main text. It looks like there is expression in non-liver and neural cells in the zebrafish assays. This is fine but needs to be explained to the reader.

We have updated the main text to note that there is substantial signal due to autofluorescence in the yolk sac. We have updated the legends in supplementary figures 33 and 34 as well to note the faint autofluorescence signal in the yolk sac in the empty vector negative controls as well.

Line 486:

“Remarkably, we detected minimal off-target expression in non-targeted cell types *outside the autofluorescent yolk sac*.”

R2C13. “We confirmed cortex specific expression with focal activity occurring in the neurons at neocortical layer 6 and at subplate neurons (Figure 4e-g, Supplementary Figure 37a,b).”

Please explain the link between the cell line used for the MPRA assays (the neuroblastoma cell line) and the cell types in which the enhancer is found beyond the fact that they are neural. Why would one expect these CREs to be only expressed within neocortical layer 6?

We should clarify that we had no *a priori* expectation that these CREs would have specificity to any anatomy (beyond neuronal identity of SK-N-SH). The confirmation was with respect to the observation

that we had already observed expression in prenatal cerebral cortices and then continued to observe expression in the brain postnatally, not with respect to an expectation that SK-N-SH would be related specifically to neocortical layer 6. We agree that the original language could be misconstrued as intentional targeting of layer 6 was achieved and hope the alternative wording conveys that the sub-anatomical specificity was simply an observation.

Line 504:

*“We **observed** cortex-specific expression **is maintained in postnatal mice**, with focal activity occurring in the neurons at neocortical layer 6 and at subplate neurons (**Figure 4e-g, Supplementary Figure 37a,b**).”*

R2C14. *“Due to the intractability of fully searching sequence space, CODA cannot assuredly identify global specificity maxima, but our exhaustive evaluation of natural sequences demonstrates the design methods we used can identify synthetic sequences that regularly outperform natural ones with 1000-fold greater efficiency compared to previous methods using a zero-order Markov approach (Supplementary Figure 38) 40,41 .”*

Again the sequences outperform in designing enhancers that drive expression specifically in transformed cell lines. I understand the use of outperform in different fields but this needs to be clear, so it is not misleading.

We thank the reviewer for their careful examination of the text and their insightful suggestion for the language used in our discussion. We have added text to both clarify our use of the term outperform and restate context of the comparison between synthetic and natural sequences.

Line 562:

*“Due to the intractability of fully searching sequence space, CODA cannot assuredly identify global specificity maxima, but our exhaustive evaluation of natural sequences demonstrates the design methods we used can identify synthetic sequences that regularly outperform natural ones **in achieving the specificity objectives of this study** with 1000-fold greater efficiency compared to previous methods using a zero-order Markov approach (**Supplementary Figure 38**)^{40,41} .”*

R2C15. *“Synthetic sequences designed by CODA easily outperform natural sequences in driving cell type-specific gene expression in a reporter system, which suggests that novel functions can be programmed into CREs and interpreted by human cells.”*

.... In driving cell line specific gene expression, which suggests that synthetic enhancer can be programmed into CREs and interpreted by transformed human cell lines. Again this is likely to be the case when you use a cancer cell line to do the assays and needs to be stated for the reader.

We have updated this sentence to be consistent with the above-discussed changes in language specifying that large-scale comparisons of synthetic vs natural sequences are limited in scope to transformed cell lines. We have split this sentence into two ideas to better reflect this finding is not tested at scale *in vivo*. In doing so, we reframed the second point, focusing on the fact that in general synthetic sequences can be interpreted by biological systems, and that programming functionality into them is possible.

Line 557:

“Synthetic sequences designed by CODA easily outperform natural sequences in driving cell type-specific gene expression in a reporter system assayed across transformed cell lines. We further found a selected subset of synthetic CREs could regulate transgene expression in analogous tissues in mouse and zebrafish, uncovering the possibility that novel functionalities can be programmed into CREs which are interpretable by cell lines in vitro and related cells and tissues in vivo.”

R2C16. *“The dearth of natural sequences capable of achieving exquisite specificity in a desired cell type in our study highlights the difficulty of using human genomic sequences to achieve non-natural objectives for which evolution may not have acted on.”*

My comments relating to this have not been addressed sufficiently and this needs to be spelled out to the reader. The synthetic elements will of course do better as the focus is on elements that are driving expression in transformed cell lines which the genome has not evolved to do.

We now have a better understanding of the reviewer’s concern and hope the new edits address it. The intended purpose of the sentence was to convey to the reader that our results could be an indication of the potential difficulty of using only natural sequences to achieve other non-natural goals that also lie outside evolutionary pressures, and encourage exploration of synthetically designed CREs for such goals. We realize that the sentence as originally written could be misinterpreted as if, based on our results in transformed cell lines, we were concluding that natural sequences will universally underperform in achieving other non-natural objectives, which is not our belief. We have edited the sentence to better convey our view that focusing solely on human genomic sequences may present challenges when pursuing non-natural objectives, and that therapeutic applications could benefit from the exploration of synthetic CREs, avoiding any definitive conclusion.

In that vein, it is most relevant to note that it was not assured that our synthetic sequences would work at all in either cell lines or *in vivo*. There were no assurances from prior literature to suggest natural sequences, especially those identified via highly specific DHS signals, achieve less cell type specificity in K562, HepG2 and SK-N-SH than sequences designed by a given CNN in combination with generative algorithms. There was substantial concern that the synthetic CREs would fall outside the accurate predictive power of the model and their empirical activity would differ greatly from the predictions. The expected quantitative levels of specificity of synthetic elements were extreme compared to the training data. Prior to validating the sequences, we had substantial concern that the synthetic CREs contained some pathology that would invalidate the predictions and result in no activity or specificity. This is the reason we deployed multiple sequence design algorithms and tested >50k synthetic elements. However, most worked for our specific objective, highlighting that the model had learned useful features of CRE logic.

In the current version of the manuscript, we address evolution twice. Once early in the **Main** on line 43 where we note that “sequences generated by evolution represent only a small subset of possible genetic sequences”, and the second is this statement of issue in the reviewer’s feedback. In it, we pose evolution’s contrasting role in shaping natural sequence function with the myriad of artificial transgenic applications being deployed in medicine and biotechnology whose goals are entirely distinct from natural selection. We have now made the following revision to make the above points more clear and to hopefully address the reviewer’s concern.

Line 571:

“The dearth of natural sequences capable of achieving exquisite specificity in the cell lines studied here attests to the potential challenges of using human genomic sequences to achieve goals for which evolution may not have optimized. Transgenic applications, such as gene therapies that require tissue, cell type, or diseased cell state specificity, will likely benefit from design and validation of synthetic CREs with programmable functions. Without human input, CODA deploys unique combinations of strongly on-target activating and off-target repressing TFs within a short sequence that are not commonly found in the human genome, to yield highly specific synthetic CREs.”

R2C17. *“This suggests that our models have learned a component of the foundational rules governing CREs and possess the ability to extrapolate this knowledge to unobserved or rarely observed syntax combinations.”*

Replace syntax with TFs combinations.

In accordance with our manuscript-wide reconsideration of the term ‘syntax’, this statement has been changed. We have included the change here again for completeness:

Line 578:

“This suggests that our models have learned a component of the foundational rules governing CREs, and possess the ability to extrapolate this knowledge to rarely observed TF combinations.”

R2C18. *“we were able to identify natural sequences in the genome with moderate proficiency for cell-specific activity, albeit to a lesser degree than synthetics.”*

we were able to identify sequences in the genome with moderate proficiency for cell-specific activity, albeit to a lesser degree than synthetics in the transformed cell lines.

We agree the proposed clarification is helpful and have made the change consistent with this reviewer's concerns around cell line language.

Line 583:

“[...] we were able to identify natural sequences in the genome with moderate proficiency for cell-specific activity, albeit to a lesser degree than synthetics across the transformed cell lines studied here.”

Reviewer #3

The authors do a very thorough job addressing my concerns and those of the other reviewers. In particular, the comparisons to other methods (Enformer, MPRA-DracoNN) provide better benchmarking. The new TF-MoDISco results improve the confidence that the models are learning the relevant regulatory code. However, there are still two major concerns. First, there are still key details missing in the transfer learning procedure, including whether potential circularity. Second, the manuscript still relies heavily on MPRA experiments that have still not been adequately described. The motivation behind Malinois and its role in the manuscript are made much more clear by the additions to the text. However, it also raises new questions:

We appreciate the reviewer's positive comments about our revised submission, and we are pleased to have addressed their concerns thoroughly. We include new analyses and data in the responses below that we hope will fully address the two remaining concerns raised by the reviewer. Specifically, we provide extended methods on the transfer learning procedure and show it had minimal impact on model performance. We have also cited the pre-printed manuscript on the underlying MPRA data which fully describes their generation and quality control (Siraj et al. 2024). DOI: <https://doi.org/10.1101/2024.05.05.592437>

R3C1. Given the improvement that transfer learning provides, how does Basset on its own do in predicting MPRA activity? This is critical because it gets to whether the neural network model requires MPRA data or whether open chromatin is sufficient.

To address the reviewer's question, we generated MPRA activity predictions from two additional model types: 1) using only the parent Basset weights used for transfer learning and 2) a Malinois-like model where all of its weights were randomly initialized (and thus did not use parent Basset weights). The Basset-only model predictions for oligos in the test set had low correlation with empirical MPRA measurements (Spearman's ρ for K562: 0.28, HepG2: 0.30, SK-N-SH: 0.29; **Reviewer Figure R3C1a**). This may be expected, as the Basset only-model is directly predicting DNase accessibility, which empirically displays weak correlation to MPRA activity. Our standard Malinois model, which includes the weights transferred from Basset, performed much better, (Spearman's ρ for K562: 0.81, HepG2: 0.83, SK-N-SH: 0.83; **Reviewer Figure R3C1b**). However, when we use a CNN similar to Malinois with randomly initialized weights, rather than transfer learning, performance remains high (Spearman's ρ for K562: 0.81, HepG2: 0.83, SK-N-SH: 0.83; **Reviewer Figure R3C1c**).

Our results show that chromatin models are insufficient to predict MPRA alone. This highlights the importance of using data directly from the reporter assays and the unimportance of pre-existing knowledge of open chromatin when large MPRA datasets are available. In the subsequent reviewer response below (R3C2), we describe that although transfer learning is not necessary for high model performance, the improved stability of training is valuable for the model development process.

Reviewer Figure R3C1. Comparison of predictions from Basset, Malinois, and another CNN trained on MPRA data. (a) Chromatin accessibility predictions from Basset in K562, HepG2, and SK-N-SH (y-axis) are more weakly correlated with empirical MPRA measurements (x-axis). Basset was trained as a binary classifier using DNase peaks so predicted probabilities are plotted in the logit space so a comparable range to MPRA activity is covered. This monotonic transformation will not impact Spearman's ρ . (b) The comparison of Malinois predictions with MPRA measurements in the test set from Figure 1d are reproduced here. (c) Predictions from the best alternate MPRA model from our post-hoc experiment using Bayesian Optimization for hyperparameter selection (Supplementary Figure 3, Methods: Model fitting), which did not use transfer learning from Basset, were also compared to empirical MPRA measurements. This model achieves comparable performance to Malinois without transfer learning.

R3C2. How much of Malinois' performance boost relative to MPRA-DracoNN and Enformer are due to transfer learning?

We thank the reviewer for raising this important question, and have data that can help resolve it in **Supplementary Figure 3b** and **Supplementary Table 3**. There we compare the performance of models with randomly initialized weights and models using transfer learning with Basset during the Bayesian hyperparameter optimization search. Although the best performing model with randomly initialized weights closely compares to Malinois (**Reviewer Figure R3C1b-c**), it seems that inheriting the weights of the convolutional layers from Basset offers more robust model fitting, making the model less sensitive to other hyperparameter choices. In practice, the robustness introduced by transfer

learning markedly reduced the costs and time that needed to be dedicated to model development and hyperparameter optimization. As a result, when benchmarking on synthetic elements, the top model with randomly initialized parameters (Pearson's r 0.81-0.91, Reviewer Figure R3C3c) performs similarly to Malinois (Pearson's r 0.81-0.92, Reviewer Figure R3C3b) and was more accurate than both MPRA-DracoNN (max Pearson's r 0.10-0.48) and Enformer (max Pearson's r 0.38-0.74) in predicting MPRA activity.

We now highlight in the main text that **high-performance models can be trained by fitting directly on MPRA data without chromatin accessibility-informed transfer learning** in the main text:

Line 121:

“Malinois accurately models episomal CRE activity across cell types. For sequences held out from training (62,582 elements on chromosomes 7 and 13), Malinois predictions in K562, HepG2, and SK-N-SH correlate highly with empirical activity measurements (Pearson's r 0.88-0.89; Spearman's ρ 0.81-0.83) (Figure 1d) and demonstrate cell specificity on par with experimental results (Supplementary Figure 4). Randomly initialized versions of Malinois that did not deploy transfer-learning from chromatin accessibility also performed well in practice, though were not favored during Bayesian Optimization of hyperparameters (Pearson's r 0.88-0.89; Spearman's ρ 0.81-0.83, Supplementary Figure 8c, Supplementary Table 3, Methods).”

R3C3. The authors have addressed my original concerns of circularity, but new ones have been introduced. It seems like the parameters from the Bassett architecture, which is trained on a global library of open chromatin, have been used to initialize the models. This could be then be introducing circularity across many of the loci.

We appreciate the reviewer's attentive examination and appreciate their interest in ensuring our methodology avoids circularity and is well described. We describe here and in our revised manuscript how we aimed to avoid circularity.

We note that the data split method used to train and validate Bassett upon its publication in 2016 was purely random, irrespective of chromosome. Standard practices have moved towards splitting data based on chromosomes. It would be an impractical undertaking to ensure there are no conflicts between these two splits.

That said, the results above (R3C1, R3C2) showing similarly high performance Malinois models without transfer learning, strongly suggest there is no significant leakage, or circularity, due to transfer learning. If test set information leakage was propagating through the transfer learning process, we would expect Malinois' performance to be unachievable without transfer learning due to the unfair advantage.

Moreover, because the primary problem arising from information leakage would be unexpected poor generalization to new data, the synthetic sequences designed in this study, which do not map to the genome, provide a validation set devoid of any circularity due to data split choices and directly test generalizability. If Malinois test set performance was inflated due to information leakage, we would expect a randomly initialized model with comparable test set performance to Malinois to provide better predictions for synthetic sequences. Instead, Malinois predicts synthetic sequences with comparable

accuracy to a randomly initialized model, showing generalizability is not adversely impacted. We provide detailed scatter plots of synthetic element testing for the reviewers in **Reviewer Figure R3C3**.

Reviewer Figure R3C3. Accuracy of Basset, Malinois, and a randomly initialized CNN trained on MPRA data when modeling the activity of synthetic elements. (a) Chromatin accessibility predictions from Basset in K562, HepG2, and SK-N-SH (y-axis) for synthetic oligos are only correlated with empirical MPRA measurements (x-axis) for K562. (b) Malinois predictions (y-axis) for synthetic oligos are compared to MPRA measurements (x-axis). (c) Predictions from the randomly initiated model referenced in **Reviewer Figure R3C1c** (y-axis) for synthetic oligos show similar correlations to MPRA measurements (x-axis) as Malinois.

We also now include this entirely new analysis in **Supplementary Figure 8c** showing Malinois generalization to non-genomic synthetic elements is comparable to a randomly initiated MPRA model with comparable test split performance, which we reproduce below.

New Supplementary Figure 8c. Example sequence generation trajectories and assessment of impacts of transfer learning on generalizability. (c) A scatter plot summarizing the predictive performance of the parent Basset model used for transfer learning, a randomly initialized model fitted to MPRA data without transfer learning, and Malinois. Performance is measured using the test split (oligos derived from chr7 and chr13; x-axis) and synthetic elements (y-axis) to calculate correlation with empirical MPRA measurements. Correlations are calculated separately for cell type. Note, x-shaped markers are overlapping o-shaped markers for each cell type.

Lastly, we believe there are several theoretical factors that prevent indirect leakage from the Basset training data from impacting Malinois performance:

1. Predicting chromatin accessibility is a divergent enough task from predicting MPRA activity that the inherited weights will need to change to accommodate new information.
2. Inherited layers from Basset at the start of training only account for part of the model, and are upstream of additional branched linear layers that were randomly initialized preventing the model from providing meaningful prediction without fitting to new data.
3. The Basset training data was collected from the union of DNaseI peaks from 164 cell types, while the MPRA oligos were collected based largely on genetic association studies with most oligos residing outside DNaseI peaks. This would further reduce the overlap in the exact sequences that are used in each of the two training sets.

References

- Agarwal, Vikram, Fumitaka Inoue, Max Schubach, Beth K. Martin, Pyaree Mohan Dash, Zicong Zhang, Ajuni Sohota, et al. 2023. "Massively Parallel Characterization of Transcriptional Regulatory Elements in Three Diverse Human Cell Types." *bioRxiv : The Preprint Server for Biology*, March. <https://doi.org/10.1101/2023.03.05.531189>.
- Avsec, Žiga, Melanie Weilert, Avanti Shrikumar, Sabrina Krueger, Amr Alexandari, Khyati Dalal, Robin Fropf, et al. 2021. "Base-Resolution Models of Transcription-Factor Binding Reveal Soft Motif Syntax." *Nature Genetics* 53 (3): 354–66.
- Bourges, Christophe, Abigail F. Groff, Oliver S. Burren, Chiara Gerhardinger, Kaia Mattioli, Anna Hutchinson, Theodore Hu, et al. 2020. "Resolving Mechanisms of Immune-Mediated Disease in Primary CD4 T Cells." *EMBO Molecular Medicine* 12 (5): e12112.
- Brown, Ashley R., Grant A. Fox, Irene M. Kaplow, Alyssa J. Lawler, Badoi N. Phan, Morgan E. Wirthlin, Easwaran Ramamurthy, et al. 2022. "An in Vivo Massively Parallel Platform for Deciphering Tissue-Specific Regulatory Function." *bioRxiv*. <https://doi.org/10.1101/2022.11.23.517755>.
- Chan, Ya-Chien, Eike Kienle, Martin Oti, Antonella Di Liddo, Maria Mendez-Lago, Dominik F. Aschauer, Manuel Peter, et al. 2023. "An Unbiased AAV-STARR-Seq Screen Revealing the Enhancer Activity Map of Genomic Regions in the Mouse Brain in Vivo." *Scientific Reports* 13 (1): 6745.
- Deng, Chengyu, Sean Whalen, Marilyn Steyert, Ryan Ziffra, Pawel F. Przytycki, Fumitaka Inoue, Daniela A. Pereira, et al. 2023. "Massively Parallel Characterization of Psychiatric Disorder-Associated and Cell-Type-Specific Regulatory Elements in the Developing Human Cortex." *bioRxiv : The Preprint Server for Biology*, February. <https://doi.org/10.1101/2023.02.15.528663>.
- Grant, Charles E., Timothy L. Bailey, and William Stafford Noble. 2011. "FIMO: Scanning for Occurrences of a given Motif." *Bioinformatics* 27 (7): 1017–18.
- Hrvatín, Sinisa, Christopher P. Tzeng, M. Aurel Nagy, Hume Stroud, Charalampia Koutsoumpa, Oren F. Wilcox, Elena G. Assad, et al. 2019. "A Scalable Platform for the Development of Cell-Type-Specific Viral Drivers." *eLife* 8 (September). <https://doi.org/10.7554/eLife.48089>.
- Kim, Daniel S., Viviana I. Risca, David L. Reynolds, James Chappell, Adam J. Rubin, Namyoung Jung, Laura K. H. Donohue, et al. 2021. "The Dynamic, Combinatorial Cis-Regulatory Lexicon of Epidermal Differentiation." *Nature Genetics* 53 (11): 1564–76.
- Lagunas, Tomas, Jr, Stephen P. Plassmeyer, Anthony D. Fischer, Ryan Z. Friedman, Michael A. Rieger, Din Selmanovic, Simona Sarafinowska, et al. 2023. "A Cre-Dependent Massively Parallel Reporter Assay Allows for Cell-Type Specific Assessment of the Functional Effects of Non-Coding Elements in Vivo." *Communications Biology* 6 (1): 1151.
- Linder, Johannes, Divyanshi Srivastava, Han Yuan, Vikram Agarwal, and David R. Kelley. 2023. "Predicting RNA-Seq Coverage from DNA Sequence as a Unifying Model of Gene Regulation." *bioRxiv*. <https://doi.org/10.1101/2023.08.30.555582>.
- Penzar, Dmitry, Daria Nogina, Elizaveta Noskova, Arsenii Zinkevich, Georgy Meshcheryakov, Andrey Lando, Abdul Muntakim Rafi, Carl de Boer, and Ivan V. Kulakovskiy. 2023. "LegNet: A Best-in-Class Deep Learning Model for Short DNA Regulatory Regions." *Bioinformatics* 39 (8). <https://doi.org/10.1093/bioinformatics/btad457>.
- Rafi, Abdul Muntakim, Daria Nogina, Dmitry Penzar, Dohoon Lee, Danyeong Lee, Nayeon Kim, Sangyeup Kim, et al. 2024. "Evaluation and Optimization of Sequence-Based Gene Regulatory Deep Learning Models." *bioRxiv : The Preprint Server for Biology*, February. <https://doi.org/10.1101/2023.04.26.538471>.
- Siraj, Layla, Rodrigo I. Castro, Hannah Dewey, Susan Kales, Thanh Thanh L. Nguyen, Masahiro Kanai, Daniel Berenzy, et al. 2024. "Functional Dissection of Complex and Molecular Trait Variants at Single Nucleotide Resolution." *bioRxiv*. <https://doi.org/10.1101/2024.05.05.592437>.

Reviewer Reports on the Second Revision:

Referee #2 (Remarks to the author):

This manuscript represents a huge amount of work and has some novel aspects, most notably the design of neural enhancers that are active in vertebrates. I appreciate the authors' attempts to address my comments. The manuscript is very dense, and still oversells and stretches the findings and impacts. While there have been some changes to the tone of the statements, the manuscript still suggests that they can use this approach to design tissue specific enhancer for therapeutics. Yet the approach cannot target specific cell types such as a specific subpopulation of neurons, rather the approach can only target broad categories of cell types such as neural or liver. Thus for therapeutics there is a way to go, so the pitch oversells the significance of the data. If they did want to use this for therapeutic they'd need to be able to target a specific neural population such as glutamatergic neurons of a specific structure within the brain. Given that in zebrafish 2/3 of their CREs are active in a variety of neural subtypes and then in mouse 1/2 of these CREs drives expression only within the cortex it doesn't seem like there is a handle on how to get neural specificity with the precision needed to deliver therapeutics. It is great to see for the K562 optimized CREs that specificity remained even within the A569 and HCT116 cells, however the specificity of Hep and SK optimized CREs is less compelling as these CREs also show expression in the A549 and HCT116 cell lines.

Comments:

for each cell line we identified 4,000 'DHS-natural' sequences with cell type-specific chromatin accessibility and overlapping H3K27ac signals (12,000 total)

I would like to see how cell type specific these DHS peaks are, they say they take the top 4,000 but they don't really justify how cell type specific those 4k are or why they chose 4k. There is a lot of data in many cell types that could be used in a supplementary figure to illustrate how tissue-specific these features are.

We consistently observed that disrupting blocks of positive contribution led to a decrease in predicted activity, while disrupting blocks of negative contribution resulted in an increase (Supplementary Figure 21, Methods).

This is very circular, they determined which bp the model thinks are important and mutated them. Mutating them makes predicted activity lower, this is expected. They need to validate this experimentally.

The section: *Complex semantic architectures are syntactically differentially deployed in natural and synthetic sequences*

This section is very technical and dense. There is a lot of computational analysis that is not validated. I'm unclear what this section adds to the paper.

They claim that *synthetic enhancers have higher program content and higher program heterogeneity*. This is shown in Fig S 28b, they need to do some stats on this section. This whole section is hard to understand, and they don't really define what these metrics are. It would be helpful to choose a few, explain them well, and try to discuss their relevance. It is unclear to me what this finding adds to the paper or how this finding can inform our ability to design tissue specific enhancers.

Supplementary Figure 30. MPRA models for A549 and HCT116 predict synthetic CREs. Additional MPRA measurements were made in A549 and

While for K562 optimized CREs there is clearly specificity for the cell line of choice, this is less true for the HEPG2 and SK optimized sequences, as these CREs also show expression in A549 and HCT116 cells. For SK-N-SH optimized CREs there is the least specificity seen, the axis has a different scale in G and the activity of the SK-N-SH optimized CREs is only slightly higher in the SK cell line than in A549 and HCT116. Therefore, they do not attain the same levels of specificity as when they were only testing CREs in the k562, HepG2 and SK lines. But in the text they say they are successful. I think they should remark on the fact that when they extended their model these two additional cell lines they were successful for k562 optimized CREs, but Hep G2 and SK were less specific. This indicates that in some cases it may be difficult to generalize the model to other cell lines without further training. Also, they should do stats if they are going to make claims about this data supporting their model's generalizability.

Sequences designed for neuronal specificity showed similar success (2 of 3), driving expression in a subset of neuronal cell types. There needs to be some discussion of the limitations here. By this I mean, what specificity is actually achieved. I would have expected the CREs to be active in all neural cells or in specific subset of neural cell types most similar to the SK cell line. This is not what is seen, instead there is generalized neural expression that varies between embryos and the two enhancers.

We observed specific expression for neuronal #1 (N1) with localized expression in the developing cortex and no additional expression observed elsewhere (Supplementary Figure 36a,b). To localize the expression patterns further within the cortex, we repeated the reporter assay with the N1 CRE and performed in situ staining of the whole brain at 5 weeks postnatal (Figure 4d, Supplementary Figure 36c-h). We observed cortex-specific expression is maintained in postnatal mice, with focal activity occurring in the neurons at neocortical layer 6 and at subplate neurons (Figure 4e-g, Supplementary Figure 37a,b).

Of the two CREs active in the zebrafish, one is active in the mouse, but appears to be active in only a very specific and different region of the brain to what was seen in the zebrafish. Why would the expression be in cortex alone here, and why was only one of the two CREs active in mouse. Why would the expression be in the neocortical layer 6 and at subplate neurons, rather than in other neural cell types or

populations of cells. There needs to be some discussion of this and the limitations of the approach, especially given the pitch highlighting the significance related to designing enhancers for therapeutics.

Additionally, while I understand that this is a novel approach, I wonder if validation of 1 of two CREs is sufficient for the claims they are making. Ideally I'd like to see more CREs tested and validated, the del Almeida et al tests 8 enhancers per cell type in drosophila. While I understand that this is in mouse, they could test more than 2, and could test more in zebrafish.

For the liver enhancers they tested in the zebrafish, these seem to have more uniform expression, although it is hard to know which liver cell type these enhancers are active within. I would like to know if these were inactive in mouse when tested or untested. If untested, I wonder why they did not choose to test these given that the liver is a less heterogenous cell type than neural cells found within the CNS and PNS. Liver CREs would have been an easier system to validate conservation of activity across species and the ability to design tissue-specific CREs than the neural enhancers and this omission concerns me.

Our results suggest that CREs designed for tissue-specific targeting can work across species, even in the brain, which has been an ongoing challenge to target with viral-based delivery approaches. An integrated framework leveraging human cell lines in conjunction with whole organism models may thus be a viable approach to rapidly identify CREs to execute novel functions in humans.

If the goal is to find tissue-specificity with certain neural cell types then the enhancers need to be active within a particular type of neural cells. Yet in the study, only neural is selected, and the particular neural cell types vary widely between embryos, enhancers and species in the 2 CREs validated. While this framework could work it has not been demonstrated by the data in this manuscript.

We successfully deployed CODA for cell type specificity

The ability to get specificity is lower for the SK and HEP optimized sequences when considering the A549 and HCT116 cell lines suggesting that there are limitations in this approach to find CREs with specificity. I have also discussed this in terms of *in vivo* specificity above.

Minor comments

Fig S26 title needs some attention.

type: token ratio – is very technical and it would be good to make this more understandable

Figure S3. All y axes should be labelled.

Stats should be shown in the main figures when claiming significant enrichments, for example: Figure 1d, could they please show the stats in the main figure. They mention they have done an MWU, but it would be good to see the stats in the main figure.

Fig S27. I get that topics are technical terms, but the paper may be more understandable if they tried getting away from technical language. Instead of calling these topics, perhaps the authors could name these "TF vocabularies" or "TF Vocab 1", "TF vocab 2", ...

Referee #3 (Remarks to the Author):

The authors have address all of my comments. This study is really important for the broader scientific community and represents a substantial advance.

Referee #3 (Remarks on code availability):

The code is well documented.

Author Rebuttals to Second Revision:

Reviewer #2

This manuscript represents a huge amount of work and has some novel aspects, most notably the design of neural enhancers that are active in vertebrates. I appreciate the authors' attempts to address my comments. The manuscript is very dense, and still oversells and stretches the findings and impacts. While there have been some changes to the tone of the statements, the manuscript still suggests that they can use this approach to design tissue specific enhancer for therapeutics. Yet the approach cannot target specific cell types such as a specific subpopulation of neurons, rather the approach can only target broad categories of cell types such as neural or liver. Thus for therapeutics there is a way to go, so the pitch oversells the significance of the data. If they did want to use this for therapeutic they'd need to be able to target a specific neural population such as glutamatergic neurons of a specific structure within the brain. Given that in zebrafish 2/3 of their CREs are active in a variety of neural subtypes and then in mouse 1/2 of these CREs drives expression only within the cortex it doesn't seem like there is a handle on how to get neural specificity with the precision needed to deliver therapeutics. It is great to see for the K562 optimized CREs that specificity remained even within the A569 and HCT116 cells, however the specificity of Hep and SK optimized CREs is less compelling as these CREs also show expression in the A549 and HCT116 cell lines.

We appreciate the reviewer's acknowledgement of the significant effort and novel contribution of our work. We also recognize their concern about our claims of what the work has achieved.

As we described in our previous revision and responses, we do not believe, or intend to convey, that this manuscript provides an out-of-the-box solution to shortcomings in existing gene therapy vectors. We simply aimed to provide context and motivation for the reader, that the methodology outlined here could be a generalizable blueprint that applies to diverse experimental disease models and, in those contexts, could eventually be used to develop therapeutic products. Accordingly, we aimed to provide background about the connection to therapeutics conveyed as forward looking statements that suggest possible future work.

During revisions, we edited or removed all sentences that the reviewer suggested and further carefully revised the manuscript accordingly. In review of the current manuscript, we are not able to identify sentences where we claim that we can use our approach in its current state for therapeutics. However, in an abundance of caution against overinterpretation, we also removed the following rationale that motivated selection of our cell lines of choice:

“These well-studied cell types are ideal for high-throughput method development and can provide useful insight for the growing body of experimental gene therapies that target blood cells⁵⁰⁻⁵³ and neurons⁵⁴, but that can induce toxicity in the liver⁵⁵⁻⁵⁷.”

If there are any sentences remaining that we have overlooked that imply the results here can be *directly* integrated into a therapeutic, we will be happy to address them.

We also appreciate that, in retrospect, it may appear obvious that synthetic elements will drive patterns of expression mirroring *a priori* deep learning predictions; however, we want to highlight that this was far from a guaranteed outcome as the synthetic elements in this study are fundamentally different from natural CREs in their informational content. Much of our manuscript details the systematic differences in TF motif vocabulary (e.g., by lexical analysis) and semantic structures (i.e., by topic modeling) between synthetic and natural elements. Additionally, other groups have deployed large language models, which can parse the functional subregions of the genome with high accuracy, to demonstrate that our synthetic CREs resemble random sequences more closely than genomic elements (Lal et al., 2024; Nguyen et al., 2023). It was therefore highly possible that our sequence optimization processes may be adversarially exploiting pathologies in Malinois, leading us to expect a far lower success rate than we observed. We tested thousands of synthetic elements in our study to ensure we were well powered to make comparisons in the case of that outcome. We find it remarkable that despite these potential pitfalls, synthetic CREs reliably drive expression patterns over three cell lines with a tight correlation to their predicted function. While there is undoubtedly more work to be done throughout the field to refine synthetic CRE design for individual drugs, this work represents an important step towards that goal.

R2C1. for each cell line we identified 4,000 ‘DHS-natural’ sequences with cell type-specific chromatin accessibility and overlapping H3K27ac signals (12,000 total)

I would like to see how cell type specific these DHS peaks are, they say they take the top 4,000 but they don't really justify how cell type specific those 4k are or why they chose 4k. There is a lot of data in many cell types that could be used in a supplementary figure to illustrate how tissue-specific these features are.

Analysis of DHS specificity

We chose 4,000 CREs to test from each cell type (12,000 total) due to size constraints in the MPRA, but aimed to have enough to make well-powered comparisons across synthetic and natural categories. Using uniformly processed epigenetic datasets from the ENCODE project, we selected DHS sites that also overlapped H3K27ac, an orthogonal marker for active CREs, to ensure high-confidence CRE elements. Specifically, we took 159,277, 130,520, and 155,722 DHS peaks and overlapped with 51,343, 50,759, and 69,317 H3K27ac peaks, resulting in 61,280, 49,093, 43,244 DHS+H3K27ac peaks in K562, HepG2, SK-N-SH, respectively. To identify initially specific DHS peaks, for each cell type we identified DHS+H3K27ac peaks that had no DHS peak overlap in the other two cell types. For each cell type, using the DHS-coordinates of the on-target cell type, we calculated the total DHS signal in each cell type. We then transformed the DHS signal into \log_2 space to mirror \log_2 scale MPRA data and selected the top 4,000 peaks that maximize minGap. We have updated the **Methods** section describing our selection criteria, and describing in greater detail our rationale for selection and improving overall clarity. We also provide a visual representation of DHS score distributions for the reviewer below as the new **Supplementary Figure 6**.

Line 906:

“DHS-natural. To identify CREs broadly replicating across experimental approaches, using a uniformly processed dataset from ENCODE, we first selected DNase peaks from each of the three cell lines (K562, HepG2, and SK-N-SH). To further select for active CREs we subsetted DHS peaks that intersect with H3K27ac peaks from the same cell type. For each cell type, we then identified cell-type specific peaks by requiring a DHS+ H3K27ac+ peak had no overlap with a DHS peak in the other two cell types. For these DHS-H3K27ac peaks, in each cell type, we scored the K562, HepG2, and SK-N-SH DHS signal in the peak coordinates of the target cell type. We then selected the top 4,000 peaks with the highest MinGap calculated using \log_2 -space DHS signal, mirroring our efforts to maximize MinGap of \log_2 -space MPRA activity with other CREs.”

Supplementary Figure 6: DHS specificity as measured by MinGap of \log_2 of DHS signal counts for specific peaks, and the selected 4,000 peaks for MPRA. Boxes demarcate the 25th, 50th, and 75th percentile values, while whiskers indicate maximum and minimum (left-to-right, top-to-bottom $n=35,894$; 4,000; 27,310; 4,000; 20,773; 4,000).

Uniform analysis of transcriptional activity across 10 tissue types

Similarly to the reviewer, we were also interested in leveraging the breadth of data across many cell types to understand how specificity of the elements we tested by MPRA would fare when considering additional tissue types. We simulated CRE activity using an orthogonal model after insertion upstream of a reporter gene inserted in the mouse H11 locus. This mimicked taking a natural CRE out of its native genomic context and in a specific, uniform reporter context, that also happened to match our in-vivo experiments. We then used Enformer, a state-of-the-art model for prediction of chromatin accessibility and other markers of transcriptional activation that has been accepted as sufficient to make claims about cell type specificity (Taskiran et al., 2024), to generate predictions across 10 tissues. We then quantified specificity of these predictions using MinGap for targeting spleen, liver, and brain for K562-, HepG2-, and SK-N-SH-targeting CREs, respectively. This analysis is summarized in **Extended Data Figure 8e** and demonstrates significant differences in specificity between natural and synthetic CREs.

Extended Data Figure 8. Enformer based prioritization of oligos for in vivo tests. (e) Enformer-based cell type matched tissue-specific transcriptional activation predictions (K562 matched to spleen, HepG2 matched to liver, SK-N-SH matched to adult brain). Stars indicate family-wise error rate corrected p -values $< 1e-4$ (In each trio of boxes, $n=4,000$; $4,000$; $12,000$ elements for the DHS, Malinois, and synthetic groups, respectively).

R2C2. We consistently observed that disrupting blocks of positive contribution led to a decrease in predicted activity, while disrupting blocks of negative contribution resulted in an increase (Supplementary Figure 21, Methods).

This is very circular, they determined which bp the model thinks are important and mutated them. Mutating them makes predicted activity lower, this is expected. They need to validate this experimentally.

Contribution score methods are a mathematical tool used to provide a hypothesis of why a deep learning model generates a prediction, regardless of whether its prediction is accurate or not. In turn, the goal of the contribution-block ablation study was to build trust in sampled integrated gradients, the mathematical method we have developed to obtain contribution scores. This approach consistently identified sequence fragments that increase or decrease Malinois predictions of activity from an individual CRE in the various cell lines. We agree that an empirical validation via MPRA would additionally test how positive and negative contribution blocks affect sequence activity. However, comprehensively evaluating all blocks with the necessary power would require testing a number of

oligos exceeding what has been investigated so far in this entire study. For example, our *in silico* analysis required testing 5M sequences which is unfeasible experimentally. Furthermore, this experiment could conflate assessment of model accuracy with technical validation of contribution scores requiring additional experimental and mathematical study to resolve. Ultimately, we deemed this outside the scope of this study and agreed with reviewer 1's opinion that *in silico* methods would be suitable for subsequence functional characterization. We point out that our general approach using contribution scores from a well-benchmarked model to interpret subsequence function is standard in the field (Avsec, Agarwal, et al., 2021; Avsec, Weilert, et al., 2021; Linder et al., 2023), and to our knowledge a well-powered empirical validation as proposed here would be the first of its kind.

We have edited the text to better reflect the views above and moved it to **Supplementary Note 4**:

*“In order to validate that our contribution score method accurately reflects how single nucleotides impact model predictions, we systematically disrupted sequence segment blocks of positive, negative, and neutral contributions (**Methods**). We consistently observed that disrupting blocks of positive contribution led to a decrease in predicted activity, while disrupting blocks of negative contribution resulted in an increase (**Supplementary Note 4 - Figure 1, Methods**). This alignment with expected effects supports the suitability of the contribution score method to interpret model predictions.”*

R2C3. The section: *Complex semantic architectures are syntactically differentially deployed in natural and synthetic sequences*

This section is very technical and dense. There is a lot of computational analysis that is not validated. I'm unclear what this section adds to the paper.

They claim that *synthetic enhancers have higher program content and higher program heterogeneity*. This is shown in Fig S 28b, they need to do some stats on this section. This whole section is hard to understand, and they don't really define what these metrics are. It would be helpful to choose a few, explain them well, and try to discuss their relevance. It is unclear to me what this finding adds to the paper or how this finding can inform our ability to design tissue specific enhancers.

This section uses topic modeling, a well-known language analysis methodology used to study word co-occurrence, to distinguish how motifs are combined in different sequence groups. In this case, topic modeling identifies groups of co-occurring motifs and quantifies how these groups are distributed throughout sequences, allowing for more flexible detection of motif combinations beyond single motifs or pairs performed earlier in the manuscript. This provides our work a high level analysis of how patterns of motifs differ between element sets.

We agree with the reviewer that **Supplementary Figure 15b** (previously **Supplementary Figure 28b**) was lacking crucial information to understand the plots. We appreciate the comment and have added the missing labels to the y axes and color bars, and included their descriptions in the figure legend.

Supplementary Figure 15b. Distribution of individual program fraction, normalized by total program content for 12 programs assessed by NMF decomposition. Sequences are grouped by design methodology (x-axis) and intended target cell type (hue). Inset slider indicates aggregate program function over K562, HepG2, and SK-N-SH (average repressive function indicated by blue, averages clipped within +/-1 range). Boxes demarcate the 25th, 50th, and 75th percentile values, while whiskers indicate the outermost point within 1.5 times the interquartile range from the edges of the boxes.

Additionally, we have added label descriptions in the legend of **Supplementary Figure 16a-d** (previously **Supplementary Figure 29a-d**) as well.

“Supplementary Figure 16. Overall program usage. (a) Distribution of total program coefficients for sequences in different design groups, indicating the total amount of information encoded in each element. (b) Heterogeneity of program coefficients for each sequence measured by entropy. Higher entropy suggests greater diversity of programs used in each CRE. (c) Aggregating activating program content corresponding to the correct target cell type. High values indicate a

greater proportion of information encoded in the CREs is dedicated to enhancing transcription in the target cell. (d) Same as c, except repressing programs. Higher values indicate a greater proportion of information encoded in the CREs is dedicated to repressing transcription in off-target cells.”

We note that reviewer 1 had multiple suggestions during the first rounds of revisions about the importance of this section and suggestions to improve it. Due to the observational nature of NMF and in accordance with Reviewer 1’s concerns about overinterpretation, we removed our original detailed statistical analysis on the data in **Supplementary Figure 15 and 16**. These were present in lines 410-420 of our original submission.

We reproduce our explanation to reviewer 1 of the technical challenges to providing experimental validation of our NMF analysis, and our agreement that statistical analysis on these observational data could be overinterpreted.

Submission 1: Reviewer 1 Comment 15:

Constructing sequences from NMF programs and testing with Malinois

Our original intention was for NMF to serve as observational analysis to show broad patterns of differences between groups of sequences. However, the reviewer raises an interesting question that we agree should be evaluated if we were using NMF programs to define cell specificity. We agree we lack the data to establish a causal relationship between program content and sequence function (e.g. cell type specificity). While it is clear how to embed individual motifs into sequences, as we have done in response to R1C10, we find embedding programs into sequences non-trivial. Our method of featurizing sequences and computing NMF programs is not end-to-end, making it difficult to generate sequences with well controlled program embeddings. One way to overcome this is similar to the approach applied by Taskiran et al. which used a model trained on program assignments as the oracle for designing synthetic CREs which can then be evaluated for specificity (Taskiran et al., 2024). There is no straightforward way to do that without training a new model to approximate our NMF results and applying sequence design techniques. We look forward to future studies attempting to develop methods that could directly assess how well program assignments reflect regulatory programs at-scale.

We have now updated the manuscript to clarify our NMF is intended only to provide a broad overview of relationships between sequences tested in our study. **To limit overinterpretation, we have removed our detailed analysis of differences in the content of specific programs in different groups of sequences.** We instead focus on higher level patterns of synthetic elements deploying activating and repressing programs to a higher degree than natural sequences, which is in agreement with the analysis of individual activators and repressors.

R2C4. While for K562 optimized CREs there is clearly specificity for the cell line of choice, this is less true for the HEPG2 and SK optimized sequences, as these CREs also show expression in A549 and HCT116 cells. For SK-N-SH optimized CREs there is the least specificity seen, the axis has a different scale in G and the activity of the SK-N-SH optimized CREs is only slightly higher in the SK cell line than in A549 and HCT116. Therefore, they do not attain the same levels of specificity as when they were only testing CREs in the k562, HepG2 and SK lines. But in the text they say they are successful. I think they should remark on the fact that when they extended their model these two additional cell lines they were successful for k562 optimized CREs, but Hep G2 and SK were less specific. This indicates that in some cases it may be difficult to generalize the model to other cell lines without further training. Also, they should do stats if they are going to make claims about this data supporting their model's generalizability.

We note that the cell lines A549 and HCT116 were not part of the initial experimental design, that is, the synthetic sequences were not explicitly optimized by CODA to be inactive in those cell lines. While we extended models to generate predictions in these cell lines retrospectively, the generative pipeline as a whole was not extended. The comparisons were performed due to a shared curiosity with the reviewer (round 1, R2C2) with respect to the generalizability of the models. We do not claim that training in the three cell lines used in our initial study is enough to generalize off-target repression to all cell types. As discussed in our initial response to the reviewer, new technologies are needed to extend generalizability to many cell types, especially *in vivo*, and is a focus of our current research program. Indeed, in the discussion, we point towards more data being needed to effectively target new cell types.

Regarding interpretation of the results when adding new cell lines, while the intensity of cell type-specific objective function goes down, we note that activity in the targeted cell line remains higher than the new cell lines. The insets of **Supplementary Figure 17e-g** quantify the percentage of sequences that have highest activity in the target cell types before and after the additional cell lines are considered. Sequences designed by Fast SeqProp and Simulated Annealing maintain maximum activity in the intended cell line even when considering additional cell types with high consistency, which we consider a successful outcome given CODA was unaware of the new cell types during the design stage. Furthermore, the synthetic elements maintain a significantly higher minGap than natural elements even when considering A549 and HCT116 in the calculations, which is now quantified in **Supplementary**

Figure 17h. We do, however, appreciate the caveat the reviewer points out regarding reduced MinGap intensity and include it in a revised description of the result.

Line 288:

“To determine if specificity is maintained when adding new cell lines, we trained additional models for A549 (lung epithelial cancer) and HCT116 (colon epithelial cancer), observing that synthetic CREs retained maximum predicted activity in their target cell type over A549 and HCT116, especially those generated using Fast SeqProp, albeit with reduced MinGap (Supplementary Figure 17).”

Supplementary Figure 17h legend:

“...For each set of comparisons made using activity predictions for the same collection of cells, synthetic elements are predicted to maintain significantly higher average MinGap than any natural group both with and without A549 and HCT116 being considered in the calculations ($p\text{-adj} < 10^{-300}$ for all pairwise comparisons, Tukey's HSD test).”

R2C5. Sequences designed for neuronal specificity showed similar success (2 of 3), driving expression in a subset of neuronal cell types. There needs to be some discussion of the limitations here. By this I mean, what specificity is actually achieved. I would have expected the CREs to be active in all neural cells or in specific subset of neural cell types most similar to the SK cell line. This is not what is seen, instead there is generalized neural expression that varies between embryos and the two enhancers.

We understand the reviewer's perspective that without full context a reader may incorrectly assume synN1 was intentionally designed with neural subtype specificity, and overinterpret the capabilities of our existing CRE design capabilities. We have added context to the discussion conveying our surprise regarding that specific result, and clarify to the reader that more work is needed for precise sub-tissue targeting. We reproduce the text below, but first answer your questions.

We expect that the neuronal *in vivo* CREs designed by CODA should have specificity driven by two features, an analogous neural cell type to the SK-N-SH line, and lack of activity in cell lines analogous to K562 and HEPG2. It is important to make the distinction that “analogous cell lines” is with respect to a shared regulatory program. As these are not primary cell lines or tissue samples, we don't expect perfect matching to cells in humans, let alone other species. However, we expect some sharing of regulatory programs resulting in overlap in the specific TFs driving transcriptional activity in these cell lines with TFs driving activity in the tissues these cell lines arose from. Our baseline expectation for success was overlap in expression with any neuronally related tissue, and lack of expression in liver or blood tissues. This prior hypothesis differs from the reviewer's as our initial hypothesis did not bound the number of neural cells expression was expected in. We are unaware of the exact analogous neural cell type of the SK cell line, and as such are unsure that we can say definitively that CREs are seen to be active in a subset of neural cells most similar to SK cells. Furthermore, it is unsurprising that our CREs have incidental specificity to subtypes of neurons, given our cell lines likely do not capture the transcriptional diversity of *all* neurons, nor have we directed CODA to utilize regulatory programs ubiquitous across neurons.

With regards to heterogeneity of expression patterns, zebrafish experiments were conducted without germline integration using a transposon system. Chimeric, non-uniform expression even between identical cell types is a common well known artifact of the assays. For all animal experiments, we have used standardized scoring approaches for these transgenics that are well established by leaders of the field (PMID: 27768887 and by the Vista Enhancer Browser (<https://enhancer.lbl.gov/>)). Lastly, we think the idea of specifying cell subtypes is an interesting question that we are currently investigating, but one likely served best by work in primary cell, *in vivo* reporters, and advances in modeling. We have updated language to address these ideas.

Line 385:

“We were surprised that our neuronal synN1 CRE, designed from a single transformed SK-N-SH cell line, exhibited highly specific sub-cortical expression in mice. Further research is needed to develop optimal strategies to translate in vitro models to precise targeting in vivo.”

R2C6. We observed specific expression for neuronal #1 (N1) with localized expression in the developing cortex and no additional expression observed elsewhere (Supplementary Figure 36a,b). To localize the expression patterns further within the cortex, we repeated the reporter assay with the N1 CRE and performed *in situ* staining of the whole brain at 5 weeks postnatal (Figure 4d, Supplementary Figure 36c h). We observed cortex-specific expression is maintained in postnatal mice, with focal activity occurring in the neurons at neocortical layer 6 and at subplate neurons (Figure 4e-g, Supplementary Figure 37a,b).

Of the two CREs active in the zebrafish, one is active in the mouse, but appears to be active in only a very specific and different region of the brain to what was seen in the zebrafish. Why would the expression be in cortex alone here, and why was only one of the two CREs active in mouse. Why would the expression be in the neocortical layer 6 and at subplate neurons, rather than in other neural cell types or populations of cells. There needs to be some discussion of this and the limitations of the approach, especially given the pitch highlighting the significance related to designing enhancers for therapeutics.

Additionally, while I understand that this is a novel approach, I wonder if validation of 1 of two CREs is sufficient for the claims they are making. Ideally I'd like to see more CREs tested and validated, the del Almeida et al tests 8 enhancers per cell type in drosophila. While I understand that this is in mouse, they could test more than 2, and could test more in zebrafish.

For the liver enhancers they tested in the zebrafish, these seem to have more uniform expression, although it is hard to know which liver cell type these enhancers are active within. I would like to know if these were inactive in mouse when tested or untested. If untested, I wonder why they did not choose to test these given that the liver is a less heterogenous cell type than neural cells found within the CNS and PNS. Liver CREs would have been an easier system to validate conservation of activity across species and the ability to design tissue-specific CREs than the neural enhancers and this omission concerns me.

While we agree with the reviewer that unraveling the causal factors underlying cortical specificity is interesting future work, this degree of specificity was incidental to the cell lines and *in silico* techniques used in the study and not a core result. For background, we expect that tissue specificity to be driven by the unique combination of activating and repressing factors, and that SK-N-SH cells, for which these neuronal enhancers were designed, will have a higher overlap in the expression of these TF to neuronal cells than to other cells. Differences across species, specific cell types, and developmental timing, will

likely result in further divergence in TFs between those expressed in SK-N-SH cells and the tissues surveyed in our study. As SK-N-SH cells represent just one cell type compared to the complexity of an *in vivo* system, it is not unexpected that CREs functional in SK-N-SH may be specific to only a subset of cells *in vivo*, and different SK-N-SH-specific CREs designed by CODA may use different regulatory programs each having a different neuronal specificity *in vivo*. Our description of the specificity for N2 to layer 6 neurons is evidence of succeeding in designing a CRE with function in mammals to *any* neuronal tissue, rather specifically to the cellular subtype of layer 6 neurons. We have attempted to clarify this in the updated manuscript.

More broadly, as we noted in R2C1, it was not our intention for our manuscript to be a “pitch” with respect to therapeutics. Our discussions about therapeutic enhancers are forward-looking statements that we believe help contextualize our work for the reader. They are the long-term goals that motivate us. We are not making claims that we have designed therapeutic CREs or created a hardened approach for therapeutic development. Instead, the three manuscripts preprinted in 2023 demonstrated for the first time the success of synthetic CREs and our work is the first to do so at-scale in humans and evaluates the resulting sequences in a vertebrate system.

As the reviewer noted themselves, mouse experiments are quite different from drosophila experiments, and we believe testing additional sequences is beyond the scope of this paper for multiple reasons. First, these tests are extremely costly, time-consuming, and would require a separate manuscript to properly communicate the results. In addition, the *in vivo* experiments are not intended to be experimental validation of the CODA pipeline; this was the purpose of the large-scale MPRA. The 6 sequences tested using our *in vivo* experiments were performed to provide supplementary insights into how some generated sequence might work within a whole organism. We thus did not derive any definitive claims from the mouse or zebrafish tests.

Regarding the reviewer’s proposal that liver CREs would have been easier to validate, we want to clarify that this is not the case given our validation pipeline. We first took a conservative and efficient approach to validate a sequence in the mouse by performing embryonic lacZ reporter screens on F0 transgenic animals. This technique is not compatible with the testing of HepG2-specific CREs due to the pigmentation of the developing liver that prevents visualization of the X-gal substrate. This, combined with the general interest in reducing off-target transgenic effects in the liver, led us to test neuronal CREs, leaving the candidate liver-targeting elements untested. After receiving positive results at E14.5, we used the same targeting vectors to produce the immunohistochemistry images of 5-week old brains.

R2C7. Our results suggest that CREs designed for tissue-specific targeting can work across species, even in the brain, which has been an ongoing challenge to target with viral-based delivery approaches. An integrated framework leveraging human cell lines in conjunction with whole organism models may thus be a viable approach to rapidly identify CREs to execute novel functions in humans.

If the goal is to find tissue-specificity with certain neural cell types then the enhancers need to be active within a particular type of neural cells. Yet in the study, only neural is selected, and the particular neural cell types vary widely between embryos, enhancers and species in the 2 CREs validated. While this framework could work it has not been demonstrated by the data in this manuscript.

The text noted here is in the third to last paragraph in the discussion where we are contextualizing the results and providing a forward-looking hypothesis on what may be feasible (ex. *“may thus be a viable approach”*). We fully agree that our experiments do not conclusively demonstrate that our current approach can be used as presented to overcome a specific and targeted gene delivery challenge which, to our knowledge, we do not state otherwise in the manuscript. We simply aimed to convey that our results, in combination with Taskiran et al. and de Almeida et al., can provide critical justification for future application of machine-guided engineering frameworks, including our own, to increasingly complex and resource intensive experimental systems that more comprehensively model the requirements of a specific goal.

In order to ensure this is clear, we have re-written the statement in the discussion to further emphasize that additional work is required.

Line 383:

“These findings show it is feasible for CREs with novel functionalities developed in vitro to maintain specificity in analogous tissues in vivo. We were surprised that our neuronal synN1 CRE, designed from a single transformed SK-N-SH cell line, exhibited highly specific sub-cortical expression in mice. Further research is needed to develop optimal strategies to translate in vitro models to precise targeting in vivo. An integrated framework that combines human cell lines with whole organism experimental models

may be an effective approach to rapidly identify CREs capable of accomplishing novel functions in humans.”

R2C8. We successfully deployed CODA for cell type specificity

The ability to get specificity is lower for the SK and HEP optimized sequences when considering the A549 and HCT116 cell lines suggesting that there are limitations in this approach to find CREs with specificity. I have also discussed this in terms of in vivo specificity above.

As we note in R2C4, we did not directly deploy CODA to design sequences with off-target expression in A549 and HCT116. We do however conclusively achieve specificity across the studied cell lines K562, HepG2, and SK-N-SH. Our work does not intend to imply that leveraging MPRA in these three cell lines is enough to induce off-target repression in all other cell types other than the target. A better assessment of the limits of our approach to additional cell lines would be to test by MPRA sequences that have been explicitly designed using models trained in these additional cell types, which would be an interesting and suitable research question, but unfortunately beyond the scope of the proposed work. Nonetheless, while the intensity of specificity went down in A549 and HCT116 for HepG2- and SK-N-SH-optimized sequences, the targeted cell line is still highest. We made the specificity objectives more clear throughout the paper in the previous response to this reviewer, relating to these three cell lines using reporter assays specifically.

R2C9. Fig S26 title needs some attention. type: token ratio – is very technical and it would be good to make this more understandable

We apologize for not providing a suitable title for **Supplementary Figure 26**, we have updated it to read “**Lexical analysis of motif content**” (now **Supplementary Note 2 - Figure 4**). We have also moved this analysis to Supplementary Note 2 to allow further explanation of types and tokens which are commonly employed in language analysis.

R2C10. Figure S3. All y axes should be labelled.

We thank the reviewer for bringing this to our attention. We have updated the y-labels accordingly.

R2C11. Stats should be shown in the main figures when claiming significant enrichments, for example: Figure 1d, could they please show the stats in the main figure. They mention they have done an MWU, but it would be good to see the stats in the main figure.

Given the high number of data points used in this analysis ($n = 62,582$), any correlation coefficient above 0.01 will be significant (i.e., $p < 0.05$) and above 0.31 will cause underflow when calculating p-values (i.e., $p < 10^{-308}$) so we inadvertently omitted p-values for correlation analysis when n is very large (e.g., above 10,000 data points). We have included in the legend that $p < 10^{-300}$ for **Figure 1d**. In accordance with this update, we also include the same p-value indication for **Figure 1e** in the legend. We note **Figure 1f** is a visual representation of signal tracks at DHS peaks to appreciate concordance in the data, but was not used directly to make a statistical comparison. However, a closely related statistical comparison is presented in **Extended Data Figure 2c**.

R2C12. Fig S27. I get that topics are technical terms, but the paper may be more understandable if they tried getting. Away from technical language. Instead of calling these topics, perhaps the authors could name these “TF vocabularies” or “TF Vocab 1”, “TF vocab 2”, ...

We apologize for the oversight in the “Topic” labels in **Supplementary Figure 14** (previously **Supplementary Figure 27**) and have corrected these to read “Program” which is in keeping with the rest of the terminology related to NMF in our manuscript. While we agree that diverging from the standard use of the term could help unfamiliar readers grasp the concepts more easily, we think that being consistent with the original terminology is important to connect readers to related literature. In our writing we try to achieve a balance of using established language with providing simple descriptions of

the analysis/results. We believe this helps maintain a shared vocabulary within a field while making our work accessible to those outside the field. To be consistent with the terminology used in gene expression analyses referred to in the **Methods**, we have now ensured exclusive use of the term “programs” except for one the instance in the **Methods** used to explain the conceptual similarity in language topics and gene expression programs. To aid readers unfamiliar to these concepts we include a more detailed description when introducing the term.

Line 267:

“These programs describe co-occurring TF vocabularies found in the elements we tested.”

References:

- Avsec, Ž., Agarwal, V., Visentin, D., Ledsam, J. R., Grabska-Barwinska, A., Taylor, K. R., Assael, Y., Jumper, J., Kohli, P., & Kelley, D. R. (2021). Effective gene expression prediction from sequence by integrating long-range interactions. *Nature Methods*, *18*(10), 1196–1203.
- Avsec, Ž., Weilert, M., Shrikumar, A., Krueger, S., Alexandari, A., Dalal, K., Fropf, R., McAnany, C., Gagneur, J., Kundaje, A., & Zeitlinger, J. (2021). Base-resolution models of transcription-factor binding reveal soft motif syntax. *Nature Genetics*, *53*(3), 354–366.
- Lal, A., Gunsalus, L., Gupta, A., Biancalani, T., & Eraslan, G. (2024). Polygraph: A Software Framework for the Systematic Assessment of Synthetic Regulatory DNA Elements. In *bioRxiv* (p. 2023.11.27.568764). <https://doi.org/10.1101/2023.11.27.568764>
- Linder, J., Srivastava, D., Yuan, H., Agarwal, V., & Kelley, D. R. (2023). Predicting RNA-seq coverage from DNA sequence as a unifying model of gene regulation. In *bioRxiv* (p. 2023.08.30.555582). <https://doi.org/10.1101/2023.08.30.555582>
- Nguyen, E., Poli, M., Faizi, M., Thomas, A., Birch-Sykes, C., Wornow, M., Patel, A., Rabideau, C., Massaroli, S., Bengio, Y., Ermon, S., Baccus, S. A., & Ré, C. (2023). HyenaDNA: Long-Range Genomic Sequence Modeling at Single Nucleotide Resolution. *ArXiv*. <https://www.ncbi.nlm.nih.gov/pubmed/37426456>
- Taskiran, I. I., Spanier, K. I., Dickmänken, H., Kempynck, N., Pančíková, A., Ekşi, E. C., Hulselmans, G., Ismail, J. N., Theunis, K., Vandepoel, R., Christiaens, V., Mauduit, D., & Aerts, S. (2024). Cell-type-directed design of synthetic enhancers. *Nature*, *626*(7997), 212–220.